# A Principled Path to Fitted Distributional Evaluation

**Sungee Hong**
Texas A&M University

**Jiayi Wang**
University of Texas at Dallas

**Zhengling Qi**
George Washington University

**Raymond K. W. Wong**
Texas A&M University

## Abstract

In reinforcement learning, distributional off-policy evaluation (OPE) focuses on estimating the return distribution of a target policy using offline data collected under a different policy. This work focuses on extending the widely used fitted Q-evaluation—developed for expectation-based reinforcement learning—to the distributional OPE setting. We refer to this extension as fitted distributional evaluation (FDE). While only a few related approaches exist, there remains no unified framework for designing FDE methods. To fill this gap, we present a set of guiding principles for constructing theoretically grounded FDE methods. Building on these principles, we develop several new FDE methods with convergence analysis and provide theoretical justification for existing methods, even in non-tabular environments. Extensive experiments, including simulations on linear quadratic regulators and Atari games, demonstrate the superior performance of the FDE methods.

## 1 Introduction

In reinforcement learning (RL), the return, defined as the cumulative sum of (discounted) rewards, is a fundamental measure for how well the underlying policy performs. Traditional RL assesses policies by calculating the expected return, whereas distributional RL [5] focuses on the full distribution of return, providing a richer and more complete understanding of the policy behavior. Leveraging distributional properties beyond the expectation (e.g., risk, multimodality) enables the handling of more general tasks, such as risk assessment [27] and risk-sensitive policy learning [e.g., 15, 32]. Therefore, distributional RL has found applications across various domains [e.g., 8, 4, 64, 17, 12].

In this paper, we focus on addressing off-policy evaluation (OPE) problems within distributional RL, referred to as distributional OPE. The goal is to estimate the return distribution of a target policy using data collected under a different policy known as the behavior policy. Many distributional OPE methods extend existing OPE techniques from traditional RL, including temporal difference (TD) methods [e.g., 7, 14, 65, 52] and model-based approaches [e.g., 31]. Bellman residual minimization, a well-studied method in traditional RL [e.g., 33, 21, 48], was recently adapted for distributional OPE by [26]. Another key methodology, fitted Q-evaluation (FQE)—an iterative algorithm extensively analyzed in traditional RL [e.g., 10, 69, 46, 59]—has also been recently extended to the distributional setting [32, 63]. In this work, we focus on the distributional extension of FQE.

Unlike FQE, where the discrepancy function (squared $\ell_2$-loss) is defined over real numbers, distributional extensions operate directly on distributions, making the choice of discrepancy non-trivial. Although statistical distances are natural candidates for measuring discrepancy between distributions, some well-known distances (e.g., total variation distance) perform poorly. Existing distributional extensions of FQE [32, 63] are developed based on specific discrepancy functions. [32] utilizes the $p$-powered Wasserstein-$p$ metric, whereas [63] is based on log-likelihood (closely related to Kullback–Leibler (KL) divergence). A general guideline for selecting appropriate discrepancy functions

in FDE remains unclear, leaving practitioners without systematic guidance for developing valid FQE extensions tailored to their specific needs. In this paper, we address this gap by proposing a unified framework for FDE that clarifies the role of discrepancy function selection and enables analysis under a broad class of discrepancy functions and general conditions (including non-tabular settings, i.e., inifinite state-action space). Along the way, we establish several guiding principles and provide theoretical support for our framework. Beyond these general principles, we present concrete examples of discrepancy functions derived from our framework, accompanied by their statistical analyses, offering readily applicable discrepancy functions for practitioners. Additionally, our general framework gives rise to FDE methods that overcome some challenges faced by existing distributional OPE approaches (beyond those FQE extensions). In particular, many of such FDE methods support multi-dimensional returns, which are incompatible with widely used quantile-based methods [e.g., 15, 32, 47]. Furthermore, we provide examples of FDE methods with theoretical guarantees for unbounded distributions, in contrast to conventionally assumed bounded distributions [e.g., 63, 45].

Our key contributions are summarized as follows. (1) *Principled Framework for Fitted Distributional Evaluation:* We introduce guiding principles for constructing theoretically grounded extensions of FQE to the distributional OPE problems, which we term fitted distributional evaluation (FDE) methods. In particular, we show that functional Bregman divergences are natural candidates for the discrepancy measures within this framework. See Section 2; (2) *Survey of Valid Discrepancy Measures:* Leveraging the proposed framework, we derive many examples and discover novel discrepancy measures that have not previously been considered as objective functions for distributional OPE. See Tables 1 and 2; (3) *Unified Statistical Convergence Analysis:* We provide a comprehensive convergence analysis covering a broad class of FDE methods in tabular (i.e., finite state-action space) and non-tabular settings. This significantly expands the set of distributional OPE methods with theoretical guarantees in non-tabular scenarios. See Section 3.

## 2 Fitted distributional evaluation in off-policy settings

Consider a homogeneous Markov decision process $(\mathcal{S}, \mathcal{A}, \tilde{p}, \gamma)$ where $\mathcal{S}$ and $\mathcal{A}$ refer to the state and action spaces, $\tilde{p}$ is the transition probability, and $\gamma \in [0, 1)$ is the discount factor. More specifically, the transition probability $\tilde{p}$ represents the conditional distribution of reward $R \in \mathbb{R}^d$ and next state $S' \in \mathcal{S}$ conditional on the current state $S \in \mathcal{S}$ and action $A \in \mathcal{A}$. We shall focus on the evaluation of a stationary target policy $\pi : \mathcal{S} \to \Delta(\mathcal{A})$ under infinite-horizon setting. Consider a trajectory $\{(S_t, A_t, R_t)\}_{t \geq 0}$ generated by iteratively sampling from the target policy and the transition probability: $A_t \sim \pi(\cdot|S_t)$ and $R_t, S_{t+1} \sim \tilde{p}(\cdot|S_t, A_t)$ (and some initial distribution of $S_0$). The return of the trajectory is defined by $\sum_{t \geq 0} \gamma^t R_t$. Unlike traditional RL which focuses on the policy value, i.e., the expectation of return, distributional RL considers the whole distribution of return. Analogously to Q-function estimation in traditional RL, the policy evaluation problem in distributional RL, known as distributional policy evaluation, aims to estimate the conditional distribution of return $\Upsilon_\pi \in \mathcal{P}^{\mathcal{S} \times \mathcal{A}}$ (with $\mathcal{P} \subseteq \Delta(\mathbb{R}^d)$ as a convex set of probability measures[1]), where $\Upsilon_\pi(s, a) \in \mathcal{P}$ refers to the probability measure of $\sum_{t \geq 0} \gamma^t R_t$ under $\pi$ starting from the initial state-action pair $s, a$. We are interested in distributional policy evaluation in the off-policy setup, where data are collected under a different policy, resulting in a distributional mismatch. More specifically, we shall assume that the data consisting of tuples in the form $(S, A, R, S')$ that are generated by $S \sim \mu$, $A \sim b(\cdot|S)$ with $b$ being the behavior policy, and followed by $(R, S') \sim \tilde{p}(\cdot|S, A)$. We further let $(S, A) \sim \rho = \mu \times b$.

### 2.1 Fitted distributional evaluation algorithm

In traditional RL, fitted Q-evaluation (FQE) [e.g., 10, 30, 58, 29, 46, 59] is a popular approach to estimate $Q$-function, i.e., $Q_\pi(s, a) := \mathbb{E}\{\sum_{t \geq 0} \gamma^t R_t | s, a\}$. It is motivated by the Bellman equation

$$Q_\pi(s, a) = \mathbb{E}_{R, S' \sim \tilde{p}(\cdot|s, a), A' \sim \pi(\cdot|S')} \left\{ R + \gamma \cdot Q_\pi(S', A') \right\} =: (\mathcal{B}^\pi Q_\pi)(s, a), \quad s, a \in \mathcal{S} \times \mathcal{A}, \quad (1)$$

where $\mathcal{B}^\pi : \mathbb{R}^{\mathcal{S} \times \mathcal{A}} \to \mathbb{R}^{\mathcal{S} \times \mathcal{A}}$ is known as the Bellman operator. Based on the contractive property of $\mathcal{B}^\pi$ with respect to $L_\infty$-norm, iterative application of the Bellman operator yields convergence towards

---

[1]For our objective functions to be well-defined, $\mathcal{P}$ should be closed under push-forward mapping with respect to affine maps: $g(x) = r + \gamma x$ for any $r$ in the support of the reward distribution. See Appendix C.1.2.

the unique target, i.e., $\lim_{T\to\infty}(\mathcal{B}^\pi)^T Q_0 = Q_\pi$ in $L_\infty$-norm for an initial value $Q_0$. However, the Bellman operator is often unknown and so is the evaluation of the RHS in (1), rendering this approach impractical. Instead, given a data set $\mathcal{D} = \{(s_i, a_i, r_i, s'_i)\}_{i=1}^N$, FQE aims to iteratively approximate $Q_t \approx \mathcal{B}^\pi Q_{t-1}$ by minimizing the mean squared error

$$Q_t = \arg\min_{Q\in\mathcal{Q}} \frac{1}{|\mathcal{D}|} \sum_{(s,a,r,s')\in\mathcal{D}} \left[ Q(s,a) - \underbrace{\left\{ r + \gamma \cdot \mathbb{E}_{A'\sim\pi(\cdot|s')}\{Q_{t-1}(s',A')\} \right\}}_{\text{sample-approximated RHS of } \mathcal{B}^\pi Q_{t-1}(s,a)} \right]^2 \quad (t \geq 1), \quad (2)$$

where $\mathcal{Q} \subseteq \mathbb{R}^{\mathcal{S}\times\mathcal{A}}$ is a chosen function class. Succinctly stated, FQE consists of iteratively solving regression problems with predictions $Q(s,a)$ and the Bellman backup $r + \gamma\cdot\mathbb{E}_{A'\sim\pi(\cdot|s')}(Q_{t-1}(s',A'))$, where the squared $\ell_2$-loss is used to measure the discrepancy.

We aim to extend FQE for distributional policy evaluation, where the quantity of interest is the whole conditional distribution $\Upsilon_\pi \in \mathcal{P}^{\mathcal{S}\times\mathcal{A}}$ instead of the conditional mean $Q_\pi \in \mathbb{R}^{\mathcal{S}\times\mathcal{A}}$. To this end, we define the distributional Bellman operator $\mathcal{T}^\pi : \mathcal{P}^{\mathcal{S}\times\mathcal{A}} \to \mathcal{P}^{\mathcal{S}\times\mathcal{A}}$ [7] by

$$(\mathcal{T}^\pi\Upsilon)(s,a) := \int_{\mathbb{R}^d\times\mathcal{S}\times\mathcal{A}} (g_{r,\gamma})_\# \Upsilon(s',a') \mathrm{d}\pi(a'|s') \mathrm{d}\tilde{p}(r,s'|s,a), \quad (3)$$

where $(g_{r,\gamma})_\# : \mathcal{P} \to \mathcal{P}$ is the push-forward mapping with respect to the function $g_{r,\gamma}(x) := r + \gamma x$, that maps the distribution of any random vector $X$ to the distribution of $r + \gamma X$. Analogously to (1), $\Upsilon_\pi$ is the solution to the distributional Bellman equation: $\Upsilon = \mathcal{T}^\pi\Upsilon$. As in FQE, we would like to form a sequence $\Upsilon_t \approx \mathcal{T}^\pi\Upsilon_{t-1}$ based on the data. In a similar spirit of FQE, given a single observation $(s, a, r, s')$, we compute the distributional Bellman backup

$$\Psi_\pi(r, s', \Upsilon_{t-1}) := \int_{\mathcal{A}} (g_{\gamma,r})_\# \Upsilon_{t-1}(s',a') \mathrm{d}\pi(a'|s') \in \mathcal{P}, \quad (4)$$

as the target. Then we aim to optimize the prediction $\Upsilon(s,a)$ such that some discrepancy measure is minimized. Specifically, consider an appropriate mapping $\mathfrak{d} : \mathcal{P} \times \mathcal{P} \to [0,\infty)$, we can then formulate a distributional extension of FQE by performing iterative minimization:

$$\Upsilon_t = \arg\min_{\Upsilon\in\mathcal{M}} \frac{1}{|\mathcal{D}|} \sum_{(s,a,r,s')\in\mathcal{D}} \mathfrak{d}\{\Upsilon(s,a), \Psi_\pi(r,s',\Upsilon_{t-1})\}, \quad t \geq 1, \quad (5)$$

where $\mathcal{M} \subseteq \mathcal{P}^{\mathcal{S}\times\mathcal{A}}$ is a chosen set of conditional distributions. We call the resulting method *fitted distributional evaluation* (FDE). See Algorithm 1 in Appendix A for its full algorithm. In passing, [32] also discussed a distributional FQE extension with the same name, but there are some key differences as will be explained in Remark 2.1.

In FQE, it is straightforward to choose squared $\ell_2$-loss for minimization as the Bellman backup is real number. However, in FDE, the distributional Bellman backup is a distribution. Thus, to construct a FDE algorithm, a key question is:

*How to choose $\mathfrak{d}$ correctly to build a theoretically grounded FDE?*

This question is indeed non-trivial. While many known statistical distances are natural candidates for measuring discrepancy between two probabilities, a number of them fail. For instance, total variation distance can perform poorly as shown in Section 4 (also see Figure 1). A major reason is that (the expected-extended or supremum-extended) total variation distance does not lead to contraction of the distributional Bellman operator [5]. See Section 2.3 for more details. This may not be surprising, as the core motivation behind FQE is the contraction argument, which likely extends to its distributional counterpart, FDE. However, a metric that guarantees contraction of the distributional Bellman operator may also fail. One example is the (expectation-extended or supremun-extended) Wasserstein distance due to biased gradient [7]. The issue originates from the sample approximation based on the Bellman backup in (5), which will be explained in Section 2.4. In Section 2.2, we will outline several principles to choose a valid discrepancy $\mathfrak{d}$, based on a motivating error-bound analysis.

*Remark* 2.1. Two existing methods share a similar construction with FDE. Crucially, both focus exclusively on specific objective functions, without any general guideline on how to select an appropriate divergence $\mathfrak{d}$. First, [32] formulated their objective using powered Wasserstein-$p$ metric (see their Equation (4)). Instead of applying the distributional Bellman backup $\Psi_\pi$ (4) based on a

single sample, [32] estimates $\mathcal{T}^\pi$ by leveraging the entire data to approximate the transition dynamics with conditional empirical distributions. However, this approach generally fails to yield consistent estimates in non-tabular settings (i.e., $|\mathcal{S} \times \mathcal{A}| = \infty$), introducing potential bias. The second method is fitted likelihood evaluation (FLE) [63], which is based on log-likelihood. Unlike FDE, FLE uses only a single draw from the distribution Bellman backup, leading to information loss. Moreover, FLE is restricted to the cases where the return distributions have densities. Theoretically, their statistical convergence rate (see their Corollary 4.14) is also slower than ours; see details in Section 3.2.

## 2.2 A motivating error-bound analysis

To motivate a theoretically grounded FDE method, we begin with an error-bound analysis. Firstly, the metric under which the distributional Bellman operator is contractive plays a crucial role.

**Definition 2.2.** (Contraction-inducing metric) Let $\tilde{\eta}$ be a metric over $\mathcal{P}^{\mathcal{S} \times \mathcal{A}}$. We call it a contraction-inducing metric if $\mathcal{T}^\pi$ is $\zeta$-contraction with respect to $\tilde{\eta}$, i.e., $\tilde{\eta}(\mathcal{T}^\pi \Upsilon_1, \mathcal{T}^\pi \Upsilon_2) \leq \zeta \cdot \tilde{\eta}(\Upsilon_1, \Upsilon_2)$ for a constant $\zeta \in (0, 1)$.

We will focus exclusively on contraction-inducing metrics under which $\mathcal{T}^\pi$ is contractive. Given a contraction-inducing metric $\tilde{\eta}$, we can derive the inequality for any sequence $\Upsilon_0, \ldots, \Upsilon_T \in \mathcal{P}^{\mathcal{S} \times \mathcal{A}}$ (see Appendix C.2):

$$\tilde{\eta}(\Upsilon_T, \Upsilon_\pi) \leq \sum_{t=1}^{T} \zeta^{T-t} \cdot \underbrace{\tilde{\eta}(\Upsilon_t, \mathcal{T}^\pi \Upsilon_{t-1})}_{\text{iteration-level error}} + \underbrace{\zeta^T \cdot \tilde{\eta}(\Upsilon_0, \Upsilon_\pi)}_{\text{shrinks to zero as } T \to \infty} , \tag{6}$$

which provides an upper bound for $\tilde{\eta}(\Upsilon_T, \Upsilon_\pi)$, referred to as the $\tilde{\eta}$-error of $\Upsilon_T$. Roughly speaking, the second term $\zeta^T \cdot \tilde{\eta}(\Upsilon_0, \Upsilon_\pi)$ in the error upper bound (6) becomes negligible as we increase $T$, due to $\zeta \in (0, 1)$. Then, to ensure small $\tilde{\eta}$-error of $\Upsilon_T$, it suffices to additionally require that $\tilde{\eta}(\Upsilon_t, \mathcal{T}^\pi \Upsilon_{t-1})$ converges. This is the essential goal of the minimization (5). To build a successful FDE method, we lay out the following principles to choose a valid pair $(\tilde{\eta}, \mathfrak{d})$:

(P1) $\tilde{\eta}$ is a contraction-inducing metric;

(P2) For any given $\tilde{\Upsilon} \in \mathcal{P}^{\mathcal{S} \times \mathcal{A}}$, the unique[2] minimizer of the population objective $F(\Upsilon; \tilde{\Upsilon}) := \mathbb{E}_{S,A \sim \rho, R, S' \sim \tilde{p}(\cdot|S,A)} \{\mathfrak{d}(\Upsilon(S, A), \Psi_\pi(R, S', \tilde{\Upsilon}))\}$ is $\Upsilon = \mathcal{T}^\pi \tilde{\Upsilon}$.

(P3) For any $\tilde{\Upsilon} \in \mathcal{P}^{\mathcal{S} \times \mathcal{A}}$, there exists a function $g : \mathbb{R} \to [0, \infty)$ such that: (i) $g(\xi) = 0$ and $g$ is continuous at $\xi = \min_\Upsilon F(\Upsilon; \tilde{\Upsilon})$; (ii) $\tilde{\eta}(\Upsilon, \mathcal{T}^\pi \tilde{\Upsilon}) \leq g(F(\Upsilon; \tilde{\Upsilon}))$ for any $\Upsilon \in \mathcal{P}^{\mathcal{S} \times \mathcal{A}}$.

(P2) ensures that the minimizer of the population objective $F(\Upsilon; \Upsilon_{t-1})$ of (5) is the target $\mathcal{T}^\pi \Upsilon_{t-1}$. In addition, (P3) indicates that sufficiently small value of $F(\Upsilon; \Upsilon_{t-1})$ implies closeness between $\Upsilon$ and $\mathcal{T}^\pi \Upsilon_{t-1}$ with respect to the metric $\tilde{\eta}$. In the following subsections, we will center our discussion around these three principles. With the systematic construction based on these principles, we are able to construct a wide range of FDEs, even with choices of $\mathfrak{d}$ that have never been used in the current literature of distributional RL (see Table 2). The above principles do not address the finite-sample error, but we will provide a unified statistical analysis for a broad class of FDEs in Section 3.

## 2.3 Contraction-inducing metrics

In this subsection, we will focus on the choices of contraction-inducing metrics (P1). To broaden the choices of valid discrepancy $\mathfrak{d}$ (see (P3)), we describe two classes of contraction-inducing metrics over $\mathcal{P}^{\mathcal{S} \times \mathcal{A}}$. The first class comprises supremum-extended metrics (7), which are well studied in the literature [e.g., 42, 40, 6], whereas the second class consists of expectation-extended metrics (8), which have received significantly less attention but are particularly useful for large or continuous state-action spaces.

**Supremum extension** Given a probability metric $\eta$ over $\mathcal{P}$, its supremum-extended metric is defined as

$$\eta_\infty(\Upsilon_1, \Upsilon_2) := \sup_{s,a} \left\{ \eta(\Upsilon_1(s, a), \Upsilon_2(s, a)) \right\}, \quad \Upsilon_1, \Upsilon_2 \in \mathcal{P}^{\mathcal{S} \times \mathcal{A}}. \tag{7}$$

---

[2]Uniqueness is only required to hold almost surely over the data generating distribution $\rho$.

If the individual probability metric $\eta$ satisfies (i) *scale-sensitivity* (*c*-sensitivity), (ii) *location-insensitivity* (regularity) and (iii) *q-convexity*, collectively denoted by (S-L-C) with definitions deferred to Appendix C.4, then $\eta_\infty$ is a contraction-inducing metric with $\zeta = \gamma^c$ (proof in Appendix C.3.2). Our result is a slight extension from Theorem 4.25 of [6], not requiring that $R$ and $S'$ to be independent conditioned on $S, A$. We have listed three examples that satisfy (S-L-C) in Table 3 of Appendix C.4, along with a survey of other well-known probability metrics that fail to satisfy (S-L-C). Examples include total variation distance (TVD) whose supremum extension is not guaranteed to be a contraction-inducing metric [5].

However, controlling supremum-extended metric (7) requires uniform control of probability metrics over all state-action pairs. For large or infinite state-action space, supremum-extended metrics can be challenging to control, which limits the choice of discrepancy $\mathfrak{d}$ in view of (P3). This is related to the fundamental difficulty for using an expectation-based criterion (or the sample version in (5)) to control a supermum-based quantity (7). Indeed, existing analyses of statistical error bounds in $\eta_\infty$ for distributional OPE methods mainly focus on tabular setting ($|\mathcal{S} \times \mathcal{A}| < \infty$) [e.g., 47, 22, 45].

**Expectation extension**    In view of the above explanation, we also introduce expectation-extended metrics which are more compatible with the expectation-based criterion (P3). Given a probability metric $\eta$ over $\mathcal{P}$ and a parameter $q \geq 1$, an expectation-extended metric is defined as

$$\bar{\eta}_{d_\pi, q}(\Upsilon_1, \Upsilon_2) := \left[ \mathbb{E}_{S, A \sim d_\pi} \left\{ \eta^{2q}\big(\Upsilon_1(S, A), \Upsilon_2(S, A)\big) \right\} \right]^{\frac{1}{2q}}, \tag{8}$$

where $d_\pi \in \Delta(\mathcal{S} \times \mathcal{A})$ is defined as $d_\pi = (1 - \gamma)^{-1} \sum_{h=1}^\infty \gamma^{h-1} d_\pi^h$ with $d_\pi^h(E) := \mathbb{P}((S_h, A_h) \in E | S_0, A_0 \sim \rho, A_t \sim \pi(\cdot|A_t)$ for $t \geq 1)$. The distribution $d_\pi$ has appeared commonly in the RL literature [e.g., 39, 68, 66], and is important for ensuring contraction-inducing property as in Theorem 2.3 below. Note that the supremum-extended metric $\eta_\infty$ (7) can be regarded as a special case of $\bar{\eta}_{d_\pi, q}$ when $q \to \infty$ (under appropriate conditions of $d_\pi$). Expectation-extended metrics (based on possibly different distributions in the expectation) have been recently used for distributional OPE [63, 26]. Here, we provide a new result that facilitates the construction of a contraction-inducing expectation-extended metric as follows, with the corresponding proof given in Appendix C.3.1.

**Theorem 2.3.** *Suppose that a probability metric $\eta$ over $\mathcal{P}$ satisfies (S-L-C) with convexity parameter $q \geq 1$ and scale-sensitivity parameter $c > 1/(2q)$ (see (15) of Appendix C.4). Then the expectation-extended metric $\bar{\eta}_{d_\pi, q}$ is a contraction-inducing metric with $\zeta = \gamma^{c - \frac{1}{2q}}$ defined in Definition 2.2.*

## 2.4   Discrepancy measures

We will now provide some guideline on the choice of discrepancy $\mathfrak{d}$. Based on (P2), we would ask how to choose $\mathfrak{d}$ such that, for any $\Upsilon \in \mathcal{P}^{\mathcal{S} \times \mathcal{A}}$,

$$\mathcal{T}^\pi \Upsilon = \arg \min_{\Upsilon' \in \mathcal{P}^{\mathcal{S} \times \mathcal{A}}} \mathbb{E}_{S, A \sim \rho, R, S' \sim \tilde{p}(\cdot|S,A)} \left\{ \mathfrak{d}\big(\Upsilon'(S, A), \Psi_\pi(R, S', \Upsilon)\big) \right\}, \tag{9}$$

where the minimizer is unique up to almost surely equivalence. Note that $\mathbb{E}\{\Psi_\pi(R, S', \Upsilon)|S = s, A = a\} = \mathcal{T}^\pi \Upsilon(s, a)$ for any $(s, a) \in \mathcal{S} \times \mathcal{A}$. As such, we hope that this conditional expectation of random measure is the minimizer of the expected discrepancy. In the case of FQE where the discrepancy is defined between two scalar values (see (2)), it is well known that minimizing the expected squared loss yields the conditional expectation. However, extending this property to settings where the discrepancy is defined between two measures is less obvious. Nevertheless, we show that a broad family of discrepancies—the functional extension of Bregman divergences—does satisfy this desirable property. This result significantly expands the possible construction of FDE. The formal definition of functional Bregman divergence [43] is technically involved and thus deferred to Definition C.1 of Appendix C.1.1. Before further discussion of functional Bregman divergences, we present the following key result, which we prove in Appendix C.1.2:

**Theorem 2.4.** *A functional Bregman divergence $\mathfrak{d}$ satisfies (9) for any $\rho \in \Delta(\mathcal{S} \times \mathcal{A})$, transition $p$, and target policy $\pi$.*

Despite the technically involved definition of functional Bregman divergence, it has a close relationship with strictly proper scoring rule, which has been broadly studied in statistics literature. A scoring rule $S(\cdot, *) : \mathcal{P} \times \Omega_\mathcal{X} \to \mathbb{R}$ (with $\Omega_\mathcal{X}$ being the corresponding support space of $\mathcal{P}$, say $\mathbb{R}^d$) is strictly

proper if $\bar{S}(Q,Q) \geq \bar{S}(P,Q)$ where $\bar{S}(P,Q) := \int_{\Omega_{\mathcal{X}}} S(P,x)\mathrm{d}Q(x)$ [23] with equality holding only when $P = Q$. Given a strictly proper scoring rule $S$, we can always build a functional Bregman divergence by letting $\mathfrak{d}(P,Q) = \bar{S}(Q,Q) - \bar{S}(P,Q) \geq 0$, and vice versa (see Definition 3.8 and Theorem 4.1 of [43]). This linkage, together with Theorem 2.4, provides justifications to many examples that have been used in distributional RL, including logarithmic scoring rule [e.g., 63, 61] that corresponds to Kullback–Leibler (KL) divergence, and squared maximum mean discrepancy (MMD) with specific kernels [40]. More interestingly, we also find (and analyze) various examples that have never been used in distributional RL, including squared MMD with additional kernels and $L_2$ distance based on density functions (see Table 2). Finally, our result also provides justifications for adopting other strictly proper scoring rules (e.g., survival, spheric, Hyvärinen, Tsallis, Brier scoring rules) [e.g., 23, 43, 16], which are not analyzed in this work.

*Remark* 2.5. With appropriate differentiability condition, the property (9) also implies that expected gradient of $\mathfrak{d}(\Upsilon(S, A), \Psi_\pi(R, S', \Upsilon_{t-1}))$ (with respect to $\Upsilon$) becomes zero at $\Upsilon = \mathcal{T}^\pi \Upsilon_{t-1}$. This is a crucial unbiased gradient property for building TD-based or more general gradient-based algorithms. Despite our focus on FDE, we note that Theorem 2.4 also provides justifications for building such algorithms (e.g., TD update based on squared-MMD in [40, 67]) via functional Bregman divergence.

Next, we discuss the last principle (P3). Unlike (P1) and (P2), which involve a single quantity (either $\tilde{\eta}$ or $\mathfrak{d}$), (P3) requires establishing an appropriate relationship between the contraction-inducing metric $\tilde{\eta}$ and the discrepancy $\mathfrak{d}$. Since each discrepancy $\mathfrak{d}$ has its own relationship with different probability metrics $\eta$ (prior to their extension to $\tilde{\eta}$), the discussion of (P3) becomes specific to each pair $(\tilde{\eta}, \mathfrak{d})$. While a certain level of generalization is possible—for the squared form of some probability metric (i.e., $\mathfrak{d} = m^2$), $\tilde{\eta}$ can be controlled under conditions such as Assumption 3.2—this framework does not accommodate other divergences (e.g., KL divergence). Additionally, depending on the cardinality of $\mathcal{S} \times \mathcal{A}$, different forms of $\tilde{\eta}$ may be employed (e.g., $\tilde{\eta} = \eta_\infty$ or $\tilde{\eta} = \bar{\eta}_{d_\pi, q}$). Consequently, unlike (P1) and (P2), it is challenging to establish a concise guideline to choose $(\tilde{\eta}, \mathfrak{d})$ based on (P3). Instead, we study a number of examples (Table 1 for $\tilde{\eta} = \eta_\infty$ and Table 2 for $\tilde{\eta} = \bar{\eta}_{d_\pi, q}$). Section 3 provides statistical convergence analyses of FDE methods based on different choices of $\mathfrak{d}$.

## 3 Theoretical results

First, we make the modeling in (5) explicit and write $\mathcal{M} = \mathcal{M}_\Theta := \{\Upsilon_\theta \in \mathcal{P}^{\mathcal{S} \times \mathcal{A}} : \theta \in \Theta\}$. We assume that the offline dataset $\mathcal{D} = \{(s_i, a_i, r_i, s_i')\}_{i=1}^N$ consists of $N$ independently and identically distributed draws according to: $(s_i, a_i) \sim \rho$ and $r_i, s_i' \sim \tilde{p}(\cdot|s_i, a_i)$. To simplify the theoretical analysis, we will slightly modify the objective function (5) so that we use non-overlapping subsets of data in each iteration. That is, the data is first split into $T$ equally sized partitions, i.e., $\mathcal{D} = \cup_{t \in [T]} \mathcal{D}_t$ with $|\mathcal{D}_t| = n = N/T$ (assuming that $N$ is divisible by $T$, without loss of generality), and, at the $t$-th iteration, we obtain the estimator $\hat{\theta}_{n,t} := \arg\min_{\theta \in \Theta} \hat{F}_{n,t}(\theta|\hat{\theta}_{n,t-1})$ where

$$\hat{F}_{n,t}(\theta|\hat{\theta}_{n,t-1}) := \frac{1}{n} \sum_{(s,a,r,s') \in \mathcal{D}_t} \mathfrak{d}\{\Upsilon_\theta(s,a), \Psi_\pi(r, s', \Upsilon_{\hat{\theta}_{n,t-1}})\}. \tag{10}$$

For convergence of iteration-level error in (6), i.e., $\tilde{\eta}(\Upsilon_{\hat{\theta}_{n,t}}, \mathcal{T}^\pi \Upsilon_{\hat{\theta}_{n,t-1}}) \xrightarrow{P} 0$, $\mathcal{T}^\pi \Upsilon_{\hat{\theta}_{n,t-1}}$ should be accommodated in the chosen model, which is implied by the completeness assumption:

**Assumption 3.1.** (Completeness) For $\forall \theta \in \Theta$, $\mathcal{T}^\pi \Upsilon_\theta = \Upsilon_{\theta'}$ for some $\theta' \in \Theta$.

This assumption is common in the analysis of many iteration-based RL algorithms [e.g., 13, 61, 63, 60, 24]. By Assumption 3.1, there exists a value $\theta_{*,t} \in \Theta$ such that $\Upsilon_{\theta_{*,t}} = \mathcal{T}^\pi \Upsilon_{\hat{\theta}_{n,t-1}}$.

Our results are divided into the following two subsections. In Section 3.1, we focus exclusively on divergences that can be expressed as squared metrics and consider the tabular setting. In this case, we can obtain near-minimax optimal convergence rates for various FDEs (see Table 1). In Section 3.2, we broaden the analysis to cover a wider range of divergences in both tabular and non-tabular settings (see Table 2), establishing theoretical guarantees for many FDEs under more general conditions.

### 3.1 Squared metric divergence under tabular setting

Under tabular setting (i.e., $|\mathcal{S} \times \mathcal{A}| < \infty$), we shall assume that every state-action pair can be observed with non-zero probability, which is at least $p_{\min} := \min_{s,a} \rho(s,a) > 0$, where $\rho$ represents

the probability mass function of $(S, A)$ in the data generating distribution. In this subsection, we focus on a class of functional Bregman divergences that are squared metrics, i.e., $\mathfrak{d} = m^2$ for some induced metric $m$ associated with an appropriate inner product space $(\mathcal{G}, \langle \cdot, \cdot \rangle_m)$. More specifically, we require that each probability distribution $\mu \in \mathcal{P}$ admits a unique representation $G(\cdot|\mu) \in \mathcal{G}$, and that $m(\mu_1, \mu_2) = \|G(\cdot|\mu_1) - G(\cdot|\mu_2)\|_m$ holds for any $\mu_1, \mu_2 \in \mathcal{P}$. (It is possible that some element in $\mathcal{G}$ does not correspond to any element in $\mathcal{P}$.) The inner product structure supports a stronger theoretical analysis, leading to near-minimax optimal convergence rate established in Theorem 3.3. The precise requirement (including an additional technical condition) for the metric $m$ are formally stated in Definition C.2 of Appendix C.6. We will refer to such metrics as inner-product-space (IPS) metrics. Under mild conditions, squared IPS metrics $\mathfrak{d} = m^2$ are functional Bregman divergences (see Appendix C.9.1). Examples of functional Bregman divergence $\mathfrak{d}$ that can be represented as squared IPS metrics are listed in Table 1. See Appendix C.5 for the their technical constructions.

Regarding (P3), we assume that $\eta$ can be well-bounded by $m$ in the following assumption.

**Assumption 3.2.** The probability metric $\eta$ satisfies (S-L-C). For any $\mu_1, \mu_2 \in \mathcal{P}$, there exist some constants $C_{surr} > 0$, $\delta > 0$, $\epsilon_0 \geq 0$ such that

$$\eta(\mu_1, \mu_2) \leq C_{surr} \cdot m(\mu_1, \mu_2)^\delta + \epsilon_0.$$

**Theorem 3.3.** *Suppose Assumptions 3.1 and 3.2 hold. Moreover, given a probability metric $\eta$ that satisfies (S-L-C) with convexity parameter $q \geq 1$ and scale-sensitivity parameter $c > 0$ (i.e., (15) of Appendix C.4), consider the FDE with a functional Bregman divergence $\mathfrak{d} = m^2$ where $m$ is an IPS metric (Definition C.2 of Appendix C.6). Assume that there exists $C_{max,m} \in (0, \infty)$ such that*

$$C_{max,m} \geq \sup_{\theta \in \Theta} \sup_{s,a} \|G(\cdot|\Upsilon_\theta(s,a))\|_m \quad \& \quad C_{max,m} \geq \sup_{\theta \in \Theta} \sup_{r,s'} \{\|G(\cdot|\Psi_\pi(r, s', \Upsilon_\theta))\|_m\}. \quad (11)$$

*Then for any $\delta_0 \in (0,1)$, by letting $T = \lfloor \frac{\delta}{2} \cdot \frac{1}{c} \cdot \log_{1/\gamma}(\frac{N}{\log(4|\mathcal{S} \times \mathcal{A}|/\delta_0)}) \rfloor$, we have, with probability larger than $1 - \delta_0$,*

$$\eta_\infty(\Upsilon_{\theta_T}, \Upsilon_\pi) \leq \frac{C}{1 - \gamma^c} \cdot \left( \frac{\log_{1/\gamma} N \cdot \log(\log_{1/\gamma} N)}{N} \cdot \log\left(\frac{4|\mathcal{S} \times \mathcal{A}|}{\delta_0}\right) \cdot \frac{1}{p_{\min}^2} \right)^{\delta/2} + \frac{\epsilon_0}{1 - \gamma^c},$$

*for a constant $C > 0$ that does not depend on any of $\gamma, N, |\mathcal{S} \times \mathcal{A}|, \delta_0$.*

See Appendix C.6 for its proof and more detailed bound (21). Table 1 shows valid examples of $(\mathfrak{d} = m^2, \eta)$, along with the parameters presented in Theorem 3.3. The second last column determines the convergence rate ($N^{\delta/2}$, up to a logarithmic order), which depends on the moment degree $r \geq 1$ (defined in Table 1) for those that can cover unbounded distributions. We can see that $m = l_2, d_{L_2}, \text{MMD}_{cou}$ achieves $O_P(N^{-1/(2p)})$ (up to a logarithmic order) in $\mathbb{W}_{p,\infty}$-error for bounded distributions (i.e., $r = \infty$), where $\mathbb{W}_{p,\infty}$ is the supremum extension (7) of $\mathbb{W}_p$ metric. For $p = 1$, the convergence rate is (near-)optimal, aligning with the minimax optimal convergence rate for $\mathbb{W}_{1,\infty}$-error shown in Theorem B.1 of [45] (up to a logarithmic order). To our knowledge, there is no corresponding minimax result for distributional OPE problems when $p > 1$. However, $N^{-1/(2p)}$ is comparable to the optimal convergence rate of empirical probability measure (Theorem 1 of [18]).

Table 1: Comparison of different FDE methods under tabular setting. See Appendix A for definitions of the suggested examples of $\mathfrak{d} = m^2$ and $\eta_\infty$, and the corresponding probability space $\mathcal{P}$, such that $\mathcal{M}_\Theta \subseteq \mathcal{P}^{\mathcal{S} \times \mathcal{A}}$. $p \geq 1$ is an integer, and $r > p$ is such that $\sup_{\theta \in \Theta} \sup_{s,a} \mathbb{E}_{Z \sim \Upsilon_\theta(s,a)} \{\|Z\|^r\} < \infty$.

| $\mathfrak{d}(= m^2)$ | $\eta_\infty$ | $c$ | $\delta$ | $\epsilon_0$ |
|---|---|---|---|---|
| Cramér : $l_2^2$ | $\mathbb{W}_{p,\infty}$ | 1 | $1/p$ | 0 |
| MMD-Energy : $\text{MMD}_\beta^2$ | $\text{MMD}_{\beta,\infty}$ | $\beta/2$ | 1 | 0 |
| PDF-$L_2$ : $d_{L_2}^2$ | $\mathbb{W}_{p,\infty}$ | 1 | $\frac{2(r-p)}{(d+2r)p}$ | 0 |
| MMD-Matern : $\text{MMD}_\nu^2$ | $\mathbb{W}_{p,\infty}$ | 1 | $\frac{r-p}{(d+2r)p}$ | 0 |
| MMD-RBF : $\text{MMD}_{\sigma_{\text{RBF}}}^2$ | $\mathbb{W}_{p,\infty}$ | 1 | $\frac{2(r-p)}{(d+2r)p}$ | $2^{3/2} \cdot \frac{\Gamma((p+d)/2)}{\Gamma(p/2)}$ |
| MMD-RBF-transformed : $\text{MMD}_{\sigma,f}^2$ | $\mathbb{W}_{p,\infty}$ | 1 | $1/p$ | $2^{3/2} \cdot \frac{\Gamma((p+d)/2)}{\Gamma(p/2)}$ |
| MMD-Coulomb : $\text{MMD}_{cou}^2$ | $\mathbb{W}_{p,\infty}$ | 1 | $1/p$ | 0 |

## 3.2 General divergence under possibly non-tabular setting

In this subsection, we will discuss general state-action spaces that can be either tabular or non-tabular. As explained in Section 2.3, controlling $\eta_\infty$-error (like Theorem Theorem 3.3) by an expectation-based objective function is challenging and requires strong assumptions for non-tabular settings. We now shift our focus to the more compatible (expectation-based) $\bar{\eta}_{d_\pi,q}$-bound. In this subsection, we move beyond squared IPS metrics discussed in Section 3.1 and allow $\mathfrak{d}$ to be more general functional Bregman divergence. Some standard conditions are needed to establish the convergence. Specifically, we assume that the data distribution ($\rho$) sufficiently covers the target distribution ($d_\pi$) (see Assumption C.4), and that the model class satisfies an entropy condition characterized by a complexity coefficient $\alpha \in (0,2)$ (see (23) of Assumption C.3). The larger the $\alpha$, the more complex the model class.

Table 2 summarizes the theoretical guarantee for a broad class of $\mathfrak{d}$ based on our theorems. All error bounds are developed using empirical process theory for M-estimation (see Theorem D.1). However, different classes of $\mathfrak{d}$ require separate ways to verify the required modulus condition (Condition 2 of Theorem D.1), resulting in different convergence rates. To apply the empirical process theory, we introduce a surrogate metric for the construction of entropy conditions (e.g., see $m$ in Appendices C.8 and C.15). Informally, this surrogate metric is required to be able to control the differences in the objective function. However, unlike Theorem 3.3, we do not require the divergence to be explicitly constructed from the metric, hence accommodating a broader class of divergences. Based on this, we present two general theorems. Theorem C.5 assumes that the surrogate metric is induced by a normed space, while Theorem C.11 applies to general surrogate metrics. Although Theorem C.5 supports a more limited set of divergences, it provides stronger convergence rate guarantees compared to Theorem C.11. In terms of the scope of applicable divergence choices, we have the following inclusion relationships:

$$\text{Theorem 3.3 (tabular)} \subseteq \text{Theorem C.5 (general)} \subseteq \text{Theorem C.11 (general)}.$$

However, the strength of the convergence rate guarantees follows the reverse order. Note that Theorem 3.3 only applies to the tabular case. We remark that some important divergences (e.g., KL divergence) are not covered in the aforementioned theorems, requiring separate analysis (see Corollary B.9).

Due to space limitation, we only present a simplified version of Theorem C.5 for reference. The error bound in Theorem C.11 shares a similar form.

**Theorem 3.4** (Simplified version of Theorem C.5). *Let $\eta$ be a probability metric that satisfies (S-L-C) with $q \geq 1$ and $c > 1/(2q)$. Suppose Assumptions 3.1, 3.2, C.3 and C.4 hold. By letting $T = \lfloor \frac{1}{c-\frac{1}{2q}} \cdot \frac{\delta'}{2(l-1)+\alpha} \cdot \log_{1/\gamma} N \rfloor$ with $\delta' = \min\{\delta, 1/q\}$, the corresponding FDE achieves*

$$\bar{\eta}_{d_\pi,q}(\Upsilon_{\theta_T}, \Upsilon_\pi) \leq \frac{1}{1-\gamma^{c-\frac{1}{2q}}} \cdot O_P\left\{ \left(\frac{N}{\log_{1/\gamma} N}\right)^{\frac{-\delta'}{2(l-1)+\alpha}} \right\} + \frac{1}{1-\gamma^{c-\frac{1}{2q}}} \cdot 2^{1-\frac{1}{2q}} \cdot \epsilon_0.$$

We now turn to Table 2, which provides several examples of divergences $\mathfrak{d}$. (See Appendix A for definitions and Corollaries B.1–B.9 in Appendix B for detailed bounds.) The table summarizes the pairs $(\mathfrak{d}, \bar{\eta}_{d_\pi,q})$ in Columns 1–2, along with their ability to handle unbounded return distributions, distributions without densities, and their associated return dimensionality considerations (Columns 3–5). If convergence cannot be guaranteed due to a nonzero $\epsilon_0$, the corresponding convergence rate is omitted. To the best of our knowledge, [63] is the only existing work that provides a statistical convergence analysis for distributional OPE in non-tabular settings. In comparison, our FDE method using $\mathfrak{d} = l_2^2$ (the first row of Table 2) consistently achieves faster convergence rates, as compared to the rate of their FLE method obtained in [63], across all degrees of Wasserstein metric $p \geq 1$ and model complexities $\alpha \in (0,2)$. See (12) of Appendix B.1 for details.

## 4 Experiments

We evaluate our FDE methods with baselines through two experiments[3]: linear quadratic regulator (LQR) and Atari games. LQR is a parametric setting widely studied in RL literature [e.g., 9, 63, 37].

---

[3]Codes are available at `https://github.com/hse1223/Fitted-Distributional-Evaluation.git`

Table 2: Comparison of different FDE methods under general state-action space. See Appendix A for definitions of examples of $\mathfrak{d}$ and $\bar{\eta}_{d_\pi,q}$. $p \geq 1$ is an integer, and $r > p$ satisfies $\sup_{\theta\in\Theta}\sup_{s,a}\mathbb{E}_{Z\sim\Upsilon_\theta(s,a)}\{\|Z\|^r\} < \infty$. Convergence rates are displayed up to a logarithmic order.

| $\mathfrak{d}$ | $\bar{\eta}_{d_\pi,q}$ | unbounded | density-free | $d$ | rate | reference |
|---|---|---|---|---|---|---|
| $l_2^2$ | $\overline{\mathbb{W}}_{p,d_\pi,p}$ | ✗ | ✓ | 1 | $N^{\frac{-1}{p}\cdot\frac{1}{2+\alpha}}$ | Theorem 3.4 |
| $\mathrm{MMD}^2_{\beta=1}$ | $\overline{\mathbb{W}}_{1,d_\pi,1}$ | ✓ | ✓ | $\geq 1$ | $N^{\frac{-1}{\frac{4(d+1)r}{r-1}+\alpha}}$ | Theorem 3.4 |
| $\mathrm{MMD}^2_{\beta>1}$ | $\overline{\mathrm{MMD}}_{\beta,d_\pi,1}$ | ✗ | ✓ | $\geq 1$ | $N^{\frac{-1}{2+\alpha}}$ | Theorem C.11 |
| $d_{L_2}^2$ | $\overline{\mathbb{W}}_{p,d_\pi,p}$ | ✓ | ✗ | $\geq 1$ | $N^{\frac{-2(r-p)}{(d+2r)p}\cdot\frac{1}{2+\alpha}}$ | Theorem 3.4 |
| $\mathrm{MMD}^2_{\nu}$ | $\overline{\mathbb{W}}_{p,d_\pi,p}$ | ✓ | ✗ | $\geq 1$ | $N^{\frac{-(r-p)}{(d+2r)p}\cdot\frac{1}{2+\alpha}}$ | Theorem 3.4 |
| $\mathrm{MMD}^2_{\sigma_{RBF}}$ | $\overline{\mathbb{W}}_{p,d_\pi,p}$ | ✓ | ✓ | $\geq 1$ | – | Theorem 3.4 |
| $\mathrm{MMD}^2_{\sigma,f}$ | $\overline{\mathbb{W}}_{p,d_\pi,p}$ | ✗ | ✓ | $\geq 1$ | – | Theorem 3.4 |
| $\mathrm{MMD}^2_{cou}$ | $\overline{\mathbb{W}}_{p,d_\pi,p}$ | ✗ | ✗ | $\neq 1$ | $N^{\frac{-1}{p}\cdot\frac{(6-\alpha)}{16}}$ | Theorem C.11 |
| KL | $\overline{\mathbb{W}}_{p,d_\pi,p}$ | ✗ | ✗ | $\geq 1$ | $N^{\frac{-1}{p}\cdot\frac{1}{2+\alpha}}$ | Corollary B.9 |

For the LQR experiment, we have compared FDE methods (PDFL2, RBF, Matern, Energy, KL). All of these FDE methods outperformed the baseline FLE [63]. Due to space limitations, we defer the details to Appendix E.1.

Atari games are common testbeds for distributional RL methods [e.g., 5, 15, 40, 52, 65]. We compared the proposed FDE methods with FLE [63], QRDQN [15] and IQN [14]. Among these methods, QRDQN and IQN rely on quantile-based modeling, and we adopted Gaussian mixture modeling (GMM, see details in Appendix E.2.1) to model distributions for all other methods. In our experiments, we tested different numbers of mixtures / quantiles ($M = 10, 100, 200$) and considered multiple Atari games: Atlantis, Breakout, Enduro, KunfuMaster, Pong, Qbert, SpaceInvader. We estimated $\Upsilon_\pi \in \mathcal{P}^{\mathcal{S}\times\mathcal{A}}$ in two different environments (deterministic / random reward), each having different target policies. For each environment, we collected offline data from two different behavior policies (representing strong and weak coverage), with three different sample sizes ($N = 2K, 5K, 10K$). See Appendices E.2.2 and E.2.4 for algorithmic details.

In our simulations, we have included the following methods: (i) baseline methods (FLE, QRDQN, IQN), (ii) our FDE methods (KL, Energy, PDF-L2, RBF), (iii) non-functional Bregman divergence (TVD) that we did not study. Figure 1, which contains these methods in order from left to right for four different games, shows the inaccuracy comparison with $N = 10K$ and $M = 200$ under a specific environment and behavior policy. TVD shows no improvement of accuracy through iterations, corroborating our proposal to build the objective functions based on functional Bregman divergence in Section 2.4. On the contrary, our FDE methods (particularly, KL and Energy) not only decrease the inaccuracy, but also outperform the baseline methods in most games with respect to mean inaccuracy and variance.

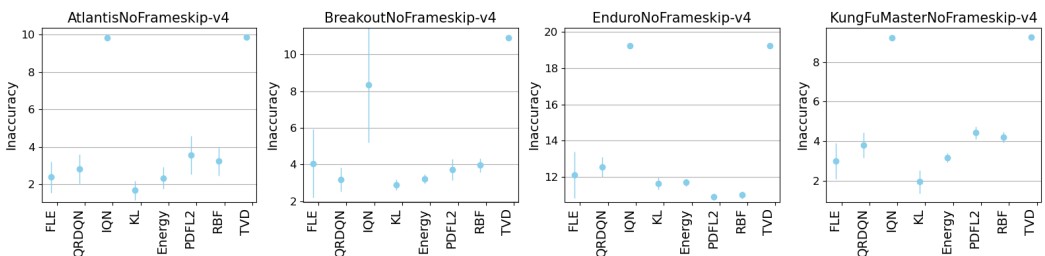

Figure 1: Mean (dots) and confidence region (mean $\pm$ STD) for 5 seeds based on $N = 10K$ samples: $\mathbb{W}_1$-inaccuracy (Y-axis) for each method (X-axis) for the games with $M = 200$. See Figures 3–5 for simulation results in all seven games.

Besides the four settings of Figure 1, we have more results for a wide range of different settings (i.e., seven games, two environments, two behavior policies, three mixture sizes and three sample sizes). Overall, FDE methods outperformed the baseline methods. More details can be found in Appendix E.2. Moveover, we also demonstrated that another functional Bregman divergence—the Hyvärinen divergence, which was not part of our main study—performs well in practice, supporting our discussion in Section 2.4. See Figures 3–5 and Tables 6–33 of Appendix E.2.5.

## 5   Discussion

A central goal of this work is to explore how to choose $\mathfrak{d}$ for constructing a theoretically sound FDE method. At a high level, different $\mathfrak{d}$ respond differently to deviation between $\Upsilon$ and $\Psi_\pi$ in (5), much like how various loss functions behave differently in empirical risk minimization. Therefore, the choice of divergence should be tailored to the specific problem at hand. To move toward a principled framework for choosing $\mathfrak{d}$ in specific applications, we believe the first step is to identify what constitutes a valid candidate. To date, this question remains largely unaddressed in the literature, and only a few valid choices have been identified - let alone systematically compared. Our work makes substantial progress on this front. While comparative analyses of certain functional Bregman divergences do exist and may provide practical guidance, we acknowledge that this work does not yet offer a comprehensive guide for selecting among them in practice.

Besides, we acknowledge several other technical limitations of our study. First, while our framework establishes the sufficiency of using functional Bregman divergence to ensure the property (9) (Theorem 2.4), the necessity remains an open question. Second, for simplicity, our theoretical analysis assumes data splitting, leading to independent samples at each iteration. It would be valuable to extend the analysis to scenarios where data are reused across iterations. Finally, our theorems rely on the completeness assumption (Assumption 3.1), which may be overly restrictive in practical settings. We could introduce a term that quantifies violations of this assumption via the inherent Bellman error [38] and incorporate it into our final bound. We leave such an extension for future work.

## Acknowledgments and Disclosure of Funding

Portions of this research were conducted with the advanced computing resources provided by Texas A&M High Performance Research Computing. The work of Jiayi Wang is partly supported by the National Science Foundation (DMS-2401272) and Texas Artificial Intelligence Research Institute (TAIRI).

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

# Table of contents in Appendix

# A   Definitions of metrics and divergences

Following is the generalized algorithm of FDE methods.

---

**Algorithm 1** Fitted distributional evaluation

---
    **Input:** Functional Bregman divergence $\mathfrak{d}$ (Tables 1–2), Model $\mathcal{M}$, Initial $\Upsilon_0 \in \mathcal{M}$, Data $\mathcal{D}$
    **Output:** $\Upsilon_T$
    **for** $t = 1$ **to** $T$ **do**
        Perform the minimization (5) to obtain $\Upsilon_t$.
    **end for**

---

For Tables 1 and 2 of Section 3, we shall define the necessary concepts needed to understand the objective functions and contraction-inducing metrics.

## A.1   Contraction-inducing metrics

Following are the individual metrics that satisfy (S-L-C) and their supremum (7) and expectation-extensions (8) (which are contraction-inducing metrics (Definition 2.2)) that we used in Tables 1 and 2.

Wasserstein-p metric is defined as follows, with $J(\mu_1, \mu_2)$ being the possible joint distributions of $\mathbf{X} \sim \mu_1$ and $\mathbf{Y} \sim \mu_2$:

$$\mathbb{W}_p(\mu_1, \mu_2) := \inf_{J(\mu_1,\mu_2)} \left( \mathbb{E}\|\mathbf{X} - \mathbf{Y}\|^p \right)^{1/p} \quad (p \geq 1).$$

Its supremum and expectation extensions are defined as follows, with their contractive factors ($\zeta$ of Definition 2.2)

$$\mathbb{W}_{p,\infty}(\Upsilon_1, \Upsilon_2) := \sup_{s,a} \mathbb{W}_p(\Upsilon_1(s,a), \Upsilon_2(s,a)) \quad \text{with } \zeta = \gamma,$$

$$\overline{\mathbb{W}}_{p,d_\pi,p}(\Upsilon_1, \Upsilon_2) := \left[ \mathbb{E}\left\{ \mathbb{W}_p^{2p}(\Upsilon_1(s,a), \Upsilon_2(s,a)) \right\} \right]^{\frac{1}{2p}} \quad \text{with } \zeta = \gamma^{1-\frac{1}{2p}}.$$

MMD-Energy is defined as the square-rooted form of following MMD-squared (with $X, X' \sim \mu_1$ and $Y, Y' \sim \mu_2$ all being indepenent)

$$\mathrm{MMD}_\beta^2(\mu_1, \mu_2) := \mathbb{E}\{k_\beta(X, X')\} + \mathbb{E}\{k_\beta(Y, Y')\} - 2 \cdot \mathbb{E}\{k_\beta(X, Y)\},$$
$$\text{with} \quad k_\beta(x, y) := \|x\|^\beta + \|y\|^\beta - \|x - y\|^\beta \quad (0 < \beta < 2).$$

Its supremum and expectation extensions are defined as follows, with

$$\mathrm{MMD}_{\beta,\infty}(\Upsilon_1, \Upsilon_2) := \sup_{s,a} \mathrm{MMD}_\beta(\Upsilon_1(s,a), \Upsilon_2(s,a)) \quad (0 < \beta < 2) \quad \text{with } \zeta = \gamma^{\frac{\beta}{2}},$$

$$\overline{\mathrm{MMD}}_{\beta,d_\pi,1}(\Upsilon_1, \Upsilon_2) := \left[ \mathbb{E}_{s,a \sim d_\pi}\left\{ \mathrm{MMD}_\beta^2(\Upsilon_1(s,a), \Upsilon_2(s.a)) \right\} \right]^{\frac{1}{2}} (1 < \beta < 2) \text{ with } \zeta = \gamma^{\frac{\beta-1}{2}}.$$

## A.2   Divergences

Functional Bregman divergences (choices suggested in Tables 1 and 2) are what determine the objective functions and their corresponding choices of $\mathcal{P}$, whose elements can be used as inputs of $\mathfrak{d}$. Our model should satisfy $\mathcal{M}_\Theta \subseteq \mathcal{P}^{\mathcal{S} \times \mathcal{A}}$.

For Cramér FDE, we use $\mathfrak{d} = l_2^2$ and corresponding $\mathcal{P}$:

$$l_2^2(\mu_1, \mu_2) := \int_{b_1}^{b_2} \left| F(z|\mu_1) - F(z|\mu_2) \right|^2 \mathrm{d}z \quad \text{with } F(\cdot|\mu) \text{ being cdf of } \mu \in \mathcal{P},$$

$$\mathcal{P} = \Delta([b_1, b_2]) := \{\text{probability measures of distributions bounded in } [b_1, b_2] \subsetneq \mathbb{R}\}.$$

For Energy FDE, we use $\mathfrak{d} = \mathrm{MMD}_\beta^2$ and corresponding $\mathcal{P}$ ($X, X' \sim \mu_1$ and $Y, Y' \sim \mu_2$ being mutually independent):

$$\mathrm{MMD}_\beta^2(\mu_1, \mu_2) := \mathbb{E}\{k_\beta(X, X')\} + \mathbb{E}\{k_\beta(Y, Y')\} - 2 \cdot \mathbb{E}\{k_\beta(X, Y)\},$$

$$\text{with} \quad k_\beta(x, y) := \|x\|^\beta + \|y\|^\beta - \|x - y\|^\beta \quad (0 < \beta < 2),$$

$$\mathcal{P} = \{\mu \in \Delta(\mathbb{R}^d) \ (d \geq 1) : \|\mathbb{E}_{X \sim \mu}\{k_\beta(X, \cdot)\}\|_{\mathcal{H}} < \infty\} \quad \text{with } \mathcal{H}_\beta \text{ being the RKHS of kernel } k_\beta.$$

For PDF-L2 FDE, we use $\mathfrak{d} = d_{L_2}^2$ and corresponding $\mathcal{P}$:

$$d_{L_2}^2(\mu_1, \mu_2) := \int_{\mathbb{R}^d} \left| f(z|\mu_1) - f(z|\mu_2) \right|^2 \mathrm{d}z \quad \text{with } f(\cdot|\mu) \text{ being pdf of } \mu \in \mathcal{P},$$

$$\mathcal{P} = \left\{ \mu \in \Delta(\mathbb{R}^d) \ (d \geq 1) : \int_{\mathbb{R}^d} f^2(z|\mu) \mathrm{d}z < \infty \right\}.$$

For MMD-Matern FDE, we use $\mathfrak{d} = \mathrm{MMD}_\nu^2$ and corresponding $\mathcal{P}$ ($X, X' \sim \mu_1$ and $Y, Y' \sim \mu_2$ being mutually independent):

$$\mathrm{MMD}_\nu^2(\mu_1, \mu_2) := \mathbb{E}\{k_\nu(X, X')\} + \mathbb{E}\{k_\nu(Y, Y')\} - 2 \cdot \mathbb{E}\{k_\nu(X, Y)\},$$

$$\text{with} \quad k_\nu(\mathbf{x}, \mathbf{y}) := \frac{2^{1-\nu}}{\Gamma(\nu)} \cdot \left( \frac{\sqrt{2\nu}\|\mathbf{x} - \mathbf{y}\|}{\sigma_{\mathrm{Mat}}} \right)^\nu \cdot K_\nu\left( \frac{\sqrt{2\nu} \cdot \|\mathbf{x} - \mathbf{y}\|}{\sigma_{\mathrm{Mat}}} \right) \text{ for some } \sigma_{\mathrm{Mat}} > 0, \nu > 0,$$

$$\mathcal{P} = \{\mu \in \Delta(\mathbb{R}^d) \ (d \geq 1) : \|\mathbb{E}_{X \sim \mu}\{k_\nu(X, \cdot)\}\|_{\mathcal{H}} < \infty\}$$

$$\text{with } \mathcal{H}_\nu \text{ being the RKHS of kernel } k_\nu.$$

For MMD-RBF FDE, we use $\mathfrak{d} = \mathrm{MMD}_{\sigma_{\mathrm{RBF}}}^2$ and corresponding $\mathcal{P}$ ($X, X' \sim \mu_1$ and $Y, Y' \sim \mu_2$ being mutually independent):

$$\mathrm{MMD}_{\sigma_{\mathrm{RBF}}}^2(\mu_1, \mu_2) := \mathbb{E}\{k_{\sigma_{\mathrm{RBF}}}(X, X')\} + \mathbb{E}\{k_{\sigma_{\mathrm{RBF}}}(Y, Y')\} - 2 \cdot \mathbb{E}\{k_{\sigma_{\mathrm{RBF}}}(X, Y)\},$$

$$\text{with} \quad k_{\sigma_{\mathrm{RBF}}}(\mathbf{x}, \mathbf{y}) = \frac{1}{(2\sqrt{\pi})^d} \cdot \sigma_{\mathrm{RBF}}^{-d} \cdot \exp\left( \frac{-1}{4\sigma_{\mathrm{RBF}}^2} \|\mathbf{x} - \mathbf{y}\|^2 \right),$$

$$\mathcal{P} = \{\mu \in \Delta(\mathbb{R}^d) \ (d \geq 1) : \|\mathbb{E}_{X \sim \mu}\{k_{\sigma_{\mathrm{RBF}}}(X, \cdot)\}\|_{\mathcal{H}} < \infty\}$$

$$\text{with } \mathcal{H}_{\sigma_{\mathrm{RBF}}} \text{ being the RKHS of } k_{\sigma_{\mathrm{RBF}}}.$$

For MMD-RBF-transformed FDE, we use $\mathfrak{d} = \mathrm{MMD}_{(\sigma, f)}^2$ and corresponding $\mathcal{P}$ (with conditions of function $f$ and definition of $I_f$ are mentioned in Equations 8 and 10 of [70]) ($X, X' \sim \mu_1$ and $Y, Y' \sim \mu_2$ being mutually independent):

$$\mathrm{MMD}_{(\sigma, f)}^2(\mu_1, \mu_2) := \mathbb{E}\{k_{(\sigma, f)}(X, X')\} + \mathbb{E}\{k_{(\sigma, f)}(Y, Y')\} - 2 \cdot \mathbb{E}\{k_{(\sigma, f)}(X, Y)\},$$

$$\text{with} \quad k_{(\sigma, f)}(\mathbf{x}, \mathbf{y}) := \exp\left( -\frac{\|\mathbf{x} - \mathbf{y}\|^2}{4\sigma^2} \right) \cdot I_f\left( \frac{\|\mathbf{x} + \mathbf{y}\|}{\sqrt{2}\sigma} \right),$$

$$\mathcal{P} = \{\mu \in \Delta(\mathbb{R}^d) \ (d \geq 1) : \|\mathbb{E}_{X \sim \mu}\{k_{(\sigma, f)}(X, \cdot)\}\|_{\mathcal{H}} < \infty\}$$

$$\text{with } \mathcal{H}_{(\sigma, f)} \text{ being the RKHS of } k_{(\sigma, f)}.$$

For MMD-Coulomb FDE, we use $\mathfrak{d} = \mathrm{MMD}_{cou}^2$ and corresponding $\mathcal{P}$ ($X, X' \sim \mu_1$ and $Y, Y' \sim \mu_2$ being mutually independent):

$$\mathrm{MMD}_{cou}^2(\mu_1, \mu_2) := \mathbb{E}\{k_{cou}(X, X')\} + \mathbb{E}\{k_{cou}(Y, Y')\} - 2 \cdot \mathbb{E}\{k_{cou}(X, Y)\},$$

$$\text{with} \quad k_{cou}(\mathbf{x}, \mathbf{y}) = \kappa_0^{(cou)}(\mathbf{x} - \mathbf{y}) := \begin{cases} -\log\|\mathbf{x} - \mathbf{y}\| & \text{if} \quad d = 2 \\ \|\mathbf{x} - \mathbf{y}\|^{2-d} & \text{if} \quad d \geq 3 \end{cases}$$

$$\mathcal{P} = \{\mu \in \Delta(\mathbb{R}^d) \ (d \geq 2) : \|\mathbb{E}_{X \sim \mu}\{k_{cou}(X, \cdot)\}\|_{\mathcal{H}} < \infty\} \text{ with } \mathcal{H}_{cou} \text{ being the RKHS of } k_{cou}.$$

For KL FDE, we use $\mathfrak{d}(\mu_1, \mu_2) = \mathrm{KL}(\mu_2, \mu_1)$ and corresponding $\mathcal{P}$:

$$\mathrm{KL}(\mu_2\|\mu_1) := \int_{\mathbb{R}^d} f(z|\mu_2) \cdot \log \frac{f(z|\mu_2)}{f(z|\mu_1)} \mathrm{d}z, \text{with } f(\cdot|\mu) \text{ being pdf of } \mu \in \mathcal{P}$$

$$\mathcal{P} = \{\mu \in \Delta(\mathcal{X}) \text{ with } \mathcal{X} \subseteq \mathbb{R}^d \ (d \geq 1) : \mu \text{ has a density } f(\cdot|\mu) > 0 \text{ on } \mathcal{X}.\}.$$

# B FDE methods for general state-action space

## B.1 Cramér FDE

We construct our objective function (10) based on Cramér distance $\mathfrak{d} = l_2^2$, and refer to this as Cramér FDE. Assuming bounded variables in $\mathbb{R}$, i.e., $Z(s, a; \theta) \in [a, b]$ for all $s, a \in \mathcal{S} \times \mathcal{A}$ and $\theta \in \Theta$, we obtain the following based on Theorem C.5, with proof and detailed bound in Appendix C.10.

**Corollary B.1.** *Under Assumptions 3.1, C.4, our estimator* (10) *based on* $\mathfrak{d} = l_2^2$ *achieves the following in bounded support in $\mathbb{R}$, say $[a, b]$.*

$$\overline{\mathbb{W}}_{p, d_\pi, p}(\Upsilon_{\theta_T}, \Upsilon_\pi) \lesssim \frac{1}{1 - \gamma^{1 - \frac{1}{2p}}} \cdot O_P\left\{\left(\frac{N}{\log_{1/\gamma} N}\right)^{\frac{-1/p}{2+\alpha}}\right\}.$$

*Here, $\lesssim$ means upper-bounded by RHS multiplied by some constant that does not depend on $\gamma$ and $N$.*

Although Cramér FDE is limited in bounded uni-dimensional distributions, we can provide direct comparison with FLE under exactly the same conditions, at least for $d = 1$. Assuming bounded return distribution with the same model complexity $\log \mathcal{N}_{[\,]}(\widetilde{\mathcal{F}}_\Theta, \|\cdot\|_{L_{1,\infty}}, \epsilon) \lesssim \epsilon^{-\alpha}$ (where $\widetilde{\mathcal{F}}_\Theta$ represents the conditional pdf family modeled by $\Theta$), Cramér FDE achieves faster convergence rate than FLE uniformly for all $p \geq 1$ and $\alpha \in (0, 2)$. See Appendix C.17.1 for its proof.

$$\text{FLE} : \overline{\mathbb{W}}_{p, d_\pi, p}(\Upsilon_{\theta_T}, \Upsilon_\pi) \lesssim \tilde{O}_P\left(N^{\frac{-1}{2p}(1-\alpha)}\right) \quad \text{VS} \quad \text{Cramér} : \overline{\mathbb{W}}_{p, d_\pi, p}(\Upsilon_{\theta_T}, \Upsilon_\pi) \lesssim \tilde{O}_P\left(N^{\frac{-1}{p(2+\alpha)}}\right).$$

(12)

Here, $\tilde{O}_P$ indicating the convergence rate, allowing up to logarithmic difference from the conventional $O_P$. We have suggested two examples for Cramér FDE, in which FLE cannot provide a valid bound due to no conditional densities. The first example is linear MDP (Appendix C.17.2) that is frequently assumed in both traditional and distributional RL [e.g., 62, 46, 61]. Cramér FDE achieves $\overline{\mathbb{W}}_{p, d_\pi, p}$-convergence rate of $\tilde{O}_P(N^{-1/(3p)})$. The second example is Linear Quadratic Regulator (Appendix C.17.3), which has been also been frequently mentioned in traditional RL [e.g., 9, 37], and was recently applied in distributional RL under finite-horizontal setting [63]. We extended this towards the infinite-horizontal setting. For bounded distributional families, Cramér FDE achieves $\overline{\mathbb{W}}_{p, d_\pi, p}$-convergence rate of $\tilde{O}_P(N^{-1/(2p)})$. For unbounded distributional families, its generalized extension Energy FDE (introduced in the following subsection) achieves $\overline{\mathbb{W}}_{1, d_\pi, 1}$-convergence rate of $\tilde{O}_P(N^{-1/8})$.

## B.2 Energy FDE

Now, we will assume multi-dimensional return ($d \geq 1$), but still bounded. Here, we use $\mathfrak{d} = \text{MMD}_\beta^2$ with $\beta \in (1, 2)$, which we name as Energy FDE. $\text{MMD}_\beta$ is the MMD (17) with the following

$$k_\beta(x, y) := \|x\|^\beta + \|y\|^\beta - \|x - y\|^\beta \quad (1 < \beta < 2).$$

Unlike the tabular setting (supported by Theorem 3.3) where we could apply $\beta \in (0, 1)$, we have to limit to $\beta \in (0, 1)$. But we can extend the result into $\beta = 1$ (second statement of Corollary B.2). Considering that $l_2^2 = \frac{1}{2}\text{MMD}_{\beta=1}^2$ for $d = 1$ [7], Energy FDE can be viewed as an extension of Cramér FDE. Proofs and detailed bounds are in Appendix C.11.

**Corollary B.2.** *Under Assumptions 3.1, C.4, our estimator* (10) *based on* $\mathfrak{d} = \text{MMD}_\beta^2$ *with $\beta \geq 1$ can achieve the following bound in bounded support of $\mathbb{R}^d$,*

$$\overline{\text{MMD}}_{\beta, d_\pi, 1}(\Upsilon_{\theta_T}, \Upsilon_\pi) \lesssim \frac{1}{1 - \gamma^{\frac{\beta-1}{2}}} \cdot O_P\left\{\left(\frac{N}{\log_{1/\gamma} N}\right)^{\frac{-1}{2+\alpha}}\right\} \quad (1 < \beta < 2),$$

$$\overline{\text{MMD}}_{1, d_\pi, 1}(\Upsilon_{\theta_T}, \Upsilon_\pi) \lesssim \left(\frac{1}{\log(1/\gamma)}\right)^{1 + \frac{1}{2+\alpha}} \cdot O_P\left\{(\log N)^{1 + \frac{2}{2+\alpha}} \cdot N^{\frac{-1}{2+\alpha}}\right\} \quad (\beta = 1).$$

This result can further be used to bound $\overline{\mathbb{W}}_{1,d_\pi,1}$-inaccuracy based on the relationship between $\mathrm{MMD}_\beta$ and $\mathbb{W}_1$ based on (36), which is introduced by [36]. Based on $\mathbb{W}_p(\mu,\nu) \leq D^{1-\frac{1}{p}} \cdot \mathbb{W}_1^{1/p}(\mu,\nu)$ (suggested by [44]) where $D \in (0,\infty)$ represents the diameter of support of bounded distributions, we can further bound $\overline{\mathbb{W}}_{p,d_\pi,p}(\Upsilon_{\theta_T}, \Upsilon_\pi) \leq D^{2(1-\frac{1}{p})} \cdot \overline{\mathbb{W}}_{1,d_\pi,1}^{1/p}(\Upsilon_{\theta_T}, \Upsilon_\pi)$. However, these are still restricted in bounded distributions.

Therefore, let us introduce the case where Energy FDE can bound the inaccuracy for unbounded distributions even without $s,a$-conditional densities, based on $\overline{\mathbb{W}}_{1,d_\pi,1}$-inaccuracy. Towards that end, we assume that the $r$-th moment ($r > 1$) of the distributions are uniformly bounded, i.e., $M_r := \sup_{\theta \in \Theta}(\mathbb{E}\|Z(s,a;\theta)\|^r)^{1/r} < \infty$. See Appendix C.15.3 for proof, which is based on the alternative Theorem C.11.

**Corollary B.3.** *Under Assumptions 3.1, C.4, our estimator* (10) *based on* $\mathfrak{d} = \mathrm{MMD}_\beta^2$ *($\beta = 1$), we have following. Assuming that* $\sup_{\theta \in \Theta} \sup_{s,a} \mathbb{E}\|Z(s,a;\theta)\| < \infty$ *and* $\sup_{s,a}\|R(s,a)\|_{\psi_2} < \infty$ *where* $R(s,a)$ *indicates the reward vector conditioned on* $s,a$*, we have*

$$\overline{\mathbb{W}}_{1,d_\pi,1}(\Upsilon_{\theta_T}, \Upsilon_\pi) \lesssim \frac{1}{1-\gamma^{1/2}} \cdot O_P\left\{ \left( \frac{1}{\log(1/\gamma)} \cdot \frac{(\log N)^2}{N} \right)^{\frac{1}{2(l-1)+\alpha}} \right\} \ with \ l = \frac{r(2d+3)-1}{r-1}.$$

Here, having finite values of higher moments (i.e., larger $r > 1$) leads to tighter convergence rate. Bounded variables ($r = \infty$) will give us $l = 2d+3$, leading to $\overline{\mathbb{W}}_{1,d_\pi,1}(\Upsilon_{\theta_T}, \Upsilon_\pi) = \tilde{O}_P(N^{\frac{-1}{4(d+1)+\alpha}})$. In case of $d = 1$ and $r = \infty$, it does not degenerate into that of Corollary B.1, since they are based upon different proof structures.

## B.3 PDF-L2 FDE

Now let us construct the objective function (10) with $\mathfrak{d} = d_{L_2}^2$, namely PDF-L2 method. By assuming existence of conditional densities, PDF-L2 method can achieve faster convergence rate for unbounded distributions than Energy FDE (Corollary B.3). Proof and detailed bound are in Appendix C.12.

**Corollary B.4.** *Under Assumptions 3.1, C.4, our estimator* (10) *based on* $\mathfrak{d} = d_{L_2}^2$ *can achieve the following bound in* $\mathbb{R}^d$*. Assuming* $M_r := \sup_{\theta \in \Theta} \sup_{s,a}(\mathbb{E}\|Z(s,a;\theta)\|^r)^{1/r} < \infty$ *with* $r > p$ *and* $\sup_{\theta \in \Theta} \sup_{z,s,a} |f_\theta(z|s,a)| < \infty$.

$$\overline{\mathbb{W}}_{p,d_\pi,p}(\Upsilon_{\theta_T}, \Upsilon_\pi) \lesssim \frac{1}{1-\gamma^{1-\frac{1}{2p}}} \cdot O_P\left\{ \left( \frac{N}{\log_{1/\gamma} N} \right)^{\frac{-1}{2+\alpha} \cdot \frac{2(r-p)}{(d+2r)p}} \right\}$$

We can see that PDF-L2 method not only provides $\overline{\mathbb{W}}_{p,d_\pi,p}$-bound for all $p \geq 1$, but also provides faster convergence rate than Energy FDE (Corollary B.3) in the case of $p = 1$, for all $r \geq 1$, $d \geq 1$, and $\alpha \in (0,2)$.

## B.4 MMD-Matern FDE

[56] has shown that MMD based on translation invariant kernels, i.e., $k(\mathbf{x}, \mathbf{y}) = \kappa_0(\mathbf{x}-\mathbf{y})$, can bound Wasserstein metrics when the distributions have smooth enough densities (see their Theorem 15). One such example is MMD-Matern, which is the MMD (17) with following kernel with parameters $\nu > 0$ and $\sigma_{\mathrm{Mat}} > 0$ (assumed to be fixed) and $K_\nu$ being modified Bessel function,

$$k_\nu(\mathbf{x}, \mathbf{y}) := \frac{2^{1-\nu}}{\Gamma(\nu)} \cdot \left( \frac{\sqrt{2\nu}\|\mathbf{x}-\mathbf{y}\|}{\sigma_{\mathrm{Mat}}} \right)^\nu \cdot K_\nu\left( \frac{\sqrt{2\nu} \cdot \|\mathbf{x}-\mathbf{y}\|}{\sigma_{\mathrm{Mat}}} \right).$$

We will treat $\sigma_{\mathrm{Mat}} > 0$ as a fixed constant since we are more interested in $\nu$. It is well-known that $\nu = 1$ corresponds to Laplace kernel and $\nu \to \infty$ corresponds to RBF kernel. It leads to following result (proof and detailed bound in Appendix C.13).

**Corollary B.5.** *Under Assumptions 3.1, C.4, our estimator* (10) *based on* $\mathfrak{d} = \mathrm{MMD}_\nu^2$ *can achieve the following bound in* $\mathbb{R}^d$*. Assuming* $M_r := \sup_{\theta \in \Theta} \sup_{s,a}(\mathbb{E}\|Z(s,a;\theta)\|^r)^{1/r} < \infty$ *with* $r > p$ *and its conditional densities having Sobolev norms* $\|f_\theta(\cdot|s,a)\|_{H^{\nu+d/2}(\mathbb{R}^d)} \leq B(< \infty)$*, we have*

$$\overline{\mathbb{W}}_{p,d_\pi,p}(\Upsilon_{\theta_T}, \Upsilon_\pi) \lesssim \frac{1}{1-\gamma^{1-\frac{1}{2p}}} \cdot O_P\left\{ \left( \frac{N}{\log_{1/\gamma} N} \right)^{\frac{-1}{2+\alpha} \cdot \frac{r-p}{(d+2r)p}} \right\}.$$

Although this can be applied for unbounded distributions as PDF-L2 method (Corollary B.4), its bound is always looser than that of PDF-L2 method, as shown in its convergence rate which is half times that of PDF-L2 method. This is since the proof of bounding Wasserstein metric by MMD-Matern is based on the following logic.

$$\mathbb{W}_p(\mu_1, \mu_2) \lesssim d_{L_2}^{\frac{2(r-p)}{(d+2r)p}}(\mu_1, \mu_2) \lesssim \mathrm{MMD}_\nu^{\frac{r-p}{(d+2r)p}}(\mu_1, \mu_2).$$

See Section 2.4 of [56] for details. This applies for all possible MMD's associated with translation-invariant kernels, other than MMD-Matern.

## B.5  MMD-RBF FDE

Since MMD-Matern that is shown above covers the case of $0 < \nu < \infty$ and requires smooth enough densities. MMD-RBF (corresponding to MMD-Matern with $\nu = \infty$) is used a lot in many machine learning areas (e.g., kernel support vector machine).

$$k_{\sigma_{\mathrm{RBF}}}(\mathbf{x}, \mathbf{y}) = \frac{1}{(2\sqrt{\pi})^d} \cdot \sigma_{\mathrm{RBF}}^{-d} \cdot \exp\left(\frac{-1}{4\sigma_{\mathrm{RBF}}^2}\|\mathbf{x} - \mathbf{y}\|^2\right) \quad \text{with } \sigma_{\mathrm{RBF}} > 0.$$

It is also used in distributional reinforcement learning (MMDRL by [40]). They only used it in simulations, without any theoretical justification due to their failure to provide a contraction-inducing metric based on MMD-RBF. Following lemma can provide justification of its practice (proof and detailed bound in Appendix C.14.2), not requiring the existence of conditional densities.

**Corollary B.6.** *Under Assumptions 3.1, C.4, assuming $M_r := \sup_{\theta \in \Theta} \sup_{s,a}(\mathbb{E}\|Z(s,a;\theta)\|^r)^{1/r} < \infty$ with $r > p$, our estimator (10) based on $\mathfrak{d} = \mathrm{MMD}_{\sigma_{\mathrm{RBF}}}^2$ with $\sigma_{\mathrm{RBF}} \stackrel{let}{=} \sigma_N > 0$ can achieve the following bound in $\mathbb{R}^d$,*

$$\overline{\mathbb{W}}_{p,d_\pi,p}(\Upsilon_{\theta_T}, \Upsilon_\pi) \lesssim \frac{1}{1 - \gamma^{1-\frac{1}{2p}}} \cdot C(p, d, \sigma_N, p) \cdot \left(\frac{C_{brack}(\sigma_N)}{\sigma_N^d}\right)^{\frac{1}{1+\alpha/2} \cdot \frac{r-p}{(d+2r)p}}$$

$$\times O_P\left\{\left(\frac{N}{\log_{1/\gamma} N}\right)^{\frac{-1}{2+\alpha} \cdot \frac{2(r-p)}{(d+2r)p}}\right\} + \underbrace{\frac{2^{\frac{5}{2} - \frac{1}{2p}}}{1 - \gamma^{1-\frac{1}{2p}}} \cdot \frac{\Gamma((p+d)/2)}{\Gamma(p/2)} \cdot \sigma_N}_{\textit{non-random quantity decreasing by } \sigma_N \to 0}.$$

*Fixing $\sigma_N = \sigma_{\mathrm{RBF}} > 0$ leads to an irreducible term at the end. Of course, we can shrink it by letting $\sigma_N \to 0$. However, other terms (e.g., $C(p, d, \sigma_N, p)$, $C_{brack}(\sigma_N)/\sigma_N^d$ can increase with $\sigma_N \to 0$, leaving the optimal rate of $\sigma_N$ to be intractable.*

Although we cannot show that MMD-RBF can shrink Wasserstein inaccuracy to zero, it can bound gaussian-smoothed Wasserstein metric $\mathbb{W}_p^\sigma(\mu, \nu) := \mathbb{W}_p(\mu * \alpha_\sigma, \nu * \alpha_\sigma)$ where $*$ indicates convolution and $\alpha_\sigma$ is the probability measure of $N(\mathbf{0}, \sigma^2 I_d)$. This is widely used as an alternative for Wasserstein metric that is difficult to be computed in high dimensions $d > 1$ [e.g., 70, 41]. Thus, we have made a separate lemma (Lemma C.9) that can accommodate many other probability distances that can bound smoothed gaussain Wasserstein metric, which includes MMD-Two-Moment.

## B.6  MMD-RBF-transformed FDE

[70] suggested an MMD (17) with the following kernel which is RBF kernel multiplied with an extra term. We will refer to its corresponding MMD as MMD-RBF-transformed, denoting it as $\mathrm{MMD}_{(\sigma, f)}$.

$$k_{(\sigma, f)}(\mathbf{x}, \mathbf{y}) := \exp\left(-\frac{\|\mathbf{x} - \mathbf{y}\|^2}{4\sigma^2}\right) \cdot I_f\left(\frac{\|\mathbf{x} + \mathbf{y}\|}{\sqrt{2}\sigma}\right).$$

The conditions of density function $f$ and definition of $I_f$ are mentioned in Equations 8 and 10 of [70]. Assuming bounded kernel $k_{(\sigma, f)}(\cdot, \cdot) < \infty$ for fixed value of $\sigma > 0$, we can achieve the following. Proof and detailed bound are in Appendix C.14.3.

**Corollary B.7.** *Under Assumptions 3.1, C.4, our estimator* (10) *based on* $\mathfrak{d} = d_{L_2}^2$ *can achieve the following bound in* $\mathbb{R}^d$, *if we have* $k_{(\sigma,f)} < \infty$ *for fixed value of* $\sigma > 0$.

$$\overline{\mathbb{W}}_{p,d_\pi,p}(\Upsilon_{\theta_T}, \Upsilon_\pi) \lesssim \frac{\sigma_N}{1 - \gamma^{1-\frac{1}{2p}}} \cdot \left( C_{max}^{(\sigma_N,f)} \cdot C_{brack}^{1/2}(\sigma_N) \right)^{\frac{1/p}{1+\alpha/2}} \cdot O_P\left\{ \left( \frac{N}{\log_{1/\gamma} N} \right)^{\frac{-1/p}{2+\alpha}} \right\}$$

$$+ \frac{1}{1 - \gamma^{1-\frac{1}{2p}}} \cdot 2^{\frac{5}{2} - \frac{1}{p}} \cdot \frac{\Gamma((p+d)/2)}{\Gamma(p/2)} \cdot \sigma_N$$

*We can let* $\sigma_N \to 0$ *as* $N \to \infty$. *However, the optimal rate is intractable.*

One example of such $f$ is generalized beta-prime distributions: $f(x) = \frac{\epsilon}{2\pi x} \cdot \left( \left(\frac{x}{\lambda}\right)^{-\epsilon} + \left(\frac{x}{\lambda}\right)^\epsilon \right)^{-1}$ with $\epsilon \in (0, d + 2p]$ and $\lambda \in (0, \infty)$, which leads to Two-moment kernel (see Definition 1 of [70]). They have shown that such $k_{(\sigma,f)}$ is bounded in their Section 3.2 for bounded distributions.

## B.7 MMD-Coulomb FDE

MMD-Coulumb is an MMD which corresponds to the following kernel.

$$k_{cou}(\mathbf{x}, \mathbf{y}) = \kappa_0^{(cou)}(\mathbf{x} - \mathbf{y}) := \begin{cases} -\log \|\mathbf{x} - \mathbf{y}\| & \text{if} \quad d = 2 \\ \|\mathbf{x} - \mathbf{y}\|^{2-d} & \text{if} \quad d \geq 3 \end{cases}$$

We can see that $k(\cdot, \cdot) < \infty$ does not hold, leading to violation of the first statement of (23). Since we cannot apply Theorem C.5, we should resort to Theorem C.11 instead, which gives us the following result for MMD-Coulomb. Proof and detailed bound are in Appendix C.15.2.

**Corollary B.8.** *Under Assumptions 3.1, C.4, our estimator* (10) *based on* $\mathfrak{d} = \mathrm{MMD}_{cou}^2$ *can achieve the following bound. Assuming bounded support in* $\mathbb{R}^d$ *($d \geq 2$) and* $\sup_{\theta \in \Theta} \sup_{s,a} \mathcal{E}_{cou}(\Upsilon_\theta(s,a)) < \infty$ *with* $\mathcal{E}_{cou}(\mu) := k_{cou}(\mathbf{x}, \mathbf{y})\mu(d\mathbf{x})\mu(d\mathbf{y})$, *we have*

$$\overline{\mathbb{W}}_{p,d_\pi,p}(\Upsilon_{\theta_T}, \Upsilon_\pi) \lesssim \frac{1}{1 - \gamma^{1-\frac{1}{2p}}} \cdot O_P\left\{ \frac{1}{\log(1/\gamma)} \cdot \left( \frac{(\log N)^2}{N} \right)^{\frac{6-\alpha}{16p}} \right\}$$

*We can further bound* $\overline{\mathbb{W}}_{p,d_\pi,p}(\Upsilon_{\theta_T}, \Upsilon_\pi) \leq D^{2(1-\frac{1}{p})} \cdot \overline{\mathbb{W}}_{1,d_\pi,1}^{1/p}(\Upsilon_{\theta_T}, \Upsilon_\pi)$ *if we assume bounded conditional distributions. Note that the condition* $\mathcal{E}(\Upsilon_\theta(s,a)) < \infty$ *does not allow nonzero probability to any points.*

## B.8 KL FDE

Interestingly enough, convergence of FLE [63] can also be shown by Theorem D.1. However, since it is based on maximium likelihood, which is not a squared metric, we could fit this into the structure of neither Theorem C.5 nor C.11. Instead, we developed a method that uses log-likelihood (or equivalently, KL divergence), based on Theorem D.1. See Appendix C.16 for its proof. Note that this is different from FLE in the sense that it fully utilizes the closed form density of $\Upsilon_{\hat{\theta}_{n,t-1}}(s', a')$ and $\pi(a'|s')$ needed for $\Psi_\pi(r, s', \Upsilon_{\hat{\theta}_{n,t-1}})$ in computing the objective function (10). This is free from MC error caused by sampling from $z \sim \Psi_\pi(r, s', \Upsilon_{\hat{\theta}_{n,t-1}})$ as [63] did.

**Corollary B.9.** *Under Assumptions 3.1, C.4, our estimator* (10) *with* $\mathfrak{d} = \mathrm{KL}$ *(Kullback-Leibler divergence) achieves the following bound. Assuming bounded distributions and existence of conditional densities, we have following with* $T = \lfloor \frac{1}{1-\frac{1}{2p}} \cdot \frac{1/p}{2+\alpha} \cdot \log_{1/\gamma} N \rfloor$,

$$\overline{\mathbb{W}}_{p,d_\pi,p}(\Upsilon_{\theta_T}, \Upsilon_\pi) \lesssim \frac{1}{1 - \gamma^{1-\frac{1}{2p}}} \cdot O_P\left\{ \left( \frac{N}{\log_{1/\gamma} N} \right)^{\frac{-1/p}{2+\alpha}} \right\}.$$

*Here, we have* $\log \mathcal{N}_{[\,]}(\widetilde{\mathcal{F}}_\Theta^{1/2}, \|\cdot\|_{L_2,\rho}, \epsilon) \leq C_{brack} \cdot \epsilon^{-\alpha}$ *for some* $\alpha \in (0,2)$ *with* $\widetilde{\mathcal{F}}_\Theta^{1/2}$ *being the squared conditional densities, i.e.,* $\widetilde{\mathcal{F}}_\Theta^{1/2} := \{f_\theta^{1/2}(\cdot | \cdots) : \theta \in \Theta\}$.

# C Technical proofs

## C.1 Functional Bregman divergence

### C.1.1 Technical definition

Bregman divergence was originally developed to measure discrepancy between two real vectors. It measures the difference between the value of the convex function $\phi$ at $\mathbf{x} \in \mathbb{R}^d$ and its first-order approximation at $\mathbf{y} \in \mathbb{R}^d$:

$$D_\phi(\mathbf{x}, \mathbf{y}) = \phi(\mathbf{x}) - \phi(\mathbf{y}) - \langle \nabla \phi(\mathbf{y}), \mathbf{x} - \mathbf{y} \rangle.$$

After defining a scalar-valued convex function $\Phi$ over a functional inner product space, they can also be extended to functional objects based on Frechet derivative [19].

$$D_\Phi(f, g) = \Phi(f) - \Phi(g) - \langle \delta\Phi(g), f - g \rangle.$$

Based on the fact that each probability measure can have its own functional representation (e.g., density functions), we can also extend functional Bregman divergence into discrepancy between two probability measures. Following is the technical definition of a functional Bregman divergence for two probability measures $P, Q \in \mathcal{P}$ (Section 3.2 of [43]).

**Definition C.1.** Linear span of $\mathcal{P}$, namely span$\mathcal{P}$, is a collection of signed measures, and let $\mathcal{U} \subseteq \text{span}\mathcal{P}$ be a convex subset that contains $\mathcal{P}$, i.e., $\mathcal{P} \subseteq \mathcal{U}$. Let $\mathcal{L}(\mathcal{P})$ be the functional space where its arbitrary element $f$ satisfies $\int_{\Omega_\mathcal{X}} |f| d\mu < \infty$ for $\forall \mu \in \mathcal{P}$ with $\Omega_\mathcal{X}$ being the support space. Now, assume a strictly convex function $\Phi : \mathcal{U} \to \mathbb{R}$ that has a subgradient $\Phi^* : \mathcal{U} \to \mathcal{L}(\mathcal{P})$ which satisfies $\Phi(Q) \geq \Phi^*(P) \cdot (Q - P) + \Phi(P)$ for all $P, Q \in \mathcal{U}$, where $f \cdot \mu := \int_{\Omega_\mathcal{X}} f d\mu$ for any $f \in \mathcal{L}(\mathcal{P})$ and $\mu \in \text{span}\mathcal{P}$. Then we can build a functional Bregman divergence $\mathfrak{d} : \mathcal{U} \times \mathcal{U} \to \mathbb{R}_+$ on $\mathcal{U}$ as follows (Definition 3.8 of [43]),

$$\mathfrak{d}(P, Q) = \Phi(Q) - \Phi^*(P) \cdot (Q - P) - \Phi(P).$$

Since functional Bregman divergence in Definition C.1 is based on a strictly convex function $\Phi$, equality $\mathfrak{d}(P, Q) = 0$ holds only when $P = Q$. This is needed to ensure that $\mathcal{T}^\pi \Upsilon$ becomes the unique minimizer of (9). Note that this is stronger than general definition of functional Bregman divergence (e.g., Definition 3.8 of [43]) that is based on standard convex function (which may not be strictly convex).

### C.1.2 Proof of Theorem 2.4

We will treat both $S, A \sim \rho$ and $R, S' \sim \tilde{p}(\cdots | S, A)$ to be random. Let $\Upsilon_1, \Upsilon_2 \in \mathcal{P}^{\mathcal{S} \times \mathcal{A}}$ be arbitrary. We obtain the following, by using the same logic that [3] used in their Theorem 1,

$$\mathbb{E}\left\{ \mathfrak{d}\big(\Upsilon_1(S, A), \Psi_\pi(R, S', \Upsilon_2)\big) \right\}$$

$$= \mathbb{E}\left\{ \Phi\big(\Psi_\pi(R, S', \Upsilon_2)\big) - \Phi\big(\Upsilon_1(S, A)\big) - \Phi^*\big(\Upsilon_1(S, A)\big) \cdot \big(\Psi_\pi(R, S', \Upsilon_2) - \Upsilon_1(S, A)\big) \right\}.$$

Letting $\Upsilon_2 = \Upsilon$ and $\Upsilon_1 = \mathcal{T}^\pi \Upsilon$, we have

$$\mathbb{E}\left\{ \Phi^*\big(\mathcal{T}^\pi \Upsilon(S, A)\big) \cdot \big(\Psi_\pi(R, S', \Upsilon) - \mathcal{T}^\pi \Upsilon(S, A)\big) \right\} =$$

$$\mathbb{E}\left\{ \Phi^*\big(\mathcal{T}^\pi \Upsilon(S, A)\big) \cdot \mathbb{E}\left\{ \Psi_\pi(R, S'; \Upsilon) - \mathcal{T}^\pi \Upsilon(S, A) \,\Big|\, S, A \right\} \right\} = 0,$$

$$\therefore \mathbb{E}\left\{ \mathfrak{d}\big(\mathcal{T}^\pi \Upsilon(S, A), \Psi_\pi(R, S', \Upsilon)\big) \right\} = \mathbb{E}\left\{ \Phi\big(\Psi_\pi(R, S', \Upsilon)\big) \right\} - \mathbb{E}\left\{ \Phi\big(\mathcal{T}^\pi \Upsilon(S, A)\big) \right\}.$$

Then, the following holds based on $\mathcal{T}^\pi \Upsilon_2(s,a) = \mathbb{E}\{\Psi_\pi(R,S',\Upsilon_2)|s,a\}$,

$$\mathbb{E}\Big\{\mathfrak{d}\big(\Upsilon_1(S,A),\Psi_\pi(R,S',\Upsilon_2)\big)\Big\} - \mathbb{E}\Big\{\mathfrak{d}\big(\mathcal{T}^\pi \Upsilon_2(S,A),\Psi_\pi(R,S',\Upsilon_2)\big)\Big\}$$

$$= \mathbb{E}\Big\{\Phi\big(\mathcal{T}^\pi \Upsilon_2(S,A)\big)\Big\} - \mathbb{E}\Big\{\Phi\big(\Upsilon_1(S,A)\big)\Big\}$$

$$- \mathbb{E}\Big\{\Phi^*\big(\Upsilon_1(S,A)\big)\cdot\big(\Psi_\pi(R,S',\Upsilon_2)-\Upsilon_1(S,A)\big)\Big\}$$

$$= \mathbb{E}\Big\{\mathfrak{d}\big(\Upsilon_1(S,A),\mathcal{T}^\pi \Upsilon_2(S,A)\big)\Big\}.$$

Therefore, we have

$$\mathbb{E}\Big\{\mathfrak{d}\big(\Upsilon_1(S,A),\Psi_\pi(R,S',\Upsilon_2)\big)\Big\} = \bar{\mathfrak{d}}_\rho(\Upsilon_1,\mathcal{T}^\pi\Upsilon_2) + \mathbb{E}\Big\{\mathfrak{d}\big(\mathcal{T}^\pi\Upsilon_2(S,A),\Psi_\pi(R,S',\Upsilon_2)\big)\Big\}. \tag{13}$$

where $\bar{\mathfrak{d}}_\rho(\Upsilon_1,\Upsilon_2) := \mathbb{E}\{\mathfrak{d}(\Upsilon_1(S,A),\Upsilon_2(S,A))\}$. Note that we definition of our empirical and population objective functions (5) and (9) already require $\Psi_\pi(r,s';\Upsilon) \in \mathcal{P}$. Otherwise, they will not be defined. Then, by convexity of $\mathcal{P}$, we have $\mathcal{T}^\pi\Upsilon(s,a) \in \mathcal{P}$, since it is a convex combination over $r,s' \sim p(\cdot|s,a)$ (see definition (3)). This leads to satisfaction of (9), since $\mathcal{P}$ is a convex set.

To ensure $\Psi_\pi(r,s';\Upsilon) \in \mathcal{P}$, we can assume that $\mathcal{P}$ is closed under push-forward mapping with respect to single-sample Bellman backup functions $\{g_{r,\gamma}\}$ for all $r \in \mathbb{R}^d$ that can be observed. For a single-sampe Bellman backup function defined as $g_{r,\gamma}(x) := r + \gamma x$ and $X \sim \mu$, its result of push-forward mapping $(g_{r,\gamma})_\# \mu$ is defined as the distribution of $r + \gamma X$. Being closed under push-forward mapping means that $(g_{r,\gamma})_\# \mu \in \mathcal{P}$ holds for any probability measure $\mu \in \mathcal{P}$ and any $r \in \mathbb{R}^d$ that can be observed. Then, for an arbitrary $\Upsilon \in \mathcal{P}^{\mathcal{S}\times\mathcal{A}}$, we can ensure that $\Psi_\pi(r,s',\Upsilon) \in \mathcal{P}$ holds for any target policy $\pi : \mathcal{S} \to \Delta(\mathcal{A})$ and observable $(r,s') \in \mathbb{R}^d \times \mathcal{S}$, by its definition (4) and convexity of $\mathcal{P}$. Then, our objective functions (5) and (9) can always be defined.

## C.2 How bounding iteration-level error leads to final inaccuracy

By Definition 2.2, we can prove (6) based on $\Upsilon_\pi = \mathcal{T}^\pi\Upsilon_\pi$:

$$\tilde{\eta}(\Upsilon_T,\Upsilon_\pi) \le \tilde{\eta}(\Upsilon_T,\mathcal{T}^\pi\Upsilon_{T-1}) + \tilde{\eta}(\mathcal{T}^\pi\Upsilon_{T-1},\Upsilon_\pi) \le \tilde{\eta}(\Upsilon_T,\mathcal{T}^\pi\Upsilon_{T-1}) + \zeta\cdot\tilde{\eta}(\Upsilon_{T-1},\Upsilon_\pi)$$

$$\le \cdots \le \sum_{t=1}^{T}\zeta^{T-t}\cdot\tilde{\eta}(\Upsilon_t,\mathcal{T}^\pi\Upsilon_{t-1}) + \zeta^T\cdot\tilde{\eta}(\Upsilon_0,\Upsilon_\pi).$$

This leads to (6).

## C.3 How extended metrics become contraction-inducing metrics

### C.3.1 Expectation-extension

Let us show that the expectation-extended (8) metric satisfies Definition 2.2.

$$\bar{\eta}_{d_\pi,q}^{2q}(\mathcal{T}^\pi\Upsilon_1,\mathcal{T}^\pi\Upsilon_2) = \mathbb{E}_{s,a\sim d_\pi}\Bigg[\Big\{\underbrace{\eta^q\big(\mathcal{T}^\pi\Upsilon_1(s,a),\mathcal{T}^\pi\Upsilon_2(s,a)\big)}_{(A)}\Big\}^2\Bigg]$$

Here, the (A) part can be bounded as follows based on Properties (15). Letting $\tilde{p}^\pi(r,s',a'|s,a) = \tilde{p}(r,s'|s,a)\cdot\pi(a'|s')$ and temporarily using it as a probability measure, we have

$$(A) = \eta^q\bigg\{\int(g_{r,\gamma})_\#\Upsilon_1(s',a')\mathrm{d}\tilde{p}^\pi(r,s',a'|s,a), \int(g_{r,\gamma})_\#\Upsilon_2(s',a')\mathrm{d}\tilde{p}^\pi(r,s',a'|s,a)\bigg\}$$

$$\le \int\eta^q\Big\{(g_{r,\gamma})_\#\Upsilon_1(s',a'),(g_{r,\gamma})_\#\Upsilon_2(s',a')\Big\}\mathrm{d}\tilde{p}^\pi(r,s',a'|s,a)$$

$$\le \gamma^{cq}\cdot\int\eta^q\big(\Upsilon_1(s',a'),\Upsilon_2(s',a')\big)\mathrm{d}\tilde{p}^\pi(r,s',a'|s,a), \tag{14}$$

where we have used the (C) property of (15) in the second line, and the (S) and (L) properties of (15) in the last line. Then, we have

$$\bar{\eta}_{d_\pi,q}^{2q}(\mathcal{T}^\pi\Upsilon_1, \mathcal{T}^\pi\Upsilon_2) \leq \gamma^{2cq} \cdot \mathbb{E}_{s,a\sim d_\pi}\left[\left\{\mathbb{E}_{s',a'\sim\tilde{p}^\pi(\cdots|s,a)}\big(\eta^q(\Upsilon_1(s',a'), \Upsilon_2(s',a'))\big)\right\}^2\right]$$

$$\leq \gamma^{2cq} \cdot \mathbb{E}_{s,a\sim d_\pi}\mathbb{E}_{s',a'\sim\tilde{p}^\pi(\cdots|s,a)}\left\{\eta^{2q}\big(\Upsilon_1(s',a'), \Upsilon_2(s',a')\big)\right\}$$

$$\leq \gamma^{2cq-1} \cdot \mathbb{E}_{s,a\sim d_\pi}\left\{\eta^{2q}\big(\Upsilon_1(s,a), \Upsilon_2(s,a)\big)\right\},$$

We acknowledge that the last line can be shown by the trick used by [63], i.e., $\mathbb{E}_{\tilde{s},\tilde{a}\sim d_\pi}\tilde{p}^\pi(s,a|\tilde{s},\tilde{a}) \leq \gamma^{-1} \cdot d_\pi(s,a)$. This finally leads to

$$\therefore \quad \bar{\eta}_{d_\pi,q}(\mathcal{T}^\pi\Upsilon_1, \mathcal{T}^\pi\Upsilon_2) \leq \gamma^{c-\frac{1}{2q}} \cdot \bar{\eta}_{d_\pi,q}(\Upsilon_1, \Upsilon_2).$$

### C.3.2   Supremum-extension

We can apply the same logic to supremum-extension (7). Taking up from (14), we have

$$(A) \leq \gamma^{cq} \cdot \sup_{s,a} \eta\big(\Upsilon_1(s,a), \Upsilon_2(s,a)\big).$$

Taking supremum over $\mathcal{S} \times \mathcal{A}$ on the LHS gives us $\eta_\infty^q(\mathcal{T}^\pi\Upsilon_1, \mathcal{T}^\pi\Upsilon_2) \leq \gamma^{cq} \cdot \eta_\infty(\Upsilon_1, \Upsilon_2)$, leading to $\eta_\infty(\mathcal{T}^\pi\Upsilon_1, \mathcal{T}^\pi\Upsilon_2) \leq \gamma^c \cdot \eta_\infty(\Upsilon_1, \Upsilon_2)$. Although Theorem 4.25 of [6] presented the same statement, their proof requires an additional assumption, i.e., $R$ and $S'$ (generated from $\tilde{p}(\cdots|s,a)$) being independent. We did not require this assumption.

### C.4   Examples of probability metric

To construct a metric $\tilde{\eta}$ over $\mathcal{P}^{\mathcal{S}\times\mathcal{A}}$ that satisfies Definition 2.2, we will assume to have a probability metric $\eta$ over $\mathcal{P}$ that satisfies the following three properties of (15) with some $c > 0$ and $q \geq 1$. We copied the terminologies from [6] (see their Definitions 4.22, 4.23, 4.24). In the first two properties, $\mathcal{L}(X)$ denotes the probability measure of random vector $X \in \mathbb{R}^d$, and $\mu_1, \mu_2 : \mathcal{X} \to \mathcal{P}, \nu \in \Delta(\mathcal{X})$ for some space $\mathcal{X}$ can be arbitrary in the third property.

**(S)** $c$-scale-sensitive : $\eta(\mathcal{L}(\gamma X), \mathcal{L}(\gamma Y)) \leq \gamma^c \cdot \eta(\mathcal{L}(X), \mathcal{L}(Y))$           (15)

**(L)** location-insensitive : $\eta(\mathcal{L}(X+z), \mathcal{L}(Y+z)) \leq \eta(\mathcal{L}(X), \mathcal{L}(Y))$   for an arbitrary $z \in \mathbb{R}^d$

**(C)** $q$-convex : $\eta^q\left\{\int\mu_1(x)\mathrm{d}\nu(x), \int\mu_2(x)\mathrm{d}\nu(x)\right\} \leq \int\eta^q\left\{\mu_1(x), \mu_2(x)\right\}\mathrm{d}\nu(x)$.

Note that if some $q \geq 1$ satisfies the Property-3, then any $q' \geq q$ also satisfies it by Jensen's inequality.

Here are the examples that satisfy the properties in (15). We specify the values of $c, q > 0$ for each $\eta$. Then, we specify the contractive factor $\zeta = \gamma^{c-\frac{1}{2q}}$ for their corresponding expectation-extended metrics (8). (Note that the supremum-extension (7) has $\zeta = \gamma^c$.)

$$\text{Wasserstein-}p \text{ metric } (\mathbb{W}_p): \quad c = 1, \quad q \geq p \quad \Rightarrow \quad \zeta = \gamma^{1-\frac{1}{2q}} \tag{16}$$

$$\text{MMD-Energy } (\mathrm{MMD}_\beta): \quad c = \frac{\beta}{2}, \quad q \geq 1 \text{ and } q > \frac{1}{\beta} \quad \Rightarrow \quad \zeta = \gamma^{\frac{1}{2}(\beta-\frac{1}{q})}$$

$$\text{CDF } L_p - \text{metric } (l_p): \quad c = \frac{1}{p}, \quad q \geq p \quad \Rightarrow \quad \zeta = \gamma^{\frac{1}{p}-\frac{1}{2q}}$$

Here are the definition of above metrics. $\mathbb{W}_p$ is the Wasserstein metric, which is defined as follows with $J(\mu_1, \mu_2)$ being the possible joint distributions of $\mathbf{X} \sim \mu_1$ and $\mathbf{Y} \sim \mu_2$,

$$\mathbb{W}_p(\mu_1, \mu_2) := \inf_{J(\mu_1,\mu_2)}\left(\mathbb{E}\|\mathbf{X}-\mathbf{Y}\|^p\right)^{1/p}.$$

It is straightforward to see Properties 1 and 2 hold with $c = 1$. Property 3 is shown to hold by [26] (see their Appendix A.3.1).

MMD (maximum-mean discrepancy) [25] associated with a kernel $k(\cdot, \cdot)$ defined as

$$\mathrm{MMD}_k^2(\mu_1, \mu_2) := \mathbb{E}\{k(X, X')\} + \mathbb{E}\{k(Y, Y')\} - 2 \cdot \mathbb{E}\{k(X, Y)\}, \tag{17}$$

with $\mathbf{X}, \mathbf{X}' \sim \mu_1$ and $\mathbf{Y}, \mathbf{Y}' \sim \mu_2$ are all independent. $\mathrm{MMD}_\beta$ is associated with Energy kernel $k_\beta(\mathbf{x}, \mathbf{y}) := \|\mathbf{x}\|^\beta + \|\mathbf{y}\|^\beta - \|\mathbf{x} - \mathbf{y}\|^\beta$ $(0 < \beta < 2)$, Properties 1 and 2 are straightforward t show with $c = \beta/2$. Property 3 can be shown with $q = 1$, since MMD can be expressed as the norm of difference between two mean embeddings, i.e., $\mathrm{MMD}_k(\mu_1, \mu_2) = \|\kappa_{\mu_1} - \kappa_{\mu_2}\|_{\mathcal{H}}$ where $\kappa_\mu$ is the mean embedding of $\mu$ and $\| \cdot \|_{\mathcal{H}}$ is the corresponding RKHS norm of RKHS $\mathcal{H}$.

$$\mathrm{MMD}_k\left( \int \mu_1(x)\mathrm{d}\nu(x), \int \mu_2(x)\mathrm{d}\nu(x) \right) = \left\| \int \kappa_{\mu_1(x)}\mathrm{d}\nu(x) - \int \kappa_{\mu_2(x)}\mathrm{d}\nu(x) \right\|_{\mathcal{H}}$$

$$\leq \int \|\kappa_{\mu_1(x)} - \kappa_{\mu_2(x)}\|_{\mathcal{H}}\mathrm{d}\nu(x) = \int \mathrm{MMD}_k(\mu_1(x), \mu_2(x))\mathrm{d}\nu(x).$$

$l_p$ is the metric between cdf's of 1-dimensional random variable. Letting $F_i$ be the cdf of $\mu_i$, we define

$$l_p(\mu_1, \mu_2) := \left( \int_{-\infty}^{\infty} |F_1(z) - F_2(z)|^p \mathrm{d}z \right)^{\frac{1}{p}}.$$

Properties 1 and 2 are shown with $c = 1/p$ by Proposition 3.2 of [42]. Property 3 can be shown by using the fact that $l_p$ is a functional $L_p$ norm between cdf's. The same logic holds for $d_{L_p}$.

Table 3 also contains the survey of other well-known probability metrics. We will skip proof for their satisfaction or non-satisfaction of each property. With $f_i$ and $F_i$ being the pdf and cdf of probability measure $\mu_i$,

$$\text{Hellinger metric: } h(\mu_1, \mu_2) := \left\{ \int_{\mathbb{R}^d} \left( \sqrt{f_1(z)} - \sqrt{f_2(z)} \right)^2 \mathrm{d}z \right\}^{1/2},$$

$$\text{PDF } L_p\text{-metric: } d_{L_p}(\mu_1, \mu_2) := \left( \int_{\mathbb{R}^d} \left| f_1(z) - f_2(z) \right|^p \mathrm{d}z \right)^{1/p},$$

$$\text{Discrepancy metric: } d_D(\mu_1, \mu_2) := \sup_{\text{closed balls } B} \left| \mu_1(B) - \mu_2(B) \right|,$$

$$\text{Kolmogorov metric: } d_K(\mu_1, \mu_2) := \sup_{x \in \mathbb{R}} \left| F_1(x) - F_2(x) \right|.$$

Table 3: Survey of metrics: If it satisfies $c$-scale-sensitive or $q$-convexity, the corresponding value of $c > 0$ or $q \geq 1$ are specified.

| | Location-insensitive | Scale-sensitive | Convex |
|---|---|---|---|
| Wasserstein-p ($\mathbb{W}_p$) | ✓ | $c = 1$ | $q \geq p$ |
| MMD-Energy ($\mathrm{MMD}_\beta$ with $0 < \beta < 2$) | ✓ | $c = \beta/2$ | $q \geq 1$ |
| MMD-Matern ($\mathrm{MMD}_\nu$ with $0 < \nu \leq \infty$) | ✓ | ✗ | $q \geq 1$ |
| CDF-$L_p$ ($l_p$) | ✓ | $c = 1/p$ | $q \geq 1$ |
| PDF-$L_p$ ($d_{L_p}$) : $p = 1$ refers to $2 \times$TVD | ✓ | ✗ | $q \geq 1$ |
| Hellinger metric | ✓ | $c = d/4$ | ✗ |
| Discrepancy metric | ✓ | ✗ | $q \geq 1$ |
| Kolmogorov metric | ✓ | ✗ | $q \geq 1$ |

In Table 3, MMD's associated with any kernel satisfy convexity with $q = 1$. However, MMD-Energy was so far the only example that we have found, which satisfies all three properties. In the definitions of Wasserstein metric and MMD-Energy (Appendix C.4), the Euclidean norm $\| \cdot \|$ used can be replaced with another norm on $\mathbb{R}^d$. See Example 11 of [23] and Proposition 3 of [49] for extensions of MMD-Energy, and [44] for extensions of Wasserstein metrics.

## C.5 Functional spaces and corresponding norms

Following are examples of the functional inner product space $(\mathcal{G}, \langle \cdot, \cdot \rangle_m)$ and its corresponding probability space $\mathcal{P}$ that we have used for our methods in Section 3.1.

For $m = l_2$ (only applying for $d = 1$), the functional representer is the cdf, denoted by $F(\cdot|\mu)$. The functional space is the functional $L_2$ space, that is $\mathcal{G} := \{F : \mathcal{X} \to \mathbb{R} : \int |F(z)|^2 \mathrm{d}z < \infty\}$ with $\mathcal{X} = [b_1, b_2] \subsetneq \mathbb{R}$ being a bounded support. The equipped inner product $\langle \cdot, \cdot \rangle_m = \langle \cdot, \cdot \rangle_{L_2}$ is defined as $\langle F_1, F_2 \rangle_{L_2} := \int_{\mathcal{X}} F_1(z)F_2(z)\mathrm{d}z$. The functional representer of $\Upsilon_\theta(s, a) \in \mathcal{P}$, namely $F_\theta(\cdot|s, a)$ will be an element of the following set $\mathcal{F}$:

$$m = l_2 \ (d = 1) \ : \ \mathcal{F} \overset{let}{=} \{F : \mathcal{X} \to [0, 1] \mid F \text{ is a cdf.}\} \subsetneq \mathcal{G},$$
$$: \ \mathcal{P} \overset{let}{=} \Delta(\mathcal{X}) := \{\text{probability measures that have support in } \mathcal{X}\}.$$

For $m = d_{L_2}$, the functional representer is the pdf, denoted by $f(\cdot|\mu)$. The functional space is the functional $L_2$ space, that is $\mathcal{G} := \{f : \mathcal{X} \to \mathbb{R} : \int |f(z)|^2 \mathrm{d}z < \infty\}$ with $\mathcal{X} \subseteq \mathbb{R}^d$ being the support. The equipped inner product $\langle \cdot, \cdot \rangle_m = \langle \cdot, \cdot \rangle_{L_2}$ is defined as $\langle f_1, f_2 \rangle_{L_2} := \int_{\mathcal{X}} f_1(z)f_2(z)\mathrm{d}z$. The functional representer of $\Upsilon_\theta(s, a) \in \mathcal{P}$, namely $f_\theta(\cdot|s, a)$ will be an element of the following set $\widetilde{\mathcal{F}}$:

$$m = d_{L_2} \ : \ \widetilde{\mathcal{F}} \overset{let}{=} \{f : \mathcal{X} \to \mathbb{R}_+ \mid f \text{ is a pdf and } \|f(\cdot|\mu)\|_{L_2} < \infty\} \subsetneq \mathcal{G},$$
$$: \ \mathcal{P} \overset{let}{=} \{\mu \in \mathcal{P} \mid \mu \text{ has a pdf and } \|f(\cdot|\mu)\|_{L_2} < \infty\}.$$

For $m = \mathrm{MMD}_k$ (maximum mean discrepancy associated with some kernel), the functional representer is the kernel mean embedding, $\kappa(\cdot|\mu) = \mathbb{E}_{Z \sim \mu}\{k(Z, \cdot)\}$. The functional space is the RKHS corresponding to the given kernel, i.e., $\mathcal{G} = \{\sum_{i=1}^n \alpha_i \cdot k(x_i, \cdot)|\alpha_i \in \mathbb{R}, x_i \in \mathcal{X}, n \in \mathbb{N}\}$ with $\mathcal{X} \subseteq \mathbb{R}^d$ being the support. The equipped inner product is the corresponding RKHS inner product, $\langle \cdot, \cdot \rangle_m = \langle \cdot, \cdot \rangle_{\mathcal{H}}$ that is defined as $\langle \sum_{i=1}^n \alpha_i k(x_i, \cdot), \sum_{j=1}^m \beta_j k(x'_j, \cdot) \rangle_{\mathcal{H}} := \sum_{i=1}^n \sum_{j=1}^m \alpha_i \beta_j k(x_i, x'_j)$. The functional representer of $\Upsilon_\theta(s, a) \in \mathcal{P}$, namely $\kappa_\theta(\cdot|s, a)$ will be an element of the following set $\mathcal{H}$:

$$m = \mathrm{MMD} \ : \ \mathcal{H} \overset{let}{=} \{\kappa : \mathcal{X} \to \mathbb{R} \mid \kappa(\cdot) = \mathbb{E}_{Z \sim \mu}\{k(Z, \cdot)\} \text{ for some } \mu \in \mathcal{P}\} \subsetneq \mathcal{G},$$
$$: \ \mathcal{P} \overset{let}{=} \{\mu \in \mathcal{P} \mid \|\kappa(\cdot|\mu)\|_{\mathcal{H}} < \infty\}.$$

### C.6 Proof for Theorem 3.3

**Definition C.2** (Inner-product-space metric). The probability metric $m$ over $\mathcal{P}$ is called an inner-product-space metric (IPS metric, in short) if there exists an inner product space $(\mathcal{G}, \langle \cdot, \cdot \rangle_m)$ where any element $\mu \in \mathcal{P}$ can be uniquely (up to equivalence under zero induced norm $\| \cdot \|_m$) represented by the corresponding element $G(\cdot|\mu) \in \mathcal{G}$ such that

(IPS1) The probability $m$ is expressed as the norm of difference between functional representers, i.e., $m(\mu_1, \mu_2) := \|G(\cdot|\mu_1) - G(\cdot|\mu_2)\|_m$, where $\| \cdot \|_m$ is the induced norm by the inner product;

(IPS2) Functional representers of probability measures are unbiased with respect to probability mixture. That is, for $\mu(\cdot) : \mathcal{Z} \to \mathcal{P}$ and arbitrary probability $P$ over $\mathcal{Z}$, we have $G(\cdot| \int \mu(z)\mathrm{d}P(z)) = \int G(\cdot|\mu(z))\mathrm{d}P(z)$.

It is possible that there exists an element in $\mathcal{G}$ that does not represent any $\mu \in \mathcal{P}$.

#### C.6.1 Outline of proof

The first step is to obtain the convergence of single iteration at $t$-th step, i.e., $\eta_\infty(\Upsilon_{\hat{\theta}_{n,t}}, \mathcal{T}^\pi \Upsilon_{\hat{\theta}_{n,t-1}}) = O_P(1/\sqrt{n})$. This is similar to Lemma 10 of [2], which demonstrates that empirical probability measure converges towards the underlying probability measure in $O_P(1/\sqrt{n})$ rate with respect to MMD. We can generalize the argument to IPS metrics (listed in Appendix C.5) that have corresponding inner product spaces of functional elements. We can form the analogy to the empirical probability measure under tabular setting to simplify the arguments.

The second step is to obtain convergence of $T$ different steps' estimations. Here, there are two things that play a role in determining $t$-th estimation, i.e., the previous iterate and the data used in the $t$-th iteration (denoted as $\mathcal{D}_t$, or the $t$-th subdataset). Although the previous iterate depends on all previous datasets, i.e., $\mathcal{D}_1, \cdots, \mathcal{D}_{t-1}$, we can utilize the independence among datasets.

### C.6.2 Proof

Assuming $|\mathcal{S} \times \mathcal{A}| < \infty$, each state-action pair can be observed with probability at least $p_{\min} := \inf_{s,a} \rho(s,a)$ where $\rho(s,a) = \mathbb{P}\{S, A = s, a\}$. We can always assume $p_{\min} > 0$ without loss of generality, since we can exclude $s, a$ with $\rho(s,a) > 0$ from $\mathcal{S} \times \mathcal{A}$. Denoting the empirical estimate of $\rho(s,a)$ and $\tilde{p}(r, s'|s, a)$ as $\hat{\rho}(s,a)$ and $\hat{p}(r, s'|s, a)$, we can copy the logic of (13) to obtain the following equivalence for $\hat{\theta}_{n,t}$ defined in (10),

$$\hat{\theta}_{n,t} = \arg \min_{\theta \in \Theta} \sum_{s,a} \hat{\rho}(s,a) \cdot \mathfrak{d}\left\{ \Upsilon_\theta(s,a), \hat{\mathcal{T}}^\pi \Upsilon_{\hat{\theta}_{n,t-1}}(s,a) \right\}.$$

Here, $\hat{\mathcal{T}}^\pi$ is defined accordingly to (3), only replacing $\tilde{p}(r, s'|s, a)$ with $\hat{p}(r, s'|s, a)$.

Now, we restrict $\mathfrak{d} = m^2$ into squared form of probability metric so that it satisfies relaxed triangular inequality, that is, $m^2(P, Q) \leq 2 \cdot (m^2(P, R) + m^2(R, Q))$. With fixed $\hat{\theta}_{n,t-1}$ and its corresponding value fo $\theta_{*,t}$ such that $\Upsilon_{\theta_{*,t}} = \mathcal{T}^\pi \Upsilon_{\hat{\theta}_{n,t-1}}$, we have following,

$$\bar{m}_{n,1}^2(\hat{\theta}_{n,t}.\theta_{*,t}) \leq 2 \cdot \bar{m}_{n,1}^2(\Upsilon_{\hat{\theta}_{n,t}}, \hat{\mathcal{T}}^\pi \Upsilon_{\hat{\theta}_{n,t-1}}) + 2 \cdot \bar{m}_{n,1}^2(\hat{\mathcal{T}}^\pi \Upsilon_{\hat{\theta}_{n,t-1}}, \Upsilon_{\theta_{*,t}})$$

$$\leq 4 \cdot \bar{m}_{n,1}^2(\Upsilon_{\theta_{*,t}}, \hat{\mathcal{T}}^\pi \Upsilon_{\hat{\theta}_{n,t-1}}) \leq 4 \cdot \sup_{s,a} m^2 \left\{ \Upsilon_{\theta_{*,t}}(s,a), \hat{\mathcal{T}}^\pi \Upsilon_{\hat{\theta}_{n,t-1}}(s,a) \right\}, \tag{18}$$

$$\text{where} \quad \bar{m}_{n,1}^2(\theta_1, \theta_2) := \frac{1}{n} \sum_{i=1}^n m^2 \left\{ \Upsilon_{\theta_1}(s_i, a_i), \Upsilon_{\theta_2}(s_i, a_i) \right\}. \tag{19}$$

Given that $\Upsilon_{\theta_{*,t}} = \mathcal{T}^\pi \Upsilon_{\hat{\theta}_{n,t-1}}$, this comes down to showing $\hat{\mathcal{T}}^\pi \Upsilon_{\hat{\theta}_{n,t-1}}(s,a)$ converges in $m$ towards $\mathcal{T}^\pi \Upsilon_{\hat{\theta}_{n,t-1}}(s,a)$ for each $s, a$. Towards that end, we will assume that each $s, a$ is observed sufficiently many times. We define the following event along with two probability vectors in $\mathbb{R}^{\mathcal{S} \times \mathcal{A}}$,

$$\Omega_0 := \left\{ \omega \in \Omega \mid \|\hat{\mathbf{p}} - \mathbf{p}\|_\infty \leq \frac{1}{2} p_{\min} \right\} \quad \text{with} \quad \mathbf{p} := (\rho(s,a)) \quad \& \quad \hat{\mathbf{p}} := (\hat{\rho}(s,a)).$$

By Lemma A.3 of [26], we have

$$\mathbb{P}(\Omega_0^c) = \mathbb{P}(\|\hat{\mathbf{p}} - \mathbf{p}\|_\infty > \frac{1}{2} p_{\min}) \leq \mathbb{P}(\|\hat{\mathbf{p}} - \mathbf{p}\| > \frac{1}{2} p_{\min}) \leq \exp(\frac{1}{4}) \cdot \exp\left(\frac{-n}{128} \cdot p_{\min}^2\right).$$

Note that conditioning on $\Omega_0$ has no effect on the randomness of $r, s' \sim \tilde{p}(\cdots|s, a)$ for each $s, a$. Denote its conditional expectation and probability as $\tilde{\mathbb{E}}(\cdots) := \mathbb{E}(\cdots|\Omega_0)$ and $\tilde{\mathbb{P}}(\cdots) := \mathbb{P}(\cdots|\Omega_0)$. Based on the IPS Properties of Definition C.2 and (11), we obtain the following for a fixed $s, a$, assuming that we have collected $\{(s, a, r_i, s_i')\}_{i=1}^{n(s,a)}$ for the $s, a$. We acknowledge that this proof is highly based on Lemma 10 of [2].

$$\tilde{\mathbb{E}}\left[ m^2 \left\{ \Upsilon_{\theta_{*,t}}(s,a), \hat{\mathcal{T}}^\pi \Upsilon_{\hat{\theta}_{n,t-1}}(s,a) \right\} \right] = \tilde{\mathbb{E}}\left[ m^2 \left\{ \Upsilon_{\theta_{*,t}}(s,a), \frac{1}{n(s,a)} \sum_{i=1}^{n(s,a)} \Psi_\pi(r_i, s_i', \Upsilon_{\hat{\theta}_{n,t-1}}) \right\} \right]$$

$$= \tilde{\mathbb{E}}\left[ \left\| G(\cdot|\Upsilon_{\theta_{*,t}}(s,a)) - G\left( \cdot \left| \frac{1}{n(s,a)} \sum_{i=1}^{n(s,a)} \Psi_\pi(r_i, s_i', \Upsilon_{\hat{\theta}_{n,t-1}}) \right. \right) \right\|_m^2 \right]$$

$$= \tilde{\mathbb{E}}\left[ \left\| \frac{1}{n(s,a)} \sum_{i=1}^{n(s,a)} \left\{ G\left( \cdot \left| \Psi_\pi(r_i, s_i', \Upsilon_{\hat{\theta}_{n,t-1}}) \right. \right) - G(\cdot|\Upsilon_{\theta_{*,t}}(s,a)) \right\} \right\|_m^2 \right]$$

$$
= \frac{1}{n(s,a)^2} \sum_{i=1}^{n(s,a)} \tilde{\mathbb{E}}\bigg[\Big\|G\big(\cdot|\Psi_\pi(r_i,s_i',\Upsilon_{\hat{\theta}_{n,t-1}})\big) - G(\cdot|\Upsilon_{\theta_{*,t}}(s,a))\Big\|_m^2\bigg]
$$

$$
+ \frac{1}{n(s,a)^2} \cdot \sum_{i \neq j} \tilde{\mathbb{E}}\bigg[\Big\langle G\big(\cdot|\Psi_\pi(r_i,s_i',\Upsilon_{\hat{\theta}_{n,t-1}})\big) - G(\cdot|\Upsilon_{\theta_{*,t}}(s,a)),
$$

$$
G\big(\cdot|\Psi_\pi(r_j,s_j',\Upsilon_{\hat{\theta}_{n,t-1}})\big) - G(\cdot|\Upsilon_{\theta_{*,t}}(s,a))\Big\rangle_m\bigg]
$$

$$
= \frac{1}{n(s,a)^2} \sum_{i=1}^{n(s,a)} \bigg\{ \tilde{\mathbb{E}}\big\|G\big(\cdot|\Psi_\pi(r_i,s_i,\Upsilon_{\hat{\theta}_{n,t-1}})\big)\big\|_m^2 + \mathbb{E}\|G(\cdot|\Upsilon_{\theta_{*,t}}(s,a))\|_m^2
$$

$$
- 2 \cdot \tilde{\mathbb{E}}\big\{\big\langle G\big(\cdot|\Psi_\pi(r_i,s_i',\Upsilon_{\hat{\theta}_{n,t-1}}),G(\cdot|\Upsilon_{\theta_{*,t}}(s,a))\big)\big\rangle_m\big\}\bigg\}
$$

$$
\leq \frac{1}{n(s,a)^2} \sum_{i=1}^{n(s,a)} \bigg\{ \tilde{\mathbb{E}}\big\|G\big(\cdot|\Psi_\pi(r_i,s_i',\Upsilon_{\hat{\theta}_{n,t-1}})\big)\big\|_m^2\bigg\} \quad \because \Upsilon_{\theta_{*,t}}(s,a) = \mathcal{T}^\pi \Upsilon_{\hat{\theta}_{n,t-1}}(s,a)
$$

$$
\leq \frac{C_{max,m}}{n(s,a)} \leq \frac{1}{n} \cdot \frac{2}{p_{\min}} \cdot C_{max,m} \quad \because n(s,a) = n \cdot \hat{\rho}(s,a) > n \cdot \frac{p_{\min}}{2} \quad \text{under } \Omega_0.
$$

Then, using $\mathbb{E}(X) \leq \{\mathbb{E}(X^2)\}^{1/2}$, we have

$$
\tilde{\mathbb{E}}\Big[m\big\{\Upsilon_{\theta_{*,t}}(s,a), \hat{\mathcal{T}}^\pi \Upsilon_{\hat{\theta}_{n,t-1}}(s,a)\big\}\Big] \leq \frac{1}{\sqrt{n}} \cdot \sqrt{\frac{2}{p_{\min}}} \cdot C_{max,m}^{1/2}.
$$

Temporarily making notation simplification $z_i = (r_i, s_i')$, we define

$$
\phi(z_1, \cdots, z_{n(s,a)}) := m\bigg\{\Upsilon_{\theta_{*,t}}(s,a), \frac{1}{n(s,a)} \sum_{i=1}^{n(s,a)} \Psi_i\bigg\} \quad \text{where } \Psi_i := \Psi_\pi(r_i, s_i', \Upsilon_{\hat{\theta}_{n,t-1}}).
$$

Since we have following where $\Psi_j' = \Psi_j$ for $j \neq i$ and $\Psi_i' = \Psi(z_i'; \Upsilon_{\hat{\theta}_{n,t-1}}, \pi)$,

$$
\big|\phi(z_1, \cdots, z_i, \cdots, z_{n(s,a)}) - \phi(z_1. \cdots, z_i', \cdots, z_{n(s,a)})\big|
$$

$$
= \bigg|m\bigg\{\Upsilon_{\theta_{*,t}}(s,a), \frac{1}{n(s,a)} \sum_{i=1}^{n(s,a)} \Psi_i\bigg\} - m\bigg\{\Upsilon_{\theta_{*,t}}(s,a), \frac{1}{n(s,a)} \sum_{i=1}^{n(s,a)} \Psi_i'\bigg\}\bigg|
$$

$$
= \bigg\|\frac{1}{n(s,a)} \sum_{i=1}^{n(s,a)} G(\cdot|\Psi_i) - \frac{1}{n(s,a)} \sum_{i=1}^{n(s,a)} G(\cdot|\Psi_i')\bigg\|_m \leq \frac{1}{n(s,a)} \sum_{i=1}^{n(s,a)} \Big\|G(\cdot|\Psi_i) - G(\cdot|\Psi_i')\Big\|_m
$$

$$
\leq \frac{2}{n(s,a)} \cdot C_{max,m}.
$$

Then, by McDiarmid's inequality (e.g., Theorem 2.9.1 of [57]), we obtain

$$
\tilde{\mathbb{P}}\bigg(\Big|m\big\{\Upsilon_{\theta_{*,t}}(s,a), \hat{\mathcal{T}}^\pi \Upsilon_{\theta_{*,t}}(s.a)\big\}\Big| \geq t\bigg) \leq 2 \cdot \exp\bigg(\frac{-2t^2}{\frac{1}{n(s,a)} \cdot 4C_{max,m}^2}\bigg)
$$

$$
\leq 2 \cdot \exp\bigg(\frac{-n \cdot p_{\min} \cdot t^2}{4C_{max,m}^2}\bigg).
$$

Now considering the conditional probabilty for $\Omega_0$, we have

$$
\sup_{s,a} m\big\{\Upsilon_{\theta_{*,t}}(s,a), \hat{\mathcal{T}}^\pi \Upsilon_{\hat{\theta}_{n,t-1}}(s,a)\big\} \leq \frac{1}{\sqrt{n}} \cdot \sqrt{\frac{2}{p_{\min}}} \cdot C_{max,m}^{1/2} + t \quad \text{with probability larger than}
$$

$$
1 - \exp(\tfrac{1}{4}) \cdot \exp(\tfrac{-n}{128} \cdot p_{\min}^2) - |\mathcal{S} \times \mathcal{A}| \cdot 2 \cdot \exp\bigg(\frac{-n \cdot p_{\min}^2 \cdot t^2}{4C_{max,m}^2}\bigg).
$$

Letting $t^2 = \frac{4C_{max,m}^2}{n \cdot p_{\min}} \cdot \log(\frac{4|\mathcal{S} \times \mathcal{A}|}{\delta_0})$ for arbitrary $\delta_0 \in (0,1)$, given that we have sufficiently large $n$ such that

$$
\exp(\tfrac{1}{4}) \cdot \exp(\tfrac{-n}{128} \cdot p_{\min}^2) \leq \tfrac{1}{2}\delta_0, \tag{20}
$$

we have following with probability larger than $1 - \delta_0$,

$$\sup_{s,a} m\{\Upsilon_{\theta_{*,t}}(s,a), \hat{\mathcal{T}}^\pi \Upsilon_{\hat{\theta}_{n,t-1}}(s,a)\} \leq \frac{1}{\sqrt{p_{\min}}} \cdot \max\{C_{max,m}, 1\} \cdot O\left\{\sqrt{\frac{1}{n} \cdot \log\left(\frac{4|\mathcal{S} \times \mathcal{A}|}{\delta_0}\right)}\right\}.$$

Note that we have following under $\Omega_0$ that leads to $\rho(s,a) \leq 2\hat{\rho}(s,a)$,

$$m_\infty^2(\hat{\theta}_{n,t}, \theta_{*,t}) \leq \sum_{s,a} \frac{\rho(s,a)}{p_{\min}} \cdot m^2\{\Upsilon_{\hat{\theta}_{n,t}}(s,a), \Upsilon_{\theta_{*,t}}(s,a)\} = \frac{1}{p_{\min}} \cdot \bar{m}_{\rho,1}^2(\hat{\theta}_{n,t}, \theta_{*,t}),$$

$$\bar{m}_{\rho,1}^2(\hat{\theta}_{n,t}, \theta_{*,t}) \leq \tilde{\mathbb{E}}_{s,a\sim\rho}\{m^2(\Upsilon_{\hat{\theta}_{n,t}}(s,a), \Upsilon_{\theta_{*,t}}(s,a))\}$$

$$\leq 2 \cdot \tilde{\mathbb{E}}_{s,a,\sim\hat{\rho}}\{m^2(\Upsilon_{\hat{\theta}_{n,t}}(s,a), \Upsilon_{\theta_{*,t}}(s,a))\} = 2 \cdot \bar{m}_{n,1}^2(\hat{\theta}_{n,t}, \theta_{*,t})$$

$$\leq 8 \cdot \sup_{s,a} m^2\{\Upsilon_{\theta_{*,t}}(s.a), \hat{\mathcal{T}}^\pi \Upsilon_{\hat{\theta}_{n,t-1}}(s,a)\} \quad \text{by (18)},$$

$$\therefore \; m_\infty(\hat{\theta}_{n,t}, \theta_{*,t}) \leq \sqrt{\frac{8}{p_{\min}}} \cdot \sup_{s,a} m\{\Upsilon_{\theta_{*,t}}(s,a), \hat{\mathcal{T}}^\pi \Upsilon_{\hat{\theta}_{n,t-1}}(s,a)\}.$$

Then, we have following with probability larger than $1 - \delta_0$,

$$m_\infty(\hat{\theta}_{n,t}, \theta_{*,t}) \leq \frac{1}{p_{\min}} \cdot \max\{C_{max,m}, 1\} \cdot O\left\{\sqrt{\frac{1}{n} \cdot \log\left(\frac{4|\mathcal{S} \times \mathcal{A}|}{\delta_0}\right)}\right\}.$$

Under Assumption 3.2, we have $\eta_\infty(\hat{\theta}_{n,t}, \theta_{*,t}) \leq C_{surr} \cdot m_\infty^\delta(\hat{\theta}_{n,t}, \theta_{*,t}) + \epsilon_0$, which allows us to bound $\eta_\infty(\Upsilon_{\theta_t}, \mathcal{T}^\pi \Upsilon_{\hat{\theta}_{n,t-1}})$ for all $t \in [T]$. Now letting $n = \frac{N}{T}$, we have following with probability larger than $1 - T \cdot \delta_0$, the inaccuracy $\eta_\infty(\Upsilon_{\theta_T}, \Upsilon_\pi)$ can be bounded by following based on (6),

$$\frac{1}{1-\zeta} \cdot C_{surr} \cdot \frac{1}{1-\zeta} \cdot C_{surr} \cdot \left\{\left(\frac{1}{p_{\min}} \cdot \max\{C_{max,m}, 1\}\right)^\delta \times\right.$$

$$O\left[\left(\frac{1}{N/T} \cdot \log\left(\frac{4|\mathcal{S} \times \mathcal{A}|}{\delta_0}\right)\right)^{\delta/2}\right]\right\} + \frac{1}{1-\zeta} \cdot \epsilon_0 + \zeta^T \cdot \eta_\infty(\Upsilon_{\theta_0}, \Upsilon_\pi) \quad \text{with} \quad \zeta = \gamma^c \quad \text{by (7)}.$$

under the premise that (20) holds with $n = N/T$. This is possible since different subdatasets $(\mathcal{D}_1, \cdots, \mathcal{D}_T)$ are non-overlapping and independently collected.

Now we choose $T = \lfloor \frac{\delta}{2} \cdot \frac{1}{c} \cdot \log_{1/\gamma}\left(\frac{N}{\log(4|\mathcal{S} \times \mathcal{A}|/\delta_0)}\right)\rfloor$, which also leads to $\zeta^T \approx \left(\frac{1}{N} \cdot \log\left(\frac{4|\mathcal{S} \times \mathcal{A}|}{\delta_0}\right)\right)^{\delta/2}$. For sufficiently large $N$, (20) holds with $n = N/T$ and $\delta_0$ replaced with $\delta_0/T$ for sufficiently large $N$. Note that we have

$$\frac{1}{N/T} \cdot \log\left(\frac{4|\mathcal{S} \times \mathcal{A}|}{\delta_0/T}\right) \approx \frac{\delta}{2N} \cdot \frac{1}{c} \cdot \log_{1/\gamma}\left(\frac{N}{\log(4|\mathcal{S} \times \mathcal{A}|/\delta_0)}\right) \cdot \left\{\log\left(\frac{4|\mathcal{S} \times \mathcal{A}|}{\delta_0}\right) + \log T\right\}$$

$$\leq \frac{\delta}{2N} \cdot \frac{1}{c} \cdot \log_{1/\gamma} N \cdot \left\{\log\left(\frac{4|\mathcal{S} \times \mathcal{A}|}{\delta_0}\right) + \log\left(\frac{\delta}{2} \cdot \frac{1}{c} \cdot \log_{1/\gamma} N\right)\right\}.$$

Thus, with probability larger than $1 - \delta_0$, we have

$$\eta_\infty(\Upsilon_{\theta_T}, \Upsilon_\pi) \leq \frac{1}{1-\gamma^c} \cdot C_{surr} \cdot \left(\frac{1}{p_{\min}} \cdot \max\{C_{max,m}, 1\}\right)^\delta \times \tag{21}$$

$$O\left(\left[\frac{\delta}{N} \cdot \frac{1}{c} \cdot \log_{1/\gamma} N \cdot \left\{\log\left(\frac{4|\mathcal{S} \times \mathcal{A}|}{\delta_0}\right) + \log\left(\frac{\delta}{c} \cdot \log_{1/\gamma} N\right)\right\}\right]^{\delta/2}\right) + \frac{1}{1-\gamma^c} \cdot \epsilon_0.$$

### C.7 Parameters of Examples in Table 1

For all other examples except for the first two examples of Table 1, the values of $C_{surr}$ and $C_{max,m}$ are the same with what we have derived in the proofs of corollaries of Appendix B. For their values, refer to the proof of each corollary of its subsections (which are shown in Appendices C.10–C.16).

Here, we will only present the first two examples of Table 1, whose values of $C_{surr}$ and $C_{max,m}$ are different from those of Appendix B.

For $m = l_2$, we need to assume that return distributions are 1-dimension and bounded, i.e., $Z(s,a;\theta) \in [a,b]$ for all $s,a \in \mathcal{S} \times \mathcal{A}$ and $\theta \in \Theta$. By Lemma C.8 and $\mathcal{E} = 2l_2^2$, we have $C_{surr} = (b-a)^{1-\frac{1}{2p}}$ and $\delta = 1/p$,

$$\frac{2}{(b-a)^{2p-1}} \cdot \mathbb{W}_p^{2p}(P,Q) \leq 2 \cdot l_2^2(P,Q), \quad \therefore \left(\frac{1}{b-a}\right)^{1-\frac{1}{2p}} \cdot \mathbb{W}_p(P,Q) \leq l_2^{1/p}(P,Q).$$

We also have $C_{max,m} = C \cdot (b-a)^{1/2}$.

For $m = \mathrm{MMD}_\beta$, we need to assume $D < \infty$ (where $D > 0$ is the diameter of support). Since the corresponding kernel can be considered as $k(\mathbf{x},\mathbf{y}) = -\|\mathbf{x}-\mathbf{y}\|^\beta$, we have $C_{max,m} = C \cdot D^{\beta/2}$. We have $C_{surr} = 1$ and $\delta = 1$ since $\eta = m$.

## C.8 Theorem C.5 and its proof

Now we shall present and prove the non-simplified version of Theorem 3.4. For our analysis, we define the population objective function and the target parameter at each iteration:

$$F_{n,t}(\theta|\hat{\theta}_{n,t-1}) := \frac{1}{n} \sum_{(s,a)\in\mathcal{Z}_t} \mathbb{E}_{R,S'\sim\tilde{p}(\cdot|s,a)} \left\{ \mathfrak{d}\Big(\Upsilon_\theta(s,a), \Psi_\pi(R,S',\Upsilon_{\hat{\theta}_{n,t-1}})\Big) \right\}. \tag{22}$$

where $\mathcal{Z}_t := \{(s,a) \mid (s,a,r,s') \in \mathcal{D}_t\}$. Note that we have $\theta_{*,t} = \arg\min_{\theta\in\Theta} F_{n,t}(\theta|\hat{\theta}_{n,t-1})$. Like Section 3.1, our objective functions should have connection with the surrogate metric $m$, although $\mathfrak{d}$ may not necessarily be the squared form of $m$ (i.e., $\mathfrak{d} \neq m^2$). With $m$ being a probability metric for $\mathcal{P}$ and extended metrics $\bar{m}_{\rho,1}, \bar{m}_{n,1}, m_\infty$ for $\mathcal{M} \subseteq \mathcal{P}^{\mathcal{S}\times\mathcal{A}}$ (defined in (24)), they should satisfy following Assumption C.3 with $\|\cdot\|_{\psi_2(n)}$ being $\mathcal{Z}_t$-conditioned subgaussian norm (defined in (30) of Appendix C.10).

**Assumption C.3** (Norm-space association). The functional Bregman divergence $\mathfrak{d}$ and probability metric $m$ over $\mathcal{P}$ form a norm-space association (NS association, in short) if there exists an norm space $(\mathcal{G}, \|\cdot\|_m)$ where any element $\mu \in \mathcal{P}$ can be uniquely (up to equivalence under zero norm $\|\cdot\|_m$) represented by the corresponding element $G(\cdot|\mu) \in \mathcal{G}$ such that

(NS1) The $\mathcal{Z}_t$-conditioned modulus $\Delta_n(\theta|\hat{\theta}_{n,t-1}) := \{\hat{F}_{n,t}(\theta|\hat{\theta}_{n,t-1}) - F_{n,t}(\theta|\hat{\theta}_{n,t-1})\} - \{\hat{F}_{n,t}(\theta_{*,t}|\hat{\theta}_{n,t-1}) - F_{n,t}(\theta_{*,t}|\hat{\theta}_{n,t-1})\}$ is a mean-zero sub-gaussian process with respect to $\bar{m}_{n,1}$. That is, for any $\mathcal{Z}_t \subset \mathcal{S} \times \mathcal{A}$ and $\hat{\theta}_{n,t-1} \in \Theta$, we have

$$\|\Delta_n(\theta_1|\hat{\theta}_{n,t-1}) - \Delta_n(\theta_2|\hat{\theta}_{n,t-1})\|_{\psi_2(n)} \leq \frac{1}{\sqrt{n}} \cdot C_{subg} \cdot \bar{m}_{n,1}(\theta_1,\theta_2).$$

(NS2) We have $F_{n,t}(\theta|\hat{\theta}_{n,t-1}) - F_{n,t}(\theta_{*,t}|\hat{\theta}_{n,t-1}) \geq \lambda \cdot \bar{m}_{n,1}^l(\theta,\theta_{*,t})$ for some $l \geq 2$. Here, the value of $\lambda > 0$ does not depend on $\mathcal{Z}_t, \theta, \hat{\theta}_{n,t-1}$ (or $\theta_{*,t}$ such that $\Upsilon_{\theta_{*,t}} = \mathcal{T}^\pi \Upsilon_{\hat{\theta}_{n,t-1}}$).

(NS3) $\bar{m}_{n,1}(\hat{\theta}_{n,t}, \theta_{*,t}) \to^P 0$ converges in a rate that does not depend on $\hat{\theta}_{n,t-1}$ (and thereby $\theta_{*,t}$) and the deterministic sequence $\mathcal{Z}_t$.

(NS4) Elements of $\mathcal{M}_\Theta \subseteq \mathcal{P}^{\mathcal{S}\times\mathcal{A}}$ has a corresponding element in $\mathcal{G}_\Theta \subseteq \mathcal{G}^{\mathcal{S}\times\mathcal{A}}$ where $\mathcal{G}$ is a functional norm space with $\|\cdot\|_m$, such that $\|G_{\mu_1} - G_{\mu_2}\|_m := m(\mu_1,\mu_2)$. With the extended norm $\|\cdot\|_{m,\infty}$ corresponding to $m_\infty$ (see (24)) and some $\alpha \in (0,2)$, it satisfies

$$\sup_{\theta\in\Theta} \sup_{z,s,a} |G_\theta(z|s,a)| < \infty \quad \& \quad \log\mathcal{N}_{[\,]}(\mathcal{G}_\Theta, \|\cdot\|_{m,\infty}, \epsilon) \leq C_{brack} \cdot \epsilon^{-\alpha}. \tag{23}$$

It is possible that there exists an element in $\mathcal{G}$ that does not represent any $\mu \in \mathcal{P}$.

Under coverage assumption (Assumption C.4), we can obtain Theorem C.5, with its proof in the subsections of Appendix C.8.

**Assumption C.4.** (Coverage) $C_{cover} := \sup_{\theta,\tilde{\theta}\in\Theta} \bar{m}^2_{d_\pi,1}(\Upsilon_\theta, \mathcal{T}^\pi \Upsilon_{\tilde{\theta}})/\bar{m}^2_{\rho,1}(\Upsilon_\theta, \mathcal{T}^\pi \Upsilon_{\tilde{\theta}}) < \infty$.

**Theorem C.5.** *Suppose Assumptions 3.1, 3.2, C.4 hold. Moreover, given a probability metric $\eta$ that satisfies (S-L-C) with convexity parameter $q \geq 1$ and scale-sensitivity parameter $c > 1/(2q)$ (i.e., (15) of Appendix C.4), consider the FDE with a functional Bregman divergence $\mathfrak{d}$ satisfying Assumption C.3 of Appendix C.8. Then, by letting $T = \lfloor \frac{1}{c-\frac{1}{2q}} \cdot \frac{\delta'}{2(l-1)+\alpha} \cdot \log_{1/\gamma} N \rfloor$ with $\delta' = \min\{\delta, 1/q\}$, we have*

$$\bar{\eta}_{d_\pi,q}(\Upsilon_{\theta_T}, \Upsilon_\pi) \leq \frac{1}{1-\gamma^{c-\frac{1}{2q}}} \cdot C_0 \cdot O_P\left\{ \left( \frac{N}{\log_{1/\gamma} N} \right)^{\frac{-\delta'}{2(l-1)+\alpha}} \right\} + \frac{1}{1-\gamma^{c-\frac{1}{2q}}} \cdot 2^{1-\frac{1}{2q}} \cdot \epsilon_0,$$

*where $C_0 > 0$ is a constant that contains the effect of the parameters $C_{surr}, C_{cover}, C_{brack}, C_{subg}$, etc (defined in (27) of Appendix C.8). Note that we need $q > 1/(2c)$ to ensure $\gamma^{c-\frac{1}{2q}} \in (0,1)$.*

### C.8.1 Outline of proof

The first step (Appendix C.8.2) is to obtain the convergence of single iteration at $t$-th step. Temporarily assuming that the state-action pairs $\mathcal{Z}_t$ are given (i.e., fixed or conditioned), we obtain the convergence with respect to the semi-metric $\bar{m}_{n,1}$ (24) that is based on $\mathcal{Z}_t$. Towards that end, we resort to standard M-estimation theory (Theorem D.1) Unlike the tabular case (Theorem 3.3), the model complexity plays a role in the convergence rate. Then, we can obtain the same convergence rate in (sample-free metric) $\bar{m}_{\rho,1}$, based on the fact that the probability metric $m$ has a corresponding functional norm space (see Theorem 2.3 of [34]).

The second step (Appendix C.8.3) is to combine the convergences of multiple iterations into the final bound for the $T$-th iteration, just like we did in Theorem 3.3 (Appendix C.6). So, it shares many similar ideas. However, unlike the previous Theorem 3.3, this requires us to handle the coverage. The inaccuracy metric $\bar{\eta}_{d_\pi,q}$ is constructed based on $d_\pi$, whereas the collected state-action pairs are generated by $\rho$. We need to take into account the misalignment between the two underlying distributions, quantified by $C_{cover}$ (Assumption C.4).

### C.8.2 Convergence in a single step

**Lemma C.6.** *Assume that our objective functions* (10) *and* (22) *and $\eta$ satisfy Assumption C.3. Conditioning on any given value of $\hat{\theta}_{n,t-1}$ (previous iteration value), under Assumptions 3.1, C.4, we have*

$$\bar{m}_{d_\pi,1}(\Upsilon_{\hat{\theta}_{n,t}}, \mathcal{T}^\pi \Upsilon_{\hat{\theta}_{n,t-1}}) \leq C^{1/2}_{cover} \cdot \left( C_{subg} \cdot C^{1/2}_{brack} \cdot \frac{\sqrt{2}^{-\alpha}}{1-\alpha/2} \right)^{\frac{1}{l-1+\alpha/2}} \cdot O_P\left( n^{\frac{-1}{2(l-1)+\alpha}} \right).$$

The proof of Lemma C.6 is as follows. Let $\mathcal{D}_t : \{(s_i, a_i, r_i, s'_i)\}^n_{i=1}$.

With the norm representation of the surrogate metric $m$, i.e., $m(\mu_1, \mu_2) := \|G_{\mu_1}(\cdot) - G_{\mu_2}(\cdot)\|_m$, then we can build its extension as follows:

empirical norm : $\bar{m}_{n,1}(\theta_1, \theta_2) = \|G_{\theta_1} - G_{\theta_2}\|_{m,n} := \left( \frac{1}{n} \sum_{i=1}^n \|G_{\theta_1}(\cdot|s_i, a_i) - G_{\theta_2}(\cdot|s_i, a_i)\|^2_m \right)^{\frac{1}{2}}$

expectation norm : $\bar{m}_{\rho,1}(\theta_1, \theta_2) = \|G_{\theta_1} - G_{\theta_2}\|_{m,\rho} = \left( \mathbb{E}_{s,a\sim\rho}\|G_{\theta_1}(\cdot|s, a) - G_{\theta_2}(\cdot|s, a)\|^2_m \right)^{\frac{1}{2}}$

supremum norm : $m_\infty(\theta_1, \theta_2) = \|G_{\theta_1} - G_{\theta_2}\|_{m,\infty} := \sup_{s,a} \|G_{\theta_1}(\cdot|s, a) - G_{\theta_2}(\cdot|s, a)\|_m.$  (24)

Define $\Theta^\delta_n := \{\theta \in \Theta : \bar{m}_{n,1}(\theta, \theta_{*,t}) \leq \delta\}$. By (NS1), we can apply Theorem 8.1.3 of [57] to obtain

$$\mathbb{E}\left\{ \sup_{\theta\in\Theta^\delta_n} \left| \Delta_n(\theta|\hat{\theta}_{n,t-1}) \right| \right\} \leq \frac{C}{\sqrt{n}} \cdot C_{subg} \cdot \int_0^\infty \sqrt{\log \mathcal{N}(\Theta^\delta_n, \bar{m}_{n,1}, \epsilon)} d\epsilon$$

$$\leq \frac{C}{\sqrt{n}} \cdot C_{subg} \cdot \int_0^\delta \sqrt{\log \mathcal{N}(\Theta^\delta_n, \bar{m}_{n,1}, \epsilon)} d\epsilon \leq \frac{C}{\sqrt{n}} \cdot C_{subg} \cdot \int_0^\delta \sqrt{\log \mathcal{N}(\Theta, m_\infty, \epsilon)} d\epsilon.$$

Let $C_{bound} := \sup_{\theta \in \Theta} \sup_{z,s,a} |G_\theta(z|s,a)| \in (0, \infty)$. Here, we let $G'_\theta(\cdot|\cdots) := \frac{1}{C_{bound}} \cdot G_\theta(\cdot|\cdots)$ and $\mathcal{G}'_\Theta := \{G'_\theta | G_\theta \in \mathcal{G}_\Theta\}$. Note that we have $G'_\theta(z|s,a) \leq 1$ and

$$\log \mathcal{N}_{[\,]}(\mathcal{G}'_\Theta, \|\cdot\|_{m,\infty}, \epsilon) = \log \mathcal{N}_{[\,]}(\mathcal{G}_\Theta, \|\cdot\|_{m,\infty}, C_{bound} \cdot \epsilon) \leq C_{brack} \cdot C_{bound}^{-\alpha} \cdot \epsilon^{-\alpha}. \quad (25)$$

Let $\bar{m}'_{n,1} := \frac{1}{C_{bound}} \cdot \bar{m}_{n,1}$, $\bar{m}'_{\rho,1} = \frac{1}{C_{bound}} \cdot \bar{m}_{\rho,1}$, $m'_\infty := \frac{1}{C_{bound}} \cdot m_\infty$, and we have $\bar{m}'_{n,1}(\theta_1, \theta_2) := \|G'_{\theta_1} - G'_{\theta_2}\|_{m,n}$, $\bar{m}'_{\rho,1}(\theta_1, \theta_2) := \|G'_{\theta_1} - G'_{\theta_2}\|_{m,\rho}$, $m'_\infty(\theta_1, \theta_2) := \|G'_{\theta_1} - G'_{\theta_2}\|_{m,\infty}$.

This leads to

$$\log \mathcal{N}(\Theta, m_\infty, \epsilon) \leq \log \mathcal{N}(\Theta, m'_\infty, \epsilon/C_{bound}) \leq \log \mathcal{N}(\mathcal{G}'_\Theta, \|\cdot\|_{m,\infty}, \epsilon/C_{bound})$$

$$\leq \log \mathcal{N}_{[\,]}(\mathcal{G}'_\Theta \leq \|\cdot\|_{m,\infty}, 2\epsilon/C_{bound}) \leq C_{brack} \cdot C_{bound}^{-\alpha} \cdot \left(\frac{2\epsilon}{C_{bound}}\right)^{-\alpha} \leq C_{brack} \cdot 2^{-\alpha} \cdot \epsilon^{-\alpha},$$

$$\therefore \mathbb{E}\left\{\sup_{\theta \in \Theta_n^\delta} |\Delta_n(\theta|\hat{\theta}_{n,t-1})|\right\} \leq \frac{C}{\sqrt{n}} \cdot C_{subg} \cdot C_{brack}^{1/2} \cdot \sqrt{2}^{-\alpha} \cdot \frac{1}{1-\alpha/2} \cdot \delta^{1-\alpha/2}.$$

We have consistency $\bar{m}_{n,1}(\hat{\theta}_{n,t}, \theta_{*,t}) \to^P 0$ by (NS3). Then, we can apply Theorem D.1 to obtain

$$\bar{m}_{n,1}(\hat{\theta}_{n,t}, \theta_{*,t}) \leq \left(C_{subg} \cdot C_{brack}^{1/2} \cdot \frac{\sqrt{2}^{-\alpha}}{1-\alpha/2}\right)^{\frac{1}{l-1+\alpha/2}} \cdot O_P\left(n^{\frac{-1}{2(l-1)+\alpha}}\right).$$

Using $\frac{1}{2+\alpha} \geq \frac{1}{2(l-1)+\alpha}$ due to $l \geq 2$, we can apply logic of Theorem 6.3.2 of [54] (or equivalently, Theorem 2.3 of [34]), based on $G'_\theta(z|s,a) \leq 1$ and (25). This gives us the same asymptotic bound for $\bar{m}_{\rho,1}(\hat{\theta}_{n,t}, \theta_{*,t})$, and further applying Assumption C.4 gives us Lemma C.6.

### C.8.3 Convergence through multiple steps

Now let us bound $\bar{\eta}_{d_\pi,q}(\hat{\theta}_{n,t}, \theta_{*,t})$ with $q \geq 1$, with Assumption 3.2. We use $(a+b)^{2q} \leq 2^{2q-1} \cdot (a^{2q} + b^{2q})$ to obtain the following with $q' \stackrel{let}{=} q\delta$:

$$\bar{\eta}_{d_\pi,q}(\hat{\theta}_{n,t}, \theta_{*,t}) = \left(\mathbb{E}\{\eta^{2q}(\Upsilon_{\hat{\theta}_{n,t}}(s,a), \mathcal{T}^\pi \Upsilon_{\hat{\theta}_{n,t-1}}(s,a))\}\right)^{\frac{1}{2q}}$$

$$\leq \mathbb{E}\left[2^{2q-1} \cdot \left\{C_{surr}^{2q} \cdot m^{2q \cdot \delta}(\Upsilon_{\hat{\theta}_{n,t}}(s,a), \Upsilon_{\theta_{*,t}}(s,a)) + \epsilon_0^{2q}\right\}\right]^{\frac{1}{2q}}$$

$$\leq 2^{1-\frac{1}{2q}} \cdot C_{surr} \cdot \left(\mathbb{E}\{m^{2q\delta}(\Upsilon_{\hat{\theta}_{n,t}}(s,a), \mathcal{T}^\pi \Upsilon_{\hat{\theta}_{n,t-1}}(s,a))\}\right)^{\frac{1}{2q}} + 2^{1-\frac{1}{2q}} \cdot \epsilon_0$$

$$= 2C_{surr} \cdot \left\{\mathbb{E}\{m^{2q'}(\Upsilon_{\hat{\theta}_{n,t}}(s,a), \mathcal{T}^\pi \Upsilon_{\hat{\theta}_{n,t-1}}(s,a)\}^{\frac{1}{2q'}}\right\}^\delta + 2^{1-\frac{1}{2q}} \cdot \epsilon_0.$$

Assume a bounded random variable $X$ such that $|X| \leq C_{maxim}$. Since $|X/C_{maxim}| \leq 1$, we have the following if $1 \leq p_1 \leq p_2$.

$$\left(\mathbb{E}\left|\frac{X}{C_{maxim}}\right|^{p_2}\right)^{\frac{1}{p_2}} \leq \left(\mathbb{E}\left|\frac{X}{C_{maxim}}\right|^{p_1}\right)^{\frac{1}{p_2}} = \left\{\left(\mathbb{E}\left|\frac{X}{C_{maxim}}\right|^{p_1}\right)^{\frac{1}{p_1}}\right\}^{\frac{p_1}{p_2}},$$

$$\therefore (\mathbb{E}|X|^{p_2})^{1/p_2} \leq C_{maxim}^{1-\frac{p_1}{p_2}} \cdot \{(\mathbb{E}|X|^{p_1})^{1/p_1}\}^{p_1/p_2}.$$

If we have $1 < q'$, letting $p_1 = 2, p_2 = 2q'$ gives us

$$\bar{\eta}_{d_\pi,q}(\hat{\theta}_{n,t}, \theta_{*,t}) \leq 2C_{surr} \cdot \left\{C_{maxim}^{1-\frac{2}{2q'}} \cdot \left(\mathbb{E}\{m^2(\Upsilon_{\hat{\theta}_{n,t}}(s,a), \mathcal{T}^\pi \Upsilon_{\hat{\theta}_{n,t-1}}(s,a))\}^{\frac{1}{2}}\right)^{\frac{2}{2q'}}\right\}^\delta$$

$$+ 2^{1-\frac{1}{2q}} \cdot \epsilon_0$$

$$= 2C_{surr} \cdot C_{maxim}^{(1-\frac{1}{q'})\delta} \cdot \bar{m}_{d_\pi,1}^{\frac{1}{q}}(\hat{\theta}_{n,t}, \theta_{*,t}) + 2^{1-\frac{1}{2q}} \cdot \epsilon_0.$$

If we have $q' \leq 1$, then we have $\bar{\eta}_{d_\pi,q}(\hat{\theta}_{n,t}, \theta_{*,t}) \leq 2C_{surr} \cdot \bar{m}_{d_\pi,1}^\delta(\hat{\theta}_{n,t}, \theta_{*,t}) + 2^{1-\frac{1}{2q}} \cdot \epsilon_0$ by Jensen's inequality (even for $q' < 1$). Thus, we have following with $\delta' = \min\{\delta, 1/q\}$,

$$\bar{\eta}_{d_\pi,q}(\hat{\theta}_{n,t}, \theta_{*,t}) \leq 2C_{surr} \cdot C_{maxim}^{\max\{(1-\frac{1}{q\delta})\delta, 0\}} \cdot \bar{m}_{d_\pi,1}^{\delta'}(\hat{\theta}_{n,t}, \theta_{*,t}) + 2^{1-\frac{1}{2q}} \cdot \epsilon_0. \quad (26)$$

Restoring the iteration index $t \in [T]$ (i.e., $\theta_t = \hat{\theta}_{n,t}$), we have following by (6),

$$\bar{\eta}_{d_\pi,q}(\Upsilon_{\theta_T}, \Upsilon_\pi) \leq \sum_{t=1}^{T} \zeta^{T-t} \cdot \bar{\eta}_{d_\pi,q}(\Upsilon_{\theta_t}, \mathcal{T}^\pi \Upsilon_{\theta_{t-1}}) + \zeta^T \cdot \bar{\eta}_{d_\pi,q}(\Upsilon_{\theta_0}, \Upsilon_\pi)$$

$$\leq 2C_{surr} \cdot C_{maxim}^{\max\{(1-\frac{1}{q\delta})\delta, 0\}} \cdot \sum_{t=1}^{T} \zeta^{T-t} \cdot \bar{m}_{d_\pi,1}^{\delta'}(\Upsilon_{\theta_t}, \mathcal{T}^\pi \Upsilon_{\theta_{t-1}}) + \frac{1}{1 - \gamma^{c-\frac{1}{2q}}} \cdot 2^{1-\frac{1}{2q}} \cdot \epsilon_0$$

$$+ \zeta^T \cdot \bar{\eta}_{d_\pi,q}(\Upsilon_{\theta_0}, \Upsilon_\pi).$$

Recall that the data (which are independently collected) are split into non-overlapping subdatasets $\mathcal{D} = \cup_{t=1}^{T} \mathcal{D}_t$ (see Section 3). This ensures that $\hat{\theta}_{n,t-1}$ can be considered as a given constant at each $t$-th iteration.

**Lemma C.7.** *With $X_{n,t} = \tilde{C} \cdot O_P(n^{-1/b})$ for some $b > 0$ which can be possibly dependent for different $t \in \mathbb{N}$, assuming $\limsup_{n\to\infty} \sup_{t\in\mathbb{N}} \mathbb{E}|n^{1/b} \cdot X_{n,t}| < \infty$, then we have $\sum_{t=1}^{T} \zeta^{T-t} \cdot X_{n,t} = \tilde{C} \cdot O_P(\frac{1}{1-\zeta} \cdot n^{-1/b})$. See Appendix D.2 for proof.*

In our case, letting $X_{n,t} = \bar{m}_{d_\pi,1}^{\delta'}(\Upsilon_{\hat{\theta}_{n,t}}, \mathcal{T}^\pi \Upsilon_{\hat{\theta}_{n,t-1}})$ (where $n$ affects $\hat{\theta}_{n,t} = \arg\min_{\theta\in\Theta} \hat{F}_{n,t}(\theta|\hat{\theta}_{n,t-1})$), we can show that the condition of Lemma C.7 can be satisfied. We will defer the proof to Appendix C.8.4 for the sake of simplicity. Then, combining with Lemma C.6 and letting $n = N/T$ with $T = \lfloor \frac{1}{c-\frac{1}{2q}} \cdot \frac{\delta'}{2(l-1)+\alpha} \cdot \log_{1/\gamma} N \rfloor$, we have

$$\bar{\eta}_{d_\pi,q}(\Upsilon_{\hat{\theta}_{n,T}}, \Upsilon_\pi) \leq \frac{1}{1 - \gamma^{c-\frac{1}{2q}}} \cdot C_0 \cdot O_P\left\{ \left( \frac{N}{\log_{1/\gamma} N} \right)^{\frac{-\delta'}{2(l-1)+\alpha}} \right\} + \frac{1}{1 - \gamma^{c-\frac{1}{2q}}} \cdot 2^{1-\frac{1}{2q}} \cdot \epsilon_0,$$

where

$$C_0 := C_{surr} \cdot C_{maxim}^{\max\{(1-\frac{1}{q\delta})\delta, 0\}} \cdot C_{cover}^{\frac{\delta'}{2}} \times \tag{27}$$
$$\left\{ C_{subg} \cdot C_{brack}^{1/2} \cdot \left( \frac{1}{c-\frac{1}{2q}} \right)^{\frac{1}{2}} \cdot \frac{\sqrt{2}^{-\alpha}}{1-\alpha/2} \cdot \left( \frac{\delta'}{2(l-1)+\alpha} \right)^{\frac{1}{2}} \right\}^{\frac{\delta'}{l-1+\alpha/2}},$$

with $C_{maxim} := \sup_{\theta,\tilde{\theta}\in\Theta} m_\infty(\Upsilon_\theta, \mathcal{T}^\pi \Upsilon_{\tilde{\theta}})$. For $C_{maxim} = \infty$, we let $C_{maxim}^0 = 1$. This validates Theorem C.5.

### C.8.4 Satisfaction of the condition of Lemma C.7 in Theorem C.5

**Stage-1: Apply peeling argument** Arbitrarily choose an integer $M \in \mathbb{N}$ and a sequence $(\delta_n)_{\geq 1}$. we can use the logic of standard proof of convergence in M-estimation (e.g., Section 5.1 of [51]). Define $\mathcal{Z}_t := \{(s_i, a_i) \mid (s_i, a_i, r_i, s_i') \in \mathcal{D}_t\}$ and $S_{n,j} := \{\theta \in \Theta : 2^{j-1} \cdot \delta_n < \bar{m}_{n,1}(\theta, \theta_{*,t}) \leq 2^j \cdot \delta_n\}$, and we can obtain the following,

$$\mathbb{P}\left( \bar{m}_{n,1}(\hat{\theta}_{n,t}, \theta_{*,t}) > 2^{M-1} \cdot \delta_n \Big| \mathcal{Z}_t \right) = \sum_{j\geq M} \mathbb{P}\left( 2^{j-1} \cdot \delta_n < \bar{m}_{n,1}(\hat{\theta}_{n,t}, \theta_{*,t}) \leq 2^j \cdot \delta_n \Big| \mathcal{Z}_t \right)$$

$$\leq \sum_{j\geq M} \mathbb{P}\left( \sup_{\theta\in S_{n,j}} \left\{ -\hat{F}_{n,t}(\theta|\hat{\theta}_{n,t-1}) + \hat{F}_{n,t}(\theta_{*,t}|\hat{\theta}_{n,t-1}) \right\} \geq 0 \Big| \mathcal{Z}_t \right)$$

$$\leq \sum_{j\geq M} \mathbb{P}\left( \sup_{\bar{m}_{n,1}(\theta,\theta_{*,t})\leq 2^j\cdot\delta_n} \left| \Delta_n(\theta|\hat{\theta}_{n,t-1}) \right| \geq \lambda \cdot 2^{l(j-1)} \cdot \delta_n^l \Big| \mathcal{Z}_t \right) \quad \text{by (NS2)}$$

$$\leq \sum_{j\geq M} \frac{1}{\lambda \cdot 2^{l(j-1)} \cdot \delta_n^l} \cdot \mathbb{E}\left\{ \sup_{\bar{m}_{n,1}(\theta,\theta_{*,t})\leq 2^j\cdot\delta_n} \left| \Delta_n(\theta|\hat{\theta}_{n,t-1}) \right| \Big| \mathcal{Z}_t \right\}$$

$$\leq \frac{C}{\lambda} \cdot \frac{C_{subg}}{2^{-l}} \cdot C_{brack}^{1/2} \cdot \frac{\sqrt{2}^{-\alpha}}{1-\alpha/2} \cdot \frac{1}{\sqrt{n}} \cdot \delta_n^{1-\alpha/2-l} \cdot \sum_{j \geq M} (2^j)^{1-\alpha/2-l} \quad \text{by (NS4)}$$

$$= \underbrace{\frac{C}{\lambda} \cdot C_{subg} \cdot C_{brack}^{1/2} \cdot \frac{\sqrt{2}^{-\alpha}}{1-\alpha/2} \cdot \frac{2^l}{1-2^{1-\alpha/2-l}}}_{\overset{let}{=} C_2} \cdot \frac{1}{\sqrt{n}} \cdot (2^M \cdot \delta_n)^{1-\alpha/2-l}$$

$$= C_2 \cdot \frac{1}{\sqrt{n}} \cdot (2^M \cdot \delta_n)^{1-\alpha/2-l}.$$

For arbitrary $\epsilon > 0$, we can choose $M \in \mathbb{N}$ and $\delta_n > 0$ be such that $2^M \cdot \delta_n = \epsilon$. Then, we have

$$\mathbb{P}(\bar{m}_{n,1}(\hat{\theta}_{n,t}, \theta_{*,t}) > \epsilon | \mathcal{Z}_t) \leq C_2 \cdot \frac{1}{\sqrt{n}} \cdot \epsilon^{1-\alpha/2-l}.$$

Now letting $\epsilon^{1-\alpha/2-l} = \frac{1}{\sqrt{n}} \cdot \tilde{\epsilon}$ for arbitrary $\tilde{\epsilon} > 0$, we have following with $b = 2(l-1) + \alpha > 2$ and $C_2 > 0$ not depending on $\hat{\theta}_{n,t-1}$ (and thereby $\theta_{*,t}$),

$$\mathbb{P}\left( n^{1/b} \cdot \bar{m}_{n,1}(\hat{\theta}_{n,t}, \theta_{*,t}) > \tilde{\epsilon} \middle| \mathcal{Z}_t \right) \leq C_2 \cdot \tilde{\epsilon}^{-b/2}.$$

Based on the above tail probability, we can bound the expectation as follows,

$$\mathbb{E}\left\{ n^{1/b} \cdot \bar{m}_{n,1}(\hat{\theta}_{n,t}, \theta_{*,t}) \middle| \mathcal{Z}_t \right\} = \int_0^\infty \mathbb{P}\left( n^{1/b} \cdot \bar{m}_{n,1}(\hat{\theta}_{n,t}, \theta_{*,t}) \geq t \middle| \mathcal{Z}_t \right) \mathrm{d}t$$

$$= 1 + \int_1^\infty \mathbb{P}\left( n^{1/b} \cdot \bar{m}_{n,1}(\hat{\theta}_{n,t}, \theta_{*,t}) \geq t \middle| \mathcal{Z}_t \right) \mathrm{d}t \leq 1 + \frac{C_2}{b/2 - 1}. \tag{28}$$

Note that the bound does not depend on $\mathcal{Z}_t$.

**Stage-2: Convert the $\bar{m}_{n,1}$-bound into $\bar{m}_{\rho,1}$-bound** First fix $c \in (0, 1)$. Based on Lemma 6.3.4 of [54], we have $L_c, \alpha_c > 0$ (that do not depend on $\hat{\theta}_{n,t-1}$) such that following holds

$$\mathbb{P}\left( \sup_{\|G_{\theta_1} - G_{\theta_2}\|_{m,\rho} \geq L_c \cdot n^{-1/b}} \left| \frac{\|G_{\theta_1} - G_{\theta_2}\|_{m,n}}{\|G_{\theta_1} - \theta_2\|_{m,\rho}} - 1 \right| \leq c \right) \geq 1 - \frac{1}{\alpha_c} \cdot \exp(-\alpha_c \cdot L_c^2 \cdot n^{\frac{b-2}{b}}).$$

Denoting the event as $\Omega_E$ and note that this only regards to the observation of $\mathcal{Z}_t$ and not the $r_i, s_i' \sim \tilde{p}(\cdots | s_i, a_i)$. Then, we have

$$\mathbb{E}\{\bar{m}_{\rho,1}(\hat{\theta}_{n,t}, \theta_{*,t})\} = \mathbb{E}\left\{ \bar{m}_{\rho,1}(\hat{\theta}_{n,t}, \theta_{*,t}) \cdot 1\left( \bar{m}_{\rho,1}(\hat{\theta}_{n,t}, \theta_{*,t}) \leq L_c \cdot n^{-1/b} \right) \right\}$$

$$+ \mathbb{E}\left\{ \bar{m}_{\rho,1}(\hat{\theta}_{n,t}, \theta_{*,t}) \cdot 1\left( \bar{m}_{\rho,1}(\hat{\theta}_{n,t}, \theta_{*,t}) > L_c \cdot n^{-1/b} \right) \right\}$$

$$\leq L_c \cdot n^{-1/b} + \mathbb{E}\left\{ \bar{m}_{\rho,1}(\hat{\theta}_{n,t}, \theta_{*,t}) \cdot 1\left( \bar{m}_{\rho,1}(\hat{\theta}_{n,t}, \theta_{*,t}) > L_c \cdot n^{-1/b} \right) \middle| \Omega_E \right\} \cdot \mathbb{P}(\Omega_E)$$

$$+ \mathbb{E}\left\{ \bar{m}_{\rho,1}(\hat{\theta}_{n,t}, \theta_{*,t}) \cdot 1\left( \bar{m}_{\rho,1}(\hat{\theta}_{n,t}, \theta_{*,t}) > L_c \cdot n^{-1/b} \right) \middle| \Omega_E^c \right\} \cdot \mathbb{P}(\Omega_E^c)$$

$$\leq L_c \cdot n^{-1/b} + \mathbb{E}\left\{ \frac{1}{1-c} \cdot \bar{m}_{n,1}(\hat{\theta}_{n,t}, \theta_{*,t}) \middle| \Omega_E \right\} + C_{maxim} \cdot \exp(-\alpha_c \cdot L_c^2 \cdot n^{\frac{b-2}{b}})$$

$$\leq \left\{ L_c + \frac{1}{1-c} \cdot \left( 1 + \frac{C_2}{b/2-1} \right) + C_3 \right\} \cdot n^{-1/b}.$$

The last line holds due to (28), since $\Omega_E$ only regards to $\mathcal{Z}_t$. Also, $C_3 > 0$ is some large value such that $C_{maxim} \cdot \exp(-\alpha_c \cdot L_c^2 \cdot n^{\frac{b-2}{b}}) \leq C_3 \cdot n^{-1/b}$ holds for sufficiently large $n$.

**Stage-3: Showing the satisfaction** We can further derive the following.

$$\mathbb{E}\{\bar{m}_{d_\pi,1}^{\delta'}(\hat{\theta}_{n,t},\theta_{*,t})\} \leq C_{cover}^{\delta'/2} \cdot \mathbb{E}\{\bar{m}_{\rho,1}^{\delta'}(\hat{\theta}_{n,t},\theta_{*,t})\}$$

$$\leq C_{cover}^{\delta'/2} \cdot \left\{\mathbb{E}\{\bar{m}_{\rho,1}(\hat{\theta}_{n,t},\theta_{*,t})\}\right\}^{\delta'} \quad \text{by Jensen's inequality } (\because \ \delta' \leq 1)$$

$$\leq C_4 \cdot n^{-\delta'/b}$$

for some constant $C_4 > 0$ that does not depend on $n$ or $t$. Then, the condition of Lemma C.7 is satisfied with degree $\tilde{b} = \frac{2(l-1)+\alpha}{\delta'}$.

## C.9 About IPS metrics

### C.9.1 How squared IPS metric becomes a functional Bregman divergence

Assume an IPS metric (Definition C.2) where $G(\cdot|P)$ can be defined for dirac delta $P = \delta_{\mathbf{x}}$. This accommodates (i) $G(\cdot|P)$ being cdf with $\langle \cdot, \cdot \rangle_{L_2}$ and (ii) $G(\cdot|P)$ being kernel mean embedding with $\langle \cdot, \cdot \rangle_{\mathcal{H}}$. See Appendix C.5 for their detailed explanation. Define the score $S(P,\mathbf{x}) := -\|G(\cdot|P) - G(\cdot|\delta_{\mathbf{x}})\|_m^2$ where $\delta_{\mathbf{x}}$ denotes dirac delta at $\mathbf{x}$. Then we have

$$-\int S(P,\mathbf{x})\mathrm{d}Q(\mathbf{x}) = \int \|G(\cdot|P) - G(\cdot|\delta_{\mathbf{x}})\|_m^2 \mathrm{d}Q(\mathbf{x})$$

$$= \mathbb{E}_{\mathbf{x}\sim Q}\left\{\|G(\cdot|P)\|_m^2 + \|G(\cdot|\delta_{\mathbf{x}})\|_m^2 - 2 \cdot \langle G(\cdot|P), G(\cdot|\delta_{\mathbf{x}})\rangle_m\right\}$$

$$= \|G(\cdot|P)\|_m^2 + \mathbb{E}_{\mathbf{x}\sim Q}\|G(\cdot|\delta_{\mathbf{x}})\|_m^2 - 2 \cdot \langle G(\cdot|P), G(\cdot|Q)\rangle_m$$

Since we have $-\int S(Q,\mathbf{x})\mathrm{d}Q(\mathbf{x}) = -\|G(\cdot|Q)\|_m^2 + \mathbb{E}_{\mathbf{x}\sim Q}\|G(\cdot|\delta_{\mathbf{x}})\|_m^2$, this leads to a valid functional Bregman divergence as follows (based on equivalence stated before (5)),

$$m^2(P,Q) = \|G(\cdot|P) - G(\cdot|Q)\|_m^2 = \int S(Q,\mathbf{x})\mathrm{d}Q(\mathbf{x}) - \int S(P,\mathbf{x})\mathrm{d}Q(\mathbf{x}).$$

Of course, there are cases where $G(\cdot|P)$ does not exist for dirac delta's, most representatively $G(\cdot|P)$ being a pdf with $\langle \cdot, \cdot \rangle_m$. However, this case is shown to make a functional Bregman divergence (Section 4.1 of [23]).

### C.9.2 How IPS Properties imply NS Properties

Under (IPS1), (IPS2) of Definition C.2, and (11), we shall show $(\mathfrak{d}, m)$ with $\mathfrak{d} = m^2$ satisfies NS Properties of Section 3.2 with $l = 2$, $\lambda = 1$, and $C_{subg} = C_{max,m}$.

(NS1) can be shown as follows. With $A_i^\theta = m^2(\Upsilon_\theta(s_i,a_i), \Psi_\pi(R_i,S_i',\Upsilon_{\hat{\theta}_{n,t-1}}))$ where $\Psi_\pi(r,s',\Upsilon) := \int_{\mathcal{A}}(g_{\gamma,r})_\#\Upsilon(s',a')\pi(\mathrm{d}a'|s') \in \mathcal{P}$ (due to convexity of $\mathcal{P}$, provided that $\mathcal{P}$ satisfies the assumption of Theorem 2.4),

$$\therefore \ |\Delta_n(\theta_1|\hat{\theta}_{n,t-1}) - \Delta_n(\theta_2|\hat{\theta}_{n,t-1})| = \left|\frac{1}{n}\sum_{i=1}^n\{A_i^{\theta_1} - \mathbb{E}(A_i^{\theta_1})\} - \frac{1}{n}\sum_{i=1}^n\{A_i^{\theta_2} - \mathbb{E}(A_i^{\theta_2})\}\right|.$$

Since we have $\|\Delta_n(\theta_1|\hat{\theta}_{n,t-1}) - \Delta_n(\theta_2|\hat{\theta}_{n,t-1})\|_{\psi_2(n)}^2 \leq C \cdot \frac{1}{n^2}\sum_{i=1}^n\|A_i^{\theta_1} - A_i^{\theta_2}\|_{\psi_2(n)}^2$ by Proposition 2.6.1 and Lemma 2.6.8 of [57], we shall bound $|A_i^{\theta_1} - A_i^{\theta_2}|$.

$$|A_i^{\theta_1} - A_i^{\theta_2}| = \left|m^2\left\{\Upsilon_{\theta_1}(s_i,a_i), \Psi_\pi(r_i,s_i',\Upsilon_{\hat{\theta}_{n,t-1}})\right\} - m^2\left\{\Upsilon_{\theta_2}(s_i,a_i), \Psi_\pi(r_i,s_i',\Upsilon_{\hat{\theta}_{n,t-1}})\right\}\right|$$

$$\leq C \cdot C_{max,m} \cdot m(\Upsilon_{\theta_1}(s_i,a_i), \Upsilon_{\theta_2}(s_i,a_i)).$$

Here, $C_{max,m} > 0$ is a constant (that depends on the corresponding kernel $k$) such that

$$\sup_{\theta,\hat{\theta}_{n,t-1}\in\Theta} \ \sup_{s,a} \sup_{r,s'} \left\{m(\Upsilon_\theta(s,a), \Psi_\pi(r,s',\Upsilon_{\hat{\theta}_{n,t-1}}))\right\} \leq C_{max,m},$$

For $m = $ MMD bounded kernel $k(\cdot,\cdot)$, we can let $C_{max,m} \overset{let}{=} C \cdot \sup_{x,y\in\mathcal{X}} k^{1/2}(x,y)$ (29)

Thus we have

$$\|\Delta_n(\theta_1|\hat{\theta}_{n,t-1}) - \Delta_n(\theta_2|\hat{\theta}_{n,t-1})\|_{\psi_2(n)} \leq \frac{C_{subg}}{\sqrt{n}} \cdot \overline{m}_{n,1}(\theta_1,\theta_2) \quad \text{with} \quad C_{subg} = C \cdot C_{max,m}.$$

(NS2) can be shown as follows,

$$F_{n,t}(\theta|\hat{\theta}_{n,t-1}) - F_{n,t}(\theta_{*,t}|\hat{\theta}_{n,t-1})$$
$$= \frac{1}{n}\sum_{i=1}^{n}\mathbb{E}\left\{m^2\big(\Upsilon_\theta(s_i,a_i),\Psi_\pi(R_i,S_i',\Upsilon_{\hat{\theta}_{n,t-1}})\big) - m^2\big(\Upsilon_{\theta_{*,t}}(s_i,a_i),\Psi_\pi(R_i,S_i',\Upsilon_{\hat{\theta}_{n,t-1}})\big)\right\}$$

The terms within the curly bracket can be rewritten as follows with temporarily letting $\Psi_i := \Psi_\pi(R_i,S_i',\Upsilon_{\hat{\theta}_{n,t-1}})$,

$$\|G(\cdot|\Upsilon_\theta(s_i,a_i)) - G(\cdot|\Psi_i)\|_m^2 - \|G(\cdot|\Upsilon_{\theta_{*,t}}(s_i,a_i)) - G(\cdot|\Psi_i)\|_m^2$$
$$= \|G(\cdot|\Upsilon_\theta(s_i,a_i))\|_m^2 - \|G(\cdot|\Upsilon_{\theta_{*,t}}(s_i,a_i))\|_m^2$$
$$- 2 \cdot \langle G(\cdot|\Upsilon_\theta(s_i,a_i)) - G(\cdot|\Upsilon_{\theta_{*,t}}(s_i,a_i)), G(\cdot|\Psi_i)\rangle_m.$$

Taking its expectation so that $\mathbb{E}(G(\cdot|\Psi)) = G(\cdot|\Upsilon_{\theta_{*,t}}(s_i,a_i))$, we have $\lambda = 1$ as follows,

$$F_{n,t}(\theta|\hat{\theta}_{n,t-1}) - F_{n,t}(\theta_{*,t}|\hat{\theta}_{n,t-1}) = \frac{1}{n}\sum_{i=1}^{n}\|G(\cdot|\Upsilon_\theta(s_i,a_i))\|_m^2 - \frac{1}{n}\sum_{i=1}^{n}\|G(\cdot|\Upsilon_{\theta_{*,t}}(s_i,a_i))\|_m^2$$

$$- \frac{2}{n}\sum_{i=1}^{n}\langle G(\cdot|\Upsilon_\theta(s_i,a_i)) - G(\cdot|\Upsilon_{\theta_{*,t}}(s_i,a_i)), G(\cdot|\Upsilon_{\theta_{*,t}}(s_i,a_i))\rangle$$

$$= \frac{1}{n}\sum_{i=1}^{n}\|G(\cdot|\Upsilon_\theta(s_i,a_i)) - G(\cdot|\Upsilon_{\theta_{*,t}}(s_i,a_i))\|_m^2 = \bar{m}_{n,1}^2(\theta,\theta_{*,t}).$$

(NS3) can be shown as follows. Let us define the following,

$$F'_{n,t}(\theta|\hat{\theta}_{n,t-1}) := \frac{1}{n}\sum_{i=1}^{n}\left\{\|G(\cdot|\Upsilon_\theta(s_i,a_i))\|_m^2\right.$$

$$\left. - 2 \cdot \langle G(\cdot|\Upsilon_\theta(s_i,a_i)) - G(\cdot|\mathcal{T}^\pi\Upsilon_{\hat{\theta}_{n,t-1}}(s_i,a_i))\rangle_m\right\},$$

$$\hat{F}'_{n,t}(\theta|\hat{\theta}_{n,t-1}) := \frac{1}{n}\sum_{i=1}^{n}\left\{\|G(\cdot|\Upsilon_\theta(s_i,a_i))\|_m^2\right.$$

$$\left. - 2 \cdot \langle G(\cdot|\Upsilon_\theta(s_i,a_i)) - G(\cdot|\Psi_\pi(r_i,s_i',\Upsilon_{\hat{\theta}_{n,t-1}}))\rangle_m\right\}.$$

Based on our derivations for $F_{n,t}(\theta|\hat{\theta}_{n,t-1}) - F_{n,t}(\theta_{*,t}|\hat{\theta}_{n,t-1})$, we can see $F_{n,t}(\theta|\hat{\theta}_{n,t-1}) - F_{n,t}(\theta_{*,t}|\hat{\theta}_{n,t-1}) = F'_{n,t}(\theta|\hat{\theta}_{n,t-1}) - F'_{n,t}(\theta_{*,t}|\hat{\theta}_{n,t-1})$. Likewise, we have $\hat{F}_{n,t}(\theta|\hat{\theta}_{n,t-1}) - \hat{F}_{n,t}(\theta_{*,t}|\hat{\theta}_{n,t-1}) = \hat{F}'_{n,t}(\theta|\hat{\theta}_{n,t-1}) - \hat{F}'_{n,t}(\theta_{*,t}|\hat{\theta}_{n,t-1})$. Then, we have the following based on our derivations for (NS2),

$$\bar{m}_{n,1}^2(\hat{\theta}_{n,t},\theta_{*,t}) = F'_{n,t}(\hat{\theta}_{n,t}|\hat{\theta}_{n,t-1}) - F'_{n,t}(\theta_{*,t}|\hat{\theta}_{n,t-1})$$

$$= \left\{F'_{n,t}(\hat{\theta}_{n,t}|\hat{\theta}_{n,t-1}) - \hat{F}'_{n,t}(\hat{\theta}_{n,t}|\hat{\theta}_{n,t-1})\right\} + \left\{\hat{F}'_{n,t}(\hat{\theta}_{n,t}|\hat{\theta}_{n,t-1}) - F'_{n,t}(\theta_{*,t}|\hat{\theta}_{n,t-1})\right\}$$

$$\leq \sup_{\theta\in\Theta}\left|F'_{n,t}(\theta|\hat{\theta}_{n,t-1}) - \hat{F}'_{n,t}(\theta|\hat{\theta}_{n,t-1})\right| + \left|\hat{F}'_{n,t}(\theta_{*,t}|\hat{\theta}_{n,t-1}) - F'_{n,t}(\theta_{*,t}|\hat{\theta}_{n,t-1})\right|$$

$$\leq \sup_{\theta\in\Theta}\left|\left\{F'_{n,t}(\theta|\hat{\theta}_{n,t-1}) - \hat{F}'_{n,t}(\theta|\hat{\theta}_{n,t-1})\right\} - \left\{F'_{n,t}(\theta_{*,t}|\hat{\theta}_{n,t-1}) - \hat{F}'_{n,t}(\theta_{*,t}|\hat{\theta}_{n,t-1})\right\}\right|$$

$$+ 2 \cdot \left|\hat{F}'_{n,t}(\theta_{*,t}|\hat{\theta}_{n,t-1}) - F'_{n,t}(\theta_{*,t}|\hat{\theta}_{n,t-1})\right|$$

$$= \sup_{\theta\in\Theta}\left|\Delta_n(\theta|\hat{\theta}_{n,t-1})\right| + 2 \cdot \left|\hat{F}'_{n,t}(\theta_{*,t}|\hat{\theta}_{n,t-1}) - F'_{n,t}(\theta_{*,t}|\hat{\theta}_{n,t-1})\right|.$$

Using subgaussianity that we have shown for (NS1), since subgaussian constant does not depend on $\hat{\theta}_{n,t-1}$, we can apply Theorem 8.1.6 of [57] (probabilistic tail for Dudley's inequality) to obtain probabilistic convergence of the term $\sup_{\theta \in \Theta} |\Delta_n(\theta|\hat{\theta}_{n,t-1})|$. Regarding the second term, we have

$$
|\hat{F}'_{n,t}(\theta_{*,t}|\hat{\theta}_{n,t-1}) - F'_{n,t}(\theta_{*,t}|\hat{\theta}_{n,t-1})| = 2 \cdot \left| \frac{1}{n} \sum_{i=1}^{n} \left\{ \langle G(\cdot|\Upsilon_{\theta_{*,t}}(s_i,a_i)), G(\cdot|\Upsilon_{\theta_{*,t}}(s_i,a_i)) \rangle_m \right. \right.
$$
$$
\left. \left. - G(\cdot|\Upsilon_{\theta_{*,t}}(s_i,a_i)), G(\cdot|\Psi_\pi(R_i,S'_i,\Upsilon_{\hat{\theta}_{n,t-1}})) \rangle \right\} \right|.
$$

Letting $X_i = \langle G(\cdot|\Upsilon_{\theta_{*,t}}(s_i,a_i)), G(\cdot|\Upsilon_{\theta_{*,t}}(s_i,a_i)) \rangle_m$, we have $|X_i| \leq C^2_{max,m}$ by Cauchy-Schwartz inequality. Then, we can apply Hoeffding's inequality for bounded random variables (e.g., Theorem 2.2.6 of [57]) to obtain probabilistic bound that does not depend on $\hat{\theta}_{n,t-1}$.

(NS4) is straightforward. However, the two statements of (23) need to be assumed.

### C.10   Proof for Corollary B.1

For the case of $(\mathfrak{d}, m, \eta) = (l_2^2, \mathbb{W}_1, \mathbb{W}_p)$, we have $q = p$, $\delta = \frac{1}{p}$, $l = 2$, $c = 1$, $C_{subg} = C$, $C_{surr} = D^{1-\frac{1}{p}}$, $\epsilon_0 = 0$.

Based on the fact that $l_2^2 = \frac{1}{2}\mathcal{E}$ [7] for $d = 1$, where energy distance $\mathcal{E} = \text{MMD}_\beta^2$ (17) with $\beta = 1$, it is equivalent to let $\mathfrak{d} = \text{MMD}_{\beta=1}^2$.

Let us first verify (NS1). We obtain the following for $F_{n,t}(\theta|\hat{\theta}_{n,t-1})$. Temporarily ignoring $\hat{\theta}_{n,t-1}$, since we have $\Delta_n(\theta_1) - \Delta_n(\theta_2) = \{\hat{F}_{n,t}(\theta_1|\hat{\theta}_{n,t-1}) - F_{n,t}(\theta_1|\hat{\theta}_{n,t-1})\} - \{\hat{F}_{n,t}(\theta_2|\hat{\theta}_{n,t-1}) - F_{n,t}(\theta_2|\hat{\theta}_{n,t-1})\}$, we can let $\Delta_n(\theta) = \hat{F}_{n,t}(\theta|\hat{\theta}_{n,t-1}) - F_{n,t}(\theta|\hat{\theta}_{n,t-1})$. We will ignore the notation $\hat{\theta}_{n,t-1}$ for the sake of convenience.

$$
\Delta_n(\theta) = 2 \cdot \Delta_n^{(1)}(\theta) + \Delta_n^{(2)}(\theta), \quad \text{where}
$$
$$
\Delta_n^{(1)}(\theta) := \frac{1}{n} \sum_{i=1}^{n} \left\{ f_i^\theta - \mathbb{E}(f_1^\theta) \right\}, \quad \Delta_n^{(2)}(\theta) := \frac{1}{n} \sum_{i=1}^{n} \left\{ g_i^\theta - \mathbb{E}(g_1^\theta) \right\}.
$$

where $f_i^\theta = \mathbb{E}\|Z_\alpha(s_i,a_i;\theta) - r_i - \gamma \cdot Z_\beta(s'_i,A'_i;\hat{\theta}_{n,t-1})\|$ and $g_i^\theta = \mathbb{E}\|Z_\alpha(s_i,a_i;\theta) - Z_\beta(s_i,a_i;\theta)\|$, with $A'_i \sim \pi(\cdot|s'_i)$. Let us only deal with the first term, since the second term can be handled via analogous approach. Letting $x_i^\theta := f_i^\theta - \mathbb{E}(f_i^\theta)$, we have $\Delta_n^{(1)}(\theta) = \frac{1}{n} \sum_{i=1}^{n} x_i$.

In order to derive its probabilistic bound, we shall bound the following term,

$$
\left\| \Delta_n^{(1)}(\theta_1) - \Delta_n^{(1)}(\theta_2) \right\|_{\psi_2(n)}^2 \leq \frac{1}{n^2} \sum_{i=1}^{n} \|x_i^{\theta_1} - x_i^{\theta_2}\|_{\psi_2(n)}^2, \quad \text{where} \tag{30}
$$

$$
\|X\|_{\psi_2(n)} := \inf \left\{ t > 0 \; ; \; \mathbb{E}_n \left\{ \exp(X^2/t^2) \right\} \leq 2 \right\}, \quad \|\mathbf{X}\|_{\psi_2(n)} := \sup_{\mathbf{x} \in \mathbb{R}^d : \|\mathbf{x}\| = 1} \|\langle \mathbf{X}, \mathbf{x} \rangle\|_{\psi_2(n)}.
$$

where the inequality holds by Proposition 2.6.1 of [57]. In defining $\|\cdot\|_{\psi_2(n)}$, $X$ and $\mathbf{x}$ mean random variable ($d = 1$) and vector ($d \geq 1$). Further using the fact that $\|\cdot\|_{\psi_2(n)}$ is a valid norm, we can derive the following where $\mathbb{E}_n := \mathbb{E}(\cdots|\mathcal{Z}_t)$,

$$
\|x_i^{\theta_1} - x_i^{\theta_2}\|_{\psi_2(n)} \leq \|f_i^{\theta_1} - f_i^{\theta_2}\|_{\psi_2(n)} + \mathbb{E}_n \|f_i^\theta - f_i^{\theta_2}\|_{\psi_2(n)},
$$

$$
\therefore \|\Delta_n^{(1)}(\theta_1) - \Delta_n^{(1)}(\theta_2)\|_{\psi_2(n)}^2 \leq \frac{2}{n^2} \sum_{i=1}^{n} \left\{ \|f_i^{\theta_1} - f_i^{\theta_2}\|_{\psi_2(n)}^2 + \mathbb{E}_n\left(\|f_i^{\theta_1} - f_i^{\theta_2}\|_{\psi_2(n)}^2\right) \right\},
$$

where the last inequality is based on Proposition 2.6.1 of [57]. Based on technique suggested in Appendix A.6 of [26], we can derive $|f_1^{\theta_1} - f_1^{\theta_2}| \leq \mathbb{W}_1\{\Upsilon_{\theta_1}(s_1,a_1), \Upsilon_{\theta_2}(s_1,a_1)\}$, which leads to following based on Example 2.5.8 of [57],

$$
\|\Delta_n^{(1)}(\theta_1) - \Delta_n^{(1)}(\theta_2)\|_{\psi_2(n)}^2 \leq \frac{C}{n^2} \cdot \sum_{i=1}^{n} \mathbb{W}_1^2(\Upsilon_{\theta_1}(s_i,a_i), \Upsilon_{\theta_2}(s_i,a_i)) = \frac{C}{n} \cdot \overline{\mathbb{W}}_{1,n}^2(\theta_1,\theta_2).
$$

Applying the same logic to $\Delta_n^{(2)}(\theta)$, we obtain $C_{subg} = C$. It is trivial to show $\Delta_n(\theta)$ is mean-zero. (NS2) can be shown as follows

$$F_{n,t}(\theta|\hat{\theta}_{n,t-1}) - F_{n,t}(\theta_{*,t}|\hat{\theta}_{n,t-1}) = \frac{1}{n}\sum_{i=1}^{n}\mathcal{E}(\Upsilon_\theta(s_i,a_i), \mathcal{T}^\pi\Upsilon_{\hat{\theta}_{n,t-1}}(s_i,a_i)).$$

Here, $\mathcal{E}$ is energy disance defined as $\mathcal{E}(P,Q) := 2\mathbb{E}\|X - Y\| - \mathbb{E}\|X - X'\| - \mathbb{E}\|Y - Y'\|$ for $X, X' \overset{iid}{\sim} Q$ and $Y, Y' \overset{iid}{\sim} P$. Using the following lemma (proof in Appendix D.3), we obtain the desired result with $\lambda = 2/(b-a)$.

**Lemma C.8.** *Assume $1 \leq p \leq r$, and let $P, Q$ be probability measures for arbitrary random vectors of $d \geq 1$. We have*

$$\mathcal{E}(P,Q) \leq 2 \cdot \mathbb{W}_p(P,Q) \leq 2 \cdot \mathbb{W}_r(P,Q).$$

*If $P, Q \in \mathcal{P}[a,b]$, we have*

$$\frac{2}{(b-a)^{2r-1}} \cdot \mathbb{W}_r^{2r}(P,Q) \leq \frac{2}{(b-a)^{2p-1}} \cdot \mathbb{W}_p^{2p}(P,Q) \leq \mathcal{E}(P,Q).$$

(NS3) can be shown by following. $\hat{\theta}_{n,t}$ is equivalent to $\hat{\theta}_{n,t} = \arg\min_{\theta\in\Theta} \hat{F}'_{n,t}(\theta|\hat{\theta}_{n,t-1})$ where $\hat{F}'_{n,t}(\theta|\hat{\theta}_{n,t-1})$ is defined as follows (instead of what we defined in Appendix C.9.2),

$$\frac{2}{n}\sum_{i=1}^{n}\mathbb{E}\|Z_\alpha(s_i,a_i;\theta) - R_i - \gamma \cdot Z_\beta(S'_i, A'_i; \hat{\theta}_{n,t-1})\| - \frac{1}{n}\sum_{i=1}^{n}\mathbb{E}\|Z_\alpha(s_i,a_i;\theta) - Z_\beta(s_i,a_i;\theta)\|,$$

with $R_i, S'_i \sim \tilde{p}(\cdots|s_i,a_i)$. Then we have following with the same logic shown in Appendix C.9.2,

$$\overline{\mathbb{W}}_{1,n,1}^2(\hat{\theta}_{n,t}, \theta_{*,t}) \lesssim F'_{n,t}(\hat{\theta}_{n,t}|\hat{\theta}_{n,t-1}) - F'_{n,t}(\theta_{*,t}|\hat{\theta}_{n,t-1})$$

$$\leq \sup_{\theta\in\Theta}\left|\Delta_n(\theta|\hat{\theta}_{n,t-1})\right| + 2\cdot\left|\hat{F}'_{n,t}(\theta_{*,t}|\hat{\theta}_{n,t-1}) - F'_{n,t}(\theta_{*,t}|\hat{\theta}_{n,t-1})\right|.$$

Just as Appendix C.9.2, we can show $|\Delta_n(\theta|\hat{\theta}_{n,t-1})| \to^P 0$ by applying Dudley's inequality (Theorem 8.1.6 of [57]) and show $|\hat{F}'_{n,t}(\theta_{*,t}|\hat{\theta}_{n,t-1}) - F'_{n,t}(\theta_{*,t}|\hat{\theta}_{n,t-1})| \to^P 0$ with Hoeffding's inequality for bounded random variables (Theorem 2.2.6 of [57]).

Now let us show (NS4). Every $\mu \in \Delta([a,b])$ has a corresponding function function $F \in \mathcal{F}$, where $\mathcal{F}$ represents the cdf family. We let $\mathcal{F}_\Theta := \{F_\theta(\cdot|\cdots) \in \mathcal{F}^{\mathcal{S}\times\mathcal{A}} \mid F_\theta(s,a) \text{ is the cdf of } \Upsilon_\theta(s,a)\}$, and we have $|F_\theta(\cdot|\cdots)| \leq 1$. Based on $\mathbb{W}_1(\mu_1,\mu_2) = \|F_1 - F_2\|_{L_1}$ with functional norm $\|F\|_{L_p} := (\int|F(z)|^p\mathrm{d}z)^{1/p}$, we can define extended norms (24) with $\|\cdot\|_m = \|\cdot\|_{L_1}$.

Lastly, by Section 2.3 of [44], we have $\mathbb{W}_p \leq D^{1-\frac{1}{p}} \cdot \mathbb{W}_1^{1/p}$, where $D$ indicates the diameter of the support of return distribution ($D = b - a$ in 1-dimensional case). This leads to $C_{surr} = (b-a)^{1-1/p}$ and $\delta = 1/p$. Then, by applying Theorem C.5. we obtain the following bound for $\overline{\mathbb{W}}_{p,d_\pi,p}(\Upsilon_{\theta_T}, \Upsilon_\pi)$,

$$\frac{(b-a)^{1-\frac{1}{p}}\cdot C_{cover}^{\frac{1}{2p}}}{1-\gamma^{1-\frac{1}{2p}}}\cdot\left\{C_{brack}^{1/2}\cdot\left(\frac{1}{1-\frac{1}{2p}}\right)^{\frac{1}{2}}\cdot\frac{\sqrt{2}^{-\alpha}}{1-\alpha/2}\cdot\left(\frac{1/p}{2+\alpha}\right)^{\frac{1}{2}}\right\}^{\frac{1/p}{1+\alpha/2}}$$

$$\times O_P\left\{\left(\frac{N}{\log_{1/\gamma}N}\right)^{\frac{-1/p}{2+\alpha}}\right\}.$$

## C.11 Proof of Corollary B.2

For the case of $(\mathfrak{d}, m, \eta) = (\text{MMD}_\beta^2, \text{MMD}_\beta, \text{MMD}_\beta)$, we have $q = 1$, $\delta = 1$, $l = 2$, $c = \frac{\beta}{2}$, $C_{subg} = C \cdot (r_{max} + z_{max})^{\beta/2} \overset{let}{=} C_{max}^{(\beta)}$ with $\sup_{\theta\in\Theta}\sup_{s,a}\|Z(s,a;\theta)\| \leq z_{max}$ and $\sup_{s,a}\|R(s,a)\| \leq r_{max}$, $C_{surr} = 1$, $\epsilon_0 = 0$. We need $\beta > 1$ for $q = 1$ as we mentioned in (16).

Any pair of $(\mathfrak{d}, m) = (\text{MMD}^2, \text{MMD})$ satisfies NS Properties ((NS1)–(NS4)) for bounded kernels (see Appendix C.9.2). Then, we can apply Theorem C.5 to obtain the following bound for $\overline{\text{MMD}}_{\beta, d_\pi, 1}(\Upsilon_{\theta_T}, \Upsilon_\pi)$.

$$\frac{C_{cover}^{1/2}}{1 - \gamma^{\frac{\beta-1}{2}}} \cdot \left\{ C_{max}^{(\beta)} \cdot C_{brack}^{\frac{1}{2}} \cdot \left( \frac{2}{\beta - 1} \right)^{\frac{1}{2}} \cdot \frac{\sqrt{2}^{-\alpha}}{1 - \alpha/2} \cdot \left( \frac{1}{2 + \alpha} \right)^{\frac{1}{2}} \right\}^{\frac{1}{1+\alpha/2}}$$

$$\times O_P \left\{ \left( \frac{N}{\log_{1/\gamma} N} \right)^{\frac{-1}{2+\alpha}} \right\}$$

If we want to obtain the result for $\beta = 1$, we let $c = 1/2$, $q = 1 + \epsilon_N$ for some monotonically decreasing sequence $\epsilon_N \to 0$ as $N \to \infty$, and $C_{max}^{(\beta=1)} = C \cdot (r_{\max} + z_{\max})^{1/2}$. Applying this to Theorem C.5, we obtain

$$\overline{\text{MMD}}_{1, d_\pi, 1}(\Upsilon_{\theta_T}, \Upsilon_\pi) \leq \overline{\text{MMD}}_{1, d_\pi, q}(\Upsilon_{\theta_T}, \Upsilon_\pi)$$

$$\leq \underbrace{C_{maxim}^{1 - \frac{1}{1+\epsilon_N}} \cdot C_{cover}^{\frac{1}{2}} \cdot \left\{ C_{max}^{(\beta=1)} \cdot C_{brack}^{1/2} \cdot \frac{\sqrt{2}^{-\alpha}}{1 - \alpha/2} \cdot \left( \frac{1/(1 + \epsilon_N)}{2 + \alpha} \right)^{1/2} \right\}^{\frac{1/(1+\epsilon_N)}{1+\alpha/2}}}_{\text{constant part}} \times$$

$$\underbrace{\frac{1}{1 - \gamma^{\frac{1}{2} - \frac{1}{2(1+\epsilon_N)}}} \cdot \left( \frac{1}{\frac{1}{2} - \frac{1}{2(1+\epsilon_N)}} \right)^{\frac{1}{2+\alpha} \cdot \frac{1}{1+\epsilon_N}} \cdot O_P \left\{ \left( \frac{N}{\log_{1/\gamma} N} \right)^{\frac{-1/(1+\epsilon_N)}{2+\alpha}} \right\}}_{\text{convergent part}}.$$

Letting $y_N := \frac{\epsilon_N}{2(1+\epsilon_N)} \to 0$, we have

$$\lim_{N \to \infty} \frac{1/y_N}{1/(1 - \gamma^{y_N})} = \lim_{x \to 0} \frac{1/x}{1/(1 - \gamma^x)} = \lim_{x \to 0} \frac{1 - \gamma^x}{x} = \log(1/\gamma). \tag{31}$$

Therefore, the convergent part can be bounded by constant multiplied by following for sufficiently large $N$,

$$\lesssim \frac{1}{\log(1/\gamma)} \cdot \left( \frac{1}{y_N} \right)^{1 + \frac{1}{2+\alpha} \cdot \frac{1}{1+\epsilon_N}} \cdot O_P \left\{ \left( \frac{N}{\log_{1/\gamma} N} \right)^{\frac{-1}{2+\alpha} \cdot \frac{1}{1+\epsilon_N}} \right\}$$

$$\leq \frac{1}{\log(1/\gamma)} \cdot (\log N)^{1 + \frac{1}{2+\alpha} \cdot \frac{1}{1+\epsilon_N}} \cdot \left( \frac{1}{\log(1/\gamma)} \right)^{\frac{1}{2+\alpha} \cdot \frac{1}{1+\epsilon_N}} \cdot (\log N)^{\frac{1}{2+\alpha} \cdot \frac{1}{1+\epsilon_N}} \cdot O_P(N^{\frac{-1}{2+\alpha} \cdot \frac{1}{1+\epsilon_N}})$$

$$\leq \left( \frac{1}{\log(1/\gamma)} \right)^{1 + \frac{1}{2+\alpha}} \cdot (\log N)^{1 + \frac{2}{2+\alpha}} \cdot O_P \left\{ \left( \frac{\log N}{N} \right)^{\frac{1}{2+\alpha}} \right\}$$

$$\leq \left( \frac{1}{\log(1/\gamma)} \right)^{1 + \frac{1}{2+\alpha}} \cdot O_P \left\{ (\log N)^{1 + \frac{3}{2+\alpha}} \cdot N^{-\frac{1}{2+\alpha}} \right\},$$

where the second last line holds by following.

$$N^{\frac{1}{1+1/\log N}} > \frac{N}{\log N} \quad \text{if and only if} \quad \frac{1}{1 + 1/\log N} \cdot \log N > \log N - \log(\log N)$$

$$\text{if and only if} \quad \log(\log N) > \frac{1}{1 + 1/\log N}. \tag{32}$$

Then, we finally have

$$\overline{\text{MMD}}_{1, d_\pi, 1}(\Upsilon_{\theta_T}, \Upsilon_\pi) \leq C_{cover}^{\frac{1}{2}} \cdot \left\{ C_{max}^{(\beta=1)} \cdot C_{brack}^{1/2} \cdot \frac{\sqrt{2}^{-\alpha}}{1 - \alpha/2} \cdot \left( \frac{1}{2 + \alpha} \right)^{1/2} \right\}^{\frac{1}{1+\alpha/2}}$$

$$\times \left( \frac{1}{\log(1/\gamma)} \right)^{1 + \frac{1}{2+\alpha}} \cdot O_P \left\{ \left( (\log N)^{1 + \frac{3}{2+\alpha}} \cdot N^{\frac{-1}{2+\alpha}} \right) \right\}.$$

## C.12 Proof for Corollary B.4

For the case of $(\mathfrak{d}, m, \eta) = (d_{L_2}^2, d_{L_2}, \mathbb{W}_p)$, we have $q = p$, $\delta = \frac{2(r-p)}{(d+2r)p}$, $l = 2$, $c = 1$, $C_{subg} = C \cdot (1 + \gamma^{-d/2}) \cdot C_{L_2}$, $C_{surr} = 2 \cdot (\max\{V_d, 1\})^{\frac{1}{2p}} \cdot M_r^{\frac{(d+2p)r}{(d+2r)p}}$, $\epsilon_0 = 0$. Here, we have $C_{L_2} := \sup_{s,a} \sup_{\theta \in \Theta} \|f_\theta(\cdot|s,a)\|_{L_2}$, $M_r := \sup_{\theta \in \Theta} \sup_{s,a} \mathbb{E}\{\|Z(s,a;\theta)\|^r\}^{1/r} < \infty$ and $V_d$ is the volume of a unit ball in $\mathbb{R}^d$.

Due to Appendix C.9.2, it suffices to show (IPS1) and (IPS2) of Definition C.2 and (11). IPS properties are trivial to show.

(11) can be shown as follows. Letting $f(\cdot|s,\pi) := \int_{a \in \mathcal{A}} f(\cdot|s,a) \mathrm{d}\pi(a|s)$, we have $\|1/\gamma^d \cdot f_{\hat{\theta}_{n,t-1}}((\cdot - r)/\gamma|s',\pi)\|_{L_2} = 1/\gamma^{d/2} \cdot \sup_{\theta \in \Theta} \sup_{s,a} \|f_\theta(\cdot|s,a)\|_{L_2}$. Thus we have $C_{max,m} = C \cdot (1 + \gamma^{-d/2}) \cdot C_{L_2}$. We can see $C_{L_2} < \infty$ by following, by letting $L := \sup_{\theta \in \Theta} \sup_{z,s,a} |f_\theta(z|s,a)| < \infty$,

$$\frac{1}{L^2}\|f_\theta(\cdot|s,a)\|_{L_2}^2 = \int_{\mathbb{R}^d} \left(\frac{f_\theta(z|s,a)}{L}\right)^2 \mathrm{d}z \leq \int_{\mathbb{R}^d} \frac{f_\theta(z|s,a)}{L} \mathrm{d}z = \frac{1}{L}, \quad \therefore \ C_{L_2} \leq L^{1/2}.$$

Lastly, by Proposition 13 of [56], we have following under $1 \leq p < r$

$$\mathbb{W}_p(\mu_1, \mu_2) \leq 2 \cdot (\max\{V_d, 1\})^{\frac{1}{2p}} \cdot M_r^{\frac{(d+2p)r}{(d+2r)p}} \cdot \{d_{L_2}(\mu_1, \mu_2)\}^{\frac{2(r-p)}{(d+2r)p}},$$

with $\mathbb{E}_{\mathbf{X} \sim \mu_1}\{\|\mathbf{X}\|^r\}^{1/r}, \mathbb{E}_{\mathbf{X} \sim \mu_2}\{\|\mathbf{X}\|^r\}^{1/r} \leq M$ and $V_d$ being the volume of $d$-dimensional unit ball. Thus, we have $\delta = \frac{2(r-p)}{(d+2r)p}$ with

$$C_{surr} = 2 \cdot (\max\{V_d, 1\})^{\frac{1}{2p}} \cdot M_r^{\frac{(d+2p)r}{(d+2r)p}}.$$

Applying Theorem C.5 gives us the following bound for $\overline{\mathbb{W}}_{p,d_\pi,p}(\Upsilon_{\theta_T}, \Upsilon_\pi)$,

$$\frac{1}{1 - \gamma^{1-\frac{1}{2p}}} \cdot \left(\max\{V_d, 1\}\right)^{\frac{1}{2p}} \cdot M_r^{\frac{(d+2p)r}{(d+2r)p}} \cdot C_{cover}^{\frac{r-p}{(d+2r)p}} \times \left\{C_{L_2} \cdot C_{brack}^{1/2} \cdot \left(\frac{1}{1 - \frac{1}{2p}}\right)^{1/2}\right.$$

$$\left. \cdot \frac{\sqrt{2}^{-\alpha}}{1 - \alpha/2} \cdot \left(\frac{2(r-p)}{(d+2r)p \cdot (2+\alpha)}\right)^{1/2}\right\}^{\frac{1}{1+\alpha/2} \cdot \frac{2(r-p)}{(d+2r)p}} \cdot O_P\left\{\left(\frac{N}{\log_{1/\gamma} N}\right)^{\frac{-1}{2+\alpha} \cdot \frac{2(r-p)}{(d+2r)p}}\right\}$$

## C.13 Proof of Corollary B.5

NS Properties ((NS1)–(NS4) of Assumption C.3) are satisfied by Appendix C.9.2, since $k_\nu$ is bounded. For the case of $(\mathfrak{d}, m, \eta) = (\mathrm{MMD}_\nu^2, \mathrm{MMD}_\nu, \mathbb{W}_p)$, we have $q = p$, $l = 2$, $c = 1$, $C_{subg} = C_{max}^{(k)} \overset{let}{=} C_{max}^{(\nu)}$. Here, we have $C_{max}^{(\nu)} < \infty$ since $k_\nu$ is a bounded function with a fixed value of $\sigma_{Mat} > 0$. Assuming that $\|f_\theta(\cdot|s,a)\|_{H^{\nu+d/2}(\mathbb{R}^d)} < B$ and $M_r < \infty$, Theorem 15 and Example 4 of [56] gives us $C_{surr} = C(B, r, M_r, s, d, p, \sigma_{Mat})$, $\epsilon_0 = 0$, $\delta = \frac{r-p}{(d+2r)p}$ in Assumption 3.2. See the detailed form of $C(B, r, M_r, s, d, p, \sigma_{Mat}) > 0$ in their Theorem 15. Then, applying Theorem C.5 gives us following bound for $\overline{\mathbb{W}}_{p,d_\pi,p}(\Upsilon_{\theta_T}, \Upsilon_\pi)$,

$$\frac{C(B, r, M_r, s, d, p, \sigma_{Mat})}{1 - \gamma^{1-\frac{1}{2p}}} \cdot C_{cover}^{\frac{r-p}{2(d+2r)p}} \cdot \left\{C_{max}^{(\nu)} \cdot C_{brack}^{1/2} \cdot \frac{\sqrt{2}^{-\alpha}}{1 - \alpha/2}\right\}^{\frac{1}{1+\alpha/2} \cdot \frac{r-p}{(d+2r)p}}$$

$$\times O_P\left\{\left(\frac{N}{\log_{1/\gamma} N}\right)^{\frac{-1}{2+\alpha} \cdot \frac{r-p}{(d+2r)p}}\right\}.$$

## C.14 Proofs based on Gaussian regularizer

Here, we introduce a method by which we can bound $\overline{\mathbb{W}}_{p,d_\pi,p}$-inaccuracy based on regularizer $\alpha_\sigma(\cdot)$ (see Definition 20 of [56]). In the following lemma, we use gaussian regularizer $\alpha_\sigma$ due to its convenience, however it can be extended to other regularizers (see their Lemma 21). Since our surrogate metric $m$ may depend on the value of $\sigma > 0$, we will denote it as $m_\sigma$.

**Lemma C.9.** *Let $\alpha_\sigma(\cdot)$ be the density (or probability measure) of $N(\mathbf{0}, \sigma^2 I_d)$ and $*$ indicate convolution. Assuming $\mathbb{W}_p(\mu * \alpha_\sigma, \nu * \alpha_\sigma) \leq C(\sigma, p) \cdot m_\sigma^\delta(\mu, \nu)$ with $p \in (0, \frac{1}{p}]$ and $m_\sigma$ satisfies (NS4), letting $\mathfrak{d} = m_\sigma^2$ achieves the following bound (with detailed bound in (33)),*

$$\overline{\mathbb{W}}_{p,d_\pi,p}(\Upsilon_{\theta_T}, \Upsilon_\pi) \lesssim \frac{C(\sigma, p)}{1 - \gamma^{1-\frac{1}{2p}}} \cdot O_P\left\{ \left( \frac{N}{\log_{1/\gamma} N} \right)^{\frac{-\delta'}{2+\alpha}} \right\} + \underbrace{\frac{2^{\frac{5}{2}-\frac{1}{2p}}}{1 - \gamma^{1-\frac{1}{2p}}} \cdot \frac{\Gamma((p+d)/2)}{\Gamma(p/2)} \cdot \sigma}_{\text{irreducible for fixed } \sigma > 0.}$$

### C.14.1 Proof for Lemma C.9

By Lemma 21 of [56], we have the following,

$$\begin{aligned}
\mathbb{W}_p(\mu, \nu) &\leq \mathbb{W}(\mu * \alpha_\sigma, \nu * \alpha_\sigma) + 2 \cdot \left( \int \|\mathbf{z}\|^p \alpha_\sigma(\mathbf{z}) \mathrm{d}\mathbf{z} \right)^{1/p} \\
&= \mathbb{W}_p(\mu * \alpha_\sigma, \nu * \alpha_\sigma) + 2^{3/2} \cdot \frac{\Gamma((p+d)/2)}{\Gamma(p/2)} \cdot \sigma \\
&\leq C(\sigma, p) \cdot \bar{m}_{\sigma,d_\pi,1}^\delta(\mu, \nu) + 2^{3/2} \cdot \frac{\Gamma((p+d)/2)}{\Gamma(p/2)} \cdot \sigma.
\end{aligned}$$

Then, we can apply Assumption 3.2 with $C_{surr} = C(\sigma, p)$, and $\epsilon_0 = 2^{3/2} \cdot \frac{\Gamma((p+d)/2)}{\Gamma(p/2)} \cdot \sigma$. Further letting $m = m_\sigma$, $l = 2$, $q = p$, $c = 1$, we can apply Theorem C.5 to obtain the following,

$$\overline{\mathbb{W}}_{p,d_\pi,p}(\Upsilon_{\theta_T}, \Upsilon_\pi) \leq \frac{1}{1 - \gamma^{1-\frac{1}{2p}}} \cdot C(\sigma, p) \cdot C_{cover}^{\frac{\delta'}{2}} \cdot \left( C_{subg} \cdot C_{brack}^{1/2} \cdot \frac{\sqrt{2}^{-\alpha}}{1 - \alpha/2} \right)^{\frac{\delta'}{1+\alpha/2}}$$

$$\cdot O_P\left\{ \left( \frac{N}{\log_{1/\gamma} N} \right)^{\frac{-\delta'}{2+\alpha}} \right\} + \frac{1}{1 - \gamma^{1-\frac{1}{2p}}} \cdot 2^{\frac{5}{2}-\frac{1}{2p}} \cdot \frac{\Gamma((p+d)/2)}{\Gamma(p/2)} \cdot \sigma. \tag{33}$$

### C.14.2 Proof for Corollary B.6

The case $(\mathfrak{d}, m, \eta) = (\text{MMD}_{\sigma_{\text{RBF}}}^2, \text{MMD}_{\sigma_{\text{RBF}}}, \mathbb{W}_p)$ satisfies Property (NS4), having $\delta = \frac{2(r-p)}{(d+2r)p} \in (0, \frac{1}{p})$, $C(\sigma_{\text{RBF}}, p) = C(p, d, r, M_r)$ (see Theorem 24 of [56] for its detailed form). We also have $C_{subg} = C_{max}^{(k)} = \frac{C}{2^{d/2} \cdot \pi^{d/4}} \cdot \sigma_{\text{RBF}}^{-d/2}$ by (29), and $C_{brack} = C_{brack}(\sigma_{\text{RBF}})$ whose dependence on $\sigma_{\text{RBF}} > 0$ is not clear. Further letting $\sigma_{\text{RBF}} = \sigma_N \to 0$ to a monotonically decreasing sequence, applying Lemma C.9 gives us

$$\overline{\mathbb{W}}_{p,d_\pi,p}(\Upsilon_{\theta_T}, \Upsilon_\pi) \leq \frac{C(p, d, \sigma_N, p)}{1 - \gamma^{1-\frac{1}{2p}}} \cdot C_{cover}^{\frac{r-p}{d+2r}} \cdot \left( \frac{C_{brack}(\sigma_N)^{1/2}}{2^{d/2} \cdot \pi^{d/4} \cdot \sigma_N^{d/2}} \cdot \frac{\sqrt{2}^{-\alpha}}{1 - \alpha/2} \right)^{\frac{1}{1+\alpha/2} \cdot \frac{2(r-p)}{(d+2r)p}}$$

$$\cdot O_P\left\{ \left( \frac{N}{\log_{1/\gamma} N} \right)^{\frac{-1}{2+\alpha} \cdot \frac{2(r-p)}{(d+2r)p}} \right\} + \frac{1}{1 - \gamma^{1-\frac{1}{2p}}} \cdot 2^{\frac{5}{2}-\frac{1}{2p}} \cdot \frac{\Gamma((p+d)/2)}{\Gamma(p/2)} \cdot \sigma_N.$$

### C.14.3 Proof for Corollary B.7

The case $(\mathfrak{d}, m, \eta) = (\text{MMD}_{(\sigma,f)}^2, \text{MMD}_{(\sigma,f)}, \mathbb{W}_p)$ satisfies (NS4), having $\delta = \frac{1}{p}$, $C(\sigma, p) = 2^{1-\frac{1}{p}} \cdot \sigma$ by Theorem 2 of [70], which can be applied since the kernel $k_{(\sigma,f)}$ is bounded. We also have $C_{subg} = C_{max}^{(k)} \overset{let}{=} C_{max}^{(\sigma,f)}$, which is bounded due to $k_{(\sigma,f)}(\cdot, \cdot) < \infty$ by (29), along with $C_{brack} = C_{brack}(\sigma)$ whose dependence on $\sigma > 0$ is not clear. Further letting $\sigma = \sigma_N \to 0$ to a monotonically decreasing sequence, applying Lemma C.9 gives us

$$\overline{\mathbb{W}}_{p,d_\pi,p}(\Upsilon_{\theta_T}, \Upsilon_\pi) \leq \frac{\sigma_N}{1 - \gamma^{1-\frac{1}{2p}}} \cdot C_{cover}^{\frac{1}{2p}} \cdot \left( C_{max}^{(\sigma_N,f)} \cdot C_{brack}^{1/2}(\sigma_N) \cdot \frac{\sqrt{2}^{-\alpha}}{1 - \alpha/2} \right)^{\frac{1/p}{1+\alpha/2}}$$

$$\cdot O_P\left\{ \left( \frac{N}{\log_{1/\gamma} N} \right)^{\frac{-1/p}{2+\alpha}} \right\} + \frac{1}{1 - \gamma^{1-\frac{1}{2p}}} \cdot 2^{\frac{5}{2}-\frac{1}{p}} \cdot \frac{\Gamma((p+d)/2)}{\Gamma(p/2)} \cdot \sigma_N.$$

Note that for fixed $\sigma > 0$, we have $C_{max}^{(\sigma,f)} < \infty$ as long as $k_{(\sigma,f)}(\cdot, \cdot) < \infty$ holds.

## C.15 Alternative for violation of (NS4)

In Theorem C.5, (NS4) may not hold. There may not be a corresponding functional norm space (e.g., Wasserstein for $d \geq 2$) or the functional element may be unbounded, that is, $\sup_{\theta \in \Theta} \sup_{z,s,a} G_\theta(z|s,a) = \infty$. For such examples, we should replace (NS3) and (NS4) of Section 3.2 with the following (MS3) and (MS4). We can maintain (NS1) and (NS2) from NS properties, renaming them (MS1) and (MS2).

**Assumption C.10** (Metric-space association)**.** The functional Bregman divergence $\mathfrak{d}$ and probability metric $m$ over $\mathcal{P}$ form metric-space association (MS association, in short) if there exists an metric space $(\mathcal{G}, m)$ where any element $\mu \in \mathcal{P}$ can be uniquely (up to equivalence under zero metric $m$) represented by the corresponding element $G(\cdot|\mu) \in \mathcal{G}$ such that

(MS1) The $\mathcal{Z}_t$-conditioned modulus $\Delta_n(\theta|\hat{\theta}_{n,t-1}) := \{\hat{F}_{n,t}(\theta|\hat{\theta}_{n,t-1}) - F_{n,t}(\theta|\hat{\theta}_{n,t-1})\} - \{\hat{F}_{n,t}(\theta_{*,t}|\hat{\theta}_{n,t-1}) - F_{n,t}(\theta_{*,t}|\hat{\theta}_{n,t-1})\}$ is a mean-zero sub-gaussian process with respect to $\bar{m}_{n,1}$. That is, for any $\mathcal{Z}_t \subset \mathcal{S} \times \mathcal{A}$ and $\hat{\theta}_{n,t-1} \in \Theta$, we have

$$\|\Delta_n(\theta_1|\hat{\theta}_{n,t-1}) - \Delta_n(\theta_2|\hat{\theta}_{n,t-1})\|_{\psi_2(n)} \leq \frac{1}{\sqrt{n}} \cdot C_{subg} \cdot \bar{m}_{n,1}(\theta_1, \theta_2).$$

(MS2) We have $F_{n,t}(\theta|\hat{\theta}_{n,t-1}) - F_{n,t}(\theta_{*,t}|\hat{\theta}_{n,t-1}) \geq \lambda \cdot \bar{m}_{n,1}^l(\theta, \theta_{*,t})$ for some $l \geq 2$. Here, the value of $\lambda > 0$ does not depend on $\mathcal{Z}_t, \theta, \hat{\theta}_{n,t-1}$ (or $\theta_{*,t}$ such that $\Upsilon_{\theta_{*,t}} = \mathcal{T}^\pi \Upsilon_{\hat{\theta}_{n,t-1}}$).

(MS3) $\bar{m}_{\rho,1}(\hat{\theta}_{n,t}, \theta_{*,t}) \to^P 0$ converges in a rate that does not depend on $\hat{\theta}_{n,t-1}$ (or $\theta_{*,t}$).

(MS4) There exists a constant $C_{met} \in (0, \infty)$ and $\alpha \in (0, 2)$ such that
$$\mathrm{diam}(\Theta, m_\infty) < \infty \quad \& \quad \log \mathcal{N}(\Theta, m_\infty, \epsilon) \leq C_{met} \cdot \epsilon^{-\alpha}.$$

It is possible that there exists an element in $\mathcal{G}$ that does not represent any $\mu \in \mathcal{P}$.

Under these alternative conditions, we can obtain the following theorem, which is proved in Appendix C.15.1.

**Theorem C.11.** *Suppose Assumptions 3.1, 3.2, C.4, and $C_3 < \infty$ for Lemma C.12. Moreover, given a probability metric $\eta$ that satisfies (S-L-C) with convexity parameter $q \geq 1$ and scale-sensitivity parameter $c > 1/(2q)$ (i.e., (15) of Appendix C.4), consider the FDE with a functional Bregman divergence $\mathfrak{d}$ satisfying Assumption C.10 of Appendix C.15. Then, by letting $T = \lfloor \frac{1}{c - \frac{1}{2q}} \cdot \frac{\delta'}{2\beta_1} \cdot \log_{1/\gamma} N \rfloor$ with $\delta' = \min\{\delta, 1/q\}$, we have*

$$\bar{\eta}_{d_\pi, q}(\Upsilon_{\theta_T}, \Upsilon_\pi) \leq \frac{1}{1 - \gamma^{c - \frac{1}{2q}}} \cdot C_0 \cdot O_P \left\{ \left( \frac{1}{\log(1/\gamma)} \cdot \frac{(\log N)^2}{N} \right)^{\frac{\delta'}{2\beta_1}} \right\} + \frac{2^{1 - \frac{1}{2q}} \cdot \epsilon_0}{1 - \gamma^{c - \frac{1}{2q}}},$$

$$C_0 := C_{surr} \cdot C_{maxim}^{\max\{(1 - \frac{1}{q\delta})\delta, 0\}} \cdot C_{cover}^{\frac{\delta'}{2}} \cdot C_B' \quad \& \quad \beta_1 = \max \left\{ \frac{l}{\frac{3}{2} - \frac{1}{4}\alpha}, l - 1 + \frac{\alpha}{2} \right\}$$

*where $C_B'$ is specified in (35). Note that we need $q > 1/(2c)$ to ensure $\gamma^{c - \frac{1}{2q}} \in (0, 1)$.*

The outline of the proof is very similar to that of Theorem C.5, which is shown in Appendix C.8.1. We first derive the convergence of single iteration, and then later combine multiple convergences altogether. However, the important difference is that we cannot employ the same technique that ensures the same convergence rate for $\bar{m}_{n,1}$ and $\bar{m}_{\rho,1}$, since it can violate (23) that is required by Theorem C.5. That is, the functional representation may not be bounded or there may not be a corresponding functional norm space. Therefore, we resort to an alternative theoretical tool (Lemma C.12, with its proof in Appendix D.4) that can possibly lead to a slower convergence rate.

**Lemma C.12.** *Assume Assumption 3.1 and $C_3 > 0$ defined below has finite value. For any $\beta > 0$, we have some sequence $\delta_n = \max\{\delta_n^{(1)}, \delta_n^{(2)}\}$ such that $\delta_n \to 0$ as $n \to \infty$ and following $\phi(\cdot)$ mentioned in Theorem D.1. Here, $\delta_n^{(1)}$ and $\delta_n^{(2)}$ are defined as follows:*

$$\delta_n^{(1)} := \left( \frac{2C_1}{\sqrt{n}} \right)^{1/\beta} \quad \& \quad \delta \geq \delta_n^{(2)} \text{ implies } \frac{C}{\sqrt{n}} \cdot C_{subg} \cdot C_{ent} \cdot \delta^{\frac{\alpha_0}{2}\beta'} \geq C_3 \cdot \exp \left( \frac{-n \cdot \delta^{2\beta}}{4C_2^2} \right),$$

$$\phi(\delta) := C_\phi \cdot \delta^{\frac{\alpha_0}{2}\beta'} \quad \text{with} \quad C_\phi := C \cdot C_{subg} \cdot C_{ent} \quad \& \quad \beta' = \min\{\beta, 2\},$$

*Here are the constants:*

$$C_{\text{ent}} = C_{\text{met}}^{1/2}/(1-\alpha/2) \quad \& \quad \alpha_0 = 1 - \alpha/2 \in (0,1),$$

$$C_1 := \frac{C}{1-\alpha/2} \cdot C_{\text{met}}^{1/2} \cdot \text{diam}(\Theta; m_\infty)^{2-\alpha/2} \quad \& \quad C_2 := C \cdot \text{diam}(\Theta; m_\infty)^2.$$

$C_3 > 0$ *is the finite constant that uniformly bounds the modulus* $\tilde{\Delta}_n(\theta|\hat{\theta}_{n,t-1})$ *with* $F(\theta|\hat{\theta}_{n,t-1}) := \mathbb{E}_{S,A\sim\rho,R,S'\sim\tilde{p}(\cdots|S,A)}(\mathfrak{d}\{\Upsilon_\theta(S,A),\Psi_\pi(R,S',\Upsilon_{\hat{\theta}_{n,t-1}})\})$

$$\tilde{\Delta}_n(\theta|\hat{\theta}_{n,t-1}) := \{\hat{F}_{n,t}(\theta|\hat{\theta}_{n,t-1}) - F(\theta|\hat{\theta}_{n,t-1})\} - \{\hat{F}_{n,t}(\theta_{*,t}|\hat{\theta}_{n,t-1}) - F(\theta_{*,t}|\hat{\theta}_{n,t-1})\},$$

$$C_3 := \sup_{\theta,\hat{\theta}_{n,t-1}\in\Theta} \sup_{s,a,r,s'} |\tilde{\Delta}_n(\theta|\hat{\theta}_{n,t-1})| < \infty.$$

### C.15.1 Proof for Theorem C.11

In (MS2), we take expectation on both sides. Then, the LHS becomes $F(\theta|\hat{\theta}_{n,t-1}) - F(\theta_{*,t}|\hat{\theta}_{n,t-1})$, where the unconditioned (not conditioned on $\mathcal{Z}_t$ unlike (22)) objective function is defined as follows:

$$F(\theta|\hat{\theta}_{n,t-1}) := \mathbb{E}\left(\mathfrak{d}\{\Upsilon_\theta(S,A),\Psi_\pi(R,S',\Upsilon_{\hat{\theta}_{n,t-1}})\}\right) \quad \& \quad \theta_{*,t} = \arg\min_{\theta\in\Theta} F(\theta|\hat{\theta}_{n,t-1}).$$

For the RHS, we have following,

$$\mathbb{E}(\bar{m}_{n,1}^l(\theta,\theta_{*,t})) = \mathbb{E}\{(\bar{m}_{n,1}^2(\theta,\theta_{*,t}))^{l/2}\} \geq \{\mathbb{E}(\bar{m}_{n,1}^2(\theta,\theta_{*,t}))\}^{l/2} \quad \text{by Jensen's inequality}$$

$$= \left\{\mathbb{E}\left(\frac{1}{n}\sum_{i=1}^n m^2(\Upsilon_\theta(s_i,a_i),\Upsilon_{\theta_{*,t}}(s_i,a_i))\right)\right\}^{l/2} = \{\bar{m}_{\rho,1}^2(\theta,\theta_{*,t})\}^{l/2} = \bar{m}_{\rho,1}^l(\theta,\theta_{*,t}).$$

Thus we have $F(\theta|\hat{\theta}_{n,t-1}) - F(\theta_{*,t}|\hat{\theta}_{n,t-1}) \geq \lambda \cdot \bar{m}_{\rho,1}^l(\theta,\theta_{*,t})$, by which we can replace (MS2).

Now, we shall combine this with (MS3) and Lemma C.12 to apply Theorem D.1. We should find $\delta_n$ and $\beta > 0$ that satisfies following with $\beta' = \min\{\beta, 2\}$,

$$C_\phi \cdot \delta_n^{\frac{\alpha_0}{2}\beta'} \leq \sqrt{n} \cdot \delta_n^l, \quad \delta_n \geq \left(\frac{2C_1}{\sqrt{n}}\right)^{1/\beta}, \quad \frac{C}{\sqrt{n}} \cdot C_{subg} \cdot C_{\text{ent}} \cdot \delta_n^{\frac{\alpha_0}{2}\beta'} \geq C_3 \cdot \exp\left(\frac{-n\cdot\delta_n^{2\beta}}{4C_2^2}\right). \tag{34}$$

The first condition of (34) can be rewritten as follows,

$$\delta_n \geq \left(\frac{C_\phi}{\sqrt{n}}\right)^{\frac{1}{l-\alpha_0\beta'/2}} \quad \Rightarrow \quad \text{requires to confirm } \alpha_0\beta' < 2l.$$

Remember that $\beta > 0$ is a free variable that we choose. There can be two cases: (i) $l \leq 2 + \alpha_0$ and (ii) $l > 2 + \alpha_0$. We will select $\beta$ differently for each case (based on the first two conditions of (34)), and we will hereafter denote its selected value as $\beta_0$.

Let us consider the first case. To meet the first two conditions, we solve $l = \alpha_0\beta'/2 = \beta$. Temporarily asuming $\beta \leq 2$ (which also needs to be confirmed later), we obtain $\beta_0 = l/(1+\alpha_0/2)$. We can confirm that $l \leq 2 + \alpha_0$ implies $\beta_0 \leq 2$, and can see $\alpha_0\beta' = \alpha_0\beta_0 < 2l$.

In the second case, we let $\beta_0 = 2$. This leads to $\beta' = 2$, and the first two conditions of (34) can be summarized as follows,

$$\delta_n \geq \left(\frac{C_\phi}{\sqrt{n}}\right)^{\frac{1}{l-\alpha_0}} \quad \& \quad \delta_n \geq \left(\frac{2C_1}{\sqrt{n}}\right)^{1/2} \quad \Rightarrow \quad \therefore \delta_n \geq \left(\frac{C_\phi}{\sqrt{n}}\right)^{\frac{1}{l-\alpha_0}}.$$

We confirm that $\alpha_0\beta' = 2\alpha_0 < 2(l-2) < 2l$ holds. Incorporating Cases (i) and (ii), we let

$$\beta_0 = \min\left\{\frac{l}{1+\alpha_0/2}, 2\right\} \leq 2.$$

With the selected $\beta_0$, the first two conditions of (34) become following,

$$\text{Case (i) } l \leq 2 + \alpha_0 \ : \ \delta_n \geq \max\left\{C_\phi, 2C_1\right\}^{1/\beta_0} \cdot \left(\frac{1}{\sqrt{n}}\right)^{1/\beta_0} \quad \text{with} \quad \beta_0 = \frac{l}{1 + \alpha_0/2},$$

$$\text{Case (ii) } l > 2 + \alpha_0 \ : \ \delta_n \geq \left(\frac{C_\phi}{\sqrt{n}}\right)^{\frac{1}{l - \alpha_0}} \quad \text{for sufficiently large } n \in \mathbb{N}.$$

Since $l \leq 2 + \alpha_0$ is equivalent to $l - \alpha_0 \leq \frac{l}{1 + \alpha_0/2}$, we have

$$\delta_n \geq \left(\max\{C_\phi, 2C_1\} \cdot \frac{1}{\sqrt{n}}\right)^{1/\beta_1} \quad \text{with} \quad \beta_1 = \max\left\{\frac{l}{1 + \alpha_0/2}, l - \alpha_0\right\}.$$

Now we select $\delta_n$ as follows,

$$\delta_n = \underbrace{\left(\max\left\{C_\phi, 2C_1, (2C_2)^{\beta_1/\beta_0}\right\}\right)^{1/\beta_1}}_{\overset{let}{=}C_A} \cdot \left(\sqrt{\frac{\log n}{n}}\right)^{1/\beta_1},$$

and we can confirm that third statement of (34) holds as follows. Note that the selected value of $\beta$ is $\beta_0$, not $\beta_1$. Using some constant $C_{\text{LHS}} > 0$ and $\beta_0 \leq \beta_1$ along with $C_A^{2\beta_0} \geq 4C_2^2$, we have

$$\text{LHS} = \frac{C_{\text{LHS}}}{\sqrt{n}} \cdot \delta_n^{\frac{\alpha_0\beta_0}{2}} = \frac{C_{\text{LHS}}}{\sqrt{n}} \cdot \left(\sqrt{\frac{\log n}{n}}\right)^{\frac{1}{\beta_1} \cdot \frac{\alpha_0\beta_0}{2}} \geq C_{\text{LHS}} \cdot (\log n)^{\alpha_0/4} \cdot \left(\frac{1}{\sqrt{n}}\right)^{1 + \frac{\alpha_0}{2}},$$

$$\text{RHS} = C_3 \cdot \exp\left(\frac{-n \cdot \delta_n^{2\beta_0}}{4C_2^2}\right) = C_3 \cdot \exp\left\{\frac{-n \cdot C_A^{2\beta_0}}{4C_2^2} \cdot \left(\frac{\log n}{n}\right)^{\beta_0/\beta_1}\right\}$$

$$\leq C_3 \cdot \exp(-\log n) = \frac{C_3}{n}.$$

Since $\alpha_0 \in (0, 1)$ as specified in Lemma C.12, we can see that LHS is larger than RHS for sufficiently large $n$, satisfying the third statement of (34).

Now we can apply peeling argument (Theorem D.1) to obtain the following,

$$\bar{m}_{\rho,1}(\hat{\theta}_{n,t}, \theta_{*,t}) = O_P(\delta_n) = C_\phi^{\frac{1}{l-1+\alpha}} \cdot C_A \cdot O_P\left\{\left(\frac{\log n}{n}\right)\right\}^{\frac{1}{2\beta_1}},$$

$$\therefore \ \bar{m}_{d_\pi,1}(\hat{\theta}_{n,t}, \theta_{*,t}) = C_{cover}^{1/2} \cdot \max\left\{C_\phi, 2C_1, (2C_2)^{\beta_1/\beta_0}\right\}^{1/\beta_1} \cdot O_P\left\{\left(\frac{\log n}{n}\right)^{\frac{1}{2\beta_1}}\right\}.$$

We can copy the logic of Appendix C.8.3 that we used to prove Theorem C.5. We can apply Lemma C.7 (with satisfaction of its condition deferred in Appendix D.5). We can let

$$\delta' = \min\{\delta, \frac{1}{q}\} \quad \& \quad C'_B := \max\left\{C_\phi, 2C_1, (2C_2)^{\beta_1/\beta_0}\right\}^{\delta'/\beta_1},$$

and further let

$$T = \left\lfloor \frac{1}{c - \frac{1}{2q}} \cdot \frac{\delta'}{2\beta_1} \cdot \log_{1/\gamma} N \right\rfloor \quad \Rightarrow \quad \left(\frac{1}{N}\right)^{\frac{\delta'}{2\beta_1}} \approx \zeta^T.$$

Ignoring the flooring effect, this leads to

$$\frac{\log n}{n} = \frac{\log(N/T)}{N/T} \leq \frac{T}{N} \cdot \log N = \frac{1}{c - \frac{1}{2q}} \cdot \frac{\delta'}{2\beta_1} \cdot \frac{1}{\log(1/\gamma)} \cdot \frac{(\log N)^2}{N}.$$

Then, applying the same logic as Appendix C.8.3 (extending towards multiple iterations), we finally obtain bound $\bar{\eta}_{d_\pi, q}(\Upsilon_{\theta_T}, \Upsilon_\pi)$ by following,

$$\frac{1}{1 - \gamma^{c - \frac{1}{2q}}} \cdot C_{surr} \cdot C_{maxim}^{\max\{(1 - \frac{1}{q\delta})\delta, 0\}} \cdot C_{cover}^{\frac{\delta'}{2}} \cdot C_B' \cdot O_P\left\{ \left( \frac{1}{\log(1/\gamma)} \cdot \frac{(\log N)^2}{N} \right)^{\frac{\delta'}{2\beta_1}} \right\}$$

$$+ \frac{1}{1 - \gamma^{c - \frac{1}{2q}}} \cdot 2^{1 - \frac{1}{2q}} \cdot \epsilon_0,$$

$$\text{with} \quad \beta_1 = \max\left\{ \frac{l}{\frac{3}{2} - \frac{1}{4}\alpha}, l - 1 + \frac{\alpha}{2} \right\} \quad \& \quad \beta_0 = \min\left\{ \frac{l}{\frac{3}{2} - \frac{1}{4}\alpha}, 2 \right\},$$

$$\text{and} \quad C_B' := \max\left\{ C_\phi, 2C_1, (2C_2)^{\beta_1/\beta_0} \right\}^{\delta'/\beta_1} \cdot \left\{ \frac{1}{c - \frac{1}{2q}} \cdot \frac{\delta'}{2\beta_1} \right\}^{\frac{\delta'}{2\beta_1}}. \tag{35}$$

where $C_\phi$, $C_1$, $C_2$ are defined in Lemma C.12.

### C.15.2 Proof for Corollary B.8

For the case of $(\eth, m, \eta) = (\mathrm{MMD}_{cou}^2, \mathrm{MMD}_{cou}, \mathbb{W}_p)$, we have $q = p$, $\delta = 1/p$, $l = 2$, $c = 1$, $C_{subg} = C_{max}^{(k)} = C(1 + \gamma^{1 - d/2}) \cdot \sup_{\theta \in \Theta} \sup_{s,a} \mathcal{E}_{cou}^{1/2}(\Upsilon_\theta(s, a)) \overset{let}{=} C_{max}^{(cou)}$, $C_{surr} = C \cdot D^{1 - \frac{1}{2p}} \cdot V_d^{\frac{1}{2p}}$.

(MS1) and (MS2) are straightforward from Appendix C.9.2. Note that $C_{max}^{(k)} = C_{max}^{(cou)}$ (29) can be derived as follows (for unbounded kernel $k_{cou}$). Based on $\|\kappa_\mu\|_{\mathcal{H}}^2 = \langle \kappa_\mu, \kappa_\mu \rangle_{\mathcal{H}} = \mathbb{E}\{k_{cou}(\mathbf{X}, \mathbf{X}')\}$ with $\mathbf{X}, \mathbf{X}' \overset{iid}{\sim} \mu$, we can use $\|\kappa_\mu\|_{\mathcal{H}} = \mathcal{E}_{cou}^{1/2}(\mu)$ to obtain the following,

$$\mathrm{MMD}_{cou}\left\{ \Upsilon_\theta(s, a), \Psi_\pi(r, s', \Upsilon_{\hat{\theta}_{n,t-1}}) \right\} \leq C \cdot \left\{ \mathcal{E}_{cou}^{1/2}(\Upsilon_\theta(s, a)) + \mathcal{E}_{cou}^{1/2}\{\Psi_\pi(r, s', \Upsilon_{\hat{\theta}_{n,t-1}})\} \right\},$$

$$\mathcal{E}_{cou}^{1/2}(\Psi_\pi(r, s', \Upsilon_{\hat{\theta}_{n,t-1}})) \leq \sum_{a' \in \mathcal{A}} \pi(a'|s') \cdot \|\kappa_{(g_{r,\gamma})\#\Upsilon_{\hat{\theta}_{n,t-1}}(s', a')}\|_{\mathcal{H}}$$

$$= \sum_{a' \in \mathcal{A}} \pi(a'|s') \cdot \mathcal{E}_{cou}^{1/2}((g_{r,\gamma})\#\Upsilon_{\hat{\theta}_{n,t-1}}(s', a')).$$

Since we have the following based on definition of $\kappa_0^{(cou)}$,

$$\mathcal{E}_{cou}\left\{ (g_{r,\gamma})\#\Upsilon_{\hat{\theta}_{n,t-1}}(s', a') \right\} = \mathbb{E}\left\{ \kappa_0^{(cou)}(\gamma Z_\alpha(s', a'; \hat{\theta}_{n,t-1}) - \gamma Z_\beta(s', a'; \hat{\theta}_{n,t-1})) \right\}$$

$$= \gamma^{2-d} \cdot \mathcal{E}_{cou}(\Upsilon_{\hat{\theta}_{n,t-1}}(s', a')).$$

This gives us $C_{max}^{(k)} = C_{max}^{(cou)}$.

(MS3) can be shown analogously with Section C.10. (MS4) regards the model complexity that we assume. $C_3 > 0$ can be shown to be bounded by using $\sup_{\theta \in \Theta} \sup_{s,a} \mathcal{E}_{cou}(\Upsilon_\theta(s, a)) < \infty$, which is assumed in Corollary B.8.

We also obtain $C_{surr} = C \cdot D^{1 - \frac{1}{2p}} \cdot V_d^{\frac{1}{2p}}$ where $V_d$ is the volume of a unit ball in $\mathbb{R}^d$, by Theorem 1.1 of [11] as follows,

$$\mathbb{W}_1(\mu, \nu) \leq C \cdot D^{1/2} \cdot V_d^{1/2} \cdot \mathrm{MMD}_{cou}(\mu, \nu),$$

$$\mathbb{W}_p(\mu, \nu) \leq D^{1 - \frac{1}{p}} \cdot \mathbb{W}_1^{1/p}(\mu, \nu) \leq C \cdot D^{1 - \frac{1}{2p}} \cdot V_d^{\frac{1}{2p}} \cdot \mathrm{MMD}_{cou}^{1/p}(\mu, \nu).$$

Then applying this into Theorem C.11 gives us following bound, since we have $\beta_1 = 2/(\frac{3}{2} - \frac{1}{4}\alpha)$,

$$\overline{\mathbb{W}}_{p, d_\pi, p}(\Upsilon_{\theta_T}, \Upsilon_\pi) \leq \frac{D^{1 - \frac{1}{2p}} \cdot V_d^{\frac{1}{2p}}}{1 - \gamma^{1 - \frac{1}{2p}}} \cdot C_{cover}^{\frac{1}{2p}} \cdot C_B' \cdot O_P\left\{ \frac{1}{\log(1/\gamma)} \cdot \left( \frac{(\log N)^2}{N} \right)^{\frac{6 - \alpha}{16p}} \right\},$$

$$C_B' := \frac{1}{1 - \alpha/2} \cdot \left( \max\left\{ C_{max}^{(cou)} \cdot C_{met}^{1/2}, C_{met}^{1/2} \cdot \mathrm{diam}(\Theta; \mathbb{W}_{p,\infty})^{2 - \alpha/2}, \mathrm{diam}(\Theta; \mathbb{W}_{p,\infty})^2 \right\} \right)^{\frac{6 - \alpha}{8p}},$$

where $C_B'$ is defined accordingly to (35).

### C.15.3 Proof for Corollary B.3

For the case of $(\eth, m, \eta) = (\mathrm{MMD}^2_{\beta=1}, \mathbb{W}_1, \mathbb{W}_1)$, we have $q = 1$, $\delta = 1$, $l = \frac{r(2d+3)-1}{r-1}$, $c = 1$, $C_{subg} = C$, $C_{surr} = 1$, $\epsilon_0 = 0$. We need $\beta > 1$ for $q = 1$ as we mentioned in (16).

(MS1) holds, as we have already shown in Appendix C.10. This can be copied since $l_2^2$ is a special case of $\mathrm{MMD}^2_{\beta=1}$ for $d = 1$. However, (MS2) should be changed as follows, based on Proposition 17 of [36]: when $M_r < \infty$, we have

$$C(d, r, \beta, M_r) \cdot \mathrm{MMD}^{\rho'}_\beta(\mu, \nu) \geq \mathbb{W}_1(\mu, \nu) \quad \text{with } \rho' = \frac{r-1}{r(d+\beta/2+1)-\beta/2}. \tag{36}$$

Refer to their Proposition 17 for the definition of constant $C(d, r, \beta, M_r) > 0$. This can be developed into

$$F_{n,t}(\theta|\hat{\theta}_{n,t-1}) - F_{n,t}(\theta_{*,t}|\hat{\theta}_{n,t-1}) = \overline{\mathrm{MMD}}^2_{\beta,n,1}(\theta, \theta_{*,t})$$

$$= \frac{1}{n} \sum_{i=1}^n \mathrm{MMD}^2_\beta(\Upsilon_\theta(s_i, a_i), \Upsilon_{\theta_{*,t}}(s_i, a_i))$$

$$\geq C(d, r, \beta, M_r)^{-2/\rho'} \cdot \frac{1}{n} \sum_{i=1}^n \left\{ \mathbb{W}_1^2(\Upsilon_\theta(s_i, a_i), \Upsilon_{\theta_{*,t}}(s_i, a_i)) \right\}^{1/\rho'}$$

$$\geq C(d, r, \beta, M_r)^{-2/\rho'} \cdot \left\{ \frac{1}{n} \sum_{i=1}^n \mathbb{W}_1^2(\Upsilon_\theta(s_i, a_i), \Upsilon_{\theta_{*,t}}(s_i, a_i)) \right\}^{1/\rho'} \quad \because \frac{1}{\rho'} > 2$$

$$= C(d, r, \beta, M_r)^{-2/\rho'} \cdot \overline{\mathbb{W}}^{2/\rho'}_{1,n,1}(\theta, \theta_{*,t}).$$

Since we are limiting to $\beta = 1$, (MS2) is satisfied with $\lambda = C(d, r, \beta, M_r)^{-2/\rho'}$ and $l = \frac{2}{\rho'} = \frac{r(2d+3)-1}{r-1} \geq 2d + 3$.

(MS3) can be shown analogously to C.10. In the part of applying Hoeffding's inequality (generalized version for unbounded distributions, e.g., Theorem 2.6.2), we should assume $\sup_{\theta \in \Theta} \sup_{s,a} \mathbb{E}\|Z(s, a; \theta)\| < \infty$ and $\sup_{s,a} \|R(s, a)\|_{\psi_2} < \infty$.

(MS4) is what we assume. Finiteness of $C_3 > 0$ can be shown by using the same trick that holds between energy distance and Wassertein-1 metric (as we mentioned in Appendix C.10),

$$|\tilde{\Delta}_n(\theta|\hat{\theta}_{n,t-1})| \leq |\hat{F}_{n,t}(\theta|\hat{\theta}_{n,t-1}) - \hat{F}_{n,t}(\theta_{*,t}|\hat{\theta}_{n,t-1})| + |F(\theta|\hat{\theta}_{n,t-1}) - F(\theta_{*,t}|\hat{\theta}_{n,t-1})|$$

$$\leq \left| \frac{2}{n} \sum_{i=1}^n (f_i^\theta - f_i^{\theta_{*,t}}) - \frac{1}{n} \sum_{i=1}^n (g_i^\theta - g_i^{\theta_{*,t}}) \right| + \left| 2 \cdot \mathbb{E}(f_1^\theta - f_1^{\theta_{*,t}}) - \mathbb{E}(g_1^\theta - g_1^{\theta_{*,t}}) \right|$$

$$\leq 8 \cdot \sup_{\theta \in \Theta} \mathbb{W}(\theta; \theta_{*,t}) \leq 8 \cdot \mathrm{diam}(\Theta; \mathbb{W}_{1,\infty}) < \infty \quad \text{by (MS4)}.$$

Then, we can apply Theorem C.11 to obtain the following bound, since $\beta_1 = l - 1 + \alpha/2$.

$$\overline{\mathbb{W}}_{1,d_\pi,1}(\Upsilon_{\theta_T}, \Upsilon_\pi) \leq \frac{1}{1-\gamma^{1/2}} \cdot C_{cover}^{1/2} \cdot O_P\left\{ \left( \frac{1}{\log(1/\gamma)} \cdot \frac{(\log N)^2}{N} \right)^{\frac{1}{2(l-1)+\alpha}} \right\},$$

$$C_B' := \left( \frac{1}{1-\alpha/2} \max\left\{ 1, C_{met}^{1/2} \cdot \mathrm{diam}(\Theta; \mathbb{W}_{1,\infty})^{2-\alpha/2}, \mathrm{diam}(\Theta; \mathbb{W}_{1,\infty})^{\frac{l-1+\alpha/2}{2}} \right\} \right)^{\frac{1}{l-1+\alpha/2}}.$$

### C.16 Proof for Corollary B.9

We acknowledge that the following proof is based upon Chapter 7 of [20] and Section 6.4 of [51]. Throughout this proof, we will resort to the following trick shown by [51],

$$|\sqrt{2s} - \sqrt{s+t}| \leq |\sqrt{s} - \sqrt{t}| \leq (1+\sqrt{2}) \cdot |\sqrt{2s} - \sqrt{s+t}| \quad \text{for } \forall s, t \geq 0.$$

Now, let us define the following functions. Define $\mathbb{E}_{s,a}$ to be expectation taken with respect to $s, a \sim \rho$, $\mathbb{E}_{r,s'}$ to be expectation taken with respect to $r, s' \sim \tilde{p}(\cdots|s, a)$, $\mathbb{E}_{z' \sim f_{\hat{\theta}_{n,t-1}}}(\cdots|s',\pi)$ to be expectation

taken with respect to $z' \sim f_{\hat{\theta}_{n,t-1}}(s', \pi)$, where $f_{\hat{\theta}_{n,t-1}}(\cdot|s', \pi) := \int f_{\hat{\theta}_{n,t-1}}(\cdot|s', a')\mathrm{d}\pi(a'|s')$.

$$M(\theta) := \mathbb{E}_{s,a}\mathbb{E}_{r,s'}\mathbb{E}_{z \sim f_{\hat{\theta}_{n,t-1}}(\cdots|s',\pi)}\left\{ -\log\left(\frac{f_\theta(r+\gamma z'|s,a) + f_{\theta_{*,t}}(r+\gamma z'|s,a)}{2 \cdot f_{\theta_{*,t}}(r+\gamma|s,a)}\right.\right.$$
$$\left.\left.\times \mathbf{1}\big(f_{\theta_{*,t}}(r+\gamma z'|s,a) > 0\big)\right)\right\},$$

$$\hat{M}_n(\theta) := \frac{1}{n}\sum_{i=1}^{n}\mathbb{E}_{z_i \sim f_{\hat{\theta}_{n,t-1}}(\cdot|s_i',\pi)}\left\{ -\log\frac{f_\theta(r_i+\gamma z_i'|s_i,a_i) + f_{\theta_{*,t}}(r_i+\gamma z_i'|s_i,a_i)}{2 \cdot f_{\theta_{*,t}}(r_i+\gamma z_i'|s_i,a_i)}\right\}.$$

Note that it is more appropriate to write it as $M(\theta|\hat{\theta}_{n,t-1})$ and $\hat{M}_n(\theta|\hat{\theta}_{n,t-1})$. However, we will take $\hat{\theta}_{n,t-1} \in \Theta$ (along with the corresponding $\theta_{*,t}$) as given, and simplify it as $M(\theta) = M(\theta|\hat{\theta}_{n,t-1})$ and $\hat{M}_n(\theta) = \hat{M}_n(\theta|\hat{\theta}_{n,t-1})$. Since the inner part of $\mathbb{E}_{s,a}$ of $M(\theta)$ is equivalent to $\mathrm{KLD}(f_{\theta_{*,t}}(\cdot|s,a)\|\frac{f_\theta(\cdot|s,a)+f_{\theta_{*,t}}(\cdot|s,a)}{2}) \geq 0$ where equality holds under $\theta = \theta_{*,t}$, $\theta_{*,t}$ becomes the minimizer of $M(\theta)$ with $M(\theta_{*,t}) = 0$.

### C.16.1 Quadratic lower bound

For fixed $s, a$, denote $p_0(\cdot) = f_{\theta_{*,t}}(\cdot|s,a)$ and $p(\cdot) = f_\theta(\cdot|s,a)$. Then, the terms inside $\mathbb{E}_{s,a}$ is simplified as follows with $q = \frac{1}{2}(p + p_0)$,

$$\mathbb{E}_{z \sim p_0}\left\{ -\log\frac{p(z)+p_0(z)}{2p_0(z)} \cdot \mathbf{1}\big(p_0(z) > 0\big)\right\} = \mathbb{E}_{z \sim p_0}\left\{ -\log\frac{q(z)}{p_0(z)} \cdot \mathbf{1}\big(p_0(z) > 0\big)\right\}$$

$$\geq 2 \cdot \int_{p_0 > 0}\left(1 - \sqrt{\frac{q(z)}{p_0(z)}}\right)p_0(z)\mathrm{d}z \qquad \because \frac{1}{2}\log x < \sqrt{x} - 1$$

$$= 2 \cdot \left(1 - \int_{p_0 > 0}\sqrt{p_0(z)} \cdot \sqrt{q(z)}\mathrm{d}z\right) \geq \int_{p_0 > 0}\left(p_0(z) + q(z) - 2 \cdot \sqrt{p_0(z)} \cdot \sqrt{q(z)}\right)\mathrm{d}z$$

$$= \int_{p_0 > 0}\left(\sqrt{p_0(z)} - \sqrt{q(z)}\right)^2\mathrm{d}z = \frac{1}{2} \cdot \int_{p_0 > 0}\left(\sqrt{2p_0(z)} - \sqrt{(p_0 + p)(z)}\right)^2\mathrm{d}z$$

$$\geq \frac{1}{2} \cdot \frac{1}{(1+\sqrt{2})^2} \cdot \int_{p_0 > 0}\left(\sqrt{p_0(z)} - \sqrt{p(z)}\right)^2\mathrm{d}z = \frac{1}{(1+\sqrt{2})^2} \cdot h_0^2(p_0, p),$$

where $h_0^2(\mu_1, \mu_2) := \frac{1}{2}\int_{p_0 > 0}(\sqrt{f_1(z)} - \sqrt{f_2(z)})^2\mathrm{d}z$ where $f_1, f_2$ are corresponding pdf's of $\mu_1, \mu_2 \in \mathcal{P}$. Allowing abuse of notation, we will allow putting densities instead of probabilty measures for the inputs.

Then, since $M(\theta_{*,t}) = 0$, we have the following where $\bar{h}_{0,\rho,1}$ is defined accordingly to (8),

$$M(\theta) - M(\theta_{*,t}) = \mathbb{E}_{s,a}\mathbb{E}_{z \sim f_{\theta_{*,t}}(\cdot|s,a)}\left\{ -\log\frac{f_\theta(z|s,a) + f_{\theta_{*,t}}(z|s,a)}{2 \cdot f_{\theta_{*,t}}(z|s,a)} \cdot \mathbf{1}\big(f_{\theta_{*,t}}(z|s,a) > 0\big)\right\}$$

$$\geq \mathbb{E}_{s,a \sim \rho}\left\{ \frac{1}{(1+\sqrt{2})^2} \cdot h_0^2\big(f_{\theta_{*,t}}(\cdot|s,a), f_\theta(\cdot|s,a)\big)\right\} = \frac{1}{(1+\sqrt{2})^2} \cdot \bar{h}_{0,\rho,1}^2(\theta_{*,t}, \theta).$$

Here we have $\bar{h}_{0,\rho,1}^2(\theta_1, \theta_2) := \mathbb{E}_{s,a \sim \rho}\{h_0^2(\Upsilon_{\theta_1}(s,a), \Upsilon_{\theta_2}(s,a))\}$ with $h_0^2(\Upsilon_{\theta_1}(s,a), \Upsilon_{\theta_2}(s,a))$ takes the integral over the support of $\Upsilon_{\theta_{*,t}}(s,a)$, or $f_{\theta_{*,t}}(\cdot|s,a) > 0$.

### C.16.2 Convergence of modulus

We recall the following Theorem 6.13 of [51].

**Theorem C.13.** *For any class $\mathcal{F}$ of measurable functions $f : \mathcal{X} \to \mathbb{R}$ with $\|f\|_{P_0,B} \leq \delta$, with $X_i \overset{iid}{\sim} P_0$, we have following with $\lesssim$ indicating "bounded by some multiplicative constant,"*

$$\mathbb{E}\left\{\sup_{f \in \mathcal{F}} \left| \frac{1}{n} \sum_{i=1}^{n} f(X_i) - \mathbb{E}f(X_i) \right| \right\} \lesssim \frac{1}{\sqrt{n}} \cdot J_{[\,]}(\delta, \mathcal{F}, \|\cdot\|_{P_0,B}) \cdot \left( 1 + \frac{J_{[\,]}(\delta, \mathcal{F}, \|\cdot\|_{P_0,B})}{\delta^2 \sqrt{n}} \right),$$

$$J_{[\,]}(\delta, \mathcal{F}, \|\cdot\|) := \int_0^\delta \sqrt{\log \mathcal{N}_{[\,]}(\epsilon, \mathcal{F} \cup \{0\}, \|\cdot\|)} d\epsilon \leq \int_0^\delta \sqrt{1 + \log \mathcal{N}_{[\,]}(\epsilon, \mathcal{F}, \|\cdot\|)} d\epsilon,$$

$$\|f\|_{P_0,B}^2 := 2 \cdot \mathbb{E}_{X \sim P_0}\left\{ \exp(|f(X)|) - 1 - |f(X)| \right\}.$$

Let us define the following.

$$m_\theta(s, a, r, s') := \mathbb{E}_{z' \sim f_{\hat{\theta}_{n,t-1}}(\cdot|s',\pi)}\left\{ -\log\left( \frac{f_\theta(r + \gamma z'|s,a) + f_{\theta_{*,t}}(r + \gamma z'|s,a)}{2 \cdot f_{\theta_{*,t}}(r + \gamma z'|s,a)} \right.\right.$$
$$\left.\left. \times \mathbf{1}\big(f_{\theta_{*,t}}(r + \gamma z'|s,a) > 0\big) \right) \right\},$$

$$\therefore \hat{M}_n(\theta) = \frac{1}{n} \sum_{i=1}^{n} m_\theta(s_i, a_i, r_i, s_i') = \frac{1}{n} \sum_{i=1}^{n} \left\{ m_\theta(s_i, a_i, r_i, s_i') - m_{\theta_{*,t}}(s_i, a_i, r_i, s_i') \right\},$$

$$M(\theta) = \mathbb{E}_{s,a,r,s'}\left\{ m_\theta(s, a, r, s') \right\} = \mathbb{E}\left\{ m_\theta(s, a, r, s') - m_{\theta_{*,t}}(s, a, r, s') \right\}.$$

We also define $\mathcal{M}_\Theta := \{m_\theta(\cdots) : \theta \in \Theta\} = \{m_\theta(\cdots) - m_{\theta_{*,t}}(\cdots) : \theta \in \Theta\}$, since we have $m_{\theta_{*,t}}(\cdots) = 0$. Since $m_\theta(s, a, r, s') \leq \log 2$ and $x < \log 2$ implies $\exp(|x|) - 1 - |x| \leq 4 \cdot (\exp(-x/2) - 1)^2$, we have the following,

$$\|m_\theta\|_{P_0,B}^2 \leq 2 \cdot \mathbb{E}_{s,a}\mathbb{E}_{r,s'}\left\{ 4 \cdot \left( \exp\big(-m_\theta(s, a, r, s')/2\big) - 1 \right)^2 \right\}$$

$$= 8 \cdot \mathbb{E}_{s,a \sim \rho}\mathbb{E}_{r,s'}\mathbb{E}_{z' \sim f_{\hat{\theta}_{n,t-1}}(\cdot|s,a)}\left\{ \left( \sqrt{\frac{f_\theta(r + \gamma z'|s,a) + f_{\theta_{*,t}}(r + \gamma z'|s,a)}{f_{\theta_{*,t}}(r + \gamma z'|s,a)}} \right)^2 \right.$$
$$\left. \times \mathbf{1}\big(f_{\theta_{*,t}}(z|s,a) > 0\big) \right\}.$$

Note that $\mathbb{E}_{r,s' \sim \tilde{p}(\cdots|s,a)}\mathbb{E}_{z' \sim f_{\hat{\theta}_{n,t-1}}(\cdot|s,a)}$ can be reduced into $\mathbb{E}_{z \sim f_{\theta_{*,t}}(\cdot|s,a)}$. Again, letting $p_0(\cdot) = f_{\theta_{*,t}}(\cdot|s,a)$ and $p(\cdot) = f_\theta(\cdot|s,a)$ for fixed $s,a$, along with $q = (p + p_0)/2$, the term within $\mathbb{E}_{s,a}$ can be rewritten as follows,

$$\mathbb{E}_{z \sim p_0}\left\{ \left( \sqrt{\frac{q(z)}{p_0(z)}} - 1 \right)^2 \cdot \mathbf{1}\big(p_0(z) > 0\big) \right\} = \int_{p_0 > 0} \left( \sqrt{\frac{q(z)}{p_0(z)}} - 1 \right)^2 p_0(z) dz$$

$$= \int_{p_0 > 0} \left( \sqrt{q(z)} - \sqrt{p_0(z)} \right)^2 dz = \int_{p_0 > 0} \left( \sqrt{\frac{p_0 + p}{2}}(z) - \sqrt{p_0(z)} \right)^2 dz$$

$$= \frac{1}{2} \cdot \int_{p_0 > 0} \left( \sqrt{p_0 + p}(z) - \sqrt{2p_0(z)} \right)^2 dz \leq \frac{1}{2} \int_{p_0 > 0} \left( \sqrt{p(z)} - \sqrt{p_0(z)} \right)^2 dz = h_0^2(p_0, p).$$

Then, this leads to $\|m_\theta\|_{P_0,B} \leq 2\sqrt{2} \cdot \bar{h}_{0,\rho,1}(\theta_{*,t}, \theta)$, meaning that $\bar{h}_{0,\rho,1}(\theta_{*,t}, \theta) \leq \frac{\delta}{2\sqrt{2}}$ implies $\|m_\theta\|_{P_0,B} \leq \delta$. Then, applying Theorem C.13 gives us the following with modulus constructed as $\Delta_n(\theta|\hat{\theta}_{n,t-1}) = (\hat{M}_n(\theta) - M(\theta)) - (\hat{M}_n(\theta_{*,t}) - M(\theta_{*,t}))$,

$$\mathbb{E}\left\{ \sup_{\theta \in \Theta : \bar{h}_{0,\rho,1}(\theta_{*,t},\theta) < \frac{\delta}{2\sqrt{2}}} \left| (\hat{M}_n(\theta) - M(\theta)) - (\hat{M}_n(\theta_{*,t}) - M(\theta_{*,t})) \right| \right\}$$

$$\leq \frac{C}{\sqrt{n}} \cdot J_{[\,]}(\delta, \mathcal{M}_\Theta, \|\cdot\|_{P_0,B}) \cdot \left( 1 + \frac{J_{[\,]}(\delta, \mathcal{M}_\Theta, \|\cdot\|_{P_0,B})}{\delta^2 \sqrt{n}} \right). \tag{37}$$

Now let us simplify the term $J_{[\,]}(\delta, \mathcal{M}_\Theta, \|\cdot\|_{P_0,B})$. Assume a bracketing pair $m_p(\cdots) \leq m_q(\cdots)$, which may not be elements of $\mathcal{M}_\Theta$. That is, we have $p(\cdot|\cdots) \geq q(\cdot|\cdots)$ (where we can assume $q(\cdot|\cdots) \geq 0$ but not necessarily conditional probability densities), and

$$m_p(s,a,r,s') = \mathbb{E}_{z'\sim f_{\hat{\theta}_{n,t-1}}(\cdot|s,a)}\left\{ -\log\left(\frac{p(z|s,a) + f_{\theta_{*,t}}(z|s,a)}{2\cdot f_{\theta_{*,t}}(z|s,a)}\cdot \mathbf{1}\big(f_{\theta_{*,t}}(z|s,a) > 0\big)\right)\right\},$$

$$m_q(s,a,r,s') = \mathbb{E}_{z'\sim f_{\hat{\theta}_{n,t-1}}(\cdot|s,a)}\left\{ -\log\left(\frac{q(z|s,a) + f_{\theta_{*,t}}(z|s,a)}{2\cdot f_{\theta_{*,t}}(z|s,a)}\cdot \mathbf{1}\big(f_{\theta_{*,t}}(z|s,a) > 0\big)\right)\right\}.$$

The term $\|m_p(\cdots) - m_q(\cdots)\|_{P_0,B}$ is defined as follows,

$$\mathbb{E}_{s,a}\mathbb{E}_{r,s'}\left[\left\{\exp\big(|m_p(s,a,r,s') - m_q(s,a,r,s')|\big) - 1 - |m_p(s,a,r,s') - m_q(s,a,r,s')|\right\}^2\right].$$

Employing the same trick that we have used before, the term inside $\mathbb{E}_{s,a}$ can be bounded as follows with $p_0(z) = f_{\theta_{*,t}}(\cdot|s,a)$ for fixed $s,a$,

$$\leq 4 \cdot \mathbb{E}_{z\sim p_0}\left\{\left(\sqrt{\frac{p(z) + p_0(z)}{q(z) + p_0(z)}} - 1\right)^2 \cdot \mathbf{1}\big(p_0(z) > 0\big)\right\}$$

$$= 4 \cdot \mathbb{E}_{z\sim p_0}\left\{\left(\sqrt{\frac{p_1(z)}{p_2(z)}} - 1\right)^2 \cdot \mathbf{1}\big(p_0(z) > 0\big)\right\} \quad \text{with } p_1 = p + p_0,\ p_2 = q + p_0$$

$$= 4 \cdot \int_{p_0 > 0}\left(\sqrt{\frac{p_1(z)}{p_2(z)}} - 1\right)^2 p_0(z)\mathrm{d}z \leq 4 \cdot \int_{p_0 > 0}\left(\sqrt{\frac{p_1(z)}{p_2(z)}} - 1\right)^2 \cdot p_2(z)\mathrm{d}z$$

$$= 8 \cdot \int_{p_0 > 0}\left(\sqrt{\frac{p + p_0}{2}(z)} - \sqrt{\frac{q + p_0}{2}(z)}\right)^2 \mathrm{d}z$$

$$\leq 8 \cdot \int_{p_0 > 0}\left(\sqrt{p(z)} - \sqrt{q(z)}\right)^2 \mathrm{d}z = 16 \cdot h_0^2(p,q)$$

Then, we have $\|m_p(\cdots) - m_q(\cdots)\|_{P_0,B} \leq 4 \cdot \|\sqrt{p(\cdot|\cdots)} - \sqrt{q(\cdot|\cdots)}\|_{L_2,\rho}$. $\|\cdot\|_{L_2,\rho}$ is defined accordingly to (24). Thus we have the following,

$$\log\mathcal{N}_{[\,]}(\mathcal{M}_\Theta, \|\cdot\|_{P_0,B}, \epsilon) \leq \log\mathcal{N}_{[\,]}(\widetilde{\mathcal{F}}_\Theta^{1/2}, \|\cdot\|_{L_2,\rho}, 4/\epsilon).$$

Then, this leads to following by assumption,

$$J_{[\,]}(\delta, \mathcal{M}_\Theta, \|\cdot\|_{P_0,B}) \leq \int_0^\delta \sqrt{\log\mathcal{N}_{[\,]}(\widetilde{\mathcal{F}}_\Theta^{1/2}, \|\cdot\|_{L_2,\rho}, \epsilon/2)}\mathrm{d}\epsilon \leq C \cdot C_{brack}^{1/2}\cdot\frac{1}{1-\alpha/2}\cdot\delta^{1-\alpha/2}.$$

We will later select $\delta_n$ such that $C \cdot C_{brack}^{1/2}\cdot\frac{1}{1-\alpha/2}\cdot\delta_n^{1-\alpha/2} = \sqrt{n}\cdot\delta_n^2$ and apply Theorem D.1.

### C.16.3 Consistence

Before we apply Theorem D.1, let us show $\bar{h}_{0,\rho,1}(\hat{\theta}_{n,t}, \theta_{*,t}) \to^P 0$. By Lemma 4.2 of [20], we have following,

$$\bar{h}_{0,\rho,1}^2(\hat{\theta}_{n,t}, \theta_{*,t}) = \mathbb{E}_{s,a\sim\rho}\left\{h_0^2\big(f_{\hat{\theta}_{n,t}}(\cdot|s,a), f_{\theta_{*,t}}(\cdot|s,a)\big)\right\}$$

$$\leq 2(1+\sqrt{2})^2 \cdot \mathbb{E}_{s,a\sim\rho}\left\{h_0^2\left(\frac{f_{\hat{\theta}_{n,t}} + f_{\theta_{*,t}}}{2}(\cdot|s,a), f_{\theta_{*,t}}(\cdot|s,a)\right)\right\}$$

$$\leq (1+\sqrt{2})^2 \cdot \big|M(\hat{\theta}_{n,t}) - \hat{M}_n(\hat{\theta}_{n,t})\big|.$$

The first inequality holds by following. Letting $p$ be an arbitrary density and $\bar{p} = \frac{1}{2}(p + p_0)$, we have

$$h_0^2(\bar{p}, p_0) = \frac{1}{2}\int_{p_0>0}\left(\sqrt{\frac{p+p_0}{2}(z)} - \sqrt{p_0(z)}\right)^2 \mathrm{d}z = \frac{1}{4}\int_{p_0>0}\left(\sqrt{(p+p_0)(z)} - \sqrt{2p_0(z)}\right)^2\mathrm{d}z$$

$$\geq \frac{1}{4(1+\sqrt{2})^2}\cdot\int_{p_0>0}\left(\sqrt{p(z)} - \sqrt{p_0(z)}\right)^2\mathrm{d}z = \frac{1}{2(1+\sqrt{2})^2}\cdot h_0^2(p,p_0).$$

The second inequality above holds due to following. Note that $\hat{\theta}_{n,t}$ is defined as follows by (10) with $\mathfrak{d}(P,Q) = \mathrm{KLD}(Q\|P)$ and we have following based on concavity of logarithm function, i.e., $\log \frac{x+y}{2} \geq \frac{1}{2}\log x + \frac{1}{2}\log y$ for $x, y > 0$.

$$\hat{\theta}_{n,t} := \arg\min_{\theta \in \Theta} \frac{1}{n} \sum_{i=1}^{n} \mathbb{E}_{z_i' \sim f_{\hat{\theta}_{n,t-1}}(\cdot|s_i',\pi)} \left\{ -\log f_\theta(r_i + \gamma z_i'|s_i, a_i) \right\},$$

$$\log \frac{f_{\hat{\theta}_{n,t}}(z|s,a) + f_{\theta_{*,t}}(z|s,a)}{2 \cdot f_{\theta_{*,t}}(z|s,a)} \cdot \mathbf{1}\big(f_{\theta_{*,t}}(z|s,a) > 0\big) \geq \frac{1}{2} \log \frac{f_{\hat{\theta}_{n,t}}(z|s,a)}{f_{\theta_{*,t}}(z|s,a)} \cdot \mathbf{1}\big(f_{\theta_{*,t}}(z|s,a) > 0\big).$$

Then this leads to following which justifies the inequality,

$$0 \leq \frac{1}{n} \sum_{i=1}^{n} \mathbb{E}_{z_i' \sim f_{\hat{\theta}_{n,t-1}}(\cdot|s_i',\pi)} \left\{ \log f_{\hat{\theta}_{n,t}}(r_i + \gamma z_i'|s_i, a_i) - \log f_{\theta_{*,t}}(r_i + \gamma z_i'|s_i, a_i) \right\}$$

$$\leq \frac{2}{n} \sum_{i=1}^{n} \mathbb{E}_{z_i' \sim f_{\hat{\theta}_{n,t-1}}(\cdot|s_i',\pi)} \left\{ \log \left( \frac{f_{\hat{\theta}_{n,t}}(r_i + \gamma z_i'|s_i, a_i) + f_{\theta_{*,t}}(r_i + \gamma z_i'|s_i, a_i)}{2 \cdot f_{\theta_{*,t}}(r_i + \gamma z_i'|s_i, a_i)} \right. \right.$$

$$\left. \left. \times \mathbf{1}\big(f_{\theta_{*,t}}(r_i + \gamma z_i'|s_i, a_i) > 0\big) \right) \right\}$$

$$\leq 2 \cdot (-\hat{M}_n(\hat{\theta}_{n,t}) + M(\hat{\theta}_{n,t})) + 2 \cdot \mathbb{E}_{s,a,r,s'} \left\{ -m_{\hat{\theta}_{n,t}}(s,a,r,s') \right\}$$

$$\leq 2 \cdot (-\hat{M}_n(\hat{\theta}_{n,t}) + M(\hat{\theta}_{n,t})) - 4 \cdot \mathbb{E}_{s,a} \left\{ h_0^2 \left( \frac{f_{\theta_{*,t}}(\cdot|s,a) + f_{\hat{\theta}_{n,t}}(\cdot|s,a)}{2}, f_{\theta_{*,t}}(\cdot|s,a) \right) \right\}.$$

The last line above holds, since the term $\mathbb{E}_{s,a,r,s'}\{-m_{\hat{\theta}_{n,t}}(s,a,r,s')\}$ can be bounded using the following fact,

$$\int \log \frac{\bar{p}(z)}{p_0(z)} \cdot \mathbf{1}\big(p_0(z) > 0\big) \cdot p_0(z)\mathrm{d}z \leq 2 \cdot \int \left( \sqrt{\frac{\bar{p}(z)}{p_0(z)}} - 1 \right) p_0(z)\mathrm{d}z \quad \because \frac{1}{2}\log x \leq \sqrt{x} - 1$$

$$= 2 \cdot \int_{p_0 > 0} (\sqrt{\bar{p}(z)} \cdot \sqrt{p_0(z)})\mathrm{d}z - 2 \leq -\int_{p_0 > 0} \left( \sqrt{p_0(z)} - \sqrt{\bar{p}(z)} \right)^2 \mathrm{d}z = -2 \cdot h_0^2(\bar{p}, p_0).$$

Using $|M(\hat{\theta}_{n,t}) - \hat{M}_n(\hat{\theta}_{n,t})| = \Delta_n(\theta|\hat{\theta}_{n,t-1})$ since $\hat{M}_n(\theta_{*,t}) = M(\theta_{*,t}) = 0$, we can show the following by Markov's inequality and (37), with probability bound not depending on $\hat{\theta}_{n,t-1}$,

$$\mathbb{P}\big(|\Delta_n(\theta|\hat{\theta}_{n,t-1})| \geq \epsilon\big) \leq \frac{1}{\epsilon} \cdot \mathbb{E}\left\{ \sup_{\theta \in \Theta} \left|\Delta_n(\theta|\hat{\theta}_{n,t-1})\right| \right\} \to^P 0,$$

as long as $\mathrm{diam}(\widetilde{\mathcal{F}}_\Theta^{1/2}, \|\cdot\|_{L_2,\rho}) < \infty$.

### C.16.4 Finalization

As we mentioned at the end of Appendix C.16.2, we let

$$\delta_n = \left( \frac{C_0^2}{n} \right)^{\frac{1}{2+\alpha}} \quad \text{where } C_0 = C \cdot C_{brack}^{1/2} \cdot \frac{1}{1 - \alpha/2},$$

$$\therefore \bar{h}_{0,\rho,1}(\hat{\theta}_{n,t}, \theta_{*,t}) = C_{brack}^{\frac{1}{2+\alpha}} \cdot \left( \frac{1}{1 - \alpha/2} \right)^{\frac{1}{1+\alpha/2}} \cdot O_P\big(n^{\frac{-1}{2+\alpha}}\big).$$

Now let us bound further bound Wasserstein metric by $h_0$. Considering two densities $p$ and $p_0$, we have

$$\int_{p_0 > 0} |p_0(z) - p(z)|\mathrm{d}z = \int_{p_0 > 0} |\sqrt{p_0(z)} - \sqrt{p(z)}| \cdot |\sqrt{p_0(z)} + \sqrt{p(z)}|\mathrm{d}z$$

$$\leq \left( \int_{p_0 > 0} |\sqrt{p_0(z)} - \sqrt{p(z)}|^2 \right)^{1/2} \cdot \left( \int_{p_0 > 0} |\sqrt{p_0(z)} + \sqrt{p(z)}|^2 \right)^{1/2} \leq 4 \cdot h_0(p_0, p).$$

This allows us to bound TVD metric as follows,

$$\int |p_0(z) - p(z)|\mathrm{d}z \leq \int_{p_0>0} |p_0(z) - p(z)|\mathrm{d}z + \int_{p>0} |p_0(z) - p(z)|\mathrm{d}z$$

$$\leq 3 \cdot \int_{p_0>0} |p_0(z) - p(z)|\mathrm{d}z \leq 12 \cdot h_0(p_0, p).$$

Based on $\mathbb{W}_p(P, Q) \leq D \cdot \mathrm{TVD}^{1/p}(P, Q)$ shown in Lemma C.6 of [63], this leads to $\mathbb{W}_p(p_0, p) \leq C \cdot D \cdot h_0^{1/p}(p_0, p)$. This leads to following,

$$\overline{\mathbb{W}}_{p,d_\pi,p}(\hat{\theta}_{n,t}, \theta_{*,t}) \leq C \cdot D \cdot C_{cover}^{\frac{1}{2p}} \cdot \bar{h}_{0,\rho,1}^{1/p}(\hat{\theta}_{n,t}, \theta_{*,t})$$

$$\leq C \cdot D \cdot C_{cover}^{\frac{1}{2p}} \cdot C_{brack}^{\frac{1/p}{2+\alpha}} \cdot \left(\frac{1}{1 - \alpha/2}\right)^{\frac{1/p}{1+\alpha/2}} \cdot O_P\left(n^{\frac{-1/p}{2+\alpha}}\right).$$

Letting $T = \lfloor \frac{1}{1 - \frac{1}{2p}} \cdot \frac{1/p}{2+\alpha} \cdot \log_{1/\gamma} N \rfloor$, we apply $n = N/T$ based on (6), and we obtain the following bound for $\overline{\mathbb{W}}_{p,d_\pi,p}(\Upsilon_{\theta_T}, \Upsilon_\pi)$,

$$\frac{D}{1 - \gamma^{1 - \frac{1}{2p}}} \cdot C_{cover}^{\frac{1}{2p}} \cdot C_{brack}^{\frac{1/p}{2+\alpha}} \cdot \left(\frac{1}{1 - \alpha/2}\right)^{\frac{1/p}{1+\alpha/2}} \cdot \left(\frac{1/p}{2+\alpha} \cdot \frac{1}{1 - \frac{1}{2p}}\right)^{\frac{1/p}{2+\alpha}}$$

$$\times O_P\left\{ \left(\frac{N}{\log_{1/\gamma} N}\right)^{\frac{-1/p}{2+\alpha}} \right\}.$$

Of course, we should show that condition of Lemma C.7 is satisfied before we apply (6). Its proof is analogous to Appendix D.5 that we used in proving Theorem C.11.

### C.17 Comparison and examples on Cramér FDE

#### C.17.1 Comparison between Cramér FDE and FLE

Although FLE [63] did not suggest a probability bound for $\overline{\mathbb{W}}_{p,d_\pi,p}$-inaccuracy, we can utilize their key results (e.g., their Lemma 4.9) to build such bound. With probability larger than $1 - \delta$, we have the following with in a single iteration based on $\mathcal{D}_t$ with $|\mathcal{D}_t| = n$:

$$\overline{\mathbb{W}}_{p,d_\pi,p}(\Upsilon_{\theta_t}, \mathcal{T}^\pi \Upsilon_{\hat{\theta}_{n,t-1}}) \leq C_{cover}^{1/2p} \cdot \frac{\sqrt{d}}{1 - \gamma} \cdot \left[ \frac{10}{n} \cdot \log\left\{ \mathcal{N}_{[\,]}\left(\widetilde{\mathcal{F}}_\Theta, \|\cdot\|, (n|\mathcal{Y}|)^{-1}\right) \Big/ \delta \right\} \right]^{1/2p}.$$

Here, $\|\cdot\|$ can be any norm that can be defined on $\widetilde{\mathcal{F}}_\Theta$. For fair comparison, let us use

$$\|f(\cdot|\cdots)\|_{L_1,\infty} := \sup_{s,a} \int_{-\infty}^\infty |f(z|s,a)|\mathrm{d}z,$$

which is what we already defined in (NS4) (or Appendix C.10). Skipping algebraic details, this leads to following with appropriate choice of $T$ with respect to the size of whole data $N = |\mathcal{D}|$,

$$\overline{\mathbb{W}}_{p,d_\pi,p}(\Upsilon_{\theta_T}, \Upsilon_\pi) \lesssim \tilde{O}_P\left(N^{\frac{-1}{2p}(1-\alpha)}\right) \quad \text{under} \quad \log \mathcal{N}_{[\,]}(\widetilde{\mathcal{F}}_\Theta, \|\cdot\|_{L_1,\infty}, \epsilon) \lesssim \epsilon^{-\alpha}.$$

Assuming the same condition, we can also bound $\log \mathcal{N}_{[\,]}(\mathcal{F}_\Theta, \|\cdot\|_{L_1,\infty}, \epsilon)$, where $\mathcal{F}_\Theta$ represents the conditional cdf family, as we have mentioned in Appendix C.10. Letting $M = \log \mathcal{N}_{[\,]}(\widetilde{\mathcal{F}}_\Theta, \|\cdot\|_{L_1,\infty}, \epsilon)$, there exists pairs of functions $\{[l_i(\cdot|\cdots), u_i(\cdot|\cdots)]\}_{i=1}^M$ that serve as $\epsilon$-bracketing of $\widetilde{\mathcal{F}}_\Theta$. Then, let us define

$$L_i(y|s,a) := \int_a^y l_i(z|s,a)\mathrm{d}z \quad \& \quad U_i(y|s,a) := \int_a^y u_i(z|s,a)\mathrm{d}z.$$

Now arbitrarily pick $F_\theta(\cdot|\cdots) \in \mathcal{F}_\Theta$, and let its corresponding conditional pdf as $f_\theta(\cdot|\cdots)$. There exists a bracketing pair $[l_k(\cdot|\cdots), u_k(\cdot|\cdots)]$ such that $l_k(\cdot|\cdots) \leq f_\theta(\cdot|\cdots) \leq u_k(\cdot|\cdots)$ and

$\|l_k(\cdot|\cdots) - u_k(\cdot|\cdots)\|_{L_1,\infty} \leq \epsilon$. Then, not only do we have $L_k(\cdot|\cdots) \leq F_\theta(\cdot|\cdots) \leq U_k(\cdot|\cdots)$, but also

$$\|L_k(\cdot|\cdots) - U_k(\cdot|\cdots)\|_{L_1,\infty} = \sup_{s,a} \int_a^b \left|L_k(y|s,a) - U_k(y|s,a)\right| \mathrm{d}y \leq$$

$$\sup_{s,a} \int_a^b \int_a^y \left|l_k(z|s,a) - u_k(z|s,a)\right| \mathrm{d}z\mathrm{d}y \leq (b-a) \cdot \|l_k(\cdot|\cdots) - u_k(\cdot|\cdots)\|_{L_1,\infty} \leq (b-a) \cdot \epsilon.$$

This further implies $\log \mathcal{N}_{[\,]}(\mathcal{F}_\Theta, \|\cdot\|_{L_1,\infty}, (b-a)\cdot\epsilon) \leq \log \mathcal{N}_{[\,]}(\widetilde{\mathcal{F}}_\Theta, \|\cdot\|_{L_1,\infty}, \epsilon) \lesssim \epsilon^{-\alpha}$, thereby satisfying the condition of (NS4). This allows us to apply Corollary B.1, yielding $\overline{\mathbb{W}}_{p,d_\pi,p}(\Upsilon_{\theta_T}, \Upsilon_\pi) = \tilde{O}_P(N^{-\frac{1}{p(2+\alpha)}})$.

### C.17.2    Example 1: Linear MDP

Let us assume the following model with $\Theta = (\mathcal{F})^K$, where $\mathcal{F}$ is a collection of cdf's of random variables bounded in $[a,b]$. Here, we can see $\theta$ as an element $\mathbf{H}(\cdot) = (H_1(\cdot), \cdots, H_K(\cdot))^\intercal$ with each $H_k \in \mathcal{F}$.

$$\mathcal{M}_\Theta := \left\{ \Upsilon_\theta \in (\mathcal{P}[a,b])^{\mathcal{S}\times\mathcal{A}} \text{ with } \Upsilon_\theta(s,a) \text{ has cdf } F_\theta(z|s,a) = \langle\phi(s,a), \mathbf{H}(z)\rangle \,\Big|\, \mathbf{H} \in (\mathcal{F})^K \right\},$$

$$\text{where} \quad \langle\phi(s,a), \mathbf{1}_K\rangle = 1 \quad \& \quad \phi_k(s,a) \geq 0 \text{ for } \forall s,a \in \mathcal{S}\times\mathcal{A},$$

$$\Sigma_\phi := \mathbb{E}_{s,a\sim\rho}\{\phi(s,a), \phi^\intercal(s,a)\} \text{ is non-singular.} \tag{38}$$

Note that tabular setting ($|\mathcal{S}\times\mathcal{A}| < \infty$) is a special case of linear MDP model. This is the case with $\phi_k(s,a) = \mathbf{1}(s,a = (s,a)_k)$ with $k = 1, \cdots, |\mathcal{S}\times\mathcal{A}|$.

Assumption 3.1 is satisfied when the transition is expressed as a linear combination of features $\tilde{p}(r,s'|s,a) = \langle\phi(s,a), \theta(r,s')\rangle$ with $\theta(r,s') = (\theta_1(r,s'), \cdots, \theta_K(r,s'))^\intercal$ being valid densities over $\mathbb{R}\times\mathcal{S}$. Also assume Assumption C.4.

For (NS4), we can use Theorem 2.7.5 of [55] to bound $\log\mathcal{N}_{[\,]}(\mathcal{F}_\Theta, \|\cdot\|_{\infty,L_1}, \epsilon)$. Letting $Q = \text{Unif}[a,b]$, we have the following,

$$\log\mathcal{N}_{[\,]}(\mathcal{F}, \|\cdot\|_{L_1}, \epsilon) = \log\mathcal{N}_{[\,]}(\mathcal{F}, \|\cdot\|_{L_1(Q)}, \frac{\epsilon}{b-a}) \leq C(b-a)\cdot\epsilon^{-1}.$$

where we have used $\|F_1 - F_2\|_{L_1} = (b-a)\cdot\|F_1 - F_2\|_{L_1(Q)}$ in the first equality.

Arbitrarily choose $\Upsilon_{\mathbf{H}_1}, \Upsilon_{\mathbf{H}_2} \in \mathcal{M}_\Theta$ and their corresponding cdf's $F_{\mathbf{H}_1}(\cdot|s,a) = \langle\phi(s,a), \mathbf{H}_1(\cdot)\rangle$ and $F_{\mathbf{H}_2}(\cdot|s,a) = \langle\phi(s,a), \mathbf{H}_2(\cdot)\rangle$. Then, we can derive the following with $\mathcal{F}[a,b]$ being the cdf's of bounded random variables in $[a,b]$,

$$\log\mathcal{N}_{[\,]}(\mathcal{F}_\Theta, \|\cdot\|_{L_1,\infty}, \epsilon) \leq K\cdot\log\mathcal{N}_{[\,]}(\mathcal{F}[a,b], \|\cdot\|_{L_1}, \epsilon) \leq CK(b-a)\cdot\epsilon^{-1}.$$

The first inequality can be verified as follows. Let the bracketing pairs of $\mathcal{F}$ be $[l_1(\cdot), u_1(\cdot)], \cdots [l_M(\cdot), u_M(\cdot)]$ where $M = \log_{[\,]}(\mathcal{F}, \|\cdot\|_1, \epsilon)$. Make $K$ copies for each pair, so that we have $[l_1^{(k)}(\cdot), u_1^{(k)}(\cdot)], \cdots [l_M^{(k)}(\cdot), u_M^{(k)}(\cdot)]$ for each $k \in [K]$. This means that we can construct $M^K$ tuples of

$$\left([l_{j_1}^{(1)}, u_{j_1}^{(1)}], \cdots, [l_{j_K}^{(K)}, u_{j_1}^{(K)}]\right) \quad \text{where } j_1, \cdots, j_K \in [M].$$

Pick an arbitrary $F(\cdot|\cdots) \in \mathcal{F}_\Theta$, and it has corresponding $\mathbf{H} = (H_1, \cdots, H_K)$ such that $F(\cdot|s,a) = \langle\phi(s,a), \mathbf{H}(\cdot)\rangle$. Since $H_1(\cdot) \in [l_{j_1}^{(1)}, u_{j_1}^{(1)}], \cdots, H_K(\cdot) \in [l_{j_K}^{(K)}, u_{j_K}^{(K)}]$ for some $j_1, \cdots, j_K \in [M]$. Then, letting $F_l(\cdot|s,a) = \sum_k \phi_k(s,a)l_{j_k}^{(k)}(\cdot)$ and $F_u(\cdot|s,a) = \sum_k \phi_k(s,a)u_{j_k}^{(k)}(\cdot)$, we have $F(\cdot|\cdots) \in [F_l(\cdot|\cdots), F_u(\cdot|\cdots)]$. We eventually have

$$\|F_l(\cdot|\cdots) - F_u(\cdot|\cdots)\|_{L_1,\infty} = \sup_{s,a} \int_a^b \left|F_u(z|s,a) - F_l(z|s,a)\right| \mathrm{d}z$$

$$\leq \sup_{s,a} \int_a^b \sum_{k=1}^K \phi_k(s,a)\cdot\left|u_{j_k}^{(k)}(z) - l_{j_k}^{(k)}(z)\right| \mathrm{d}z \leq \sup_{k\in[K]} \|u_{j_k} - l_{j_k}\|_{L_1} \leq \epsilon,$$

which leads to $\mathcal{N}_{[\,]}(\mathcal{F}_\Theta, \|\cdot\|_{L_1,\infty}, \epsilon) \leq \{\mathcal{N}_{[\,]}(\mathcal{F}, \|\cdot\|_{L_1}, \epsilon)\}^K$. Note that we have $\sup_{\theta\in\Theta}\sup_{s,a}\sup_z |F_\theta(z|s,a)| \leq 1$ and can let $C_{brack} = CK(b-a)$, $\alpha = 1$ in (NS4). Eventually this leads to $\tilde{O}_P(N^{-1/(3p)})$ $\overline{\mathbb{W}}_{p,d_\pi,p}$-convergence in Corollary B.1. However, FLE cannot suggest a meaningful bound since they assume the existence of conditional densities, which may not hold in our cdf-based model.

### C.17.3 Example 2: Linear Quadratic Regulator

We will denote state-action pairs as $x, a \in \mathcal{S} \times \mathcal{A} = \mathbb{R}^{d_x} \times \mathbb{R}^{d_a}$, its conditional reward as $R(x,a)$, and subsequent stater as $x' = x'(x,a)$. We assume the following setting

$$x'(x,a) = Ax + Ba \quad \& \quad R(x,a) = x^\intercal Q x + a^\intercal R a + \epsilon_R \quad \& \quad \pi(x) = Kx,$$
$$\text{where } \epsilon_R \text{ can follow any fixed distribution.} \tag{39}$$

Temporarily, we will assume bounded distribution for $\epsilon_R$ (although this can be relaxed for Energy FDE). Letting $\epsilon_{\text{ret}} := \sum_{t\geq 1} \gamma^{t-1}\epsilon_{R,t}$ with $\epsilon_{R,t}$ being iid copies of $\epsilon_R$, we assume the following model,

$$\mathcal{M}_\Theta := \left\{ \Upsilon_\theta \in \big(\mathcal{P}(\mathbb{R})\big)^{\mathcal{S}\times\mathcal{A}} \quad \text{where } \Upsilon_\theta(x,a) \text{ is the probability for } Z_\theta(x,a) = \right. \tag{40}$$

$$\left. x^\intercal M_1 x + a^\intercal M_2 x + a^\intercal M_3 a + \epsilon_{\text{ret}} \quad \text{with } \theta = (M_1, M_2, M_3) \in \mathbb{R}^{d_x\times d_x} \times \mathbb{R}^{d_a\times d_x} \times \mathbb{R}^{d_a\times d_a} \right\}.$$

Throughout the proof, we shall assume the analogous statements that [63] suggested in their Lemma B.3, bounding $\|x\|, \|a\|, \|M_1\|_F, \|M_2\|_F, \|M_3\|_F$.

Assumption 3.1 can be satisfied with appropriate modeling of $\Theta \subseteq \mathbb{R}^{d_x\times d_x} \times \mathbb{R}^{d_a\times d_x} \times \mathbb{R}^{d_a\times d_a}$. By following the same logic shown in D.9 of [63], we can present an example. Skipping all calculations, $\mathcal{T}^\pi Z_\theta(x,a) \sim \mathcal{T}^\pi \Upsilon_\theta(x,a)$ follows a distribution $\mathcal{T}^\pi Z_\theta(x,a) = \mathbb{E}\{\mathcal{T}^\pi Z_\theta(x,a)\} + \epsilon_{\text{ret}}$ with,

$$\mathbb{E}\left\{\mathcal{T}^\pi Z_\theta(x,a)\right\} = x^\intercal\left\{Q + \gamma(A^\intercal M_1 A + A^\intercal K^\intercal M_2 A + A^\intercal K^\intercal M_3 K A)\right\}x$$

$$+ x^\intercal\left\{\gamma \cdot \left(A^\intercal(M_1 + M_1^\intercal)B + A^\intercal(K^\intercal M_2 + M_2^\intercal K)B + A^\intercal K^\intercal(M_3 + M_3^\intercal)KB\right)\right\}a$$

$$+ a^\intercal\left\{R + \gamma \cdot \left(B_1^\intercal M_1 B + B^\intercal K^\intercal M_2 B + B^\intercal K^\intercal M_3 K B\right)\right\}a$$

Let our model be $\Theta = \{(M_1, M_2, M_3) : \|M\|_F, \|M_2\|_F, \|M_3\|_F \leq m\}$, $\mathcal{T}^\pi \Upsilon_\theta \in \mathcal{M}_\Theta$ holds if the following conditions are satisfied,

$$\|Q\|_F + \gamma \cdot \|A\|_F^2 \cdot (1 + \|K\|_F + \|K\|_F^2) \cdot m \leq m$$

$$\gamma \cdot \|A\|_F \cdot \|B\|_F \cdot \{1 + 2\|K\|_F + \|K\|_F^2\} \cdot 2m \leq m$$

$$\|R\|_F + \gamma \cdot \|B\|_F^2 \cdot \{1 + \|K\|_F + \|K\|_F^2\} \cdot m \leq m.$$

Assumption C.4 can be simplified as follows. For $X = \mu_x + \epsilon_{\text{ret}}$ and $Y = \mu_y + \epsilon_{\text{ret}}$, we have the following with $P_X, P_Y$ being their probability measures and $F_X, F_Y$ being their cdf's,

$$\mathbb{W}_1(P_X, P_Y) = \int_{-\infty}^{\infty} |F_X(z) - F_Y(z)|\mathrm{d}z = |\mu_y - \mu_x| \cdot 1 = |\mu_y - \mu_x|.$$

Note that the second equality holds, since $F_Y$ is only a location shift of $F_X$, i.e., $F_Y(\cdot) = F_X(\cdot - (\mu_y - \mu_x))$. This leads to the following with $\theta_A = (M_1^{(A)}, M_2^{(A)}, M_3^{(A)})$ and $\theta_B = (M_1^{(B)}, M_2^{(B)}, M_3^{(B)})$,

$$\mathbb{W}_1\{\Upsilon_{\theta_A}(x,a), \Upsilon_{\theta_B}(x,a)\} =$$

$$x^\intercal(M_1^{(A)} - M_1^{(B)})x + a^\intercal(M_2^{(A)} - M_2^{(B)})x + a^\intercal(M_3^{(A)} - M_3^{(B)})a = \langle \phi(x,a), \theta_{AB} \rangle,$$

where $\phi(x,a) := (x^\intercal, a^\intercal)^\intercal \otimes (x^\intercal, a^\intercal)^\intercal$ and some $\theta \in \mathbb{R}^{(d_x+d_a)^2}$. Using the fact that $\mathbb{W}_1\{\Upsilon_{\theta_A}(x,a), \Upsilon_{\theta_B}(x,a)\} = \theta_{AB}^\intercal \phi(x,a)\phi(x,a)^\intercal \theta_{AB}$, we obtain

$$\sup_{\theta_1,\theta_2\in\Theta} \frac{\overline{\mathbb{W}}_{1,d_\pi,1}^2(\Upsilon_{\theta_1}, \mathcal{T}^\pi \Upsilon_{\theta_2})}{\overline{\mathbb{W}}_{1,\rho,1}^2(\Upsilon_{\theta_1}, \mathcal{T}^\pi \Upsilon_{\theta_2})} \leq \sup_{\theta_{AB}\in\mathbb{R}^{(d_x+d_a)^2}, \theta_{AB}\neq 0} \frac{\theta_{AB}^\intercal \Sigma_{\phi,d_\pi} \theta_{AB}}{\theta_{AB}^\intercal \Sigma_{\phi,\rho} \theta_{AB}},$$

where $\Sigma_{\phi,d_\pi} := \mathbb{E}_{x,a\sim d_\pi}\{\phi(x,a)\phi(x,a)^\intercal\} \quad \& \quad \Sigma_{\phi,\rho} := \mathbb{E}_{x,a\sim\rho}\{\phi(x,a)\phi(x,a)^\intercal\}.$

This is the same result that [63] provided in their Lemma B.2.

Lastly, let us bound the term $\log \mathcal{N}_{[\,]}(\mathcal{F}_\Theta, \|\cdot\|_{\infty,1}, \epsilon)$ for (NS4). First, we define

$$\mathcal{F}_\Theta^\mu := \left\{ \mu_\theta : \mathcal{S} \times \mathcal{A} \to \mathbb{R} \ \middle| \ \mu_\theta(x,a) = \mathbb{E}_{Z_\theta(x,a) \sim \Upsilon_\theta(x,a)}\big\{ Z_\theta(x,a) \big\} \right\},$$

$$\text{with} \quad \|\mu_{\theta_1} - \mu_{\theta_2}\|_\infty := \sup_{x,a \in \mathcal{S} \times \mathcal{A}} \left| \mu_{\theta_1}(x,a) - \mu_{\theta_2}(x,a) \right|$$

We can see $\mathcal{N}_{[\,]}(\mathcal{F}_\Theta, \|\cdot\|_{L_1,\infty}, \epsilon) \leq \mathcal{N}_{[\,]}(\mathcal{F}_\Theta^\mu, \|\cdot\|_\infty, \epsilon)$ for the following reason. Letting $M = \mathcal{N}_{[\,]}(\mathcal{F}_\Theta^\mu, \|\cdot\|_\infty, \epsilon)$, there exist bracketing pairs $\{[l_i(\cdot), u_i(\cdot)]\}_{i=1}^M$ such that

$$\forall \mu_\theta \in \mathcal{F}_\Theta^\mu \quad \exists i \in [M] \quad \text{such that} \quad \|l_i(\cdot) - u_i(\cdot)\|_\infty \leq \epsilon.$$

Now consider the bracketing pairs $[F_{l_i}(\cdot|\cdots), F_{u_i}(\cdot|\cdots)]$ $(i = 1, \cdots, M)$. For an arbitrary $F_\theta(\cdot|\cdots) \in \mathcal{F}_\Theta$, its corresponding expectation function satisfies $l_i(x,a) \leq \mu_\theta(x,a) \leq u_i(x,a)$ for some $i \in [M]$. As we have previously mentioned, all $F_\theta(\cdot|x,a)$ are results of location-shifts of the cdf of $\epsilon_{\text{ret}}$. This leads to $F_\theta(\cdot|\cdots) \in [F_{l_i}(\cdot|\cdots), F_{u_i}(\cdot|\cdots)]$ for some $i \in [M]$, and it satisfies

$$\|F_{l_i}(\cdot|\cdots) - F_{u_i}(\cdot|\cdots)\|_{L_1,\infty} = \|l_i(\cdot) - u_i(\cdot)\|_\infty \leq \epsilon.$$

Thus, we have $\mathcal{N}_{[\,]}(\mathcal{F}_\Theta, \|\cdot\|_{L_1,\infty}, \epsilon) \leq \mathcal{N}_{[\,]}(\mathcal{F}_\Theta^\mu, \|\cdot\|_\infty, \epsilon)$. Further assuming $\|x\| \leq D_x$ and $\|a\| \leq D_a$, we have

$$\|\mu_{\theta_A} - \mu_{\theta_B}\|_\infty = \sup_{x,a} \left| x^\mathsf{T}(M_1^{(A)} - M_2^{(B)})x + a^\mathsf{T}(M_2^{(A)} - M_2^{(B)})x + a^\mathsf{T}(M_3^{(A)} - M_3^{(B)})a \right|$$

$$\leq \max\{D_x, D_a\}^2 \cdot \left\{ \|M_1^{(A)} - M_1^{(B)}\|_{op} + \|M_2^{(A)} - M_2^{(B)}\|_{op} + \|M_3^{(A)} - M_3^{(B)}\|_{op} \right\}$$

$$\leq 3D_0^2 \cdot d_3(\theta_A, \theta_B),$$

where $d_3(\theta_A, \theta_B) := \max\{\|M_1^{(A)} - M_1^{(B)}\|_F, \|M_2^{(A)} - M_2^{(B)}\|_F, \|M_3^{(A)} - M_3^{(B)}\|_F\}$ and $D_0 := \max\{D_x, D_a\}$. Then, we can bound our target as follows,

$$\mathcal{N}_{[\,]}(\mathcal{F}_\Theta, \|\cdot\|_{L_1,\infty}, \epsilon) \leq \mathcal{N}_{[\,]}(\mathcal{F}_\Theta^\mu, \|\cdot\|_\infty, \epsilon) = \mathcal{N}(\mathcal{F}_\Theta^\mu, \|\cdot\|_\infty, \frac{\epsilon}{2}) \leq \mathcal{N}(\Theta, d_3, \frac{\epsilon}{6D_0^2})$$

$$\leq \mathcal{N}(\Theta_1, \|\cdot\|_F, \frac{\epsilon}{6D_0^2}) \times \mathcal{N}(\Theta_2, \|\cdot\|_F, \frac{\epsilon}{6D_0^2}) \times \mathcal{N}(\Theta_3, \|\cdot\|_F, \frac{\epsilon}{6D_0^2}),$$

where $\Theta_i := \{M_i : \|M_i\|_F \leq m\}$. Since $\Theta_1, \Theta_2, \Theta_3$ with $\|\cdot\|_F$ can be perceived as subsets of Euclidean spaces $\mathbb{R}^{d_x d_x}, \mathbb{R}^{d_a d_x}, \mathbb{R}^{d_a d_a}$ with Euclidean norms $\|\cdot\|$, we can use volume comparison lemma

$$\log \mathcal{N}(A, \|\cdot\|, \epsilon) \leq p \cdot \log\left( \frac{3 \cdot \text{diam}(A, \|\cdot\|)}{\epsilon} \right) \quad \text{for } A \in \mathbb{R}^p, \quad \forall \epsilon \in (0, \text{diam}(A, \|\cdot\|)).$$

This leads to the following,

$$\log \mathcal{N}_{[\,]}(\mathcal{F}_\Theta, \|\cdot\|_{L_1,\infty,}, \epsilon) \leq d_x^3 d_a^3 \cdot \log\left( \frac{18 \cdot D_0^2 \cdot m}{\epsilon} \right).$$

Since our term can be bounded by inverse of polynomial $\mathcal{N}_{[\,]}(\mathcal{F}_\Theta, \|\cdot\|_{L_1,\infty}, \epsilon) \lesssim \epsilon^{-\alpha}$ with any $\alpha > 0$, we can let $\alpha_N = 1/\log N$.

If we assume bounded distributions of $\epsilon_{\text{rew}}$, then we can apply Corollary B.1 with the same trick (32), Cramér FDE obtains $\tilde{O}_P(N^{-1/(2p)})$ of $\overline{\mathbb{W}}_{p,d_\pi,p}$-convergence. If we assume unbounded distributions, then Energy FDE obtain $\tilde{O}_P(N^{-1/8})$ of $\overline{\mathbb{W}}_{1,d_\pi,1}$-inaccuracy by Corollary B.3. For FLE [63], unless we have a standard parametric distribution for $\epsilon_{\text{ret}}$ (e.g., truncated gaussian), we cannot obtain the bound, since we cannot relate $\|f_{\theta_A}(\cdot|\cdots) - f_{\theta_B}(\cdot|\cdots)\|_{L_1,\infty}$ and $d_3(\theta_A, \theta_B)$.

# D Auxiliary proofs for Appendix C

## D.1 Supporting theorem

Following is a standard convergence theorem in M-estimation. We slightly adapted it from Theorem 6.1 of [51], only modifying $l = 2$ into generalized $l \geq 2$. So we will skip the proof.

**Theorem D.1.** *(Adapted version of Theorem 6.1 of [51]) Assume that $F_{n,t}(\cdot) : \Theta \to \mathbb{R}$ has a minimizer $\theta_*$. Let $\tilde{\delta}_n \geq 0$ be a sequence that satisfies the following for $\delta \geq \tilde{\delta}_n$:*

*1. If $d_n(\theta, \theta_*) \in (\frac{\delta}{2}, \delta]$, then $F_{n,t}(\theta) - F_{n,t}(\theta_*) \geq \lambda \cdot \delta^l$.*

*2. We have $\phi(\delta) = C_0 \cdot \delta^{\alpha_0}$ with some $\alpha_0 \in (0, l)$ such that following holds,*

$$\mathbb{E}\left[ \sup_{\theta \in \Theta : d_n(\theta, \theta_*) \leq \delta} \left| \left\{ \hat{F}_{n,t}(\theta | \hat{\theta}_{n,t-1}) - F_{n,t}(\theta | \hat{\theta}_{n,t-1}) \right\} \right. \right.$$
$$\left. \left. - \left\{ \hat{F}_{n,t}(\theta_* | \hat{\theta}_{n,t-1}) - F_{n,t}(\theta_* | \hat{\theta}_{n,t-1}) \right\} \right| \right] \leq \frac{\phi(\delta)}{\sqrt{n}}.$$

*Further assume that we have $\delta_n$ and some estimator such that*

$$\phi(\delta_n) \leq \sqrt{n} \cdot \delta_n^l \quad \& \quad \delta_n \geq \tilde{\delta}_n \quad \& \quad \hat{\theta}_n \to^P \theta_* \text{ in } d_n.$$

*Then, we have $d_n(\hat{\theta}_n, \theta_*) = O_P(\delta_n)$. If $\tilde{\delta}_n = 0$, then we have $\delta_n = C_0^{\frac{1}{l-\alpha_0}} \cdot n^{\frac{-1}{2(l-\alpha_0)}}$.*

## D.2 Proof of Lemma C.7

Without loss of generality, we can assume $\tilde{C} = 1$. We let $Z_{n,t} := n^{1/b} \cdot X_{n,t}$, and $C = \limsup_{n \to \infty} \sup_{t \in \mathbb{N}} \mathbb{E}|n^{1/b} \cdot X_{n,t}| < \infty$,

$$\mathbb{P}\left( \sum_{t=1}^{T} \zeta^{T-t} \cdot Z_{n,t} \geq \epsilon \right) \leq \frac{\mathbb{E}|\sum_{t=1}^{T} \zeta^{T-t} \cdot Z_{n,t}|}{\epsilon} \leq \frac{1}{1-\zeta} \cdot \frac{1}{\epsilon} \cdot \sup_{t \in [T]} \mathbb{E}|Z_{n,t}|.$$

Then, letting $\epsilon = \frac{C}{1-\zeta} \cdot \tilde{\epsilon}$, we have

$$\mathbb{P}\left( \sum_{t=1}^{T} \zeta^{T-t} \cdot Z_{n,t} \geq \frac{C}{1-\zeta} \cdot \tilde{\epsilon} \right) \leq \frac{1}{\tilde{\epsilon}}.$$

Thus we have $\sum_{t=1}^{T} \zeta^{T-t} \cdot X_{n,t} = O_P(\frac{1}{1-\zeta} \cdot n^{-1/b})$. More generally, we have $\sum_{t=1}^{T} \zeta^{T-t} \cdot X_{n,t} = \tilde{C} \cdot O_P(\frac{1}{1-\zeta} \cdot n^{-1/b})$.

## D.3 Proof of Lemma C.8

Showing the upper bound is easy. Let $X, X' \overset{iid}{\sim} Q$ and $Y, Y' \overset{iid}{\sim} P$. Based on technique shown in Appendix A.6 of [26], we can derive

$$\mathcal{E}(P, Q) = \left\{ \mathbb{E}\|X - Y\| - \mathbb{E}\|X - X'\| \right\} + \left\{ \mathbb{E}\|X - Y\| - \mathbb{E}\|Y - Y'\| \right\}$$
$$\leq \mathbb{W}_1(P, Q) + \mathbb{W}_1(P, Q) = 2 \cdot \mathbb{W}_1(P, Q).$$

Due to Jensen's inequality, it is straightforward to see $\mathbb{W}_p(P, Q) \leq \mathbb{W}_r(P, Q)$ for $p < r$.

Now let us show the lower bound. We assume that $X \sim P$ and $Y \sim Q$ have bounded support $[a, b]$, and we denote their cdf's as $F_P$ and $F_Q$. Letting $U \sim \text{Unif}[a, b]$, we have

$$\mathbb{W}_1(P, Q) = (b-a) \cdot \|F_P(U) - F_Q(U)\|_{L_1(\mathbb{P})} \quad \text{where } \|X\|_{L_p(\mathbb{P})} := (\mathbb{E}|X|^p)^{\frac{1}{p}}$$
$$\leq (b-a) \cdot \|F_P(U) - F_Q(U)\|_{L_2(\mathbb{P})} \quad \text{by Jensen's inequality}$$
$$= (b-a)^{1/2} \cdot l_2(P, Q),$$

where $l_2^2(P, Q) := \int_\infty^\infty |F_P(x) - F_Q(x)|^2 \mathrm{d}x$ denotes Cramér distance. Since Energy distance is equivalent to Cramér distance in 1D domain [7], that is $\frac{1}{2}\mathcal{E}(P, Q) = l_2^2(P, Q)$, this leads to

$$\mathcal{E}(P, Q) = 2 \cdot l_2^2(P, Q) \geq \frac{2}{b - a} \cdot \mathbb{W}_1^2(P, Q).$$

Using $\mathbb{W}_p^p(P, Q) \geq= (b - a)^{p-r} \cdot \mathbb{W}_r^r(P, Q)$ that was shown by [44] (see their Section 2.3), we can show the lower bound.

## D.4  Proof of Lemma C.12

### D.4.1  Conditioned event

Let $\hat{\theta}_{n,t-1} \in \Theta$ be fixed. Then, we have a unique value of $\theta_{*,t} = \arg\min_{\theta \in \Theta} F(\theta | \hat{\theta}_{n,t-1})$. Based on this, we define the following:

$$X_i^\theta := m^2\{\Upsilon_\theta(s_i, a_i), \Upsilon_{\theta_{*,t}}(s_i, a_i)\} \quad \text{where } s_i, a_i \overset{iid}{\sim} \rho, \quad \& \quad Y_i^\theta := X_i^\theta - \mathbb{E}(X_i^\theta),$$
$$\Theta_\delta = N_{\bar{m}_{\rho,1}}(\theta_{*,t}, \delta) := \{\theta \in \Theta \, : \, \bar{m}_{\rho,1}(\theta, \theta_{*,t}) < \delta\}.$$

With $\Omega$ being the probability space, we define the following event based on $\beta \in (0, 2]$ whose exact value will later be determined,

$$\Omega_\delta := \left\{\omega \in \Omega \, : \, \sup_{\theta \in \Theta_\delta} \left|\frac{1}{n} \sum_{i=1}^n X_i^\theta(\omega) - \mathbb{E}(X_1^\theta)\right| \leq \delta^\beta\right\} \tag{41}$$
$$= \left\{\omega \in \Omega \, : \, \sup_{\theta \in \Theta_\delta} \left|\frac{1}{n} \sum_{i=1}^n Y_i^\theta(\omega) - Y_i^{\theta_{*,t}}(\omega)\right| \leq \delta^\beta\right\} \quad (\because Y_i^{\theta_{*,t}} = 0).$$

Prior to calculating its probability $\mathbb{P}(\Omega_\delta)$, let us first show its sub-gaussianity. Since $\|\cdot\|_{\psi_2}$ is a norm (Example 2.5.7 of [57]), we can use its convexity to derive

$$\|Y_1^{\theta_1} - Y_1^{\theta_2}\|_{\psi_2} \leq \|X_1^{\theta_1} - X_1^{\theta_2}\|_{\psi_2} + \mathbb{E}\|X_1^{\theta_1} - X_1^{\theta_2}\|_{\psi_2}.$$

Assume that $\theta_1, \theta_2 \in \Theta_\delta$. Then, we have

$$|X_1^{\theta_1} - X_1^{\theta_2}| = \left|m^2\{\Upsilon_{\theta_1}(s_1, a_1), \Upsilon_{\theta_{*,t}}(s_1, a_1)\} - m^2\{\Upsilon_{\theta_2}(s_1, a_1), \Upsilon_{\theta_{*,t}}(s_1, a_1)\}\right|$$

$$= \left|m\{\Upsilon_{\theta_1}(s_1, a_1), \Upsilon_{\theta_{*,t}}(s_1, a_1)\} - m\{\Upsilon_{\theta_2}(s_1, a_1), \Upsilon_{\theta_{*,t}}(s_1, a_1)\}\right|$$

$$\times \left|m\{\Upsilon_{\theta_1}(s_1, a_1), \Upsilon_{\theta_{*,t}}(s_1, a_1)\} + m\{\Upsilon_{\theta_2}(s_1, a_1), \Upsilon_{\theta_{*,t}}(s_1, a_1)\}\right|$$

$$\leq 2 \cdot \mathrm{diam}(\Theta_\delta; m_\infty) \cdot m_\infty(\theta_1, \theta_2),$$

where $\mathrm{diam}(\Theta_\delta; m_\infty) := \sup_{\theta_1, \theta_2 \in \Theta_\delta} m_\infty(\theta_1, \theta_2)$. This leads to $\|Y_1^{\theta_1} - Y_1^{\theta_2}\|_{\psi_2} \leq C \cdot \mathrm{diam}(\Theta_\delta; m_\infty) \cdot m_\infty(\theta_1, \theta_2)$. Then, by applying Theorem 8.1.6 of [57], we have the following bound for arbitrary $u > 0$,

$$\sup_{\theta \in \Theta_\delta} \left|\frac{1}{n} \sum_{i=1}^n (Y_i^\theta - Y_i^{\theta_{*,t}})\right| \leq \frac{C}{\sqrt{n}} \cdot \mathrm{diam}(\Theta_\delta; m_\infty) \cdot \left\{\int_0^\infty \sqrt{\log \mathcal{N}(\Theta_\delta, m_\infty, \epsilon)}\mathrm{d}\epsilon\right.$$

$$\left. + u \cdot \mathrm{diam}(\Theta_\delta; m_\infty)\right\} \quad \text{with probability larger than } 1 - 2 \cdot \exp(-u^2).$$

Using the constants defined in Lemma C.12, we can let $u = \frac{1}{2} \cdot \sqrt{n} \cdot \delta^\beta / C_2$ to show that the following holds with probability larger than $1 - 2 \cdot \exp(-n \cdot \delta^{2\beta}/4C_2^2)$,

$$\sup_{\theta \in \Theta_\delta} \left|\frac{1}{n} \sum_{i=1}^n (Y_i^\theta - Y_i^{\theta_{*,t}})\right| \leq \frac{C_1}{\sqrt{n}} + \frac{1}{2}\delta^\beta \quad \text{for } \forall \delta > 0.$$

Now, we assume the following:

$$\text{Criterion-1: } \delta \geq \delta_n^{(1)} := \left(\frac{2C_1}{\sqrt{n}}\right)^{1/\beta}. \tag{42}$$

Since we have $\frac{1}{2}\delta^\beta \geq \frac{C_1}{\sqrt{n}}$, we have

$$\mathbb{P}(\Omega_\delta) = \mathbb{P}\left\{\sup_{\theta \in \Theta_\delta}\left|\frac{1}{n}\sum_{i=1}^n (Y_i^\theta - Y_i^{\theta_{*,t}})\right| \leq \delta^\beta\right\} \geq 1 - 2 \cdot \exp(-n \cdot \delta^{2\beta}/4C_2^2). \tag{43}$$

Under $\Omega_\delta$, we have the following. Suppose $\theta \in \Theta_\delta$. In other words, $\bar{m}_{\rho,1}(\theta, \theta_{*,t}) < \delta$. Then, we have

$$\left|\bar{m}_{n,1}^2(\theta, \theta_{*,t}) - \bar{m}_{\rho,1}^2(\theta, \theta_{*,t})\right| = \left|\frac{1}{n}\sum_{i=1}^n X_i^\theta - \mathbb{E}(X_1^\theta)\right| \leq \delta^\beta, \quad \text{where}$$

$$\bar{m}_{n,1}(\theta_1, \theta_2) := \left[\frac{1}{n}\sum_{i=1}^n m^2\{\Upsilon_{\theta_1}(s_i, a_i), \Upsilon_{\theta_2}(s_i, a_i)\}\right]^{1/2}.$$

This leads to the following with $\beta' = \min\{\beta, 2\}$,

$$\bar{m}_{n,1}^2(\theta, \theta_{*,t}) \leq \bar{m}_{\rho,1}^2(\theta, \theta_{*,t}) + \left|\bar{m}_{n,1}^2(\theta, \theta_{*,t}) - \bar{m}_{\rho,1}^2(\theta, \theta_{*,t})\right| \leq \delta^2 + \delta^\beta \leq 2 \cdot \delta^{\beta'},$$

where we assumed $\delta \in (0, 1)$ without loss of generality. Then, we have the following,

$$\text{Under } \Omega_\delta, \quad \text{for } \forall \theta \in \Theta_\delta, \quad \bar{m}_{n,1}(\theta, \theta_{*,t}) \leq \sqrt{2} \cdot \delta^{\beta'/2} \text{ holds.}$$

### D.4.2 Concentration of modulus

Assume that $\mathcal{Z}_t := \{(s_i, a_i)\}_{i=1}^n$ is given (fixed). However, there still exists randomness in the transition $r_i, s_i' \sim \tilde{p}(\cdots|s_i, a_i)$. For this conditional probability, denote its corresponding conditional expectation as $\mathbb{E}_{\mathcal{Z}_t}(\cdots) := \mathbb{E}(\cdots|\mathcal{Z}_t)$ and its corresponding sub-gaussian norm as $\|\cdot\|_{\psi_2(n)}$. Then, by Dudley's inequality (Theorem 8.1.3 of [57]) and (MS1), we obtain following,

$$\mathbb{E}_{\mathcal{Z}_t}\left\{\sup_{\theta \in \Theta_\delta}|\tilde{\Delta}_n(\theta|\hat{\theta}_{n,t-1})|\right\} \leq \mathbb{E}_{\mathcal{Z}_t}\left\{\sup_{\theta \in \Theta : \bar{m}_{n,1}(\theta, \theta_{*,t}) \leq \sqrt{2} \cdot \delta^{\beta'/2}}|\tilde{\Delta}_n(\theta|\hat{\theta}_{n,t-1})|\right\} \quad \text{under } \Omega_\delta \text{ (41)}$$

$$\leq \frac{C \cdot C_{subg}}{\sqrt{n}} \cdot \int_0^\infty \sqrt{\log\mathcal{N}(N_{\bar{m}_{n,1}}(\theta_{*,t}, \sqrt{2} \cdot \delta^{\beta'/2}), \bar{m}_{n,1}, \epsilon)}\mathrm{d}\epsilon$$

$$\leq \frac{C \cdot C_{subg}}{\sqrt{n}} \cdot \int_0^{\sqrt{2}\cdot\delta^{\beta'/2}} \sqrt{\log\mathcal{N}(N_{\bar{m}_{n,1}}(\theta_{*,t}, \sqrt{2} \cdot \delta^{\beta'/2}), \bar{m}_{n,1}, \epsilon)}\mathrm{d}\epsilon$$

$$\leq \frac{C \cdot C_{subg}}{\sqrt{n}} \cdot \int_0^{\sqrt{2}\cdot\delta^{\beta'/2}} \sqrt{\log\mathcal{N}(\Theta, m_\infty, \epsilon)}\mathrm{d}\epsilon.$$

By Assumption of Lemma C.12, we have $0 < C_3 < \infty$. Then, we have the following,

$$\mathbb{E}\left\{\sup_{\theta \in \Theta_\delta}|\tilde{\Delta}_n(\theta|\hat{\theta}_{n,t-1})|\right\}$$

$$= \mathbb{E}\left\{\sup_{\theta \in \Theta_\delta}|\tilde{\Delta}_n(\theta|\hat{\theta}_{n,t-1})| \,\Big|\, \Omega_\delta\right\} \cdot \mathbb{P}(\Omega_\delta) + \mathbb{E}\left\{\sup_{\theta \in \Theta_\delta}|\tilde{\Delta}_n(\theta|\hat{\theta}_{n,t-1})| \,\Big|\, \Omega_\delta^c\right\} \cdot \mathbb{P}(\Omega_\delta^c)$$

$$\leq \mathbb{E}\left\{\sup_{\theta \in \Theta_\delta}|\tilde{\Delta}_n(\theta|\hat{\theta}_{n,t-1})| \,\Big|\, \Omega_\delta\right\} + \mathbb{E}\left\{\sup_{\theta \in \Theta_\delta}|\tilde{\Delta}_n(\theta|\hat{\theta}_{n,t-1})| \,\Big|\, \Omega_\delta^c\right\} \cdot 2\exp\left(\frac{-n \cdot \delta^{2\beta}}{4C_2^2}\right) \quad \text{by (43)}$$

$$\leq \frac{C \cdot C_{subg}}{\sqrt{n}} \cdot \int_0^{\sqrt{2}\cdot\delta^{\beta'/2}} \sqrt{\log\mathcal{N}(\Theta, m_\infty, \epsilon)}\mathrm{d}\epsilon + C_3 \cdot \exp\left(\frac{-n \cdot \delta^{2\beta}}{4C_2^2}\right)$$

$$\leq \frac{C \cdot C_{subg}}{\sqrt{n}} \cdot C_{ent} \cdot \delta^{\frac{\alpha_0}{2}\beta'} + C_3 \cdot \exp\left(\frac{-n \cdot \delta^{2\beta}}{4C_2^2}\right).$$

Then, we set another criterion as below,

Criterion-2: $\delta \geq \delta_n^{(2)}$,    where $\delta_n^{(2)}$ is such that $\delta > \delta_n^{(2)}$ implies

$$\frac{C}{\sqrt{n}} \cdot C_{subg} \cdot C_{\mathrm{ent}} \cdot \delta^{\frac{\alpha_0}{2}\beta'} \geq C_3 \cdot \exp\left(\frac{-n \cdot \delta^{2\beta}}{4C_2^2}\right). \tag{44}$$

This leads to following bound, whose RHS does not depend on $\hat{\theta}_{n,t-1} \in \Theta$,

$$\mathbb{E}\left\{\sup_{\theta \in \Theta_\delta} |\tilde{\Delta}_n(\theta|\hat{\theta}_{n,t-1})|\right\} \leq \frac{2C}{\sqrt{n}} \cdot C_{subg} \cdot C_{\mathrm{ent}} \cdot \delta^{\frac{\alpha_0}{2}\beta'}. \tag{45}$$

## D.5    Satisfaction of the condition of Lemma C.7 in Theorem C.11

Before bounding $\bar{\eta}_{d_\pi,q}(\Upsilon_{\theta_T}, \Upsilon_\pi)$ based on (6), we first apply Lemma C.7 by letting $X_{n,t} = \bar{m}_{d_\pi,1}^{\delta'}(\Upsilon_{\theta_t}, \mathcal{T}^\pi \Upsilon_{\hat{\theta}_{n,t-1}})$. Proving $\limsup_{n \to \infty} \sup_{t \in \mathbb{N}} \mathbb{E}|n^{1/b} \cdot X_{n,t}| < \infty$ is the analogous to Appendix C.8.4. We do not need Stage 2 of Appendix C.8.4. Skipping the details, we can repeat Stage 1 of Appendix C.8.4 to obtain the following, with $\tilde{\Delta}_n(\theta|\hat{\theta}_{n,t-1})$ defined in Lemma C.12:

$$\mathbb{P}\left\{\bar{m}_{\rho,1}(\hat{\theta}_{n,t}, \theta_{*,t}) \geq 2^{M-1}\delta_n\right\} \leq \sum_{j \geq M} \frac{1}{\lambda \cdot 2^{l(j-1)} \cdot \delta_n^l} \cdot \mathbb{E}\left\{\sup_{\bar{m}_{\rho,1}(\theta, \theta_{*,t}) \leq 2^j \cdot \delta_n} \left|\tilde{\Delta}_n(\theta|\hat{\theta}_{n,t-1})\right|\right\}.$$

Using (45) and letting $\beta' = \min\{\frac{l}{1+\alpha_0/2}, 2\}$ with $\alpha_0 = 1 - \alpha/2$ (as we chose within the proof of Theorem C.11 of Appendix C.15.1), we can take up the above inequality as

$$\mathbb{P}\left\{\bar{m}_{\rho,1}(\hat{\theta}_{n,t}, \theta_{*,t}) \geq 2^{M-1}\delta_n\right\} \leq \frac{C}{\sqrt{n}} \cdot C_{subg} \cdot C_{\mathrm{ent}} \cdot \frac{2^l}{1 - 2^{\frac{\alpha_0\beta'}{2}-l}} \cdot (2^M \cdot \delta_n)^{\frac{\alpha_0\beta'}{2}-l}.$$

After that, copying the remaining logic of Appendix C.8.4 will give us the desired result.

# E  Experiment Details

## E.1  Details in Linear Quadratic Regulator

LQR is a parametric environment with deterministic transitions and Gaussian noise in the rewards. It can be verified that Assumption C.4 holds given the chosen behavior policy, and Assumptions 3.1 is satisfied under the appropriate model space, both of which we will present. We will compare the performance of the baseline method FLE [63] with our proposed FDE method using different functional Bregman divergences: Energy ($\beta = 1$), Laplace (Matern with $\nu = 1$), RBF, PDF-L2, KL. The evaluation is based on the $\overline{\mathbb{W}}_{1,d_\pi,1}$-inaccuracy.

The proposed FDE methods using different functional Bregman divergences outperform FLE in terms of both mean inaccuracy and stability (as measured by standard deviation), as visualized in the inaccuracy graphs of Figure 2. This performance gap is expected, as FLE does not make use of the closed-form density of $\Psi_\pi(r, s', \Upsilon_{\hat{\theta}_{n,t-1}})$, making it suffer from large Monte Carlo errors. See Table 4 in Appendix E.1.4 for detailed inaccuracy results.

### E.1.1  Data collection and model

Our offline data are collected as follows. States $x = (r \cdot \cos\theta_x, r \cdot \sin\theta_x)^\intercal$ are generated with $r \sim \text{Unif}[0, 1]$, $\theta_x \sim \text{Unif}[0, 2\pi]$. Given the state, action is generated by behavior policy $b(a|x) = 1/5$ with $a = \text{Rot}(\theta_a)x$ where $\text{Rot}(\theta_a)$ is the (counter-clockwise) rotation matrix with angle $\theta_a \sim \text{Unif}\{0, \frac{2\pi}{5}, \frac{4\pi}{5}, \frac{6\pi}{5}, \frac{8\pi}{5}\}$. Since the all $x$ within the unit circle has positive density value and $\text{Rot}(\theta_a) = K$ holds with positive probability, Assumption C.4 is satisfied.

We assume deterministic target policy (which, for the time being, can be regarded as $\pi : \mathcal{S} \to \mathcal{A}$ with abuse of notation). With Gaussian noise added to the reward $\epsilon_R \sim N(0, \sigma_0^2)$, our goal is to estimate $\Upsilon_\pi \in \mathcal{P}^{\mathcal{S} \times \mathcal{A}}$. States $x \in \mathcal{S}$ and actions $a \in \mathcal{A}$ are both generated from a continuous subset of $\mathbb{R}^2$ (i.e., $\mathcal{S}, \mathcal{A} \subseteq \mathbb{R}^2$). Matrices $A, B, Q, R \in \mathbb{R}^{2 \times 2}$ that control the environment dynamics (corresponding to $\tilde{p}(r, s'|s, a)$) are as follows (but unknown in the simulations).

$$x'(x, a) = Ax + Ba \quad \& \quad R(x, a) = x^\intercal Q x + a^\intercal R a + \epsilon_R \quad \& \quad \pi(x) = Kx,$$

$$A = \begin{bmatrix} 0.6 & 0 \\ 0 & 0.8 \end{bmatrix} \quad \& \quad B = \begin{bmatrix} 0.2 & 0 \\ 0 & 0.1 \end{bmatrix} \quad \& \quad Q = \begin{bmatrix} 4 & 1 \\ 1 & 4 \end{bmatrix} \quad \& \quad R = \begin{bmatrix} 2 & 1 \\ 1 & 2 \end{bmatrix}.$$

Letting $K$ be the identity matrix, we assumed $\gamma = 0.99$ and $\sigma_0 = 1$.

Return distribution model based on (40) of Appendix C.17.3 can be written as follows,

$$\mathcal{M}_\Theta := \left\{ \Upsilon_\theta \in (\mathcal{P}(\mathbb{R}))^{\mathcal{S} \times \mathcal{A}} \quad \text{where } \Upsilon_\theta(x, a) \text{ is the probability for } Z_\theta(x, a) = \right.$$

$$\left. x^\intercal M_1 x + a^\intercal M_2 x + a^\intercal M_3 a + N\left(0, \frac{\sigma_0^2}{1 - \gamma^2}\right) \quad \text{with} \quad \theta = (M_1, M_2, M_3) \in \mathbb{R}^{2 \times 2 \times 3} \right\}.$$

Note that this setup satisfies Assumption 3.1 (proof in Appendix C.17.3).

### E.1.2  Data splitting

We applied the same splitting rule of data $\mathcal{D} = \cup_{t=1}^T \mathcal{D}_t$ by selecting the following $T$. We acknowledge that we did not strictly apply the rule of Theorem C.5 that will lead to asymptotic convergence with the suggested rate.

1. We used the rule of Theorem C.5 with the parameters of Energy FDE with inaccuracy measured with Wasserstein-1 metric ($l = 5, \delta = 1, c = 1, q = 1$). We also used $\alpha = 0$ since it is a parametric model.

2. We divided it with a constant $C_{divide} > 0$ and then choose the floored integer as follows. Since larger discount rate shall require bigger subdatasets (i.e., larger $|\mathcal{D}_t|$), we let $C_{divide} = 5$ for $\gamma = 0.99$.

$$T = \left\lfloor \frac{1}{C_{divide}} \times \frac{1}{c - \frac{1}{2q}} \cdot \frac{\min\{\delta, 1/q\}}{2(l-1) + \alpha} \cdot \log_{1/\gamma} N \right\rfloor$$

3. Allocate the same number of samples into each of the $T$ subdatasets, and then put all the remaining samples to the last one $\mathcal{D}_T$.

### E.1.3 Closed form for gaussian distributions

Since we have gaussian conditional densities, we have used the following closed form. For MMD that we have used (Energy, Laplace, RBF), we have the following with $k(x, y) = k_0(x - y)$ with mutually independent $X, X' \sim P$ and $Y, Y' \sim Q$,

$$\text{MMD}_k(P, Q) := \mathbb{E}\{k_0(X - X')\} + \mathbb{E}\{k_0(Y - Y')\} - 2 \cdot \mathbb{E}\{k_0(X - Y)\}.$$

Defining $K_0(\mu, \sigma^2) := \mathbb{E}_{Z \sim N(\mu, \sigma^2)}\{k_0(Z)\}$, we have following for Energy distance which has $k_0(y) = -|y|$,

$$K_0(\mu, \sigma^2) = - \left[ \sigma \sqrt{\frac{2}{\pi}} \exp\left(-\frac{\mu^2}{2\sigma^2}\right) + |\mu| \left(1 - 2\Phi\left(-\frac{|\mu|}{\sigma}\right)\right) \right].$$

RBF kernel with $k_0(y) = \exp(-y^2/4\sigma_{\text{RBF}}^2)$ have the following.

$$K_0(\mu, \sigma^2) = \exp\left(-\frac{\mu^2}{4\sigma_{\text{RBF}}^2 + 2\sigma^2}\right) \bigg/ \sqrt{1 + \frac{\sigma^2}{2\sigma_{\text{RBF}}^2}}$$

Laplace kernel with $k_0(y) = \exp(-|y|/\sigma_{\text{Lap}})$ has

$$K_0(\mu, \sigma^2) = \exp\left(\frac{\sigma^2}{2\sigma_{\text{Lap}}^2} - \frac{\mu}{\sigma_{\text{exp}}}\right) \cdot \Phi\left(\frac{\mu}{\sigma} - \frac{\sigma}{\sigma_{\text{Lap}}}\right)$$
$$+ \exp\left(\frac{\sigma^2}{2\sigma_{\text{Lap}}^2} + \frac{\mu}{\sigma_{\text{Lap}}}\right) \cdot \Phi\left(-\frac{\mu}{\sigma} - \frac{\sigma}{\sigma_{\text{Lap}}}\right).$$

For PDF-L2, we used the following formula for $\phi(\cdot|\mu, \sigma^2)$ represents the density function of $N(\mu, \sigma^2)$,

$$\left\|\phi(\cdot|\mu_1, \sigma_1^2) - \phi(\cdot|\mu_2, \sigma_2^2)\right\|_{L_2}^2 = \frac{1}{\sqrt{4\pi\sigma_1^2}} + \frac{1}{\sqrt{4\pi\sigma_2^2}} - \frac{2}{\sqrt{2\pi(\sigma_1^2 + \sigma_2^2)}} \cdot \exp\left(-\frac{(\mu_1 - \mu_2)^2}{2(\sigma_1^2 + \sigma_2^2)}\right).$$

For KL Divergence, we have

$$\text{KL}(N(\mu_1, \sigma_1^2) \parallel N(\mu_2, \sigma_2^2)) = \log\frac{\sigma_2}{\sigma_1} + \frac{\sigma_1^2 + (\mu_1 - \mu_2)^2}{2\sigma_2^2} - \frac{1}{2}.$$

Within our simulations, we have used fixed values of $\sigma_{\text{Lap}} = \sigma_{\text{RBF}} = 1$.

### E.1.4 Simulation results

We have applied L-BFGS-B algorithm in solving the minimization problem of (5) at each iteration $t \in [T]$. For the initial value of L-BFGS-B algorithm, we always used the previous iterate $\hat{\theta}_{n,t-1}$. Simulation results are visualized in Figure 2, with details in Table 4.

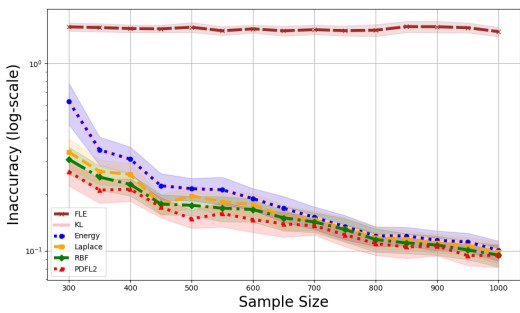

Figure 2: $\overline{\mathbb{W}}_{1,d_\pi,1}$-inaccuracy (Y-axis: logarithmic scale) for different sample sizes $N$ (X-axis) through 50 simulations. Shaded areas are (mean $\pm$ STD$/\sqrt{50}$) regions for each method, with thick lines being the means.

Table 4: Mean $\overline{\mathbb{W}}_{1,d_\pi,1}$-inaccuracy over 50 simulations (standard deviation in parentheses)

| $N$ | Energy | Laplace | RBF | PDF-L2 | KL | FLE |
|---|---|---|---|---|---|---|
| 300 | 0.627 | 0.3361 | 0.3074 | 0.2643 | 0.6302 | 1.5668 |
| | (0.5522) | (0.2142) | (0.1776) | (0.1493) | (0.5493) | (0.2838) |
| 350 | 0.3453 | 0.2648 | 0.2471 | 0.2105 | 0.3453 | 1.5537 |
| | (0.2095) | (0.1396) | (0.1287) | (0.1096) | (0.2093) | (0.2734) |
| 400 | 0.3082 | 0.2557 | 0.2261 | 0.2128 | 0.3078 | 1.5348 |
| | (0.1758) | (0.1294) | (0.1056) | (0.1025) | (0.1755) | (0.2568) |
| 450 | 0.2221 | 0.1771 | 0.1773 | 0.1710 | 0.2222 | 1.5293 |
| | (0.1268) | (0.0782) | (0.0765) | (0.0744) | (0.1269) | (0.2485) |
| 500 | 0.2143 | 0.1962 | 0.1747 | 0.1480 | 0.2143 | 1.5597 |
| | (0.1037) | (0.0946) | (0.0723) | (0.0574) | (0.1037) | (0.3352) |
| 550 | 0.2114 | 0.1821 | 0.1686 | 0.1577 | 0.2115 | 1.4932 |
| | (0.1250) | (0.0951) | (0.0913) | (0.0841) | (0.1251) | (0.2865) |
| 600 | 0.1896 | 0.1766 | 0.1658 | 0.1466 | 0.1897 | 1.5288 |
| | (0.0889) | (0.0890) | (0.0829) | (0.0759) | (0.0891) | (0.2571) |
| 650 | 0.1683 | 0.1518 | 0.1496 | 0.1392 | 0.1684 | 1.4921 |
| | (0.0936) | (0.0809) | (0.0780) | (0.0749) | (0.0936) | (0.3214) |
| 700 | 0.1507 | 0.1459 | 0.1424 | 0.1358 | 0.1506 | 1.5143 |
| | (0.0733) | (0.0628) | (0.0611) | (0.0517) | (0.0733) | (0.3056) |
| 750 | 0.1343 | 0.1313 | 0.1297 | 0.1215 | 0.1343 | 1.4949 |
| | (0.0702) | (0.0682) | (0.0656) | (0.0585) | (0.0702) | (0.3226) |
| 800 | 0.1198 | 0.1177 | 0.1146 | 0.1091 | 0.1198 | 1.5040 |
| | (0.0512) | (0.0553) | (0.0576) | (0.0517) | (0.0512) | (0.3862) |
| 850 | 0.1199 | 0.1150 | 0.1100 | 0.1047 | 0.1199 | 1.5714 |
| | (0.0479) | (0.0470) | (0.0476) | (0.0484) | (0.0479) | (0.3766) |
| 900 | 0.1136 | 0.1098 | 0.1072 | 0.1059 | 0.1136 | 1.5689 |
| | (0.0493) | (0.0450) | (0.0448) | (0.0429) | (0.0494) | (0.3886) |
| 950 | 0.1113 | 0.1063 | 0.1010 | 0.0940 | 0.1113 | 1.5502 |
| | (0.0448) | (0.0405) | (0.0404) | (0.0381) | (0.0448) | (0.3319) |
| 1000 | 0.1003 | 0.0983 | 0.0947 | 0.0944 | 0.1003 | 1.4774 |
| | (0.0438) | (0.0467) | (0.0452) | (0.0463) | (0.0438) | (0.2979) |

## E.2 Details in Atari games

### E.2.1 Deep Neural Network structure

Our model consists of two parts: (i) CNN layers that reduce $4 \times 84 \times 84$ image into $512$ features, (ii) multiple layers that transform $512$ features into $|\mathcal{A}| \times M \times 3$. Part (i) is identical to the structure of [35] that first applied DQN in Atari games. After training the optimal model via DQN (with the same network structure with [35]) for ourselves, we copied this part to (i), and then freezed it, assuming that this part already performs good recognition of the image. Part (ii) contains the components that we trained in our methods. To make the model sufficiently complex, we included multiple hidden layers that reduce the $512$ features into $512 \to 450 \to 400 \to 350 \to 300 \to 250 \to 200 \to 150 \to 128 \to |\mathcal{A}| \times (M \times 3)$, with each layer containing ReLU. For a given input image, the model outputs three parameters (weight, mean, variance) for each gaussian mixture component, for every action as follows,

$$\Upsilon_\theta(s,a) = \sum_{m=1}^{M} w_m(s,a;\theta) \cdot N(\mu_m(s,a;\theta), \sigma_m^2(s,a;\theta)) \quad \text{where} \quad M = 10, 100, 200.$$

Here, $w_m$ represents gaussian component for the GMM model, which together sum up to 1, i.e., $\sum_{m=1}^{M} w_m(s,a;\theta) = 1$.

For quantile-based methods, we preserve the same layers, except that the output distribution has $M$ quantiles instead of $(M \times 3)$ parameters. That is, this consists of multiple hidden layers that reduce the $512$ features into $512 \to 450 \to 400 \to 350 \to 300 \to 250 \to 200 \to 150 \to 128 \to |\mathcal{A}| \times M$, to form the following distributions based on dirac-delta's $\delta_m$.

$$\Upsilon_\theta(s,a) = \frac{1}{m} \sum_{m=1}^{M} \delta_m(s,a;\theta) \quad \text{where} \quad M = 10, 100, 200.$$

Note that GMM modeling can (asymptotically) accommodate dirac delta based modeling with sufficiantly small variance values $\sigma_m^2(s,a;\theta) \approx 0$ and equal weights $w_m(s,a;\theta) = 1/M$.

### E.2.2 Algorithmic details

Our goal is to estimate $\Upsilon_\pi \in \mathcal{P}^{\mathcal{S} \times \mathcal{A}}$. Here, the target policy is $\pi = \pi_{\epsilon_{\text{tar}}}^*$, which is the epsilon-greedy variant (e.g., see Section 2.2 of [53]) of DQN-trained policy $\pi_*$ [35] with $\epsilon = \epsilon_{\text{tar}}$. We collected offline data ($N = 2K, 5K, 10K$) through the trajectory of an agent following a behavior policy $b = \pi_{\epsilon_{\text{beh}}}^*$ with $\epsilon_{\text{beh}}$. For most cases, we let $\epsilon_{\text{beh}} > \epsilon_{\text{tar}}$ so as to satisfy Assumption C.4. In all simulations, we applied FDE methods based on Algorithm 1 with $T = 50$. Minimization of (5) of Algorithm 1 is done stochastically by Adam. In each $t$-th iteration ($t = 1, \cdots, T$), we ran 1000 stochastic gradient updates, each based on a batch of 32 randomly selected samples. For practicality, unlike LQR simulations shown in Appendix E.1.2, we did not use data splitting for each iteration, but instead reused samples throughout multiple iterations.

When measuring the inaccuracy, instead of $\overline{\mathbb{W}}_{1,d_\pi,1}(\Upsilon_{\theta_T}, \Upsilon_\pi)$ that requires heavy computation to approximate, we computed $\mathbb{W}_1(\Upsilon_{\theta_T}^{d_\pi}, \Upsilon_\pi^{d_\pi})$, where $\Upsilon^{d_\pi} := \int_{\mathcal{S} \times \mathcal{A}} \Upsilon(s,a) \mathrm{d}d_\pi(s,a) \in \mathcal{P}$. Here, $d_\pi$ is approximated with 1000 pre-sampled observations of state-action pairs. These are sampled by Algorithm 2, which is a commonly used strategy of sampling $s, a \sim d_\pi$ (e.g., see Algorithm 1 of [1]). Using the pre-sampled state-action pairs $s^{(m)}, a^{(m)}$ ($m = 1, \cdots, 1000$), we sample $z^{(m)} \sim \Upsilon_{\theta_T}(s^{(m)}, a^{(m)})$ for each $m$, by which we approximate $\Upsilon_{\theta_T}^{d_\pi}$. Based on the same $s^{(m)}, a^{(m)}$, we can also sample $z^{(m)} \sim \Upsilon_\pi(s^{(m)}, a^{(m)})$ by forming a long enough trajectory starting from initial state-action pair $s^{(m)}, a^{(m)}$ and adding the consecutive rewards. This gives us approximation of $\Upsilon_\pi^{d_\pi}$.

Although our theorems (Theorems 3.4, C.11) give us bounds for $\bar{\eta}_{d_\pi,q}(\Upsilon_{\theta_T}, \Upsilon_\pi)$, this can also lead to bound for $\eta(\Upsilon_{\theta_T}^{d_\pi}, \Upsilon_\pi^{d_\pi})$. This is due to the following inequality based on convexity of (15) and (16) in Appendix C.4,

$$\mathbb{W}_1(\Upsilon_{\theta_T}^{d_\pi}, \Upsilon_\pi^{d_\pi}) \leq \overline{\mathbb{W}}_{1,d_\pi,1}(\Upsilon_{\theta_T}, \Upsilon_\pi).$$

**Algorithm 2** Sampling $m$ state–action pairs from $d_\pi$

---

1: **Input:** Initial state distribution $\mu$, target policy $\pi$, discount rate $\gamma$, transition $P(\cdot|s,a)$
2: **Output:** State-action pairs $\{(s^{(m)}, a^{(m)})\}_{m=1}^{1000}$
3: **for** $m = 1$ **to** $1000$ **do**
4:     Sample $s_{\text{cur}} \sim \mu$ and $a_{\text{cur}} \sim \pi(\cdot \mid s_{\text{cur}})$
5:     $Accept \leftarrow$ **False**
6:     **while not** $Accept$ **do**
7:         Sample $U \sim \text{Unif}(0,1)$
8:         **if** $U < 1 - \gamma$ **then**
9:             $Accept \leftarrow$ **True**
10:         **else**
11:             Sample $s' \sim P(\cdot \mid s_{\text{cur}}, a_{\text{cur}})$
12:             Sample $a' \sim \pi(\cdot \mid s')$
13:             $(s_{\text{cur}}, a_{\text{cur}}) \leftarrow (s', a')$
14:         **end if**
15:     **end while**
16:     $(s^{(m)}, a^{(m)}) \leftarrow (s_{\text{cur}}, a_{\text{cur}})$
17:     **output** $(s^{(m)}, a^{(m)})$
18: **end for**

---

### E.2.3 Closed / approximated form

Based on the formula of Appendix E.1.3, we computed a single term in (10), i.e., $\mathfrak{d}(\Upsilon_\theta(s,a), \Psi_\pi(r, s', \Upsilon_{\hat{\theta}_{n,t-1}}))$, for MMD methods and PDF-L2. Since Laplace FDE had ill performance due to its numerical difficulties that we aforementioned, we excluded it. For other methods (i.e., KL, TVD, Hyvärinen divergences), we do not have a closed form objective function for GMM. Therefore, we approximated it with Monte Carlo approximation by $z'_{j_2,b} \sim^{iid} N(r + \gamma \cdot \mu_{j_2}(s', a'; \theta_{-1}), \gamma^2 \cdot \sigma^2_{j_2}(s', a'; \theta_{-1}))$ with $j_2 = 1, \cdots, M$ and $b = 1, \cdots, B$, which represents a single mixture component of $\Psi_\pi(r, s', \Upsilon_{\hat{\theta}_{n,t-1}})$.

For KL FDE, we can convert it to maximizing the expected log-likelihood, with individual term being as follows. Here, $f(\cdot|P)$ is the density of the probability measure $P$ and $\phi(\cdot|\mu, \sigma^2)$ means the density of $N(\mu, \sigma^2)$. When computing the density of $\Psi_\pi(r, s', \Upsilon_{\hat{\theta}_{n,t-1}})$, we sampled $a' \sim \pi(\cdot|s')$ instead of making use of $\pi(a'|s')$ for all $a' \in \mathcal{A}$ in every method, for the sake of computational convenience.

$$\mathbb{E}_{Z' \sim \Psi_\pi(r,s',\Upsilon_{\hat{\theta}_{n,t-1}})} \left\{ \log f(Z'|\Upsilon_\theta(s,a)) \right\} \approx$$

$$\sum_{j_2=1}^{M} w_{j_2}(s', a'; \theta_{-1}) \cdot \frac{1}{B} \sum_{b=1}^{B} \log \left\{ \sum_{j_1=1}^{M} w_{j_1}(s, a; \theta) \cdot \phi(z'_{j_2,b}; \ \mu_{j_1}(s, a; \theta), \ \sigma^2_{j_1}(s, a; \theta)) \right\}.$$

For Hyvärinen divergence, we instead maximized Hyvärinen score with individual term being $\mathbb{E}_{Z' \sim \Psi_\pi(r,s',\Upsilon_{\hat{\theta}_{n,t-1}})} \{ S_H(Z', \Upsilon_\theta(s,a)) \}$ as follows, which is computed by MC approximation in the same way (assuming GMM for $P$),

$$S_H(z, P) := \frac{\partial^2}{\partial z^2} \log f(z|P) + \frac{1}{2} \cdot \left( \frac{\partial}{\partial z} \log f(z|P) \right)^2 =$$

$$= \frac{\sum_{j=1}^{M} w_j \cdot \left\{ \frac{(z-\mu_j)^2}{\sigma_j^4} - \frac{1}{\sigma_j^2} \right\} \cdot \phi(z; \mu_j; \sigma_j^2)}{\sum_{j=1}^{M} w_j \cdot \phi(z; \mu_j, \sigma_j^2)} - \frac{1}{2} \cdot \left\{ \frac{\sum_{j=1}^{M} w_j \cdot \left( \frac{-(z-\mu_j)}{\sigma_j^2} \right) \cdot \phi(z; \mu_j, \sigma_j^2)}{\sum_{j=1}^{M} w_j \cdot \phi(z; \mu_j, \sigma_j^2)} \right\}^2.$$

For TVD, we approximated $\|f(\cdot|\Upsilon_\theta(s_i, a_i)) - f(\cdot|\Psi_\pi(r_i, s'_i, \Upsilon_{\hat\theta_{n,t-1}}, a'_i))\|_{L_1}$ by using the density ratio as follows,

$$\mathbb{E}_{Z' \sim \Psi_\pi(r, s', \Upsilon_{\hat\theta_{n,t-1}})} \left\{ \left| 1 - \frac{f(Z'|\Upsilon_\theta(s, a))}{f(Z'|\Psi_\pi(r, s', \Upsilon_{\hat\theta_{n,t-1}}))} \right| \right\} = \sum_{j_2=1}^{M} w_{j_2}(s', a'; \theta_{-1}) \times$$

$$\frac{1}{B} \sum_{b=1}^{B} \left| 1 - \frac{\sum_{j_1=1}^{M} w_{j_1}(s, a; \theta) \cdot \phi(z'_{j_2,b}|\mu_{j_1}(s, a; \theta), \sigma^2_{j_1}(s, a; \theta))}{\sum_{j_2=1}^{M} w_{j_2}(s', a'; \theta_{-1}) \cdot \phi(z'_{j_2,b}|r + \gamma \cdot \mu_{j_2}(s', a'; \theta_{-1}), \gamma^2 \cdot \sigma^2_{j_2}(s', a'; \theta_{-1}))} \right|.$$

We let $B = 100$ and $B = 50$ for deterministic transition and stochastic transition, respectively, which we will explain in the following subsection (Appendix E.2.4). We added a term $\epsilon_{\mathrm{var}} = 1.0$ to the variance $\sigma^2_j(s, a; \theta)$ in PDF-L2, KL, TVD, to prevent explosion of density values, thereby mitigating numerical instability, and used the same value for the tuning parameter of RBF FDE, i.e., $\sigma_{\mathrm{RBF}} = 1.0$. For Hyvärinen, we put $\epsilon_{\mathrm{var}} = 10.0$ since it was more prone to numerical instabilities. Even so, we could not run simulations for $M = 200$ due to numerical instabilities. The good thing about Energy FDE is that it does not require any tuning parameter, which makes it more user-friendly (and performance was good as well).

### E.2.4 Simulation settings

In our simulations, we always assumed $\gamma = 0.95$. Every time a reward is observed, we clipped it between $-10$ and $10$, and then multiplied it by a fixed constant (20 in our case). Then, we tried two different settings: (i) deterministic transition $\tilde{p}(r, s'|s, a)$ as the original Atari games, (ii) added small noise $N(0, 1)$ to every reward to make it more random (which makes it a better environment to apply distributional RL). We also tried two behavior policies for each setting, which leads to different coverage.

First, let us start with Setting-1, deterministic transition. Here, the target policy is $\pi^*_{\epsilon_{\mathrm{tar}}}$ ($\epsilon_{\mathrm{tar}}$-greedy version of DQN-trained optimal policy $\pi^*$) with $\epsilon_{\mathrm{tar}} = 0.3$. The behavior policy is $\epsilon_{\mathrm{beh}} = 0.4, 0.8$ (but $\epsilon_{\mathrm{beh}} = 0.4, 0.5$ for Enduro and Pong to prevent sparse observation of rewards). Of course, weak coverage (i.e., $\epsilon_{\mathrm{beh}}$ being a lot bigger than $\epsilon_{\mathrm{tar}}$) leads to worse performance. Results are visualized in the first two rows of Figures 3–5 for $N = 10K$ (Tables 6–19 for $N = 2K, 5K, 10K$). In the simulations, $M$ refers to the number of gaussian mixtures in GMM model and the number of quantiles in quantile-based models (QRDQN, IQN).

Second, we ran simulations on Setting-2, random transition (or reward). Here, the target policy is $\pi = \pi^*$ (deterministic policy learned by DQN) in all games except Breakout. In Breakout, we let $\pi = \pi^*_{\epsilon_{\mathrm{tar}}}$ with $\epsilon_{\mathrm{tar}} = 0.3$ to prevent the agent from being stuck at certain point (not proceeding with the game and repeating the same actions). The behavior policy has $\epsilon_{\mathrm{beh}} = 0.1, 0.5$ for all games ($\epsilon_{\mathrm{beh}} = 0.1, 0.3$ for Pong only). Results are visualized in the last two rows of Figures 3–5 for $N = 10K$ (Tables 20–33 for $N = 2K, 5K, 10K$).

In our simulations, we have tried 7 games, 3 different sample sizes ($N = 2K, 5K, 10K$), 3 different number of mixtures ($M = 10, 100, 200$), 2 different environments (random / deterministic reward), 2 different behavior policies (good or bad coverage). These amount to $7 \times 3 \times 3 \times 2 \times 2 = 252$ different settings, each leading to various shapes of return distributions. In each setting, we have applied different methodologies (FLE, QRDQN, IQM, KL, Energy, PDF-L2, RBF, TVD, Hyvärinen) under 5 different seeds. In Figures 3–5, we plotted (mean $\pm$ STD) area for each method.

### E.2.5 Simulation results

Simulation results for each of the aforementioned 252 settings are shown in Figures 3–5 and Tables 6–33. As we have stronger coverage (i.e., $\epsilon_{\mathrm{beh}}$ being closer to $\epsilon_{\mathrm{tar}}$), we achieve lower inaccuracy levels. In many cases, we could see that the inaccuracy levels generally became lower with larger sample sizes. We could also see that density modeling (based on gaussian mixture models) helps improving the accuracy, even when the reward is deterministic. This can be seen by comparing our methods (KL, Energy, PDF-L2, RBF) with quantile-based methods (QRDQN, IQN) that use the same number of $M$ (number of gaussian components in GMM or number of quantiles). Moreover, TVD does not improve the accuracy throughout iterations. This corroborates our claim in Theorem 2.4, since TVD is not a functional Bregman divergence.

Since it is difficult to compare the performances for every single setting, we summarized the performances of each method in Table 5. We have grouped the 252 settings into four categories based on (i) deterministic / random reward and (ii) strong / weak coverage. In each category (which consists of 84 settings), we measured the rank values of nine methods (with 1 being the best and 9 being the worst) based on their mean inaccuracy in each setting, and then recorded the mean of rank values. Our FDE methods (KL, Energy, Hyvärinen) showed the highest three accuracies in most categories. Although KL FDE recorded the highest mean of rank in all four categories, it does not mean that we should always resort to KL FDE. There have been well-known criticism on KL divergence (e.g., unbounded divergence value, high sensitivity to the tails of distributions, necessity that one probability measure is absolutely continuous to the other, inability to consider closeness in outcome values) [e.g., 28, 50, 7].

Table 5: Mean of rank values of inaccuracy under all settings of each category. Three methods with best accuracies are boldfaced in each category.

| Category | | Method | | | | | | | | |
|---|---|---|---|---|---|---|---|---|---|---|
| Reward | Cover | FLE | QRD | IQN | KL | Energy | PDF | RBF | TVD | Hyv |
| Deterministic | Strong | 3.60 | 6.19 | 8.25 | **1.77** | **3.36** | 4.38 | 5.03 | 8.11 | **2.78** |
| Deterministic | Weak | **3.00** | 6.39 | 8.30 | **1.84** | 3.14 | 4.69 | 5.47 | 8.09 | **2.42** |
| Random | Strong | 4.42 | 5.30 | 7.00 | **3.03** | **3.15** | 4.07 | 4.07 | 8.63 | **3.59** |
| Random | Weak | 3.76 | 6.09 | 7.31 | **2.34** | **3.61** | 4.46 | 4.23 | 8.61 | **2.40** |

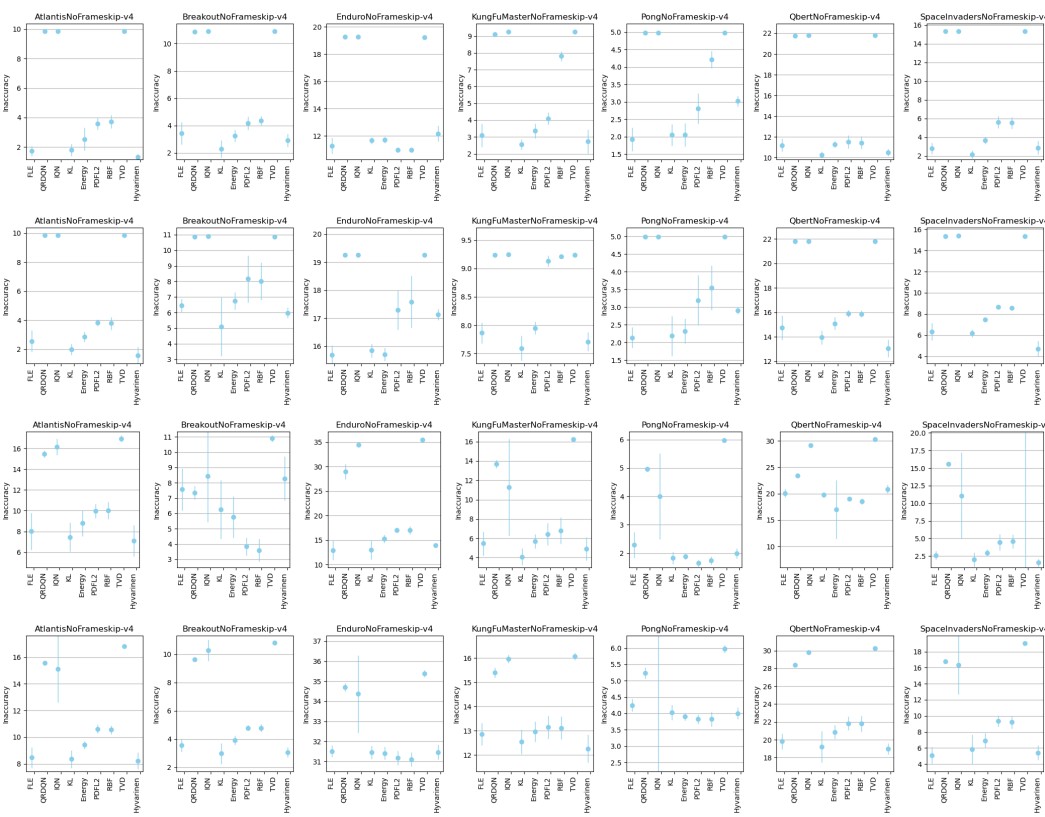

Figure 3: (Mean $\pm$ STD) of $\mathbb{W}_1(\Upsilon_\theta^{d_\pi}, \Upsilon_\pi^{d_\pi})$-inaccuracy in each games (columns) for different methods with $M = 10$, $N = 10K$: (row1) deterministic transition with strong coverage ($\epsilon_{\text{beh}} = 0.4$), (row2) deterministic transition with weak coverage ($\epsilon_{\text{beh}} > 0.4$), (row3) random transition with strong coverage ($\epsilon_{\text{beh}} = 0.1$), (row4) random transition with weak coverage ($\epsilon_{\text{beh}} > 0.1$).

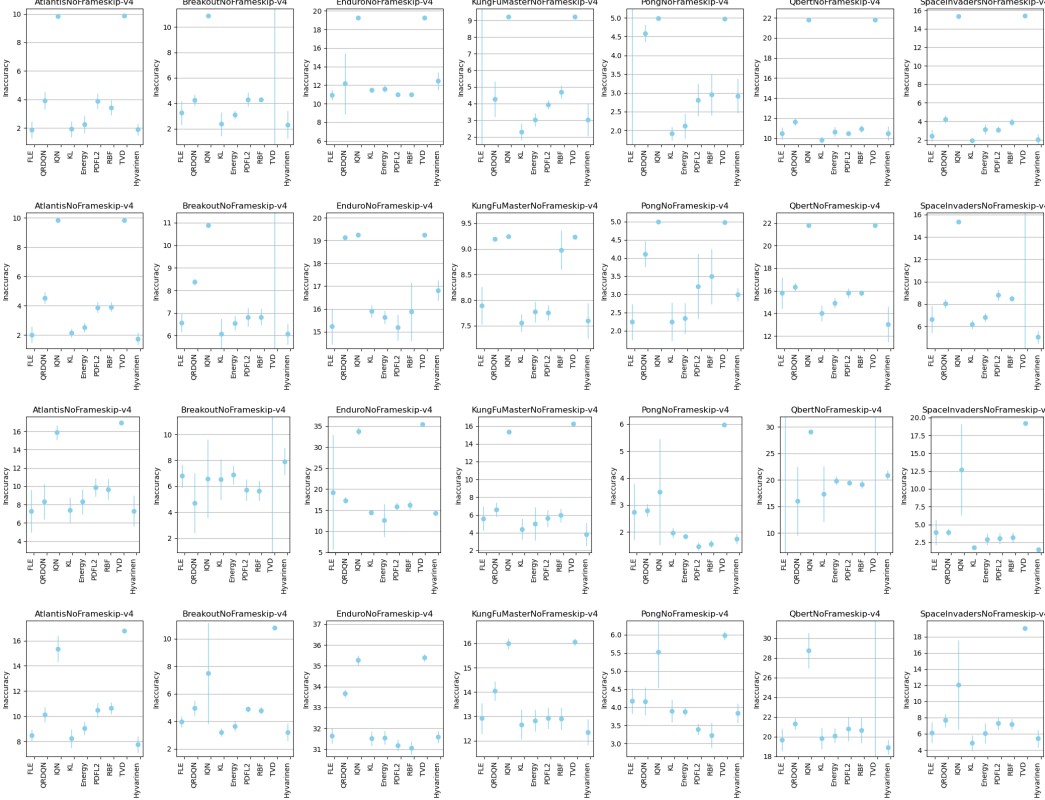

Figure 4: (Mean $\pm$ STD) of $\mathbb{W}_1(\Upsilon_\theta^{d_\pi}, \Upsilon_\pi^{d_\pi})$-inaccuracy in each games (columns) for different methods with $M = 100$, $N = 10K$: (row1) deterministic transition with strong coverage ($\epsilon_{\mathrm{beh}} = 0.4$), (row2) deterministic transition with weak coverage ($\epsilon_{\mathrm{beh}} > 0.4$), (row3) random transition with strong coverage ($\epsilon_{\mathrm{beh}} = 0.1$), (row4) random transition with weak coverage ($\epsilon_{\mathrm{beh}} > 0.1$).

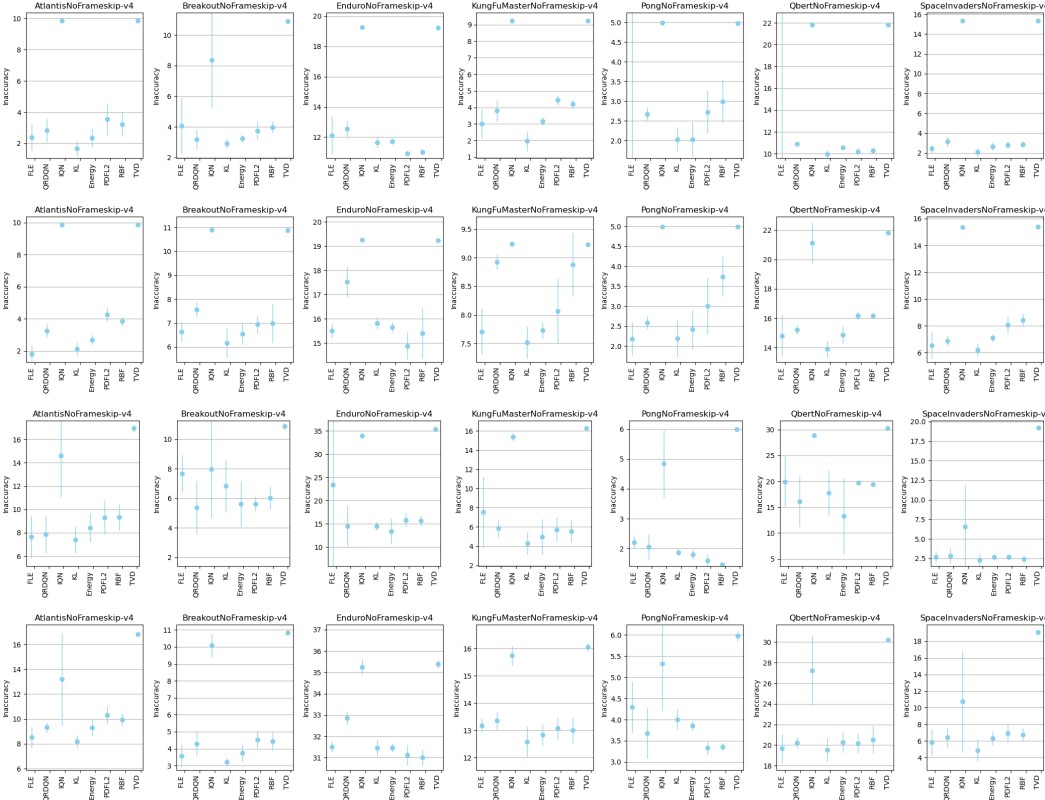

Figure 5: (Mean ± STD) of $\mathbb{W}_1(\Upsilon_\theta^{d_\pi}, \Upsilon_\pi^{d_\pi})$-inaccuracy in each games (columns) for different methods with $M = 200$, $N = 10K$: (row1) deterministic transition with strong coverage ($\epsilon_{\text{beh}} = 0.4$), (row2) deterministic transition with weak coverage ($\epsilon_{\text{beh}} > 0.4$), (row3) random transition with strong coverage ($\epsilon_{\text{beh}} = 0.1$), (row4) random transition with weak coverage ($\epsilon_{\text{beh}} > 0.1$).

Table 6: Reward-variance=0, Eps=Small, Game=AtlantisNoFrameskip-v4

| $M$ | $N$ | FLE | QRD | IQN | KLD | ENE | PDF | RBF | TVD | HYV |
|---|---|---|---|---|---|---|---|---|---|---|
|  | $2K$ | 2.02 | 9.84 | 9.86 | 2.13 | 2.54 | 3.54 | 3.7 | 9.85 | 1.3 |
| 10 | $5K$ | 1.87 | 9.84 | 9.86 | 1.51 | 2.31 | 3.6 | 3.73 | 9.85 | 1.29 |
|  | $10K$ | 1.73 | 9.84 | 9.86 | 1.8 | 2.54 | 3.59 | 3.72 | 9.86 | 1.34 |
|  | $2K$ | 2.22 | 4.13 | 9.85 | 1.71 | 2.28 | 3.54 | 3.77 | 9.87 | 1.77 |
| 100 | $5K$ | 1.88 | 3.87 | 9.85 | 1.67 | 2.29 | 3.65 | 3.44 | 9.85 | 1.77 |
|  | $10K$ | 1.85 | 3.93 | 9.85 | 1.93 | 2.26 | 3.87 | 3.42 | 9.87 | 1.88 |
|  | $2K$ | 2.27 | 2.84 | 9.85 | 1.93 | 2.4 | 3.76 | 3.63 | 9.85 | NA |
| 200 | $5K$ | 1.69 | 2.88 | 9.38 | 1.61 | 2.33 | 3.34 | 3.16 | 9.85 | NA |
|  | $10K$ | 2.39 | 2.83 | 9.85 | 1.68 | 2.35 | 3.56 | 3.24 | 9.86 | NA |

Table 7: Reward-variance=0, Eps=Small, Game=BreakoutNoFrameskip-v4

| $M$ | $N$ | FLE | QRD | IQN | KLD | ENE | PDF | RBF | TVD | HYV |
|---|---|---|---|---|---|---|---|---|---|---|
|  | $2K$ | 3.63 | 10.87 | 10.89 | 6.68 | 3.37 | 4.48 | 4.81 | 10.89 | 2.81 |
| 10 | $5K$ | 3.21 | 10.87 | 10.9 | 2.78 | 3.22 | 4.39 | 4.37 | 10.89 | 2.58 |
|  | $10K$ | 3.44 | 10.87 | 10.89 | 2.29 | 3.25 | 4.18 | 4.36 | 10.89 | 2.91 |
|  | $2K$ | 3.51 | 4.93 | 10.89 | 2.7 | 3.49 | 4.56 | 4.43 | 10.88 | 2.91 |
| 100 | $5K$ | 3.47 | 4.58 | 10.89 | 2.9 | 3.29 | 4.38 | 4.22 | 10.88 | 3.19 |
|  | $10K$ | 3.24 | 4.24 | 10.88 | 2.37 | 3.1 | 4.28 | 4.29 | >100 | 2.32 |
|  | $2K$ | 3.28 | 3.88 | 10.88 | 3.15 | 3.41 | 4.49 | 4.21 | 10.9 | NA |
| 200 | $5K$ | 3.39 | 3.65 | 10.9 | 3.02 | 3.26 | 4.11 | 3.92 | 10.9 | NA |
|  | $10K$ | 4.06 | 3.19 | 8.34 | 2.9 | 3.23 | 3.74 | 3.97 | 10.9 | NA |

Table 8: Reward-variance=0, Eps=Small, Game=EnduroNoFrameskip-v4

| $M$ | $N$ | FLE | QRD | IQN | KLD | ENE | PDF | RBF | TVD | HYV |
|---|---|---|---|---|---|---|---|---|---|---|
|  | $2K$ | 12.69 | 19.25 | 19.25 | 12.68 | 13.0 | 12.37 | 12.36 | 19.25 | 14.1 |
| 10 | $5K$ | >100 | 19.25 | 19.25 | 9.42 | 11.17 | 10.89 | 10.81 | 19.25 | 11.32 |
|  | $10K$ | 11.28 | 19.25 | 19.25 | 11.67 | 11.72 | 10.98 | 10.98 | 19.24 | 12.17 |
|  | $2K$ | 12.52 | 15.33 | 19.26 | 11.08 | 12.83 | 12.31 | 12.4 | 19.25 | 12.99 |
| 100 | $5K$ | 11.51 | 13.51 | 19.25 | 10.64 | 10.91 | 10.84 | 10.93 | 19.24 | 10.65 |
|  | $10K$ | 10.95 | 12.15 | 19.25 | 11.46 | 11.61 | 10.99 | 11.0 | 19.25 | 12.43 |
|  | $2K$ | 12.6 | 14.31 | 19.25 | 12.24 | 12.9 | 12.51 | 12.57 | 19.25 | NA |
| 200 | $5K$ | 10.47 | 12.5 | 19.26 | 10.58 | 11.1 | 10.89 | 10.85 | 19.24 | NA |
|  | $10K$ | 12.11 | 12.55 | 19.26 | 11.65 | 11.71 | 10.92 | 11.0 | 19.24 | NA |

Table 9: Reward-variance=0, Eps=Small, Game=KungFuMasterNoFrameskip-v4

| $M$ | $N$ | FLE | QRD | IQN | KLD | ENE | PDF | RBF | TVD | HYV |
|---|---|---|---|---|---|---|---|---|---|---|
| 10 | $2K$ | 2.83 | 9.04 | 9.25 | 1.84 | 2.66 | 3.61 | 4.12 | 9.23 | 1.91 |
| | $5K$ | 2.82 | 9.09 | 9.25 | 2.85 | 3.38 | 4.3 | 6.7 | 9.24 | 2.6 |
| | $10K$ | 3.1 | 9.1 | 9.26 | 2.57 | 3.36 | 4.09 | 7.8 | 9.26 | 2.72 |
| 100 | $2K$ | 2.53 | 4.1 | 9.22 | 1.42 | 2.52 | 3.3 | 3.62 | 9.26 | 2.17 |
| | $5K$ | 2.89 | 4.8 | 9.23 | 2.85 | 3.38 | 4.05 | 4.52 | 9.25 | 3.65 |
| | $10K$ | >100 | 4.26 | 9.23 | 2.29 | 3.02 | 3.94 | 4.7 | 9.23 | 3.03 |
| 200 | $2K$ | 2.74 | 3.16 | 9.24 | 2.0 | 2.63 | 3.38 | 3.49 | 9.26 | NA |
| | $5K$ | 3.73 | 4.19 | 9.24 | 2.77 | 3.43 | 4.19 | 4.44 | 9.24 | NA |
| | $10K$ | 3.01 | 3.79 | 9.23 | 1.97 | 3.17 | 4.43 | 4.2 | 9.24 | NA |

Table 10: Reward-variance=0, Eps=Small, Game=PongNoFrameskip-v4

| $M$ | $N$ | FLE | QRD | IQN | KLD | ENE | PDF | RBF | TVD | HYV |
|---|---|---|---|---|---|---|---|---|---|---|
| 10 | $2K$ | 1.79 | 4.99 | 4.99 | 1.66 | 1.79 | 2.42 | 3.93 | 4.98 | 3.2 |
| | $5K$ | 1.57 | 4.99 | 4.99 | 92.04 | 1.64 | 2.53 | 3.92 | 4.98 | 2.8 |
| | $10K$ | 1.92 | 4.99 | 4.99 | 2.05 | 2.05 | 2.81 | 4.22 | 4.98 | 3.02 |
| 100 | $2K$ | 1.53 | 4.71 | 4.99 | 1.76 | 1.62 | 2.38 | 2.41 | 4.98 | 2.74 |
| | $5K$ | 1.62 | 4.61 | 4.99 | 1.77 | 1.65 | 2.5 | 2.62 | 4.98 | 2.61 |
| | $10K$ | >100 | 4.59 | 4.99 | 1.91 | 2.12 | 2.81 | 2.96 | 4.98 | 2.92 |
| 200 | $2K$ | 1.62 | 2.75 | 4.99 | 1.87 | 1.6 | 2.32 | 2.39 | 4.98 | NA |
| | $5K$ | 4.84 | 2.62 | 4.99 | 1.76 | 1.63 | 2.42 | 2.52 | 4.99 | NA |
| | $10K$ | >100 | 2.67 | 4.99 | 2.04 | 2.04 | 2.73 | 2.99 | 4.98 | NA |

Table 11: Reward-variance=0, Eps=Small, Game=QbertNoFrameskip-v4

| $M$ | $N$ | FLE | QRD | IQN | KLD | ENE | PDF | RBF | TVD | HYV |
|---|---|---|---|---|---|---|---|---|---|---|
| 10 | $2K$ | >100 | 21.74 | 21.82 | 10.8 | 11.82 | 12.74 | 12.9 | 21.81 | 10.87 |
| | $5K$ | 10.96 | 21.76 | 21.81 | 10.79 | 11.52 | 12.5 | 12.09 | 21.83 | 10.74 |
| | $10K$ | 11.2 | 21.76 | 21.81 | 10.29 | 11.29 | 11.55 | 11.45 | 21.81 | 10.51 |
| 100 | $2K$ | 23.68 | 12.79 | 21.8 | 10.69 | 11.47 | 12.0 | 12.29 | 21.8 | 10.08 |
| | $5K$ | 10.71 | 11.94 | 18.76 | 10.34 | 11.26 | 10.7 | 11.42 | 21.81 | 10.63 |
| | $10K$ | 10.47 | 11.65 | 21.8 | 9.83 | 10.65 | 10.46 | 10.95 | 21.79 | 10.49 |
| 200 | $2K$ | 11.67 | 11.88 | 21.81 | 10.74 | 11.37 | 11.76 | 11.79 | 21.81 | NA |
| | $5K$ | 11.19 | 11.13 | 21.81 | 10.22 | 10.93 | 10.68 | 10.81 | 21.81 | NA |
| | $10K$ | >100 | 10.9 | 21.8 | 9.98 | 10.59 | 10.21 | 10.29 | 21.8 | NA |

Table 12: Reward-variance=0, Eps=Small, Game=SpaceInvadersNoFrameskip-v4

| $M$ | $N$ | FLE | QRD | IQN | KLD | ENE | PDF | RBF | TVD | HYV |
|---|---|---|---|---|---|---|---|---|---|---|
| 10 | $2K$ | 3.02 | 15.35 | 15.37 | 2.73 | 3.55 | 5.76 | 5.82 | 15.38 | 2.73 |
| | $5K$ | 2.87 | 15.35 | 15.37 | 2.16 | 3.72 | 5.79 | 5.71 | >100 | 2.2 |
| | $10K$ | 2.79 | 15.35 | 15.37 | 2.17 | 3.66 | 5.63 | 5.54 | 15.36 | 2.88 |
| 100 | $2K$ | 3.03 | 4.53 | 15.37 | 2.42 | 3.07 | 3.73 | 4.8 | 15.37 | 2.38 |
| | $5K$ | 2.58 | 4.23 | 14.07 | 2.15 | 3.22 | 3.6 | 4.31 | 15.38 | 2.31 |
| | $10K$ | 2.43 | 4.24 | 15.37 | 1.93 | 3.12 | 3.09 | 3.91 | 15.38 | 2.06 |
| 200 | $2K$ | 3.39 | 3.5 | 15.36 | 2.54 | 3.1 | 3.58 | 4.06 | 15.36 | NA |
| | $5K$ | 2.92 | 3.04 | 11.07 | 2.58 | 3.07 | 3.22 | 3.74 | 15.38 | NA |
| | $10K$ | 2.43 | 3.14 | 15.37 | 2.07 | 2.67 | 2.82 | 2.88 | 15.37 | NA |

Table 13: Reward-variance=0, Eps=Big, Game=AtlantisNoFrameskip-v4

| $M$ | $N$ | FLE | QRD | IQN | KLD | ENE | PDF | RBF | TVD | HYV |
|---|---|---|---|---|---|---|---|---|---|---|
| | $2K$ | 2.31 | 9.84 | 9.86 | 2.34 | 3.09 | 3.82 | 3.91 | 9.86 | 1.5 |
| 10 | $5K$ | 1.87 | 9.84 | 9.86 | 1.67 | 2.53 | 3.5 | 3.56 | 9.87 | 1.88 |
| | $10K$ | 2.55 | 9.84 | 9.86 | 1.98 | 2.87 | 3.82 | 3.78 | 9.85 | 1.58 |
| | $2K$ | 2.25 | 4.5 | 9.85 | 2.18 | 2.81 | 3.9 | 3.91 | 9.87 | 1.46 |
| 100 | $5K$ | 2.42 | 4.17 | 9.86 | 1.97 | 2.36 | 3.22 | 3.46 | 9.85 | 1.49 |
| | $10K$ | 2.02 | 4.54 | 9.85 | 2.14 | 2.51 | 3.88 | 3.92 | 9.85 | 1.74 |
| | $2K$ | 2.65 | 3.32 | 9.86 | 2.2 | 2.77 | 4.21 | 3.92 | 9.84 | NA |
| 200 | $5K$ | 1.8 | 3.06 | 9.86 | 1.7 | 2.39 | 3.78 | 3.78 | 9.85 | NA |
| | $10K$ | 1.81 | 3.26 | 9.85 | 2.12 | 2.7 | 4.27 | 3.84 | 9.86 | NA |

Table 14: Reward-variance=0, Eps=Big, Game=BreakoutNoFrameskip-v4

| $M$ | $N$ | FLE | QRD | IQN | KLD | ENE | PDF | RBF | TVD | HYV |
|---|---|---|---|---|---|---|---|---|---|---|
| | $2K$ | 7.14 | 10.88 | 10.9 | 6.66 | 7.29 | 9.31 | 7.95 | 10.9 | 6.72 |
| 10 | $5K$ | 6.13 | 10.88 | 10.9 | 5.6 | 6.87 | 8.25 | 8.39 | 10.89 | 6.02 |
| | $10K$ | 6.45 | 10.88 | 10.9 | 5.09 | 6.74 | 8.17 | 8.01 | 10.88 | 5.98 |
| | $2K$ | 6.66 | 8.97 | 10.89 | 6.73 | 7.08 | 7.13 | 7.37 | 10.89 | 6.28 |
| 100 | $5K$ | 6.8 | 8.24 | 10.89 | 6.13 | 6.6 | 7.15 | 7.09 | 10.9 | 6.18 |
| | $10K$ | 6.56 | 8.37 | 10.89 | 6.08 | 6.55 | 6.82 | 6.83 | >100 | 6.07 |
| | $2K$ | 7.25 | 7.98 | 10.9 | 6.75 | 7.03 | 7.21 | 7.45 | 10.88 | NA |
| 200 | $5K$ | 6.67 | 7.51 | 10.85 | 6.42 | 6.82 | 7.09 | 6.92 | 10.9 | NA |
| | $10K$ | 6.63 | 7.57 | 10.9 | 6.17 | 6.54 | 6.94 | 6.99 | 10.87 | NA |

Table 15: Reward-variance=0, Eps=Big, Game=EnduroNoFrameskip-v4

| $M$ | $N$ | FLE | QRD | IQN | KLD | ENE | PDF | RBF | TVD | HYV |
|---|---|---|---|---|---|---|---|---|---|---|
| | $2K$ | 16.85 | 19.26 | 19.25 | 16.72 | 16.65 | 16.97 | 17.78 | 19.24 | 17.6 |
| 10 | $5K$ | 15.46 | 19.26 | 19.26 | 15.33 | 15.29 | 15.34 | 16.22 | 19.24 | 16.8 |
| | $10K$ | 15.68 | 19.26 | 19.26 | 15.85 | 15.7 | 17.29 | 17.59 | 19.25 | 17.12 |
| | $2K$ | 16.79 | 18.53 | 19.25 | 16.7 | 16.7 | 16.42 | 16.88 | 19.24 | 17.2 |
| 100 | $5K$ | 15.69 | 18.5 | 19.26 | 15.42 | 15.34 | 14.79 | 15.9 | 19.24 | 16.25 |
| | $10K$ | 15.23 | 19.14 | 19.25 | 15.89 | 15.65 | 15.19 | 15.89 | 19.25 | 16.81 |
| | $2K$ | 16.99 | 17.94 | 19.25 | 16.58 | 16.73 | 16.48 | 16.75 | 19.25 | NA |
| 200 | $5K$ | 15.6 | 17.13 | 19.25 | 15.28 | 15.27 | 14.89 | 15.8 | 19.25 | NA |
| | $10K$ | 15.5 | 17.53 | 19.26 | 15.81 | 15.65 | 14.88 | 15.39 | 19.25 | NA |

Table 16: Reward-variance=0, Eps=Big, Game=KungFuMasterNoFrameskip-v4

| $M$ | $N$ | FLE | QRD | IQN | KLD | ENE | PDF | RBF | TVD | HYV |
|---|---|---|---|---|---|---|---|---|---|---|
| | $2K$ | 8.37 | 9.24 | 9.24 | 8.02 | 8.19 | 9.12 | 9.21 | 9.24 | 8.23 |
| 10 | $5K$ | 8.07 | 9.24 | 9.25 | 7.73 | 8.03 | 9.15 | 9.22 | 9.25 | 7.75 |
| | $10K$ | 7.86 | 9.24 | 9.24 | 7.6 | 7.95 | 9.13 | 9.21 | 9.24 | 7.71 |
| | $2K$ | 8.16 | 9.2 | 9.24 | 8.05 | 8.16 | 8.11 | 8.76 | 9.22 | 8.06 |
| 100 | $5K$ | 7.94 | 9.2 | 9.23 | 7.67 | 7.92 | 8.44 | 8.94 | 9.26 | 7.67 |
| | $10K$ | 7.89 | 9.2 | 9.24 | 7.55 | 7.77 | 7.75 | 8.98 | 9.23 | 7.6 |
| | $2K$ | 8.29 | 9.09 | 9.23 | 8.05 | 8.1 | 8.2 | 8.86 | 9.25 | NA |
| 200 | $5K$ | 8.0 | 9.0 | 9.24 | 7.64 | 7.94 | 8.25 | 8.75 | 9.23 | NA |
| | $10K$ | 7.71 | 8.93 | 9.24 | 7.52 | 7.73 | 8.07 | 8.88 | 9.23 | NA |

Table 17: Reward-variance=0, Eps=Big, Game=PongNoFrameskip-v4

| $M$ | $N$ | FLE | QRD | IQN | KLD | ENE | PDF | RBF | TVD | HYV |
|---|---|---|---|---|---|---|---|---|---|---|
| | $2K$ | 2.36 | 4.99 | 4.99 | 2.33 | 2.51 | 2.95 | 3.35 | 4.98 | 3.1 |
| 10 | $5K$ | >100 | 4.99 | 4.99 | 2.26 | 2.66 | 3.51 | 3.78 | 4.98 | 3.05 |
| | $10K$ | 2.13 | 4.99 | 4.99 | 2.18 | 2.32 | 3.19 | 3.54 | 4.98 | 2.9 |
| | $2K$ | 2.16 | 3.86 | 4.6 | 2.41 | 2.4 | 3.57 | >100 | 4.98 | 2.57 |
| 100 | $5K$ | 2.23 | 3.78 | 4.99 | 2.46 | 2.76 | 3.71 | 3.71 | 4.98 | 3.16 |
| | $10K$ | 2.24 | 4.11 | 4.99 | 2.25 | 2.34 | 3.22 | 3.49 | 4.98 | 3.0 |
| | $2K$ | 2.43 | 2.54 | 4.99 | 2.58 | 2.44 | 3.13 | 3.51 | 4.98 | NA |
| 200 | $5K$ | 2.43 | 2.58 | 4.99 | 2.57 | 2.55 | 3.56 | 3.71 | 4.98 | NA |
| | $10K$ | 2.19 | 2.59 | 4.99 | 2.19 | 2.42 | 3.0 | 3.74 | 4.99 | NA |

Table 18: Reward-variance=0, Eps=Big, Game=QbertNoFrameskip-v4

| $M$ | $N$ | FLE | QRD | IQN | KLD | ENE | PDF | RBF | TVD | HYV |
|---|---|---|---|---|---|---|---|---|---|---|
| | $2K$ | 14.35 | 21.8 | 21.81 | 12.08 | 14.59 | 15.56 | 15.6 | 21.81 | 13.04 |
| 10 | $5K$ | 14.27 | 21.8 | 21.82 | 12.62 | 14.72 | 15.8 | 15.63 | 21.82 | 12.88 |
| | $10K$ | 14.75 | 21.8 | 21.81 | 13.96 | 15.1 | 15.91 | 15.89 | 21.8 | 13.06 |
| | $2K$ | >100 | 15.4 | 21.81 | 14.24 | 14.31 | 15.64 | 15.43 | >100 | 12.68 |
| 100 | $5K$ | 13.95 | 15.84 | 21.81 | 13.91 | 14.7 | 15.81 | 15.7 | 21.82 | 13.22 |
| | $10K$ | 15.8 | 16.35 | 21.81 | 13.99 | 14.93 | 15.79 | 15.79 | 21.79 | 13.03 |
| | $2K$ | 14.03 | 14.72 | 20.51 | 12.97 | 14.14 | 15.45 | 15.57 | 21.81 | NA |
| 200 | $5K$ | 13.71 | 14.69 | 21.82 | 13.69 | 14.7 | 16.06 | 15.9 | 21.8 | NA |
| | $10K$ | 14.79 | 15.22 | 21.13 | 13.9 | 14.87 | 16.16 | 16.17 | 21.82 | NA |

Table 19: Reward-variance=0, Eps=Big, Game=SpaceInvadersNoFrameskip-v4

| $M$ | $N$ | FLE | QRD | IQN | KLD | ENE | PDF | RBF | TVD | HYV |
|---|---|---|---|---|---|---|---|---|---|---|
| | $2K$ | 7.26 | 15.36 | 15.38 | 7.39 | 8.12 | 9.22 | 9.34 | >100 | 5.95 |
| 10 | $5K$ | 7.24 | 15.36 | 15.37 | 6.91 | 7.81 | 8.88 | 8.94 | 15.37 | 5.8 |
| | $10K$ | 6.33 | 15.36 | 15.37 | 6.17 | 7.45 | 8.69 | 8.58 | 15.37 | 4.72 |
| | $2K$ | 7.87 | 9.29 | 14.73 | 7.45 | 7.85 | 8.95 | 8.91 | 15.39 | 5.89 |
| 100 | $5K$ | 7.26 | 9.09 | 15.37 | 6.78 | 7.59 | 9.1 | 8.65 | >100 | 5.47 |
| | $10K$ | 6.64 | 8.06 | 15.38 | 6.19 | 6.8 | 8.79 | 8.48 | >100 | 5.04 |
| | $2K$ | 7.5 | 8.13 | 15.37 | 7.41 | 7.92 | 9.0 | 8.5 | 15.38 | NA |
| 200 | $5K$ | 6.98 | 7.92 | 15.37 | 6.7 | 7.45 | 8.68 | 8.59 | 15.37 | NA |
| | $10K$ | 6.55 | 6.88 | 15.36 | 6.18 | 7.11 | 8.09 | 8.43 | 15.38 | NA |

Table 20: Reward-variance=1, Eps=Small, Game=AtlantisNoFrameskip-v4

| $M$ | $N$ | FLE | QRD | IQN | KLD | ENE | PDF | RBF | TVD | HYV |
|---|---|---|---|---|---|---|---|---|---|---|
| | $2K$ | 9.3 | 15.58 | 16.0 | 8.67 | 9.79 | 10.64 | 10.86 | >100 | 8.66 |
| 10 | $5K$ | 7.43 | 15.34 | 16.37 | 6.61 | 8.23 | 9.8 | 9.69 | 16.96 | 6.88 |
| | $10K$ | 8.01 | 15.47 | 16.14 | 7.46 | 8.78 | 9.98 | 10.01 | 16.95 | 7.09 |
| | $2K$ | 8.96 | 10.37 | 16.05 | 8.52 | 9.81 | 10.71 | 10.78 | 16.89 | 8.14 |
| 100 | $5K$ | 7.87 | 8.96 | 13.8 | 6.38 | 7.73 | 9.28 | 9.13 | 16.95 | 6.47 |
| | $10K$ | 7.27 | 8.35 | 15.89 | 7.39 | 8.34 | 9.87 | 9.68 | 16.95 | 7.31 |
| | $2K$ | 8.04 | 9.62 | 12.89 | 8.71 | 9.66 | 10.34 | 10.34 | 16.89 | NA |
| 200 | $5K$ | 7.51 | 8.09 | 16.08 | 6.91 | 7.73 | 8.47 | 8.4 | 16.95 | NA |
| | $10K$ | 7.67 | 7.87 | 14.63 | 7.41 | 8.44 | 9.3 | 9.33 | 16.95 | NA |

Table 21: Reward-variance=1, Eps=Small, Game=BreakoutNoFrameskip-v4

| $M$ | $N$ | FLE | QRD | IQN | KLD | ENE | PDF | RBF | TVD | HYV |
|---|---|---|---|---|---|---|---|---|---|---|
| | $2K$ | 7.94 | 7.45 | 8.91 | 7.96 | 6.21 | 4.47 | 4.66 | >100 | 8.2 |
| 10 | $5K$ | 7.75 | 7.23 | 8.26 | 6.41 | 6.88 | 4.97 | 4.88 | 10.89 | 8.56 |
| | $10K$ | 7.57 | 7.35 | 8.45 | 6.27 | 5.76 | 3.84 | 3.6 | 10.9 | 8.29 |
| | $2K$ | 7.62 | 4.8 | 7.58 | 7.65 | 6.48 | 3.94 | 5.52 | >100 | 7.87 |
| 100 | $5K$ | 8.17 | 6.02 | 7.09 | 7.15 | 6.34 | 6.49 | 6.32 | 10.89 | 9.81 |
| | $10K$ | 6.8 | 4.72 | 6.58 | 6.52 | 6.88 | 5.72 | 5.66 | >100 | 7.92 |
| | $2K$ | 7.14 | 5.58 | 9.33 | 7.51 | 6.38 | 5.43 | 5.9 | >100 | NA |
| 200 | $5K$ | 8.37 | 6.77 | 6.52 | 7.14 | 7.37 | 6.28 | 6.35 | 10.88 | NA |
| | $10K$ | 7.67 | 5.37 | 7.97 | 6.84 | 5.62 | 5.62 | 6.03 | 10.9 | NA |

Table 22: Reward-variance=1, Eps=Small, Game=EnduroNoFrameskip-v4

| $M$ | $N$ | FLE | QRD | IQN | KLD | ENE | PDF | RBF | TVD | HYV |
|---|---|---|---|---|---|---|---|---|---|---|
| | $2K$ | 16.47 | 30.61 | 28.15 | 16.11 | 15.65 | 19.27 | 20.04 | 35.43 | 16.65 |
| 10 | $5K$ | 12.88 | 28.69 | 27.79 | 14.16 | 13.99 | 17.56 | 17.3 | 35.44 | 13.75 |
| | $10K$ | 12.94 | 28.97 | 34.49 | 13.01 | 15.23 | 17.04 | 17.01 | 35.42 | 13.94 |
| | $2K$ | 15.29 | 20.12 | 29.65 | 17.63 | 17.02 | 18.45 | 18.9 | 35.43 | 16.91 |
| 100 | $5K$ | 13.21 | 17.4 | 25.74 | 14.44 | 13.07 | 15.42 | 14.12 | 35.43 | 14.17 |
| | $10K$ | 19.26 | 17.35 | 33.79 | 14.46 | 12.57 | 15.89 | 16.2 | 35.42 | 14.24 |
| | $2K$ | 16.47 | 18.94 | 28.05 | 17.57 | 16.25 | 18.12 | 18.14 | 35.43 | NA |
| 200 | $5K$ | 13.48 | 16.46 | 32.63 | 12.04 | 14.91 | 14.96 | 15.53 | 35.43 | NA |
| | $10K$ | 23.39 | 14.49 | 33.97 | 14.49 | 13.34 | 15.79 | 15.63 | 35.42 | NA |

Table 23: Reward-variance=1, Eps=Small, Game=KungFuMasterNoFrameskip-v4

| $M$ | $N$ | FLE | QRD | IQN | KLD | ENE | PDF | RBF | TVD | HYV |
|---|---|---|---|---|---|---|---|---|---|---|
| | $2K$ | 5.28 | 13.84 | 14.28 | 4.71 | 5.38 | 6.15 | 5.67 | 16.35 | 4.47 |
| 10 | $5K$ | 5.18 | 13.79 | 15.45 | 4.53 | 6.39 | 6.74 | 6.99 | 16.21 | 4.91 |
| | $10K$ | 5.47 | 13.67 | 11.28 | 4.08 | 5.7 | 6.41 | 6.78 | 16.26 | 4.91 |
| | $2K$ | 5.06 | 6.52 | 14.1 | 4.27 | 4.64 | 5.45 | 5.17 | 16.36 | 3.94 |
| 100 | $5K$ | 5.04 | 6.82 | 15.51 | 4.39 | 6.12 | 6.36 | 5.82 | 16.23 | 3.86 |
| | $10K$ | 5.57 | 6.61 | 15.33 | 4.4 | 5.02 | 5.61 | 5.95 | 16.25 | 3.8 |
| | $2K$ | 4.43 | 5.73 | 15.26 | 3.51 | 4.7 | 5.17 | 5.2 | 16.37 | NA |
| 200 | $5K$ | 5.29 | 5.44 | 14.88 | 5.34 | 5.33 | 6.4 | 6.19 | 16.23 | NA |
| | $10K$ | 7.53 | 5.85 | 15.38 | 4.3 | 4.99 | 5.71 | 5.54 | 16.24 | NA |

Table 24: Reward-variance=1, Eps=Small, Game=PongNoFrameskip-v4

| $M$ | $N$ | FLE | QRD | IQN | KLD | ENE | PDF | RBF | TVD | HYV |
|---|---|---|---|---|---|---|---|---|---|---|
| | $2K$ | 2.07 | 4.4 | 3.98 | 1.57 | 1.62 | 1.54 | 1.58 | 5.82 | 1.66 |
| 10 | $5K$ | 2.23 | 4.91 | 5.4 | 1.84 | 1.87 | 1.71 | 1.64 | 5.94 | 1.87 |
| | $10K$ | 2.29 | 4.97 | 4.01 | 1.83 | 1.89 | 1.65 | 1.74 | 5.98 | 1.99 |
| | $2K$ | 1.85 | 2.4 | 3.32 | 1.63 | 1.59 | 1.3 | 1.36 | 5.82 | 1.67 |
| 100 | $5K$ | 2.11 | 2.52 | 3.16 | 1.87 | 1.8 | 1.38 | 1.4 | 5.93 | 1.94 |
| | $10K$ | 2.74 | 2.8 | 3.48 | 1.97 | 1.84 | 1.46 | 1.56 | 5.98 | 1.74 |
| | $2K$ | 1.75 | 1.64 | 3.03 | 1.64 | 1.54 | 1.39 | 1.35 | 5.82 | NA |
| 200 | $5K$ | 2.35 | 1.78 | 3.39 | 1.79 | 1.86 | 1.38 | 1.4 | 5.93 | NA |
| | $10K$ | 2.2 | 2.06 | 4.84 | 1.86 | 1.8 | 1.59 | 1.46 | 5.99 | NA |

Table 25: Reward-variance=1, Eps=Small, Game=QbertNoFrameskip-v4

| $M$ | $N$ | FLE | QRD | IQN | KLD | ENE | PDF | RBF | TVD | HYV |
|---|---|---|---|---|---|---|---|---|---|---|
| 10 | $2K$ | 55.54 | 20.34 | 18.33 | 17.53 | 16.87 | 19.14 | 19.07 | 30.08 | 21.31 |
| | $5K$ | 20.71 | 18.41 | 10.83 | 20.86 | 17.7 | >100 | 19.07 | 30.23 | 22.11 |
| | $10K$ | 20.08 | 23.4 | 29.16 | 19.75 | 17.04 | 19.03 | 18.54 | 30.31 | 20.86 |
| 100 | $2K$ | >100 | 21.99 | 17.19 | 21.47 | 18.71 | 19.38 | 19.61 | >100 | 21.05 |
| | $5K$ | 21.43 | 11.46 | 28.83 | 21.11 | 14.68 | 19.5 | 19.5 | 30.22 | 21.13 |
| | $10K$ | 44.06 | 16.0 | 29.07 | 17.35 | 19.83 | 19.44 | 19.15 | >100 | 20.9 |
| 200 | $2K$ | 20.54 | 21.43 | 24.77 | 21.07 | 17.79 | 19.62 | 19.48 | 30.07 | NA |
| | $5K$ | 20.55 | 18.94 | 24.2 | 21.18 | 17.11 | 19.36 | 19.94 | >100 | NA |
| | $10K$ | 19.93 | 16.07 | 28.88 | 17.77 | 13.27 | 19.67 | 19.39 | 30.3 | NA |

Table 26: Reward-variance=1, Eps=Small, Game=SpaceInvadersNoFrameskip-v4

| $M$ | $N$ | FLE | QRD | IQN | KLD | ENE | PDF | RBF | TVD | HYV |
|---|---|---|---|---|---|---|---|---|---|---|
| 10 | $2K$ | 2.45 | 15.31 | 7.41 | 1.79 | 2.75 | 4.52 | 4.17 | >100 | 1.53 |
| | $5K$ | 2.7 | 15.73 | 10.72 | 2.65 | 3.91 | 4.95 | 5.39 | 19.05 | 2.16 |
| | $10K$ | 2.63 | 15.6 | 11.07 | 2.02 | 2.98 | 4.44 | 4.6 | >100 | 1.56 |
| 100 | $2K$ | 6.26 | 3.75 | 13.67 | 1.88 | 2.38 | 2.79 | 2.98 | 18.93 | 1.54 |
| | $5K$ | 3.1 | 4.07 | 8.68 | 2.39 | 2.81 | 3.91 | 4.18 | 19.05 | 1.68 |
| | $10K$ | 3.89 | 3.94 | 12.68 | 1.77 | 2.93 | 3.05 | 3.2 | 19.2 | 1.5 |
| 200 | $2K$ | 2.44 | 2.62 | 13.69 | 1.86 | 2.46 | 2.66 | 2.83 | 18.93 | NA |
| | $5K$ | 3.53 | 2.86 | 8.45 | 2.27 | 3.4 | 3.44 | 3.77 | >100 | NA |
| | $10K$ | 2.7 | 2.81 | 6.55 | 2.27 | 2.67 | 2.71 | 2.44 | 19.2 | NA |

Table 27: Reward-variance=1, Eps=Big, Game=AtlantisNoFrameskip-v4

| $M$ | $N$ | FLE | QRD | IQN | KLD | ENE | PDF | RBF | TVD | HYV |
|---|---|---|---|---|---|---|---|---|---|---|
| 10 | $2K$ | 8.59 | 15.19 | 15.67 | 8.08 | 9.23 | 10.4 | 10.41 | 16.47 | 7.87 |
| | $5K$ | 8.62 | 15.5 | 14.08 | 8.41 | 9.38 | 10.5 | 10.42 | >100 | 7.95 |
| | $10K$ | 8.47 | 15.57 | 15.11 | 8.35 | 9.41 | 10.62 | 10.54 | 16.81 | 8.23 |
| 100 | $2K$ | 8.02 | 9.79 | 15.67 | 7.99 | 9.06 | 10.31 | 10.17 | 16.47 | 8.11 |
| | $5K$ | 8.85 | 9.95 | 13.83 | 8.42 | 9.06 | 10.38 | 10.33 | 16.63 | 8.57 |
| | $10K$ | 8.49 | 10.13 | 15.36 | 8.24 | 9.06 | 10.49 | 10.64 | 16.8 | 7.76 |
| 200 | $2K$ | 8.29 | 9.0 | 14.87 | 8.04 | 8.86 | 9.82 | 10.08 | 16.47 | NA |
| | $5K$ | 8.99 | 8.29 | 15.71 | 7.82 | 9.09 | 9.95 | 10.09 | 16.63 | NA |
| | $10K$ | 8.54 | 9.32 | 13.2 | 8.18 | 9.28 | 10.29 | 9.96 | 16.8 | NA |

Table 28: Reward-variance=1, Eps=Big, Game=BreakoutNoFrameskip-v4

| $M$ | $N$ | FLE | QRD | IQN | KLD | ENE | PDF | RBF | TVD | HYV |
|---|---|---|---|---|---|---|---|---|---|---|
| 10 | $2K$ | 3.22 | 9.3 | 7.43 | 2.58 | 3.24 | 4.47 | 4.47 | 10.52 | 2.94 |
| | $5K$ | 3.8 | 9.44 | 10.19 | 2.72 | 3.66 | 4.87 | 4.85 | >100 | 3.27 |
| | $10K$ | 3.57 | 9.64 | 10.28 | 3.0 | 3.93 | 4.8 | 4.79 | 10.81 | 3.07 |
| 100 | $2K$ | >100 | 4.23 | 7.17 | 2.44 | 3.39 | 4.13 | 4.15 | 10.52 | 2.79 |
| | $5K$ | 3.49 | 4.67 | 8.32 | 3.32 | 3.68 | 4.27 | 4.71 | 10.73 | 3.07 |
| | $10K$ | 3.98 | 4.97 | 7.5 | 3.21 | 3.65 | 4.89 | 4.77 | 10.83 | 3.21 |
| 200 | $2K$ | 3.73 | 3.55 | 8.33 | 2.45 | 3.25 | 4.13 | 3.65 | 10.52 | NA |
| | $5K$ | 4.11 | 3.98 | 9.21 | 2.66 | 3.67 | 4.28 | 4.22 | 10.72 | NA |
| | $10K$ | 3.58 | 4.27 | 10.09 | 3.22 | 3.75 | 4.54 | 4.44 | 10.82 | NA |

Table 29: Reward-variance=1, Eps=Big, Game=EnduroNoFrameskip-v4

| $M$ | $N$ | FLE | QRD | IQN | KLD | ENE | PDF | RBF | TVD | HYV |
|---|---|---|---|---|---|---|---|---|---|---|
| 10 | $2K$ | 32.27 | 34.71 | 32.87 | 32.4 | 32.41 | 32.06 | 32.08 | 35.28 | 32.38 |
| | $5K$ | 31.56 | 34.54 | 34.43 | 31.49 | 31.36 | 31.01 | 30.99 | 35.24 | 31.42 |
| | $10K$ | 31.51 | 34.7 | 34.37 | 31.46 | 31.41 | 31.18 | 31.1 | 35.39 | 31.46 |
| 100 | $2K$ | 32.27 | 33.53 | 32.97 | 32.33 | 32.29 | 31.98 | 32.01 | 35.26 | 32.32 |
| | $5K$ | 31.47 | 33.19 | 32.92 | 31.32 | 31.37 | 30.72 | 30.86 | 35.24 | 31.38 |
| | $10K$ | 31.65 | 33.67 | 35.27 | 31.52 | 31.56 | 31.18 | 31.07 | 35.39 | 31.59 |
| 200 | $2K$ | 32.19 | 33.08 | 34.61 | 32.37 | 32.28 | 32.0 | 32.06 | 35.27 | NA |
| | $5K$ | 31.37 | 32.5 | 34.32 | 31.13 | 31.32 | 32.18 | 30.89 | 35.24 | NA |
| | $10K$ | 31.52 | 32.86 | 35.26 | 31.46 | 31.47 | 31.12 | 31.01 | 35.38 | NA |

Table 30: Reward-variance=1, Eps=Big, Game=KungFuMasterNoFrameskip-v4

| $M$ | $N$ | FLE | QRD | IQN | KLD | ENE | PDF | RBF | TVD | HYV |
|---|---|---|---|---|---|---|---|---|---|---|
| 10 | $2K$ | 12.89 | 15.41 | 14.16 | 12.92 | 13.21 | 13.4 | 13.69 | 16.17 | 12.75 |
| | $5K$ | 12.76 | 15.31 | 13.43 | 12.53 | 11.82 | 13.05 | 13.13 | 16.08 | 12.63 |
| | $10K$ | 12.86 | 15.4 | 15.97 | 12.54 | 12.97 | 13.14 | 13.12 | 16.06 | 12.26 |
| 100 | $2K$ | 13.05 | 14.31 | 15.68 | 12.89 | 13.15 | 13.16 | 13.08 | 16.16 | 12.77 |
| | $5K$ | 12.52 | 14.06 | 15.89 | 12.6 | 12.95 | 13.0 | 12.98 | 16.07 | 12.67 |
| | $10K$ | 12.92 | 14.05 | 15.99 | 12.67 | 12.82 | 12.92 | 12.9 | 16.06 | 12.35 |
| 200 | $2K$ | 12.8 | 13.64 | 15.63 | 12.85 | 13.06 | 13.16 | 13.21 | 16.17 | NA |
| | $5K$ | 12.94 | 13.37 | 15.89 | 12.55 | 12.96 | 13.06 | 12.93 | 16.07 | NA |
| | $10K$ | 13.17 | 13.36 | 15.75 | 12.59 | 12.84 | 13.08 | 13.0 | 16.06 | NA |

Table 31: Reward-variance=1, Eps=Big, Game=PongNoFrameskip-v4

| $M$ | $N$ | FLE | QRD | IQN | KLD | ENE | PDF | RBF | TVD | HYV |
|---|---|---|---|---|---|---|---|---|---|---|
| 10 | $2K$ | 3.26 | 4.99 | 3.71 | 3.08 | 3.07 | 2.8 | 2.86 | 5.94 | 3.15 |
| | $5K$ | 3.98 | 5.17 | 5.95 | 3.9 | 3.72 | 3.7 | 3.66 | 5.92 | 3.82 |
| | $10K$ | 4.25 | 5.24 | 12.39 | 4.03 | 3.9 | 3.83 | 3.82 | 5.97 | 4.01 |
| 100 | $2K$ | 3.38 | 3.44 | 4.14 | 3.1 | 3.01 | 2.31 | 2.44 | 5.95 | 3.27 |
| | $5K$ | 4.9 | 4.1 | 3.79 | 3.87 | 3.73 | 3.14 | 3.12 | 5.92 | 3.68 |
| | $10K$ | 4.17 | 4.16 | 5.52 | 3.9 | 3.88 | 3.39 | 3.23 | 5.98 | 3.84 |
| 200 | $2K$ | 3.42 | 2.91 | 1.73 | 3.06 | 2.98 | 2.36 | 2.37 | 5.94 | NA |
| | $5K$ | 3.91 | 3.67 | 4.7 | 3.64 | 3.74 | 3.1 | 3.19 | 5.92 | NA |
| | $10K$ | 4.29 | 3.68 | 5.33 | 4.0 | 3.85 | 3.34 | 3.36 | 5.98 | NA |

Table 32: Reward-variance=1, Eps=Big, Game=QbertNoFrameskip-v4

| $M$ | $N$ | FLE | QRD | IQN | KLD | ENE | PDF | RBF | TVD | HYV |
|---|---|---|---|---|---|---|---|---|---|---|
| 10 | $2K$ | 20.91 | 28.59 | 29.96 | 19.67 | 21.92 | 22.96 | 23.01 | 30.16 | 20.76 |
| | $5K$ | 20.15 | 28.36 | 29.58 | 18.3 | 20.7 | 21.67 | 21.93 | 30.11 | 19.52 |
| | $10K$ | 19.81 | 28.37 | 29.8 | 19.19 | 20.87 | 21.84 | 21.81 | 30.24 | 18.97 |
| 100 | $2K$ | 20.77 | 22.46 | 29.32 | 19.01 | 21.59 | 22.53 | 22.69 | 30.17 | 20.81 |
| | $5K$ | 20.26 | 21.62 | 29.26 | 19.7 | 20.14 | 21.3 | 21.35 | 30.11 | 19.38 |
| | $10K$ | 19.68 | 21.31 | 28.75 | 19.83 | 20.11 | 20.8 | 20.65 | >100 | 18.92 |
| 200 | $2K$ | 21.34 | 20.79 | 29.01 | 20.68 | 21.54 | 22.19 | 22.05 | 30.16 | NA |
| | $5K$ | 19.63 | 20.54 | 29.7 | 19.52 | 20.04 | 20.73 | 21.01 | 30.11 | NA |
| | $10K$ | 19.67 | 20.24 | 27.25 | 19.54 | 20.26 | 20.19 | 20.52 | 30.23 | NA |

Table 33: Reward-variance=1, Eps=Big, Game=SpaceInvadersNoFrameskip-v4

| $M$ | $N$ | FLE | QRD | IQN | KLD | ENE | PDF | RBF | TVD | HYV |
|-----|-----|-----|-----|-----|-----|-----|-----|-----|-----|-----|
| 10 | $2K$ | 6.15 | 16.76 | 11.54 | 5.91 | 7.54 | 9.86 | 9.78 | 18.89 | 5.33 |
|  | $5K$ | 6.76 | 16.75 | 9.79 | 5.61 | 7.38 | 9.4 | 9.62 | 18.91 | 5.02 |
|  | $10K$ | 5.1 | 16.81 | 16.36 | 5.81 | 6.92 | 9.34 | 9.21 | 19.06 | 5.41 |
| 100 | $2K$ | 6.35 | 7.91 | 12.26 | 5.56 | 6.25 | 8.23 | 7.71 | 18.88 | 5.98 |
|  | $5K$ | 6.55 | 7.94 | 16.3 | 5.57 | 6.61 | 7.59 | 7.86 | 18.9 | 5.14 |
|  | $10K$ | 6.14 | 7.67 | 12.07 | 4.86 | 6.07 | 7.3 | 7.18 | 19.05 | 5.41 |
| 200 | $2K$ | 6.21 | 6.8 | 10.59 | 5.4 | 6.95 | 7.87 | 7.43 | 18.88 | NA |
|  | $5K$ | 6.2 | 6.87 | 13.29 | 5.31 | 6.38 | 7.1 | 7.14 | 18.91 | NA |
|  | $10K$ | 5.83 | 6.43 | 10.75 | 4.86 | 6.33 | 6.95 | 6.78 | 19.05 | NA |

## E.3 Computation resources

Our simulations are extensive. We used high performance computing system. For LQR simulation (Appendix E.1), we have used CPU, 2GB memory. A single run of a single method under a fixed setting takes approximately 20 seconds. For Atari games' simulation (Appendix E.2), we have used GPU, 10GB memory. A single run of a single method under a fixed setting takes approximately 700 seconds.

