# OpenReview forum: "A Principled Path to Fitted Distributional Evaluation"
_NeurIPS.cc/2025/Conference — NeurIPS 2025 spotlight_

### Official Review · Reviewer_vbgr · 2025-06-29

**Clarity:** 2
**Significance:** 2
**Originality:** 2
**Rating:** 4
**Confidence:** 4

**Summary:**

The paper proposes Fitted Distributional Evaluation (FDE), a principled extension of fitted‑Q evaluation to distributional off‑policy evaluation.
First, they choose a contractive metric for the distributional Bellman operator (Section 2.3).
Second they optimize a matching functional Bregman divergence (Section 2.4).
Then they) prove that, under this metric‑loss link, the Bellman backup is the population minimiser (Theorem 2.4).
Statistically, Theorem 3.3 yields near‑minimax sample‑complexity in the tabular case, while Tables 1 and 2 list many valid metric/divergence pairs—including new ones such as PDF‑L2 and MMD‑Coulomb—and extend guarantees to continuous state–action spaces (Theorems C.4/C.11). There are also some synthetic experiments on a linear‑quadratic regulator (LQR) and seven Atari games show FDE variants (KL, Energy, PDF‑L2, RBF) reducing mean Wasserstein‑1 error compared with FLE, QR‑DQN, IQN, and a TV‑distance baseline (Figure 1).

**Questions:**

1. What happens when Assumption 3.1 fails? Can you provide a bound with an inherent Bellman‑approximation term or show an empirical study?
2. Could uniform‑stability or mixing arguments allow reuse of the full dataset instead of the per‑iteration split?
3. Which theorem, beyond Corollary B.9, ensures convergence for KL, given that Section 2.4 says the main framework excludes it?

**Ethical Concerns:**

["NO or VERY MINOR ethics concerns only"]

**Final Justification:**

The paper proposes a unified framework for distributional FQE which yields some improvements over prior FQE in theory and practice. The analysis appears fairly standard but should be valuable to the community nonetheless.

**Limitations:**

The authors acknowledge strong assumptions (completeness, data splitting, contractive metrics). Experiments also omit high‑dimensional continuous‑control domains.

**Quality:**

2

**Strengths And Weaknesses:**

## Strengths:
1. The three‑principle recipe clearly explains why certain popular distances (for example, total variation) fail and how to build safe objectives.
2. The finite‑sample analysis is tight: Theorem 3.3 matches known lower bounds for W1 in tabular MDPs and the appendix extends the result to non‑tabular settings.
3. Empirical results, though compact, are consistent with the theory and illustrate the failure of TV distance.

## Weaknesses:
1. Assumption 3.1 (completeness) is strong and rarely satisfied in deep RL; the analysis also relies on splitting the dataset across iterations (Equation 10), which hurts sample efficiency.
2. Section 2.4 states that the metric‑loss link does not accommodate KL divergence, yet KL is used in experiments; only Corollary B.9 in the appendix gives a guarantee, so the main text feels incomplete.
3. Experiments cover only one continuous‑control task and seven Atari games, with no statistical tests or hyper‑parameter ablations.
4. Dense notation and appendix‑only proofs (for example, the entire proof of Theorem 3.3) make the paper hard to digest. Especially since the appendix is more than 50 pages. I would suggest condensing the paper significantly and providing proof sketches in the main text.

---

> ### Author Rebuttal · Authors · 2025-07-31
>
> Thank you very much for your efforts. Please find our responses to your comments below.
>
> ## Weakness-1, Question-1,2: Relaxing the strong assumptions
>
> We acknowledge that both completeness (Assumption 3.1) and data splitting are limitations of the current work. While these are substantial directions that cannot be handled by the current work, we agree these limitations are worthy to be addressed. However, it is also true that these are standard limitations of many iteration-based RL algorithms. Many papers assume completeness (e.g., papers listed in Lines 247-248) and adopt similar strategies to data splitting or equivalently fresh samples in each iteration (e.g., [1], [2]) for theoretical convenience.
>
> Answer to your Question 1: When Assumption 3.1 (Bellman completeness) fails, we will have a non-negligible approximation error, similar to FQE,  at each iteration of our proposed FDE algorithm. Such an approximation error can be quantified by inherent distributional Bellman error, which can be similarly defined by adapting from the standard notion in FQE. See [3] and [4] for more details.
>
> Answer to your Question 2: Thanks for your suggestions. Due to the general choice of functional Bregman divergences, conducting a systematic study of the dependence across iterations—and thereby understanding the mixing properties—is nontrivial. To sidestep the complexities of this dependence, one possible alternative is to explore uniform convergence. However, we have not pursued these directions rigorously and therefore prefer not to make any speculative claims.
>
> ## Weakness-2, Question-3: KL divergence can be accommodated in Section 2.4.
>
> We believe there is misunderstanding. KL divergence is a functional Bregman divergence, and so Theorem 2.4 of Section 2.4 covers KL divergence. See our discussion in Lines 212-214. Therefore, KL divergence is included in our main framework of FDE. However, when it comes to obtaining statistical error bound in Section 3, we need a separate theoretical analysis for KL divergence. We proved the convergence of our FDE based on KL divergence in Corollary B.9.
>
> ## Weakness-3: Statistical tests and hyperparameter ablations
>
> We have provided standard deviations (Atari games in Figures 1, 3-5) and error bars (LQR simulations in Figure 2) for the numerical results that allow assess for statistical significance. As for hyperparameter ablations, we indeed have varied the model complexity (the number of Gaussian mixture components as detailed in Appendix E.2.1.). For the experimental setups, we have included two levels of epsilon values for behavior policies (representing weak and strong satisfaction of Assumption C.3) and two different transitions (random and deterministic). Since we evaluated 7 Atari games, our experimental setup therefore includes $2$ (behavior policies) $\times$ $2$ (transitions) $\times$ $7$ (Atari games) $=28$ configurations. Each was tested with three levels of model complexity ($M=10,100,200$; see Lines 1490-1491), resulting in a substantial number of experiments. The outcomes are visualized in Figures 3-5. We believe this comprehensive evaluation sufficiently demonstrates our main conclusion that the methods developed within our framework are working.
>
> ## Weakness-4: Proofs are all in Appendix.
>
> Thank you for your suggestion. Based on your comment, we will add proof sketch/outline. Moreover, we will add a table of content in the supplementary material for easier navigation of the supplementary material.
>
> ## References
>
> [1] Peng et al. Statistical efficiency of distributional temporal difference learning. NeurlIPS. 2024.
>
> [2] Wu et al. Distributional offline policy evaluation with predictive error guarantees. ICML. 2023.
>
> [3] Duan et al. Risk bounds and rademacher complexity in batch reinforcement learning. ICML. 2021.
>
> [4] Golowich et al. The role of inherent bellman error in offline reinforcement learning with linear function approximation. arXiv preprint. 2024.

---

### Official Review · Reviewer_NswW · 2025-07-01

**Clarity:** 1
**Significance:** 3
**Originality:** 2
**Rating:** 4
**Confidence:** 4

**Summary:**

This paper designs a framework for specifying offline distributional
policy evaluation algorithms: algorithms that estimate the distribution
over returns achieved by a given policy based on a fixed dataset.
Particularly, the paper demonstrates convergence results of their fitted
distributional policy evaluation meta-algorithm for a very broad family
of objective / loss functions. The paper concludes with a demonstration
of such methods evaluating return distributions in Atari, where the
proposed methods achieve accurate return distribution predictions as
measured by a Wasserstein metric.

**Questions:**

I split this section into two parts, namely "critical questions" and
"ancillary questions". The former includes questions which may
directly influence my score, and the latter includes questions for
clarification / curiosity.


## Critical Questions

1.  I am having trouble understanding what $m$ is in section 3.1, or at
    least what the IPS is that is being referred to. Outside of the MMD
    (where the IPS is obvious), what are examples of these IPS? Can you
    give explicit examples for e.g. KL divergence, or scoring rules
    that you outlined earlier?
2.  In Assumption 3.2, what is $\eta$? I'm assuming it's some metric
    over $\mathcal{P}$, but how does this relate to, say, $\mathfrak{d}$?
3.  In the proof of Theorem 3.3, how did you arrive at equation (33)
    from the line preceding it? How do you neglect the $\bar{m}_{n, 1}$
    term?
4.  In the experimental section, it is not entirely clear how you're
    measuring "inaccuracy". Upon reading the appendix, it says it is
    averaging 1-Wasserstein distance estimates over 100 pre-sampled
    state-action pairs. How did you sample these state-action pairs?


## Ancillary Questions

1.  In your definition of expectation extension of metrics, why do you
    prohibit the "$L^1$ version"? In other words, why are all exponents
    $2q$ and not $q$?

**Ethical Concerns:**

["NO or VERY MINOR ethics concerns only"]

**Final Justification:**

The authors' responses were satisfactory. I still maintain that the overall message could be strengthened and the motivation for the work is not especially strong, but I do appreciate the rigor and I think the distributional RL community are likely to find some of the results useful.

**Limitations:**

Limitations are addressed adequately.

**Paper Formatting Concerns:**

No major formatting issues.

**Quality:**

3

**Strengths And Weaknesses:**

## Strengths

This paper takes a really deep dive into the convergence of fitted
distributional policy evaluation methods. The results extend to a
truly vast family of metric / loss functions, and analysis ranges from
tabular to continuous state-action spaces. Additionally, the paper
treats the multidimensional reward setting, which has shown to be
challenging particularly in distributional RL.
The paper, while largely theoretical in nature, is nicely supplemented
with nontrivial experiments.

I also really appreciated the sections in the appendix that explicitly
state the convergence rates for a wide variety of divergences explicitly.


## Weaknesses

Aside from difficulties following the math due primarily to unclear
notation and undefined terms, the main criticism I have for this paper
is that the motivation is not very strong.
This paper seems very clearly parallel to [1], effectively justifying
the use of alternative loss functions for their algorithm. It's not
abundantly clear what this gets you. Reading over the paper, it looks
like:

1.  [1] is restricted to reward functions with bounded range, unlike
    this paper.
2.  The convergence rates derived in this work are favorable to those
    of [1] for certain divergences.
3.  As per the introduction, "A general guideline for selecting
    appropriate discrepancy functions in FDE remains unclear, leaving
    practitioners without systematic guidance for developing valid FQE
    extensions tailored to their specific needs". This feels fairly
    unrelatable. Besides the points mentioned above, what are other
    reasons why practitioners might feel inclined to deviate from [1]?

To be clear, I don't mean to suggest that this work has no value in
light of [1]; rather, I'm suggesting that the narrative could be
improved to make the value more clear.

Moreover, while the paper is already dense and page limits exist, I
still believe this paper deferred *a lot* of material to the
appendix. It would really be a stretch for a reader to understand the
paper (even just the theorem *statements* and experiments) without
reading the appendix.

I mention in more detail some of the specific issues I had with
following certain parts of the proofs / math throughout the remainder of the review.


## Minor Issues

1.  On line 85, it says "the Bellman operator is often unknown and so
    **as** the evaluation of the RHS (1), rendering this approach
    impractical". There's a few issues here:
    1.  There is a problem in the sentence at the "**as**", the sentence is
        broken.
    2.  $(\mathcal{B}^\pi)^\infty Q_0$ doesn't really make sense. This
        would be impossible to compute even if you know
        $\mathcal{B}^\pi$ exactly and had an oracle for evaluating
        it. The statement should really be $(\mathcal{B}^\pi)^k Q_0 \to Q_\pi$.
    3.  When $\mathcal{S}$ or $\mathcal{A}$ are infinite, even with
        perfect knowledge of the MDP parameters, $\mathcal{B}^\pi$ can
        be intractable to compute.
2.  On line 94, you define $(g\_{r,
       \gamma})\_\\#:\mathcal{P}\to\mathcal{P}$ as the push-forward measure
    of the function $g_{r, \gamma}$. I've never seen the pushforward
    defined this way before (e.g., defining the map $f_\sharp$ on
    measures), it's actually pretty clever. But then
    $(g\_{r,\gamma})\_\\#$ is technically not the push-forward measure
    (rather, $(g\_{r, \gamma})\_\\#\Upsilon(s, a)$ is). Maybe "push-forward map" instead?
3.  In the discussion starting on line 110, perhaps in Remark 2.1, the
    work of [1] should be mentioned, it is indeed highly related to
    what you're proposing here. You did cite this work elsewhere, but
    it would help contextualize in this section how your proposal differs
    from theirs. For instance, you compare to a method in Remark 2.1
    that you claim is limited to tabular MDPs, but the method of [1]
    has no such limitation.
4.  The notation $\tilde{\eta}$ for a metric on
    $\mathcal{P}^{\mathcal{S}\times\mathcal{A}}$ is really confusing,
    since $\eta$ is usually used to denote return distribution
    functions (see e.g. 2, 3).
5.  The term "contractive metric" is strange. It's not the metric
    that's contractive, it's the operator.
6.  In the discussion about MMD metrics in distributional RL on line
    213, the work of [4] discusses a broader family of MMD kernels that
    lead to contractive Bellman operators for multidimensional rewards.
7.  In Figure 1, the axis labels / tick labels are much too small.


## References

1.  Wu, Uehara & Sun (2023) Distributional Offline Policy Evaluation with Predictive Error Guarantees, ICML.
2.  Bellemare, Dabney & Munos (2017) A Distributional Perspective on Reinforcement Learning, ICML.
3.  Bellemare, Dabney & Rowland (2023) Distributional Reinforcement Learning, The MIT Press.
4.  Wiltzer, Farebrother & Gretton et al. (2024) Foundations of Multivariate Distributional Reinforcement Learning, NeurIPS.

---

> ### Author Rebuttal · Authors · 2025-07-31
>
> Thank you very much for your efforts. Please find our responses to your comments below.
>
> ## Weakness-1: Motivation; Value in light of FLE [1]
>
> Thank you for your question. As emphasized in Section 2.1, a central goal of this work is to explore how to choose $\mathfrak{d}$ in order to develop a theoretically sound FDE method. Note that [1] focuses exclusively on the log-likelihood criterion (which is closely related to KL divergence), and it \textit{does not provide direct insight} into this crucial question. To address it, we propose a theoretically motivated requirement for $\mathfrak{d}$ and show that functional Bregman divergences naturally satisfy this requirement, making them compelling candidates (see Theorem 2.4).
>
> At a high level, different divergences respond differently to deviation between $\Upsilon$ and $\Psi_\pi$ (see Equation (5)), much like how various loss functions behave differently in empirical risk minimization. Therefore, the choice of divergence should be tailored to the specific problem at hand. Relying on a single divergence (such as KL) across all settings is unlikely to be optimal. In fact, there have been well-known criticism of KL divergence (e.g., unbounded divergence value, high sensitivity to the tails of distributions, necessity that one probability measure is absolutely continuous to the other, inability to consider closeness in outcome values), as discussed in [3], [4], [5]. Even in general statistical estimations, these limitations have motivated extensive research into alternative divergences beyond KL divergence [2].
>
> To move toward a principled framework for choosing $\mathfrak{d}$ in specific applications, we believe the first step is to identify what constitutes a valid candidate. To date, this question remains largely unaddressed in the literature, and only a few valid choices have been identified - let alone systematically compared. Our work makes substantial progress on this front. While comparative analyses of certain functional Bregman divergences do exist and may provide practical guidance, we acknowledge that our paper does not yet offer a comprehensive decision-making framework for selecting among them in practice. We believe further investigation is needed to better understand the trade-offs between different functional Bregman divergences in the context of distributional OPE. Besides, identifying less common functional Bregman divergences that may be especially useful for distributional OPE is also interesting (e.g., Hyvarinen divergence which performed well in Atari games; see Tables 5-32). We believe they are important directions for future research. We will revise the paper to more clearly communicate these motivations and contributions.
>
> ## Weakness-2: Many materials in the appendix.
>
> Other reviewers have made some suggestions to improve the readability. We will revise the paper to address these comments. We plan to add more details (e.g., adding Theorem C.4 as suggested by Reviewer HoeH) to the main text. We also plan to add proof sketch/outline (as suggested by Reviewer vbgr). Besides, we will add a table of content in the supplementary material for easier navigation of the supplementary material.
>
> ## Minor issues
>
> Thank you very much for your suggestions and careful reading. Due to space limitation, we do not respond to them one by one. We will address these issues in the revised version.
>
> ## Question-1: Examples of IPS metrics
>
> In Section 3.1, the study is restricted to the case when the functional Bregman divergence is given as $\mathfrak{d}=m^2$ for some IPS metric $m$ (i.e., there exists an IPS metric such that the divergence is the squared IPS metric), and Table 1 presents examples of the functional Bregman divergence that can be written as a squared form of an IPS metric (Definition C.13). We have listed three examples of IPS metric in Appendix C.5, along with their corresponding inner products. Note that KL divergence is not a squared IPS metric and so is not included in Table 1, and the result of Section 3.1 (Theorem 3.3) does not apply to KL divergence. We have proved the convergence of FDE based on KL divergence separately in other results (Corollary B.9). (See Lines 317-318.)
>
> ## Question-2: Relationship between $\eta$ and $\mathfrak{d}$
>
> $\eta$ is a probability metric over $\mathcal{P}$, whose extension (either supremum-extension or expectation-extension of Equations 7 and 8) ensures that Bellman operator is a contraction (Definition 2.2). In our paper, we bound the inaccuracy of our estimation only based on the metric $\eta$ that satisfies Definition 2.2 (see Theorems 3.3, C.4, C.11), by using an $\mathfrak{d}$-based objective function (Equation 5). To successfully achieve this, we need a surrogate metric $m$ that links $\mathfrak{d}$ and $\eta$, which satisfies a series of properties (Assumption C.2 for Theorem C.4, Assumption C.9 for Theorem C.11) along with Assumption 3.2. Note that $m$ is used for quantifying the model complexity (see NS4 of Assumption C.2 and MS4 of Assumption C.9). One special case is using the divergence that has the form of $\mathfrak{d}=m^2$ with $m$ being an IPS metric (Definition C.13), but our results are not limited to squared IPS metics (as mentioned in Lines 310-311, 805).
>
> We remark that the divergence in Theorem C.11 requires Assumption C.9, not Assumption C.2. We will fix this typo.
>
> ## Question-3: Clarification of a proof step
>
> Thank you for your careful reading. In the line before Equation 33, there was a typo. $\bar{m}_{n,1}$ should be replaced by its squared term. Then this solves the issue. As we mentioned in Line 1186, we can follow the logic of the proof for (14) to show the inequality. We will correct it in the revised version.
>
> Before we procced, we would like to apologize for the typos listed above, which have caused unnecessary confusions. We will proofread the revised version carefully.
>
> ## Question-4: Clarification of inaccuracy measure
>
> We sampled 1000 state-action pairs from $d_\pi$ (Line 1509) according to Algorithm 1 of [6] (which we will clarify in the revised version). Since sampling from $d_\pi$ takes a long time for large $\gamma$, we saved the samples for each game (namely, pre-sampled observations) so that we can re-use them whenever we measure inaccuracy. To further clarify, we are not averaging Wasserstein-1 metric values for each pre-sampled observations of state-action pairs. Instead, we are calculating a single Wasserstein-1 metric value between the $S,A$-marginalized distributions in Line 1507 (which is also used by [1]). If necessary, we shall elaborate the algorithmic details of calculating this quantity.
>
> ## Ancillary Question 1: Can we adopt $L_1$ form of expectation-extension?
>
> Note that bounding $L_2$ version of expectation-extended metric (in the results of Theorems C.4, C.11) already implies bounding $L_1$ version of expectation-extended metric, by the fact that $L_1$ norm is bounded by $L_2$ norm for probability spaces.
>
> But, as you mentioned, there is no need to exclude $L_1$ version of expectation-extension, which can also make the Bellman operator a contraction by the same proof structure based on Appendix C.3.1.
>
> ## References
>
> [1] Wu et al. Distributional offline policy evaluation with predictive error guarantees. ICML. 2023.
>
> [2] Gneiting et al. Strictly proper scoring rules, prediction, and estimation. JASA. 2007.
>
> [3] Korba et al. Statistical and Geometrical properties of the Kernel Kullback-Leibler divergence. NeurlIPS. 2024.
>
> [4] Selten. Axiomatic characterization of the quadratic scoring rule. Experimental Economics 1.1. 1998.
>
> [5] Bellemare et al. The Cramer distance as a solution to biased wasserstein gradients. arXiv preprint. 2017.
>
> [6] Agarwal et al. On the theory of policy gradient methods: Optimality, approximation, and distribution shift. JMLR. 2021.

---

### Official Review · Reviewer_HoeH · 2025-07-02

**Clarity:** 4
**Significance:** 3
**Originality:** 3
**Rating:** 5
**Confidence:** 3

**Summary:**

This paper focuses on the fitted distributional evaluation setting in distributional RL. They propose general approaches, fitted distributional evaluation, which aim to be a class of evaluation method and thus unify many existing approahces. Convergece guarantees are provided under different specification of the proposed framework, also ranging from tabular to non-tabular. Extensive experiments are conducted to support their theoretical claims.

**Questions:**

Does the theory work in undiscounted finite horizon setting? My take is that discounting has been important in the analysis because contraction is heavily used.

**Ethical Concerns:**

["NO or VERY MINOR ethics concerns only"]

**Final Justification:**

I am keeping my positive score.

**Limitations:**

Limitations are discussed in the last section.

**Quality:**

4

**Strengths And Weaknesses:**

This paper looks at distributional evaluation from a very general perspective. It abstracts the discrepancy in Bellman error into Bregman divergences. By doing this it opens up a new class of algorithms for distributional evaluation.

Regarding theory, I have not checked the proofs (since the paper comes with extremely long appendices), but all results intuitively make sense. One suggestion is to move some more general results back to the main contents. While Theorem 3.3 might be a bit straightforward as it might be based on point-wise concentration and a union bound since visitation lower bound $p_{min}$ is assumed, which is quite strong (apologies if I am wrong), it is dominated by the more off-policy Theorem C.4 in the appendix, which I find more  interesting. Hence I would suggest moving Theorem C.4 to the main content at least. Right now it looks like many interesting results are all in the appendix.

Experiments support the theory by showing that  the proposed method performs well. One improvement could be to evaluate on multimodal distributions. Atari games are known have a single mode in return, which is a very special class of distributions though being common.

---

> ### Author Rebuttal · Authors · 2025-07-31
>
> Thank you very much for your efforts. Below are our responses to the comments you provided.
>
> ## Weakness-1: Moving general results to the main text.
>
> Thank you for your suggestion. Due to the page limit, we summarize our result in Equation 12, which accommodates other statements (Theorems C.4, C.11).  Based on your comment, we plan to add more details (e.g., Theorem C.4) to the main text.
>
> However, we would like to clarify two points. First, $p_{min}>0$ is not a strong assumption for the error bound guarantee in sup-norm metric (more specificially, $\eta_\infty$) in Theorem 3.3, since it is a strong uniform guarantee on every single state-action pair. For finitely many state action pairs (i.e., tabular setting in Theorem 3.3), $p_{min}>0$ is satisfied as long as every state-action pair has positive probability to be observed, as also assumed in [1], [2], [3]. Second, Theorem C.4 does not dominate Theorem 3.3. Although Theorem C.4 can accommodate all the functional Bregman divergences that can be applied in Theorem 3.3 (i.e., squared IPS metrics introduced in Section 3.1), they demonstrate slower convergence rates. This is explained in Lines 315-316. Indeed, Theorem 3.3 is significant as this shows near-optimal convergence rate for several divergences in tabular setting. Prior to our work, there is only one existing work [1] that shows optimal convergence rate (in tabular setting). See discussion after Theorem 3.3.
>
> ## Weakness-2: Multi-modal returns
>
> Note the return distributions depend on the choices of target policies. Moreover, we have also included experiments with modified environments with stochastic transitions (in addition to the original deterministic transitions), which obviously affect the shape of return distributions. Empirically, we observed examples that generate (marginal) return distributions that have multiple modes in our Atari game experiments.
>
> ## Question-1: Finite horizontal setting
>
> Our statistical error bounds (Theorems 3.3, C.4, C.11) do not apply to the finite horizontal settings. For finite horizontal setting, the algorithm will need to be modified. See, e.g., Section 2.1 of [4]. Note that there would be a set of return distributions defined over time, which are characterized by a series of distributional Bellman equations (which is different from that of the infinite horizon). And such relationship would be used for the finite horizontal extension, instead of the fixed-point characterization in the infinite horizontal setup.
>
> Yet, we believe that a key part of work still provides insight for the construction of such distributional extension of the FQE.
> Based on a similar argument for (P2) and in Section 2.4, functional Bregman divergences are still expected to be natural candidates for finite-horizon extension.
>
> ## References
>
> [1] Peng et al. Statistical efficiency of distributional temporal difference learning. NeurlIPS. 2024.
>
> [2] Rowland et al. An analysis of quantile temporal-difference learning. JMLR. 2024.
>
> [3] Ma et al. Conservative offline distributional reinforcement learning. NeurlIPS. 2021.
>
> [4] Wu et al. Distributional offline policy evaluation with predictive error guarantees. ICML. 2023.

---

> > ### Comment · Area_Chair_8hzp · 2025-08-05
> > **Reviewer HoeH, please respond to the authors**
> >
> > Reviewer HoeH, please respond to the authors

---

> > ### Comment · Reviewer_HoeH · 2025-08-06
> >
> > I appreciate the authors' reply and will keep my positive rating.

---

> > > ### Author Response · Authors · 2025-08-06
> > >
> > > Thank you for your response. We appreciate your time and helpful comments.

---

### Official Review · Reviewer_peoQ · 2025-07-03

**Clarity:** 3
**Significance:** 3
**Originality:** 3
**Rating:** 5
**Confidence:** 2

**Summary:**

The paper addresses the problem of distributional off-policy evaluation, and proposes a framework for designing extensions of fitted-Q evaluation to the distributional setting. Fitted distributional evaluation (FDE) replaces the regression of the scalar Bellman backup with the minimization of a discrepancy measure between a learned conditional return distribution and the distributional Bellman backup. The authors propose a set of guiding principles for choosing a discrepancy and a metric on the space of distributional Q-functions such that the resulting FDE scheme can converge with respect to this metric. The authors provide results on how to extend metrics on real numbers to contractive metrics on distributional Q-functions, in particular for expectation-extended metrics. Furthermore, the class of functional Bregman divergences is shown to be a suitable choice for a discrepancy measure. Bounds are derived for the discrepancies defined as squared metrics in the tabular setting, and for expectation-based metric extensions with general functional Bregman divergences, leading to new FDE variants. Experimental results on a linear quadratic regulator align with the derived convergence rates, whereas inaccuracies on Atari games do not closely follow the theoretical results but tend to outperform the considered baselines.

**Questions:**

Questions:

* Why was a Gaussian mixture model chosen to model the return distribution? This seems like an important design choice but I am missing a justification/discussion.

Typos:

* Line 216: “used” twice.
* Closing bracket missing in line 221.
* Article missing before “corresponding supremum-extended metric” in line 160.

**Ethical Concerns:**

["NO or VERY MINOR ethics concerns only"]

**Final Justification:**

I recommend acceptance of the submission for its principled and comprehensive treatment of distributional off-policy evaluation. The authors responded to the concerns raised by other reviewers and me regarding the high density of the main text and the long appendix which can make navigating the text difficult. The authors outlined planned changes to improve readability and minor issues with the writing.

**Limitations:**

Yes, limitations are addressed in the discussion section.

**Quality:**

3

**Strengths And Weaknesses:**

### Strengths:

* The paper is well-written, and follows a clear structure.

* The claims are well supported by the theoretical results in the paper.

* Distributional off-policy evaluation is a relevant problem for real-world applications, in particular when safety matters. The paper addresses this problem in a principled and comprehensive manner that, to the best of my knowledge, is not available yet.

* The authors offer a good discussion on the limitations of their framework, such as the focus on Bregman divergences, or the completeness assumption for the model class.

### Weaknesses:

* The paper is quite comprehensive to a point where it feels like even a very condensed version of the content barely fits into the main text. That makes it hard to read for someone not familiar with this research area.

* The results on the Atari environments are difficult to interpret as they differ a lot from game to game. It is not entirely clear to me which design choices in implementing the algorithms lead to these results. Some qualitative analysis of the learned return distributions could help form a better understanding. However, I do acknowledge that the main contribution of the paper is clearly not empirical.

* The behavior policy in the Atari experiments probably leads to quite similar behavior as the evaluated policy as they are both epsilon greedy with different epsilon parameters. It would be interesting to additionally consider distinct policies.

* The paper makes several assumptions about coverage, for example in line 255 (total coverage of the state-action space), or in line 296 (coverage of the target distribution). However, also principle (P2) looks like it requires total coverage of the state-action space as it requires the Bellman update to be the unique minimizer of the population objective. A discussion of the role of these coverage assumptions and how they could be relaxed would be interesting from a practical perspective.

---

> ### Author Rebuttal · Authors · 2025-07-31
>
> Thank you very much for your efforts. Below are our responses to your comments.
>
> ## Weakness-1: Too condensed.
>
> We agree that we have a significant amount of content to fit into the paper. Other reviewers have made some suggestions to improve the readability. We will revise the paper to address these comments. We plan to add more details (e.g., adding Theorem C.4 as suggested by Reviewer HoeH) to the main text. We also plan to add proof sketch/outline (as suggested by Reviewer vbgr). Besides, we will add a table of content in the supplementary material for easier navigation of the supplementary material.
>
> ## Weakness-2: Difficulty in interpreting the simulation results of Atari games
>
> In Figure 1, except TVD (not a functional Bregman divergence) whose inaccuracy value stayed still throughout iterations, all other methods have at least shown improvement of accuracy. This demonstrates the validity of our FDE methods that are based on functional Bregman divergences. Furthermore, we can see that our FDE methods generally have lower mean inaccuracy values (dots in Figure 1) and more stable estimation (smaller standard deviation) than the baseline methods (FLE, QRDQN, IQN). Our settings can be categorized into four groups depending on random / deterministic transition and strong / weak coverage (explained in Lines 1542-1554). In the table below, we presented the mean of the rank values (1 being the best and 9 being the worst) of inaccuracies shown in Tables 5-32. We can see that our three FDE methods (KL, Energy, Hyvarinen) outperform baseline methods (FLE, QRDQN, IQN) in most settings.
>
> | Reward type   | Coverage |  FLE | QRDQN |  IQN |   KL | Energy | PDFL2 |  RBF |  TVD | Hyvarinen |
> |---------------|----------|-----:|------:|-----:|-----:|-------:|------:|-----:|-----:|----------:|
> | Deterministic | Strong   | 3.603 | 6.190 | 8.254 | 1.778 | 3.365 | 4.381 | 5.032 | 8.111 | 2.786 |
> | Deterministic | Weak     | 3.000 | 6.397 | 8.302 | 1.841 | 3.143 | 4.698 | 5.476 | 8.095 | 2.429 |
> | Random        | Strong   | 4.429 | 5.302 | 7.000 | 3.032 | 3.159 | 4.079 | 4.079 | 8.635 | 3.595 |
> | Random        | Weak     | 3.762 | 6.095 | 7.317 | 2.349 | 3.619 | 4.460 | 4.238 | 8.619 | 2.405 |
>
> Since the learned distribution is a conditional distribution, analyzing it requires examining many distributions—one for each state-action pair—which makes it difficult to compare them and identify patterns. That said, we appreciate your suggestions.
>
> ## Weakness-3: Distinct policies in experiments.
>
> In Appendix E.2.4, we have included four pairs of target and behavior policies for each of the seven Atari games. These include epsilon values being 0.3 (target) and 0.8 (behavior) for the deterministic transition (Lines 1542-1548), or 0 (target) and 0.5 (behavior) for the random transition (Lines 1549-1554). Although the target and behavior policies are both epsilon-greedy variants for the same DQN-trained policy $\pi_*$,
> the differences between these epsilon values amount to 0.5, which is large enough to make the target and behavior policies distinct. For each target policy, we have tried two choices of epsilon values for the behavior policy (slightly larger and much larger epsilon than the epsilon value for the target policy), to empirically verify the effects of coverage constant (Assumption C.3). Indeed, larger epsilon value for the behavior policy leads to larger inaccuracy levels, since they represent weaker coverage. This can be seen by comparing Row 1 and Row 2, or comparing Row 3 and Row 4 of Figures 3, 4, and 5.
>
> ## Weakness-4: Total coverage of all state-action pairs.
>
> Thank you for your comment. In (P2), we made a mistake in our writing-the uniqueness is only required up to the support of $\rho$ (i.e., almost everywhere w.r.t. $\rho$). We will revise the paper accordingly. Therefore, the updated (P2) does not require total coverage of all state-action pairs. Without total coverage, we can still achieve the convergence in expectation-extended metrics in Section 3.2, i.e., Theorems C.4 and C.11.
>
> ## Question-1: Why Gaussian mixture model?
>
> The Gaussian mixture model (GMM) is mainly chosen as a flexible model given its ability to approximate smooth densities well with sufficiently large number of components. Since a key prior work FLE [1] also used GMM in their simulations, we simply adopt the same choice in our experiments. For our experiments, certain divergences (among the variations of our FDE method) and existing method (like FLE [1]) require existence of density, which is conveniently satisfied by the use of GMM.
>
> ## Question-2: Typos
>
> Thank you for your careful reading. We will fix them.
>
> ## References
>
> [1] Wu et al. Distributional offline policy evaluation with predictive error guarantees. ICML. 2023.

---

> > ### Comment · Area_Chair_8hzp · 2025-08-05
> > **Reviewer peoQ, please respond to the authors**
> >
> > Reviewer peoQ, please respond to the authors. --AC

---

> > ### Comment · Reviewer_peoQ · 2025-08-07
> >
> > Thank you for your detailed and comprehensive response. Please accept my apologies for my late response.
> >
> > I appreciate that you take the concerns of other reviewers and me concerning the readability of the paper seriously. I believe that the tweaking you intend to do could improve this aspect of the paper.
> >
> > Thank you for discussing the results and experimental set up for the Atari experiments. Your explanation makes sense to me now. It is easy to get a bit lost in the long appendix, however. I appreciate your plan to add a table of contents.
> >
> > Great, I am happy that our discussion could contribute to ironing out an issue in the writing.
> >
> > Your explanation of why a Gaussian mixture model was chosen makes sense to me. Thank you.
> >
> > My questions were fully adressed, thank you.

---

> > > ### Author Response · Authors · 2025-08-08
> > >
> > > Thank you for your response! It is great to hear that the rebuttal has addressed your concerns. We will revise our work according to the plan. We appreciate your helpful comments.

---

### Note · Authors · 2025-08-13

Dear AC and Reviewers,

We thank the reviewers for their constructive feedback, which has helped strengthen our paper. We also thank the AC for overseeing the review process.

We are pleased that our rebuttals addressed the reviewers' concerns. Based on the discussion, we will revise the manuscript. Among the changes we promised to do, the notable ones are moving more theoretical details to the main text, providing proof outlines and a table of content for the appendix, clarifying the motivations and contributions, and adding more discussion and details to the numerical study. We will also correct the typos and mistakes as we promised in our response. We emphasize that while these revisions will surely strengthen our paper by improving the presentation, they do not indicate any fundamental issues with the paper’s main results or contributions. Again, we are grateful for the comments that have led to these revisions.

Best,
Authors

---

### Decision · Program_Chairs · 2025-09-17

**Decision:**

Accept (spotlight)

**Comment:**

The paper considers off policy evaluation for distributional RL, extending fitted Q evaluation (FQE) to the distributional setting. The authors unify treatments of distributional FQE and allows analysis under a range of discrepancy functions and general conditions including infinite spaces. The authors analyze valid discrepancy functions, introducing novel examples, and present convergence results for tabular and non-tabular settings. Empirical evaluations are conducted for linear quadratic regulator (LQR) and Atari games.

Strengths of the paper include the detailed theoretical analysis, and the nice discussion of limitations. One reviewer also noted explicit convergences rates in the appendices. Weaknesses of the paper noted by reviewers include the density of the exposition, and the clarity of positioning with respect to the literature. Reviewers were by and large satisfied with responses to concerns about density. The question of clarity of contribution were not completely resolved, but the reviewer was nevertheless positive about the paper.

Given that reviewers were uniformly positive about the contribution of this work, I recommend acceptance.